# Problem-Parameter-Free Decentralized Bilevel Optimization

**Zhiwei Zhai**    **Wenjing Yan**[*]    **Ying-Jun Angela Zhang**
Department of Information Engineering
The Chinese University of Hong Kong
{zz024, wjyan, yjzhang}@ie.cuhk.edu.hk

## Abstract

Decentralized bilevel optimization has garnered significant attention due to its critical role in solving large-scale machine learning problems. However, existing methods often rely on prior knowledge of problem parameters—such as smoothness, convexity, or communication network topologies—to determine appropriate stepsizes. In practice, these problem parameters are typically unavailable, leading to substantial manual effort for hyperparameter tuning. In this paper, we propose **AdaSDBO**, a fully problem-parameter-free algorithm for decentralized bilevel optimization with a single-loop structure. AdaSDBO leverages adaptive stepsizes based on cumulative gradient norms to update all variables simultaneously, dynamically adjusting its progress and eliminating the need for problem-specific hyperparameter tuning. Through rigorous theoretical analysis, we establish that AdaSDBO achieves a convergence rate of $\widetilde{\mathcal{O}}\left(\frac{1}{T}\right)$, matching the performance of well-tuned state-of-the-art methods up to polylogarithmic factors. Extensive numerical experiments demonstrate that AdaSDBO delivers competitive performance compared to existing decentralized bilevel optimization methods while exhibiting remarkable robustness across diverse stepsize configurations.

## 1 Introduction

Bilevel optimization is a powerful framework widely applied in machine learning, artificial intelligence, and operations research [Camacho-Vallejo et al., 2024, Caselli et al., 2024]. In bilevel optimization, the objective is to optimize a function that is itself dependent on an optimization problem, creating a hierarchical structure of decision-making. This framework models numerous real-world problems where decisions at one level influence outcomes at another, including reinforcement learning [Hong et al., 2023, Thoma et al., 2024, Shen et al., 2025], meta-learning [Bertinetto et al., 2018, Rajeswaran et al., 2019, Ji et al., 2020], adversarial learning [Madry et al., 2017], hyperparameter optimization [Pedregosa, 2016, Franceschi et al., 2018], and imitation learning [Arora et al., 2020]. The flexibility of bilevel optimization makes it an essential tool for modeling complex systems and tackling a wide range of challenges in modern machine learning and optimization.

As datasets continue to grow and machine learning models become more complex, bilevel optimization increasingly necessitates decentralized computation paradigms [Kong et al., 2024]. Decentralized approaches distribute computation across multiple agents that communicate only with their neighbors, thereby significantly reducing communication overhead and enhancing scalability for large-scale problems. These frameworks are particularly valuable in scenarios where centralizing data is infeasible due to privacy concerns or infrastructure limitations [Zhang et al., 2019, Kayaalp et al., 2022]. Applications of decentralized bilevel optimization are prevalent in various domains, including resource allocation [Ji and Ying, 2023], collaborative decision-making [Hashemi et al., 2024], and

---

[*]Corresponding Author.

distributed machine learning [Jiao et al., 2022], where agents collaboratively solve a global bilevel problem while addressing local constraints.

Given its importance, numerous studies have explored the challenges of decentralized bilevel optimization, focusing on algorithm design [Lu et al., 2022b], convergence analysis [Wang et al., 2024], and practical applications [Lu et al., 2022a, Liu et al., 2022]. Among existing methods, double-loop frameworks have been extensively studied for their effectiveness in achieving convergence across various settings [Chen et al., 2024a, 2023]. However, these approaches are computationally expensive due to their nested structure, which requires repeatedly solving lower-level problems during each upper-level iteration. This results in significant computational and communication overhead in decentralized settings. To address these limitations, single-loop methods have emerged as a computationally efficient alternative [Zhu et al., 2024, Dong et al., 2023]. By integrating updates for both levels into a unified process, single-loop frameworks reduce overall complexity and are better suited for real-world decentralized bilevel optimization tasks.

Despite these advancements, existing decentralized bilevel optimization methods face a critical challenge: problem-specific hyperparameter tuning (e.g., stepsizes). In particular, the selection of hyperparameters in these algorithms often relies on problem-specific information, such as smoothness and strong convexity constants, the spectral gap of the graph adjacency matrix, or other topological characteristics. However, obtaining such information is typically infeasible due to physical or privacy constraints and computational limitations, especially in large-scale machine learning applications involving massive datasets. The nested structure of upper- and lower-level objectives in decentralized bilevel problems further exacerbates this challenge. As a result, extensive hyperparameter tuning remains necessary in existing methods, significantly limiting their practicality in real-world scenarios. This raises a fundamental question:

> **Can we design a single-loop decentralized bilevel optimization algorithm that eliminates reliance on problem-specific parameters while achieving comparable performance to well-tuned counterparts?**

## 1.1 Main Contributions

In this paper, we provide an affirmative answer to the above question by proposing an **Ada**ptive **S**ingle-loop **D**ecentralized **B**ilevel **O**ptimization Algorithm (AdaSDBO). AdaSDBO leverages accumulated gradient norms to dynamically adjust stepsizes per iteration, thereby eliminating the need for hyperparameter tuning. We conduct a comprehensive convergence analysis with nonconvex-strongly-convex problem settings, showing that AdaSDBO achieves performance comparable to existing well-tuned approaches. Our main contributions are summarized as follows:

- We propose AdaSDBO, the first parameter-free method for decentralized bilevel optimization with a single-loop structure. AdaSDBO employs adaptive stepsizes based on accumulated (hyper)gradient norms to update all variables simultaneously. However, due to the coupling of bilevel objectives, adaptive stepsizes in a single-loop framework must carefully orchestrate the progress of primal, dual, and auxiliary variables. Additionally, network heterogeneity in decentralized settings introduces inconsistencies in local-gradient-based adaptive stepsizes. To address these challenges, our method incorporates two key mechanisms: 1) **hierarchical stepsize design**, which respects the interdependence of different variables while preserving the autonomy of adaptive stepsizes; 2) **stepsize tracking scheme**, which synchronizes gradient-norm accumulators, effectively managing stepsize discrepancies among agents.

- We provide a comprehensive theoretical analysis, demonstrating that our algorithm eliminates the need for problem-specific hyperparameter tuning while achieving a convergence rate of $\widetilde{\mathcal{O}}\left(\frac{1}{T}\right)$, matching well-tuned counterparts [Ji et al., 2022, Dong et al., 2023] up to polylogarithmic factors. Our analysis is inspired by the two-stage framework [Xie et al., 2020, Ward et al., 2020], but uniquely addresses the intricate coupling between optimization variables and adaptive stepsizes in single-loop bilevel optimization. Furthermore, we conduct a more rigorous analysis to control the interaction between hierarchical optimization errors and network-induced discrepancies, while preserving the problem-parameter-free property.

- We conduct experiments on several machine learning problems, showing that our method performs comparably with existing well-tuned approaches on both synthetic and real-world datasets. Moreover, our method exhibits remarkable robustness across a wide range of initial stepsizes, validating the effectiveness of our adaptive stepsizes design.

Table 1: Comparison between different bilevel optimization algorithms.

$T$ denotes the number of (upper-level) iterations; $\epsilon$ is the target stationarity such that $\sum_{t=0}^{T-1} \|\nabla\Phi(\bar{x}_t)\|^2/T \leq \epsilon$; $\rho_W$ measures the connectivity of the underlying graph; $\mu$ and $L$ are the strongly convex and Lipschitz constants, respectively; $\beta$ represents the momentum parameter.

| Algorithm | Loopless | Convergence Rate$^\diamond$ | Gradient Complexity$^\dagger$ | Parameters$^\uparrow$ |
|---|---|---|---|---|
| DBO [Chen et al., 2024a] | ✗ | $\mathcal{O}\left(\frac{1}{T}\right)$ | $\mathcal{O}\left(\frac{1}{\epsilon^2}\log\left(\frac{1}{\epsilon}\right)\right)$ | $\mu, L, \rho_W$ |
| MDBO [Gao et al., 2023] | ✗ | $\mathcal{O}\left(\frac{1}{\sqrt{T}}\right)$ | $\mathcal{O}\left(\frac{1}{\epsilon^2}\log\left(\frac{1}{\epsilon}\right)\right)$ | $\mu, L, \rho_W, \beta$ |
| FSLA [Li et al., 2022a] | ✓ | $\mathcal{O}\left(\frac{1}{\sqrt{T}}\right)$ | $\mathcal{O}\left(\frac{1}{\epsilon^2}\right)$ | $\mu, L, \beta$ |
| AID [Ji et al., 2022] | ✓ | $\mathcal{O}\left(\frac{1}{T}\right)$ | $\mathcal{O}\left(\frac{1}{\epsilon}\right)$ | $\mu, L, \epsilon$ |
| SLDBO [Dong et al., 2023] | ✓ | $\mathcal{O}\left(\frac{1}{T}\right)$ | $\mathcal{O}\left(\frac{1}{\epsilon}\right)$ | $\mu, L, \rho_W$ |
| **AdaSDBO (This paper)** | ✓ | $\mathcal{O}\left(\frac{\log^4(T)}{T}\right)$ | $\mathcal{O}\left(\frac{1}{\epsilon}\log^4\left(\frac{1}{\epsilon}\right)\right)$ | None |

$^\diamond$ The convergence rate when $T \to \infty$.

$^\dagger$ The number of gradient/Jacobian/Hessian evaluations per agent to achieve $\epsilon$-accuracy when $\epsilon \to 0$.

$^\uparrow$ Stepsize-related problem-specific parameters.

## 1.2 Related Works

**Decentralized Bilevel Optimization.** Recent advancements in decentralized bilevel optimization have focused on addressing the challenges of large-scale data and leveraging the computational benefits of parallel environments. Chen et al. [2024a] proposed DBO, a general framework that incorporates convergence analysis while accounting for data heterogeneity across agents. Similarly, MA-DSBO [Chen et al., 2023] and MDBO [Gao et al., 2023] employed a double-loop framework with momentum techniques [Liu et al., 2020]. More recently, single-loop frameworks have emerged as efficient alternatives to double-loop methods. These approaches [Chen et al., 2024b, Dagréou et al., 2022, Kong et al., 2024, Zhang et al., 2023] enable approximate solutions to decentralized bilevel problems within a single iteration, significantly improving computational efficiency by reducing redundant computations. Such methods have made decentralized bilevel optimization more practical and scalable for large-scale applications. Dong et al. [2023] further introduced SLDBO, a low-complexity single-loop decentralized bilevel algorithm that leverages gradient tracking technology. Despite these advancements, existing methods rely on fixed or uniformly decaying stepsizes. Further, they require prior knowledge of problem parameters for stepsize selection. This dependency imposes additional challenges, particularly in decentralized settings where such information is often unavailable or difficult to estimate. Further details on bilevel optimization, adaptive methods, and their applications are provided in Section A.

## 1.3 Comparisons with Prior Approaches

We compare AdaSDBO with representative bilevel optimization methods, as summarized in Table 1. Notably, AdaSDBO adopts a loopless framework while achieving a convergence rate that matches the state-of-the-art results, up to a polylogarithmic factor of $\log^4(T)$. Since logarithmic factors grow significantly slower than polynomial terms, this factor is negligible relative to $T$, a common consideration in optimization research [Yang et al., 2022, Li et al., 2024]. By carefully controlling network-induced errors, AdaSDBO matches both the convergence rate and gradient complexity of its centralized counterpart AID [Ji et al., 2022], while outperforming the centralized method FSLA [Li et al., 2022a] in both metrics. Compared to decentralized approaches such as DBO [Chen et al., 2024b], MDBO [Gao et al., 2023], and SLDBO [Dong et al., 2023], AdaSDBO achieves the best-known convergence rate of $\mathcal{O}\left(\frac{1}{T}\right)$ and gradient complexity of $\mathcal{O}\left(\frac{1}{\epsilon}\right)$, while surpassing double-loop methods in gradient complexity—underscoring the efficiency of its single-loop framework. Most importantly, AdaSDBO is a completely tuning-free algorithm independent of problem parameters, which is in sharp contrast to other methods that require extensive hyperparameter tuning. This advantage significantly simplifies algorithm deployment, facilitating the implementation of bilevel optimization in diverse environments.

## 2 Algorithm Development

### 2.1 Problem Model

In this paper, we consider a networked system consisting of $n$ nodes (agents) that collectively solve the following nonconvex-strongly-convex bilevel optimization problem:

$$\min_{x \in \mathbb{R}^p} \Phi(x) = f(x, y^*(x)) := \frac{1}{n} \sum_{i=1}^{n} f_i(x, y^*(x)),$$

$$\text{s.t. } y^*(x) = \arg\min_{y \in \mathbb{R}^q} l(x, y) := \frac{1}{n} \sum_{i=1}^{n} l_i(x, y),$$

(1)

where $f(\cdot)$ represents the upper-level objective function, which is minimized with respect to $x$, subject to the constraint that $y^*(x)$ is a minimizer of the lower-level function $l(\cdot)$. Each agent $i$ has a possibly nonconvex objective function $f_i(\cdot) : \mathbb{R}^p \times \mathbb{R}^q \to \mathbb{R}$ and a strongly convex objective function $l_i(\cdot) : \mathbb{R}^p \times \mathbb{R}^q \to \mathbb{R}$ with respect to $y$. The agents are interconnected through a communication network modeled as a graph $\mathcal{G} = (\mathcal{N}, \mathcal{E})$, where $\mathcal{N} = \{1, 2, \dots, n\}$ is the set of nodes (agents), and $\mathcal{E} \subseteq \mathcal{N} \times \mathcal{N}$ is the set of edges representing communication links. An edge $(i, j) \in \mathcal{E}$ indicates a communication link between nodes $j$ and node $i$. We represent the weight matrix of the communication network $\mathcal{G}$ as $W = (w_{ij}) \in \mathbb{R}^{n \times n}$, where $w_{ij} = 0$ if $(i, j) \notin \mathcal{E}$. The following assumption is made on $W$.

**Assumption 2.1.** The matrix $W$ is doubly stochastic, i.e., $W\mathbf{1} = \mathbf{1}$, $\mathbf{1}^\top W = \mathbf{1}^\top$, and $\rho_W := \|W - \mathbf{J}\|_2^2 < 1$, where $\mathbf{J} = \mathbf{1}\mathbf{1}^\top / n$ is the averaging matrix with $n$ dimension.

A key challenge in solving the Problem (1) is the computation of the hypergradient $\nabla\Phi(x)$, which is expressed as:

$$\nabla\Phi(x) := \nabla_x f(x, y^*(x)) - \nabla_x \nabla_y l(x, y^*(x))[\nabla_y \nabla_y l(x, y^*(x))]^{-1} \nabla_y f(x, y^*(x)), \quad (2)$$

derived using the implicit function theorem [Ghadimi and Wang, 2018]. First, the lower-level minimizer $y^*(x)$ is typically not directly accessible, requiring iterative algorithms to approximate it with an estimate $\hat{y}$. Second, the expression in Eq. (2) involves the inversion of the Hessian matrix $\nabla_y \nabla_y l(x, y^*(x))$, which is computationally expensive, with the complexity of $\mathcal{O}(q^3)$. Additionally, in decentralized settings, the difficulty is further exacerbated because each agent has access only to its local problem and lacks information about the global lower-level objective $l(\cdot)$. To overcome this challenge, we introduce the following linear system:

$$\min_v r(x, \hat{y}, v) := \frac{1}{2} v^\top \nabla_y \nabla_y l(x, \hat{y}) v - v^\top \nabla_y f(x, \hat{y}), \quad (3)$$

which seeks to approximate the inversion of the Hessian matrix by $v^*(x) = [\nabla_y \nabla_y l(x, y^*(x))]^{-1} \nabla_y f(x, y^*(x))$ when $\hat{y}$ approaching $y^*(x)$, where $v \in \mathcal{V}$. An iterative algorithm is then employed to compute an approximate solution $\hat{v}$ to the Problem (3). Using the approximations $\hat{y}$ and $\hat{v}$, the $x$ is subsequently updated based on a hypergradient estimate as:

$$\bar{\nabla} f(x, \hat{y}, \hat{v}) := \nabla_x f(x, \hat{y}) - \nabla_x \nabla_y l(x, \hat{y})\hat{v}. \quad (4)$$

Building on this framework, solving the decentralized bilevel optimization problem (1) can be reformulated into three interconnected subproblems:

$$x^* = \arg\min_{x \in \mathbb{R}^p} \frac{1}{n} \sum_{i=1}^{n} f_i(x, y^*(x), \quad (5a) \qquad y^*(x) = \arg\min_{y \in \mathbb{R}^q} \frac{1}{n} \sum_{i=1}^{n} l_i(x, y), \quad (5b)$$

$$v^*(x) = \arg\min_{v \in \mathbb{R}^q} \frac{1}{n} \sum_{i=1}^{n} r_i(x, y^*(x), v). \quad (5c)$$

In (5), the subproblems (5a) and (5c) depend on the optimal solution of (5b). This naturally motivates many existing works [Chen et al., 2023, 2024a] to adopt a nested structure, where (5b) is solved up to a certain accuracy before sequentially computing (5c) and (5a). However, these nested loops result in substantial computational overhead and implementation difficulties. To address this, a single-loop framework [Dong et al., 2023, Zhu et al., 2024] has been proposed to solve the three subproblems in parallel, thereby improving computational efficiency in bilevel optimization. Nevertheless, the single-loop structure introduces significant challenges in algorithm design and convergence analysis.

Furthermore, existing decentralized bilevel optimization solutions predominantly rely on problem-specific parameters for algorithm tuning, such as smoothness and strong convexity constants, the spectral gap of the graph adjacency matrix, or other topological characteristics. Obtaining such information is often impractical due to physical or privacy constraints, as well as computational limitations—particularly in large-scale machine learning applications involving massive datasets.

In light of these challenges, this paper aims to develop a single-loop algorithm for solving Problem (5) while achieving comparable convergence guarantees without the need for hyperparameter tuning.

## 2.2 Algorithm Development

In this subsection, we propose an Adaptive Single-loop Decentralized Bilevel Optimization Algorithm (AdaSDBO) based on iterative gradient updates, as presented in Algorithm 1. Let $x_{i,t} \in \mathbb{R}^p$, $y_{i,t} \in \mathbb{R}^q$, and $v_{i,t} \in \mathbb{R}^q$ denote the iterates at agent $i$ for variables $x$, $y$, and $v$, respectively. According to Problem (5), the local gradients at each agent $i$ can be expressed as:

$$\bar{\nabla} f_i(x_{i,t}, y_{i,t}, v_{i,t}) = \nabla_x f_i(x_{i,t}, y_{i,t}) - \nabla_x \nabla_y l_i(x_{i,t}, y_{i,t}) v_{i,t},$$

$$\nabla_v r_i(x_{i,t}, y_{i,t}, v_{i,t}) = \nabla_y \nabla_y l_i(x_{i,t}, y_{i,t}) v_{i,t} - \nabla_y f_i(x_{i,t}, y_{i,t}).$$

For brevity, let $g_{i,t}^x := \bar{\nabla} f_i(x_{i,t}, y_{i,t}, v_{i,t})$, $g_{i,t}^y := \nabla_y l_i(x_{i,t}, y_{i,t})$, and $g_{i,t}^v := \nabla_v r_i(x_{i,t}, y_{i,t}, v_{i,t})$.

**Adaptive Stepsizes Design.** To eliminate the dependency on problem-specific parameters, we design the adaptive stepsizes strategy based on accumulated gradient norms. Specifically, we introduce an accumulator $m_{i,t+1}^y$ as $[m_{i,t+1}^y]^2 = [m_{i,t}^y]^2 + \|g_{i,t}^y\|^2$. Using this accumulator, the dual variable $y_{i,t}$ is updated by $y_{i,t+1} = y_{i,t} - \frac{\gamma_y}{m_{i,t+1}^y} g_{i,t}^y$, where $\gamma_y > 0$ is a control coefficient that is independent of the problem parameters. Similarly, the accumulators $m_{i,t}^x$ and $m_{i,t}^v$ are defined by the updates $[m_{i,t+1}^x]^2 = [m_{i,t}^x]^2 + \|g_{i,t}^x\|^2$ and $[m_{i,t+1}^v]^2 = [m_{i,t}^v]^2 + \|g_{i,t}^v\|^2$, respectively. However, the adaptive update rule for dual variable $y_{i,t}$ cannot be directly extended to primal variable $x_{i,t}$ and auxiliary variable $v_{i,t}$ due to their intricate interdependencies with other variables. The primary challenges are:

- **Auxiliary-Level Update**: The update of the auxiliary variable $v_{i,t}$ requires the optimal solution $y^*$ of the lower-level subproblem. Since single-loop algorithms perform only one-step updates of the variable $y_{i,t}$, the suboptimality gap $\|y_{i,t+1} - y^*\|^2$ must be carefully managed to prevent error accumulation. Thus, $v_{i,t}$ must progress no faster than $y_{i,t}$ to maintain approximation accuracy.

- **Upper-Level Update**: The primal variable $x_{i,t}$ depends on both the optimal variables $y^*$ and $v^*$, introducing additional complexity. Errors from both the lower-level and auxiliary-level updates must be considered, necessitating more conservative updates of the variable $x_{i,t}$ to align with the slower dynamics of these levels.

To overcome these challenges, we propose the following hierarchical stepsizes:

- **Auxiliary variable update** ($v_{i,t}$): We employ a stepsize inversely proportional to $\max(m_{i,t+1}^v, m_{i,t+1}^y)$, resulting in the update rule $v_{i,t+1} = v_{i,t} - \frac{\gamma_v}{\max(m_{i,t+1}^v, m_{i,t+1}^y)} g_{i,t}^v$.

- **Primal variable update** ($x_{i,t}$): We use a more conservative stepsize, inversely proportional to $m_{i,t+1}^x \max(m_{i,t+1}^v, m_{i,t+1}^y)$, yields the update $x_{i,t+1} = x_{i,t} - \frac{\gamma_x}{m_{i,t+1}^x \max(m_{i,t+1}^v, m_{i,t+1}^y)} g_{i,t}^x$.

Here, $\gamma_v > 0$ and $\gamma_x > 0$ are control coefficients.

This hierarchical stepsize design plays a critical role in balancing the progress speeds at different levels, ensuring stability and convergence in single-loop optimization. Additionally, the adaptive stepsizes based on accumulated gradient norms dynamically adjust to the local optimization geometry, enhancing the efficiency and accuracy of the algorithm without requiring problem-specific parameters.

For notational convenience, we define the stepsize variables $q_{i,t+1} := m_{i,t+1}^x \max\{m_{i,t+1}^v, m_{i,t+1}^y\}$, $u_{i,t+1} := m_{i,t+1}^y$, and $z_{i,t+1} := \max\{m_{i,t+1}^v, m_{i,t+1}^y\}$. Correspondingly, we define the following diagonal stepsize matrices as:

$$Q_{t+1} = \text{diag}\{q_{i,t+1}\}_{i=1}^n, \quad U_{t+1} = \text{diag}\{u_{i,t+1}\}_{i=1}^n, \quad Z_{t+1} = \text{diag}\{z_{i,t+1}\}_{i=1}^n.$$

Additionally, define the concatenated variable matrices as: $\mathbf{x}_t := [\dots, x_{i,t}, \dots]^\top \in \mathbb{R}^{n \times p}$, $\mathbf{y}_t := [\dots, y_{i,t}, \dots]^\top \in \mathbb{R}^{n \times q}$, and $\mathbf{v}_t := [\dots, v_{i,t}, \dots]^\top \in \mathbb{R}^{n \times q}$. We also concatenate the gradient vectors as:

$$\bar{\nabla} F(\mathbf{x}_t, \mathbf{y}_t, \mathbf{v}_t) := \left[\cdots, \bar{\nabla} f_i(x_{i,t}, y_{i,t}, v_{i,t}), \cdots\right]^\top, \quad \nabla_y L(\mathbf{x}_t, \mathbf{y}_t) := \left[\cdots, \nabla_y l_i(x_{i,t}, y_{i,t}), \cdots\right]^\top,$$

$$\nabla_v R(\mathbf{x}_t, \mathbf{y}_t, \mathbf{v}_t) = \left[\cdots, \nabla_v r_i(x_{i,t}, y_{i,t}, v_{i,t}), \cdots\right]^\top.$$

Based on these definitions, the update rules for the primal, dual, and auxiliary variables are given by:

$$\mathbf{x}_{t+1} = W\left(\mathbf{x}_t - \gamma_x Q_{t+1}^{-1} \bar{\nabla} F(\mathbf{x}_t, \mathbf{y}_t, \mathbf{v}_t)\right), \quad \mathbf{y}_{t+1} = W\left(\mathbf{y}_t - \gamma_y U_{t+1}^{-1} \nabla_y L(\mathbf{x}_t, \mathbf{y}_t)\right),$$

$$\mathbf{v}_{t+1} = \mathcal{P}_\mathcal{V}\left(W\left(\mathbf{v}_t - \gamma_v Z_{t+1}^{-1} \nabla_v R(\mathbf{x}_t, \mathbf{y}_t, \mathbf{v}_t)\right)\right),$$

where $\mathcal{P}_\mathcal{V}(\cdot)$ denotes the projection operation onto the set $\mathcal{V}$.

**Algorithm 1** Adaptive Single-Loop Decentralized Bilevel Optimization: Procedures at Each Agent $i \in [n]$

---

1: **Initialization:** $x_{i,0}, y_{i,0}, v_{i,0}, m_{i,0}^x = m_{i,0}^y = m_{i,0}^v > 0, \gamma_x = \gamma_y = \gamma_v > 0$.
2: **for** $t = 0, 1, \cdots, T-1$ **do**
3:     Compute the gradients:
    $g_{i,t}^y = \nabla_y l_i(x_{i,t}, y_{i,t}),$
    $g_{i,t}^v = \nabla_y \nabla_y l_i(x_{i,t}, y_{i,t}) v_{i,t} - \nabla_y f_i(x_{i,t}, y_{i,t}),$
    $g_{i,t}^x = \nabla_x f_i(x_{i,t}, y_{i,t}) - \nabla_x \nabla_y l_i(x_{i,t}, y_{i,t}) v_{i,t}.$
4:     Accumulate the gradient norms:
    $[m_{i,t+1}^x]^2 = [m_{i,t}^x]^2 + \|g_{i,t}^x\|^2, [m_{i,t+1}^y]^2 = [m_{i,t}^y]^2 + \|g_{i,t}^y\|^2, [m_{i,t+1}^v]^2 = [m_{i,t}^v]^2 + \|g_{i,t}^v\|^2.$
5:     Update the primal, dual, and auxiliary variables by:
    $y_{i,t+1} = y_{i,t} - \frac{\gamma_y}{m_{i,t+1}^y} g_{i,t}^y,$
    $v_{i,t+1} = v_{i,t} - \frac{\gamma_v}{\max(m_{i,t+1}^v, m_{i,t+1}^y)} g_{i,t}^v,$
    $x_{i,t+1} = x_{i,t} - \frac{\gamma_x}{m_{i,t+1}^x \max(m_{i,t+1}^v, m_{i,t+1}^y)} g_{i,t}^x.$
6:     Information exchange with neighbors:
    $\{x, y, v\}_{i,t+1} \leftarrow \sum_j w_{i,j}\{x, y, v\}_{j,t+1},$
    $\{m^x, m^y, m^v\}_{i,t+1} \leftarrow \sum_j w_{i,j}\{m^x, m^y, m^v\}_{j,t+1}.$
7:     Projection of auxiliary variable on the set $\mathcal{V}$: $v_{i,t+1} \leftarrow \mathcal{P}_{\mathcal{V}}(v_{i,t+1}).$
8: **end for**

---

**Addressing Stepsize Inconsistencies.** In decentralized bilevel settings, agents compute their adaptive stepsizes independently based on local private objective functions, and the coupling of multiple optimization variables within adaptive stepsizes amplifies their network-wide inconsistencies. These discrepancies can hinder convergence if not properly controlled. To formalize this, let $\bar{x}_t := \frac{1}{n} \sum_{i=1}^n x_{i,t}$ denote the average of all primal variables at the $t$-th iteration, and $\bar{q}_t := \frac{1}{n} \sum_{i=1}^n q_{i,t}$ represent the average of their respective stepsizes. We then define the stepsize discrepancy vector as $\tilde{\mathbf{q}}_t^{-1} := \left[ \cdots, q_{i,t}^{-1} - \bar{q}_t^{-1}, \cdots \right]^\top$. Using this definition, the update rule for $\bar{x}_t$ can be expressed as:

$$\bar{x}_{t+1} = \bar{x}_t - \gamma_x \left( \underbrace{\frac{\bar{q}_{t+1}^{-1} \mathbf{1}^\top}{n}}_{(a)} + \underbrace{\frac{(\tilde{\mathbf{q}}_{t+1}^{-1})^\top}{n}}_{(b)} \right) \bar{\nabla} F(\mathbf{x}_t, \mathbf{y}_t, \mathbf{v}_t). \tag{6}$$

In Eq. (6), term (a) resembles centralized gradient descent, while term (b) introduces an undesired perturbation caused by stepsize inconsistencies among agents. This perturbation can disrupt network consensus due to varying update rates and may lead to uncontrollable error growth, as term (b) represents accumulated gradient-norm discrepancies over iterations. Such stepsize inconsistencies similarly affect the updates of the dual and auxiliary variables.

To address this challenge, we incorporate a stepsize tracking mechanism in Algorithm 1. At each iteration, the gradient-norm accumulators $m_{i,t}^x, m_{i,t}^y,$ and $m_{i,t}^v$ are tracked by:

$$[m_{i,t+1}^b]^2 = \sum_{j=1}^n w_{ij}[m_{j,t+1}^b]^2 = \sum_{j=1}^n w_{ij} \left([m_{j,t}^b]^2 + \|g_{j,t}^b\|^2\right),$$

where $b \in \{x, y, v\}$ and $i \in [n]$. Let $\mathbf{k}_t^b := \left[\cdots, [m_{i,t}^b]^2, \cdots\right]^\top$ and $\mathbf{h}_t^b := \left[\cdots, \|g_{i,t}^b\|^2, \cdots\right]^\top$ denote the concatenated gradient accumulators and corresponding norms, respectively. The above equation can then be compactly expressed as $\mathbf{k}_{t+1}^b = W\left(\mathbf{k}_t^b + \mathbf{h}_t^b\right)$. This tracking mechanism enforces consensus on the gradient-norm accumulators before computing the adaptive stepsizes $(q_{i,t}, u_{i,t}, z_{i,t})$. By synchronizing these accumulators, it effectively bounds stepsize discrepancies among agents, preventing error accumulation while preserving the adaptive nature of the updates.

It is worth noting that the additional communication overhead of our method is modest—only scalar values (the stepsize accumulators) are exchanged—especially compared with transmitting the primal, dual, and auxiliary variables commonly communicated in decentralized bilevel methods. Some approaches [Gao et al., 2023, Chen et al., 2023, Zhu et al., 2024] also employ a gradient tracking mechanism, which involves exchanging extra tracker states with the same dimensionality as the

optimization variables. In contrast, our method adds only lightweight scalars for transmission, while offering a robust, problem-parameter-free solution for decentralized bilevel optimization.

# 3 Theoretical Analysis

## 3.1 Technical Challenges

The analysis of problem-parameter-free decentralized bilevel optimization with a single-loop structure involves several fundamental challenges:

- **Interdependent Variable Updates:** The coupling of bilevel objectives creates intricate interdependencies among the variables $(x, y, v)$, making the convergence analysis significantly more challenging compared to single-level optimization.
- **Coupled Stepsize Dynamics:** The adaptive stepsizes exhibit highly intertwined dynamics, forming a multi-stage system in which the progress of each variable directly affects the others, requiring meticulous coordination to manage these interactions effectively.
- **Accumulated Stepsize Inconsistencies:** Inconsistencies in adaptive stepsizes across agents disrupt network consensus, while their cumulative effect over iterations further exacerbates the challenge.
- **Interplay Between Optimization and Consensus Errors:** The interaction between hierarchical optimization errors and network-induced discrepancies necessitates rigorous theoretical bounds to guarantee convergence while maintaining the problem-parameter-free property.

## 3.2 Assumptions and Definitions

In this subsection, we present the standard assumptions and definitions used in our analysis.

**Assumption 3.1.** For any $i \in [n]$, the objective functions $f_i(x, y)$ and $l_i(x, y)$ are twice continuously differentiable, and $l_i(x, y)$ is $\mu$-strongly convex with respect to $y$.

**Assumption 3.2.** For any $i \in [n]$, the function $f_i(x, y)$ is $L_{f,0}$-Lipschitz continuous; the gradients $\nabla f_i(x, y)$ and $\nabla l_i(x, y)$ are $L_{f,1}$- and $L_{l,1}$-Lipschitz continuous, respectively; the second-order gradients $\nabla_x \nabla_y l_i(x, y)$ and $\nabla_y \nabla_y l_i(x, y)$ are $L_{l,2}$-Lipschitz continuous.

The above assumptions are commonly adopted in prior works, including [Zhu et al., 2024, Chen et al., 2024a, 2023, Dong et al., 2023, Ji et al., 2022].

*Remark* 3.3. Assumption 3.2 indicates that there exist constants $C_{f_x}$, $C_{f_y}$, $C_{l_{xy}}$, and $C_{l_{yy}}$ such that $\|\nabla_x f_i(x, y)\| \leq C_{f_x}$, $\|\nabla_y f_i(x, y)\| \leq C_{f_y}$, $\|\nabla_x \nabla_y l_i(x, y)\| \leq C_{l_{xy}}$, and $\|\nabla_y \nabla_y l_i(x, y)\| \leq C_{l_{yy}}$.

Define $\bar{u}_t := \frac{1}{n} \sum_{i=1}^{n} u_{i,t}$, $\bar{z}_t := \frac{1}{n} \sum_{i=1}^{n} z_{i,t}$, and recall that $\bar{q}_t := \frac{1}{n} \sum_{i=1}^{n} q_{i,t}$. We then introduce the following metrics to quantify the level of stepsize inconsistency among agents:

$$\zeta_q^2 := \sup_{i \in [n], t > 0} \frac{\left(q_{i,t}^{-1} - \bar{q}_t^{-1}\right)^2}{\left(\bar{q}_t^{-1}\right)^2}, \quad \zeta_u^2 := \sup_{i \in [n], t > 0} \frac{\left(u_{i,t}^{-1} - \bar{u}_t^{-1}\right)^2}{\left(\bar{u}_t^{-1}\right)^2}, \quad \zeta_z^2 := \sup_{i \in [n], t > 0} \frac{\left(z_{i,t}^{-1} - \bar{z}_t^{-1}\right)^2}{\left(\bar{z}_t^{-1}\right)^2},$$

$$\sigma_q^2 := \inf_{i \in [n], t > 0} \frac{\left(q_{i,t}^{-1} - \bar{q}_t^{-1}\right)^2}{\left(\bar{q}_t^{-1}\right)^2}, \quad \sigma_u^2 := \inf_{i \in [n], t > 0} \frac{\left(u_{i,t}^{-1} - \bar{u}_t^{-1}\right)^2}{\left(\bar{u}_t^{-1}\right)^2}, \quad \sigma_z^2 := \inf_{i \in [n], t > 0} \frac{\left(z_{i,t}^{-1} - \bar{z}_t^{-1}\right)^2}{\left(\bar{z}_t^{-1}\right)^2}.$$

These metrics are guaranteed to remain bounded under Assumption 3.2 and Remark 3.3.

**Definition 3.4.** An output $\bar{x}$ of an algorithm is the $\epsilon$-accurate stationary point of the objective function $\Phi(x)$ if it satisfies $\|\nabla \Phi(\bar{x})\|^2 \leq \epsilon$, where $\epsilon \in (0, 1)$.

## 3.3 Theoretical Results

In this subsection, we present the main theoretical results for the proposed Algorithm 1, with the proof sketch provided in Section B. As outlined in Section D.2, the descent behavior of the objective function $\Phi(\cdot)$ is governed by three key factors: approximation errors, consensus errors, and stepsize inconsistencies. These components are bounded in the following lemmas.

To facilitate the analysis, we first define $\bar{x}_t := \frac{1}{n} \sum_{i=1}^{n} x_{i,t}$, $\bar{y}_t := \frac{1}{n} \sum_{i=1}^{n} y_{i,t}$, and $\bar{v}_t := \frac{1}{n} \sum_{i=1}^{n} v_{i,t}$. Then, let $\bar{m}_t^x := \frac{1}{n} \sum_{i=1}^{n} m_{i,t}^x$ and $\bar{m}_t^y := \frac{1}{n} \sum_{i=1}^{n} m_{i,t}^y$ represent the average of the gradient accumulators. Additionally, denote $f(\bar{x}_t, \bar{y}_t, \bar{v}_t) := \frac{1}{n} \sum_{i=1}^{n} f_i(\bar{x}_t, \bar{y}_t, \bar{v}_t)$, $l(\bar{x}_t, \bar{y}_t) :=$

$\frac{1}{n}\sum_{i=1}^n l_i(\bar{x}_t,\bar{y}_t)$, $r(\bar{x}_t,\bar{y}_t,\bar{v}_t):=\frac{1}{n}\sum_{i=1}^n r_i(\bar{x}_t,\bar{y}_t,\bar{v}_t)$ as the corresponding aggregated functions when the variables $(\mathbf{x},\mathbf{y},\mathbf{v})$ achieve consensus to $(\bar{x},\bar{y},\bar{v})$.

From the descent lemma in Section D.2, we obtain that the approximation error $\|\nabla\Phi(\bar{x}_t) - \bar{\nabla}f(\bar{x}_t,\bar{y}_t,\bar{v}_t)\|^2$ is attributed in terms of $\mathcal{O}(\|\nabla_y l(\bar{x}_t,\bar{y}_t)\|^2)$ and $\mathcal{O}(\|\nabla_v r(\bar{x}_t,\bar{y}_t,\bar{v}_t)\|^2)$. Hence, we establish the following lemma to provide bounds for these terms associated with approximation errors during the optimization process.

**Lemma 3.5** (**Approximation Errors**). *Under Assumptions 3.1 and 3.2, for any integer $k_0 \in [0,t)$, we have $\sum_{k=k_0}^t \frac{\|\nabla_y l(\bar{x}_k,\bar{y}_k)\|^2}{\bar{m}_{k+1}^y} \leq a_5 \log(t+1)+b_5$ and $\sum_{k=k_0}^t \frac{\|\nabla_v r(\bar{x}_k,\bar{y}_k,\bar{v}_k)\|^2}{\bar{z}_{k+1}} \leq a_6 \log(t+1)+b_6$, where the constants $a_5$, $b_5$, $a_6$, and $b_6$ are defined in Eq. (108) of Section D.8.*

**Lemma 3.6** (**Accumulated Gradients**). *Under Assumptions 3.1 and 3.2, the gradient accumulators satisfy the bounds $\bar{m}_t^x \leq \mathcal{O}(\log(t))$ and $\bar{z}_t \leq \mathcal{O}(\log(t))$.*

Lemma 3.6 shows that the accumulated gradient norms for all variables $(x,y,v)$ grow at most logarithmically with respect to the iteration index $t$.

**Lemma 3.7** (**Consensus Errors**). *Suppose that Assumptions 2.1, 3.1, and 3.2 hold. Let $\Delta_t := \|\mathbf{x}_t - \mathbf{1}\bar{x}_t\|^2 + \|\mathbf{y}_t - \mathbf{1}\bar{y}_t\|^2 + \|\mathbf{v}_t - \mathbf{1}\bar{v}_t\|^2$ represent the consensus errors for all variables at the $t$-th iteration. Then, the time-averaged consensus error satisfies $\frac{1}{T}\sum_{t=0}^{T-1}\Delta_t \leq \mathcal{O}(\log(T)/T)$.*

Next, we provide the bounds for the terms associated with the stepsize inconsistencies in our analysis.

**Lemma 3.8** (**Stepsize Inconsistencies**). *Suppose Assumptions 2.1, 3.1, and 3.2 hold. Define discrepancy vectors as $\tilde{\mathbf{q}}_t^{-1}:=[\cdots,q_{i,t}^{-1}-\bar{q}_t^{-1},\cdots]^\top$, $\tilde{\mathbf{u}}_t^{-1}:=[\cdots,u_{i,t}^{-1}-\bar{u}_t^{-1},\cdots]^\top$, and $\tilde{\mathbf{z}}_t^{-1}:=[\cdots,z_{i,t}^{-1}-\bar{z}_t^{-1},\cdots]^\top$. Then, under Algorithm 1, we have that $\frac{1}{T}\sum_{t=0}^{T-1}\left\|(\tilde{\mathbf{q}}_{t+1}^{-1})^\top \bar{\nabla}F(\mathbf{x}_t,\mathbf{y}_t,\mathbf{v}_t)/n\bar{q}_{t+1}^{-1}\right\|^2$, $\frac{1}{T}\sum_{t=0}^{T-1}\left\|(\tilde{\mathbf{u}}_{t+1}^{-1})^\top \nabla_y L(\mathbf{x}_t,\mathbf{y}_t)/n\bar{u}_{t+1}^{-1}\right\|^2$, and $\frac{1}{T}\sum_{t=0}^{T-1}\left\|(\tilde{\mathbf{z}}_{t+1}^{-1})^\top \nabla_v R(\mathbf{x}_t,\mathbf{y}_t,\mathbf{v}_t)/n\bar{z}_{t+1}^{-1}\right\|^2$ are each upper-bounded by $\mathcal{O}(\log(T)/T)$.*

Combining the above results, we establish the convergence of Algorithm 1 as follows.

**Theorem 3.9.** *Under Assumptions 2.1, 3.1, and 3.2, for any positive constants $\gamma_x$, $\gamma_y$, $\gamma_v$, $m_{i,0}^x$, $m_{i,0}^y$, and $m_{i,0}^v$, the iterates generated by Algorithm 1 satisfy:*

$$\frac{1}{T}\sum_{t=0}^{T-1}\|\nabla\Phi(\bar{x}_t)\|^2 \leq \frac{2}{T}\left[\left(4\left(\frac{\Phi(\bar{x}_0)-\Phi^*}{\gamma_x}\right)+a_7\log(T)+b_7\right)(a_1\log(T)+b_1)^2\right.$$
$$\left. +C_{m^x}\left(4\left(\frac{\Phi(\bar{x}_0)-\Phi^*}{\gamma_x}\right)+a_7\log(T)+b_7\right)(a_1\log(T)+b_1)\right] = \mathcal{O}\left(\frac{\log^4(T)}{T}\right),$$

*where $\Phi^* := \inf_{x\in\mathbb{R}^p}\Phi(x) > -\infty$, and the constants $C_{m^x}$, $a_1$, $b_1$, $a_7$, and $b_7$ are defined in Eqs. (30), (70), and (126) in the Appendix.*

*Remark* 3.10. Theorem 3.9 implies that for any positive coefficients $(\gamma_x,\gamma_y,\gamma_v)$ and positive initial stepsizes $(m_{i,0}^x, m_{i,0}^y, m_{i,0}^v)$, Algorithm 1 guarantees convergence with a rate of $\mathcal{O}\left(\frac{\log^4(T)}{T}\right)$. This convergence rate matches the state-of-the-art results, up to a polylogarithmic factor of $\log^4(T)$, which is regarded negligible relative to $T$ in the optimization literature [Yang et al., 2022, Li et al., 2024].

**Corollary 3.11.** *From Theorem 3.9, to achieve an $\epsilon$-accurate stationary point, Algorithm 1 requires $T = \mathcal{O}\left(\frac{1}{\epsilon}\log^4\left(\frac{1}{\epsilon}\right)\right)$ iterations, resulting in a gradient complexity of $\mathrm{Gc}(\epsilon) = \mathcal{O}\left(\frac{1}{\epsilon}\log^4\left(\frac{1}{\epsilon}\right)\right)$.*

In sharp contrast, the theoretical convergence of existing decentralized bilevel methods heavily relies on the correct selection of hyperparameters based on problem-specific constants, such as $\mu$, $L$, and $\rho_W$. This reliance restricts their applicability, as these parameters are unknown or difficult to determine. In comparison, our approach significantly simplifies the implementation of decentralized bilevel optimization by operating in a completely problem-parameter-free manner.

## 4 Numerical Experiments

In this section, we evaluate the performance of Algorithm 1 on the hyperparameter optimization problem, as illustrated in Section E.1. Our algorithm is compared with several decentralized bilevel optimization methods, including SLDBO [Dong et al., 2023], MA-DSBO [Chen et al., 2023], MDBO [Gao et al., 2023], and DBO [Chen et al., 2024a]. Experiments are conducted on both synthetic and real-world datasets, with detailed configurations and additional results provided in Section E.

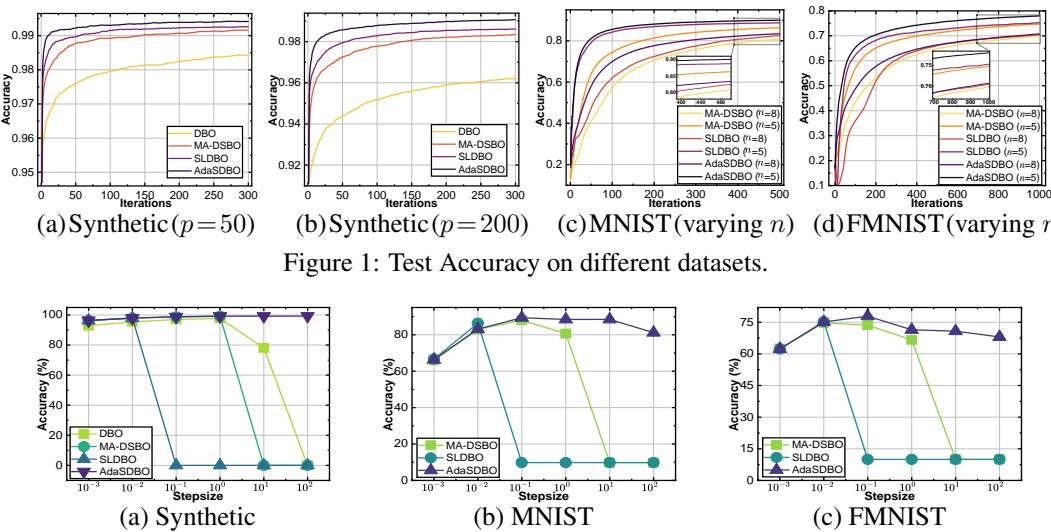

Figure 1: Test Accuracy on different datasets.

(a) Synthetic   (b) MNIST   (c) FMNIST

Figure 2: Test accuracy versus stepsize on different datasets.

## 4.1 Synthetic Data Experiments

For synthetic data, the data distribution at node $i$ follows $\mathcal{N}(0, i^2 \cdot r^2)$, where the parameter $r$ controls the level of data heterogeneity. To evaluate the advantages of AdaSDBO, we first compare different methods under low data heterogeneity conditions with $r = 1$ (Experiments under higher heterogeneity conditions are presented in Figure 4 of Section E.6). Both algorithms compute full gradients and utilize a training dataset and a testing dataset, each containing 20,000 samples. As illustrated in Figure 1 (a) and Figure 1 (b), AdaSDBO achieves faster convergence than baseline methods across different data dimensions (i.e., $p = 50$ and $p = 200$). Notably, our method consistently outperforms the double-loop frameworks DBO and MA-DSBO. These results validate both the superiority of the adaptive stepsizes design and the single-loop structure of AdaSDBO.

## 4.2 Real-World Data Experiments

We evaluate our method on the hyperparameter optimization task using the MNIST [LeCun et al., 1998] and FMNIST [Xiao et al., 2017] datasets. In Figure 1(c) and Figure 1(d), we vary the number of agents to assess scalability. The results illustrate that AdaSDBO consistently maintains a robust convergence rate across different network sizes, highlighting its scalability and robustness to variations in the number of agents. Furthermore, AdaSDBO achieves a competitive convergence rate compared to state-of-the-art methods, further corroborating its effectiveness. Additional scalability evaluations under broader network configurations are presented in Figure 8 and Figure 9 of Section E.6.

Figure 2 compares the test accuracy of various algorithms versus stepsizes on the synthetic, MNIST, and FMNIST datasets. All algorithms were evaluated over 1,000 rounds to ensure a fair comparison. To comprehensively assess robustness, the stepsizes were varied over a wide range (i.e., from $10^{-3}$ to $10^2$). It can be observed that the AdaSDBO algorithm demonstrates remarkable resilience to stepsize selection, maintaining stable performance over a substantially broader range of stepsizes compared to baseline methods. In contrast, the baseline algorithms exhibit relatively narrower regions of stable performance, underscoring the enhanced stepsize robustness of our proposed parameter-free method.

## 4.3 Decentralized Meta-Learning

We evaluate our method on decentralized meta-learning using the CIFAR-10 dataset [McMahan et al., 2017], where multiple tasks are constructed following the protocol in [Finn et al., 2017]. This approach minimizes the test loss with respect to shared parameters as the upper-level loss, while the training loss is managed by task-specific parameters at the lower level. The detailed configuration of this experiment can be found in Section E.4. To highlight the effectiveness of our approach, we compare against the state-of-the-art SLDBO method [Dong et al., 2023]. As shown in Figure 10 of Section E.6, our method achieves notably better training accuracy. This improvement stems from its problem-parameter-free design, which allows the algorithm to automatically adapt stepsizes, consistently reaching optimal convergence rates without manual tuning.

# 5 Conclusions and Limitations

In this paper, we proposed AdaSDBO, a parameter-free algorithm for decentralized bilevel optimization with a single-loop framework, supported by a rigorous finite-time convergence analysis. AdaSDBO adaptively adjusted stepsizes without relying on prior knowledge of problem parameters, achieving a convergence rate comparable to well-tuned counterparts. Extensive experiments showed that AdaSDBO delivered strong generalization performance and eliminated the need for tedious hyperparameter tuning, showcasing its potential for large-scale machine learning applications. Nevertheless, our analysis targets deterministic settings with full-gradient information and assumes a strongly convex lower-level problem. Extending the results to stochastic regimes and to generally convex lower-level objectives remains an open direction. We aim to address these limitations in future work to broaden the scope of applicability of AdaSDBO.

## Acknowledgements

This work is supported in part by the General Research Fund (project number 14214122, 14202723, 14207624), Area of Excellence Scheme grant (project number AoE/E-601/22-R), and NSFC/RGC Collaborative Research Scheme (project number CRS_HKUST603/22, CRS_HKU702/24), all from the Research Grants Council of Hong Kong.

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

# A Additional Discussion on Related Works

**Bilevel Optimization.** Bilevel optimization originated with [Bracken and McGill, 1973] and has seen significant advances in both theory and algorithms. Early methodological approaches [Hansen et al., 1992, Shi et al., 2005] predominantly addressed these problems through constrained optimization formulations, treating the inner-level problem as a parametric constraint. Contemporary research has increasingly focused on efficient gradient-based methods, which fall into three main categories: 1) approximate implicit differentiation (AID) methods [Domke, 2012, Liao et al., 2018, Ji et al., 2021, Dagréou et al., 2022], which use the implicit function theorem to approximate hypergradients; 2) iterative differentiation (ITD) methods [Maclaurin et al., 2015, Franceschi et al., 2018, Grazzi et al., 2020], which leverage automatic differentiation; 3) Neumann series-based methods [Ji et al., 2021, Yang et al., 2021], which approximate the inverse Hessian using truncated series. However, implicit differentiation in bilevel optimization requires accurate inner problem solutions for each outer variable update, leading to high computational costs in large-scale problems. To address this, the researchers proposed solving the inner problem with a fixed number of steps and computing gradients with the "backpropagation through time" technique [Shaban et al., 2019, Franceschi et al., 2017]. Nevertheless, this approach remains computationally expensive for modern machine learning models with hundreds of millions of parameters. Recently, there has been a surge of interest in using implicit differentiation to derive single-loop algorithms. Ghadimi and Wang [2018] introduced an accelerated AID method with the Neumann series. Yang et al. [2021] proposed a warm-start strategy to reduce the number of inner steps required at each iteration. Additionally, Li et al. [2022a] introduced FSLA, a fully single-loop algorithm for bilevel optimization that eliminates the need for Hessian inversion.

**Adaptive Methods.** The introduction of AdaGrad [McMahan and Streeter, 2010, Duchi et al., 2011] marked a milestone in adaptive gradient-based methods. Originally designed for online convex optimization, AdaGrad quickly evolved into a foundation for deep learning algorithms, spawning numerous variants such as Adadelta [Zeiler, 2012], RMSprop [Tieleman and Hinton, 2017], and Adam [Luo et al., 2019, Xie et al., 2024]. In particular, AdaGrad variants with normalized gradients, including AdaNGD [Levy, 2017], AcceleGrad [Levy et al., 2018], and AdaGrad-Norm [Xie et al., 2020], introduced adaptive stepsizes that eliminate the need for problem-specific parameters, establishing themselves as effective parameter-free methods. More recent refinements, such as the Lipschitzness parameter approximation [Malitsky and Mishchenko, 2019] and the restart mechanisms [Marumo and Takeda, 2024], have further enhanced both performance and robustness. Additionally, Yang et al. [2023] provided foundational insights into mainstream adaptive methods, laying the groundwork for their applications in distributed optimization [Li et al., 2024, Yan et al., 2025].

**Adaptive Minimax and Bilevel Methods.** Currently, some research has begun addressing adaptive stepsize design specifically within the minimax optimization context [Li et al., 2022b, Huang et al., 2024a,b]. However, minimax problems inherently possess simpler structural properties compared to bilevel optimization problems. The nested structure inherent in bilevel optimization introduces additional complexities due to the coupling of variables between the upper and lower levels, significantly complicating the design of adaptive stepsizes. In centralized bilevel optimization, a few adaptive methods have been proposed. For instance, Antonakopoulos et al. [2025] proposed a double-loop adaptive algorithm utilizing mirror descent, which still relies on the unknown strong convexity parameter of the lower-level function. Similarly, Yang et al. [2025] introduced a centralized adaptive method based on AdaGrad-Norm, achieving convergence rates comparable to well-tuned methods. However, tuning hyperparameters in practice is often prohibitively expensive and becomes considerably more difficult in decentralized scenarios, thus making the ability to automatically adjust the update dynamics particularly crucial [Li et al., 2024]. Nevertheless, achieving such adaptability in decentralized bilevel optimization remains profoundly challenging. Specifically, decentralized bilevel optimization necessitates carefully orchestrated updates across primal, dual, and auxiliary variables to manage the nested structure effectively. Additionally, the hierarchical structure of decentralized bilevel optimization introduces multiple coupled adaptive stepsizes, significantly amplifying heterogeneity across agents. Without meticulous coordination, the resulting variability can degrade convergence performance. Moreover, the decentralized setting requires simultaneously managing network-induced communication errors and hierarchical bilevel approximation errors. To achieve convergence rates matching those of optimally tuned methods, adaptive decentralized bilevel algorithms require an even more precise theoretical analysis, as these error sources interact intricately. To the best of our knowledge, it remains an open and challenging question on how to leverage adaptive methods to design a completely problem-parameter-free algorithm for decentralized bilevel optimization.

## B Proof Sketch

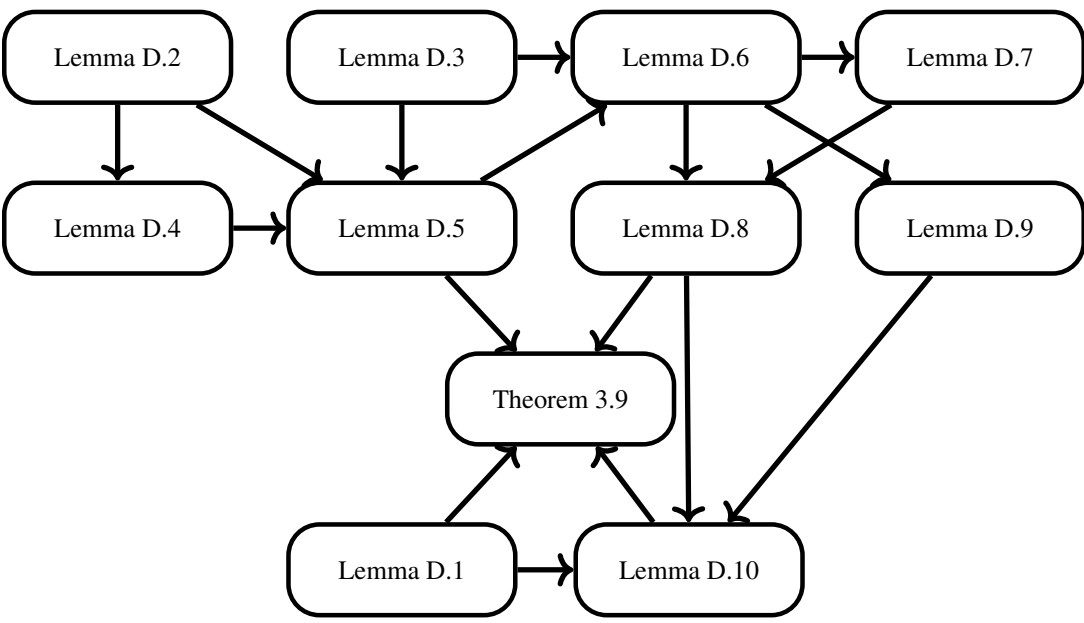

Figure 3: Structure of the proof

Figure 3 illustrates the structure of the proof. Next, we present the proof sketch for Theorem 3.9.

**Proof Sketch of Theorem 3.9:**

**Step 1:** We start by introducing the two-stage framework outlined in Lemma C.5 to examine the progression of the gradient accumulators $\bar{m}_t^x$, $\bar{m}_t^y$, and $\bar{m}_t^v$. This framework divides the iterations into two cases: when the gradient accumulators are below or exceed a predefined threshold. Using this structure, we derive two descent lemmas in Lemma D.1 for the objective function, corresponding to these two stages of $\bar{m}_t^x$.

**Step 2:** Next, we derive tail bounds for two key components in the descent Lemma D.1: $\sum_{k=k_2}^{t} \frac{\|\nabla_y L(\mathbf{x}_k, \mathbf{y}_k)\|^2}{\bar{u}_{k+1}}$ (in Lemma D.2) and $\sum_{k=k_3}^{t} \frac{\|\nabla_v R(\mathbf{x}_k, \mathbf{y}_k, \mathbf{v}_k)\|^2}{\bar{z}_{k+1}}$ (in Lemma D.3), where $k_2$ and $k_3$ represent the cutoff points corresponding to the second stage in the two-stage framework of Lemma C.5.

**Step 3:** Using the bounds derived in Step 2, we establish upper bounds for $\bar{m}_{t+1}^y$ and $\bar{z}_{t+1}$ in Lemma D.4 and Lemma D.5, respectively. Based on these results, we derive general bounds for $\sum_{k=k_0}^{t} \frac{\|\bar{\nabla} F(\mathbf{x}_k, \mathbf{y}_k, \mathbf{v}_k)\|^2}{[\bar{m}_{k+1}^x]^2}$, $\sum_{k=k_0}^{t} \frac{\|\nabla_y L(\mathbf{x}_k, \mathbf{y}_k)\|^2}{\bar{m}_{k+1}^y}$, and $\sum_{k=k_0}^{t} \frac{\|\nabla_v R(\mathbf{x}_k, \mathbf{y}_k, \mathbf{v}_k)\|^2}{\bar{z}_{k+1}}$ with $k_0 \in [0, t)$, as shown in Lemma D.6.

**Step 4:** We then analyze the consensus errors for the primal, dual, and auxiliary variables in Lemma D.7. By combining these results with Lemma D.6, we derive the upper bounds: $\sum_{k=k_0}^{t} \frac{\|\bar{\nabla} f(\bar{x}_k, \bar{y}_k, \bar{v}_k)\|^2}{[\bar{m}_{k+1}^x]^2}$, $\sum_{k=k_0}^{t} \frac{\|\nabla_y l(\bar{x}_k, \bar{y}_k)\|^2}{\bar{m}_{k+1}^y}$ and $\sum_{k=k_0}^{t} \frac{\|\nabla_v r(\bar{x}_k, \bar{y}_k, \bar{v}_k)\|^2}{\bar{z}_{k+1}}$ with respect to the consensus variables $\bar{x}_k$, $\bar{y}_k$, and $\bar{v}_k$ in Lemma D.8, where $k_0 \in [0, t)$.

**Step 5:** Additionally, in Lemma D.9, we derive an upper bound for the term associated with the stepsize-inconsistency errors in Lemma D.1. By substituting the results from Lemma D.8 and Lemma D.9 into the Descent Lemma D.1, we obtain the bound for $\bar{m}_t^x$, as presented in Lemma D.10.

**Step 6:** Finally, by combining the results from Steps 3, 4, and 5, we establish the convergence of Algorithm 1 based on the descent analysis in Lemma D.1.

# C  Proofs of Supporting Lemmas

**Lemma C.1** (Lemma 3.2 in [Ward et al., 2020]). *For any non-negative $a_1, \ldots, a_T$, and $a_1 \geq 1$, we have:*

$$\sum_{l=1}^{T} \frac{a_l}{\sum_{i=1}^{l} a_i} \leq \log\left(\sum_{l=1}^{T} a_l\right) + 1. \tag{7}$$

**Lemma C.2.** *Under Assumption 3.1 and Assumption 3.2, we have the following basic properties:*

1). *$\Phi(\bar{x})$ is $L_\Phi$-smooth with respect to $\bar{x}$, where $L_\Phi := \left(L_{f,1} + \frac{L_{l,2} C_{fy}}{\mu}\right)\left(1 + \frac{C_{l_{xy}}}{\mu}\right)^2$;*

2). *$y^*(\bar{x})$ is $L_y$-Lipschitz continuous with respect to $\bar{x}$, where $L_y := \frac{C_{l_{xy}}}{\mu}$;*

3). *The gradient estimator $\bar{\nabla} f(\bar{x}, \bar{y}, \bar{v})$ is $(L_{l,2}\|\bar{v}\| + L_{f,1})$-Lipschitz continuous with respect to $(\bar{x}, \bar{y})$ and $L_{l,1}$-Lipschitz continuous with respect to $\bar{v}$;*

4). *$\bar{\nabla} f(\bar{x}, \bar{y}, \bar{v})$ can be bounded as $\|\bar{\nabla} f(\bar{x}, \bar{y}, \bar{v})\| \leq C_{l_{xy}}\|\bar{v}\| + C_{f_x}$.*

*Proof.* The proofs of 1) and 2) can refer to [Ghadimi and Wang, 2018]. For 3), under Assumption 3.2, we have:

$$\begin{aligned}
&\|\bar{\nabla} f(\bar{x}_1, \bar{y}_1, \bar{v}) - \bar{\nabla} f(\bar{x}_2, \bar{y}_2, \bar{v})\| \\
&\leq \|\nabla_x \nabla_y l(\bar{x}_1, \bar{y}_1) - \nabla_x \nabla_y l(\bar{x}_2, \bar{y}_2)\| \cdot \|\bar{v}\| + \|\nabla_x f(\bar{x}_1, \bar{y}_1) - \nabla_x f(\bar{x}_2, \bar{y}_2)\| \\
&\leq (L_{l,2}\|\bar{v}\| + L_{f,1})(\|\bar{x}_1 - \bar{x}_2\| + \|\bar{y}_1 - \bar{y}_2\|),
\end{aligned} \tag{8}$$

and

$$\|\bar{\nabla} f(\bar{x}, \bar{y}, \bar{v}_1) - \bar{\nabla} f(\bar{x}, \bar{y}, \bar{v}_2)\| \leq \|\nabla_x \nabla_y l(\bar{x}, \bar{y})\| \cdot \|\bar{v}_1 - \bar{v}_2\| \leq L_{l,1}\|\bar{v}_1 - \bar{v}_2\|. \tag{9}$$

By Assumption 3.2, we can easily prove 4) as:

$$\|\bar{\nabla} f(\bar{x}, \bar{y}, \bar{v})\| \leq \|\nabla_x \nabla_y l(\bar{x}, \bar{y})\| \cdot \|\bar{v}\| + \|\nabla_x f(\bar{x}, \bar{y})\| \leq C_{l_{xy}}\|\bar{v}\| + C_{f_x}. \tag{10}$$

Then the proof is complete. $\square$

**Lemma C.3.** *Under Assumption 3.1 and Assumption 3.2, we have basic properties of the linear system function $r$ in Eq. (3) as follows:*

1). *$r(\bar{x}, \bar{y}, \bar{v})$ is $\mu$-strongly convex and $C_{l_{yy}}$-smooth with respect to $\bar{v}$;*

2). *$\nabla_v r(\bar{x}, \bar{y}, \bar{v})$ is $(L_{l,2}\|\bar{v}\| + L_{f,1})$-Lipschitz continuous with respect to $(\bar{x}, \bar{y})$;*

3). *$\nabla_v r(\bar{x}, \bar{y}, \bar{v})$ can be bounded as $\|\nabla_v r(\bar{x}, \bar{y}, \bar{v})\| \leq C_{l_{yy}}\|\bar{v}\| + C_{fy}$;*

4). *$v^*(\bar{x})$ in Eq. (3) can be bounded as $\|v^*(\bar{x})\| \leq \frac{C_{fy}}{\mu}$, and $v^*(\bar{x}, \bar{y}) := \arg\min_{\bar{v}} r(\bar{x}, \bar{y}, \bar{v})$ can also be bounded as $\|v^*(\bar{x}, \bar{y})\| \leq \frac{C_{fy}}{\mu}$;*

5). *$\bar{\nabla} f(\bar{x}, \bar{y}, \bar{v})$ is $\bar{L}_f$-Lipschitz continuous with respect to $(\bar{x}, \bar{y}, \bar{v})$, where $\bar{L}_f = \max\left\{\frac{C_{fy} L_{l,2}}{\mu} + L_{f,1}, L_{l,1}\right\}$;*

6). *$\nabla_v r(\bar{x}, \bar{y}, \bar{v})$ is $\bar{L}_r$-Lipschitz continuous with respect to $(\bar{x}, \bar{y}, \bar{v})$, where $\bar{L}_r = \max\left\{\frac{C_{fy} L_{l,2}}{\mu} + L_{f,1}, C_{l_{yy}}\right\}$;*

7). *$v^*(\bar{x})$ is $L_v$-Lipschitz continuous with respect to $\bar{x}$ and $v^*(\bar{x}, \bar{y})$ is $\bar{L}_v$-Lipschitz continuous with respect to $\bar{y}$, where $L_v := \left(\frac{L_{f,1}}{\mu} + \frac{C_{fy} L_{l,2}}{\mu^2}\right)(1 + L_y)$ and $\bar{L}_v := \frac{L_{f,1}}{\mu} + \frac{C_{fy} L_{l,2}}{\mu^2}$.*

*Proof.* First of all, since $\nabla_v \nabla_v r(\bar{x}, \bar{y}, \bar{v}) = \nabla_y \nabla_y l(\bar{x}, \bar{y})$, we know $\mu I \preceq \nabla_y \nabla_y l(\bar{x}, \bar{y})$. Thus, according to Assumption 3.1 and Assumption 3.2, we have:

$$\|\nabla_v \nabla_v r(\bar{x}, \bar{y}, \bar{v}_1) - \nabla_v \nabla_v r(\bar{x}, \bar{y}, \bar{v}_2)\| \leq \|\nabla_y \nabla_y l(\bar{x}, \bar{y})\|\|\bar{v}_1 - \bar{v}_2\| \leq C_{l_{yy}}\|\bar{v}_1 - \bar{v}_2\|. \tag{11}$$

Then 1) is proved.

Next, by using Lipschitz continuity in Assumption 3.2, we have:

$$
\begin{aligned}
&\|\nabla_v r(\bar{x}_1, \bar{y}_1, \bar{v}) - \nabla_v r(\bar{x}_2, \bar{y}_2, \bar{v})\| \\
&\leq \|\nabla_y \nabla_y l(\bar{x}_1, \bar{y}_1) - \nabla_y \nabla_y l(\bar{x}_2, \bar{y}_2)\|\|\bar{v}\| + \|\nabla_y f(\bar{x}_1, \bar{y}_1) - \nabla_y f(\bar{x}_2, \bar{y}_2)\| \\
&\leq (L_{l,2}\|\bar{v}\| + L_{f,1})(\|\bar{x}_1 - \bar{x}_2\| + \|\bar{y}_1 - \bar{y}_2\|).
\end{aligned}
\tag{12}
$$

Then 2) is proved.

By Assumption 3.2, we can easily prove 3) as:

$$
\|\nabla_v r(\bar{x}, \bar{y}, \bar{v})\| \leq \|\nabla_y \nabla_y l(\bar{x}, \bar{y})\|\|\bar{v}\| + \|\nabla_y f(\bar{x}, \bar{y})\| \leq C_{l_{yy}}\|\bar{v}\| + C_{f_y}.
\tag{13}
$$

Next, for $v^*(\bar{x}, \bar{y})$, we have:

$$
\nabla_v r(\bar{x}, \bar{y}, \hat{v}^*(\bar{x}, \bar{y})) = \nabla_y \nabla_y l(\bar{x}, \bar{y})\hat{v}^*(\bar{x}, \bar{y}) - \nabla_y f(\bar{x}, \bar{y}) = 0,
\tag{14}
$$

which indicates that

$$
\|\hat{v}^*(\bar{x}, \bar{y})\| = \|\left(\nabla_y \nabla_y l(\bar{x}, \bar{y})\right)^{-1} \nabla_y f(\bar{x}, \bar{y})\| \leq \|\left(\nabla_y \nabla_y l(\bar{x}, \bar{y})\right)^{-1}\|\|\nabla_y f(\bar{x}, \bar{y})\| \leq \frac{C_{f_y}}{\mu}.
\tag{15}
$$

Since $v^*(\bar{x})$ is a special case where $v^*(\bar{x}) = \hat{v}^*(\bar{x}, y^*(\bar{x}))$, 4) is proved.

By Lemma C.2, we have:

$$
\begin{aligned}
&\|\bar{\nabla}f(\bar{x}_1, \bar{y}_1, \bar{v}_1) - \bar{\nabla}f(\bar{x}_2, \bar{y}_2, \bar{v}_2)\| \\
&\leq \|\bar{\nabla}f(\bar{x}_1, \bar{y}_1, \bar{v}_1) - \bar{\nabla}f(\bar{x}_2, \bar{y}_2, \bar{v}_1)\| + \|\bar{\nabla}f(\bar{x}_2, \bar{y}_2, \bar{v}_1) - \bar{\nabla}f(\bar{x}_2, \bar{y}_2, \bar{v}_2)\| \\
&\leq (L_{l,2}\|\bar{v}_1\| + L_{f,1})(\|\bar{x}_1 - \bar{x}_2\| + \|\bar{y}_1 - \bar{y}_2\|) + L_{l,1}\|\bar{v}_1 - \bar{v}_2\| \\
&\leq \bar{L}_f(\|\bar{x}_1 - \bar{x}_2\| + \|\bar{y}_1 - \bar{y}_2\| + \|\bar{v}_1 - \bar{v}_2\|),
\end{aligned}
\tag{16}
$$

where $\bar{L}_f = \max\{\frac{C_{f_y}L_{l,2}}{\mu} + L_{f,1}, L_{l,1}\}$. Thus, 5) is proved.

For the proof of 6), we have:

$$
\begin{aligned}
&\|\nabla_v r(\bar{x}_1, \bar{y}_1, \bar{v}_1) - \nabla_v r(\bar{x}_2, \bar{y}_2, \bar{v}_2)\| \\
&= \|\nabla_y \nabla_y l(\bar{x}_1, \bar{y}_1)\bar{v}_1 - \nabla_y f(\bar{x}_1, \bar{y}_1) - (\nabla_y \nabla_y l(\bar{x}_2, \bar{y}_2)\bar{v}_2 - \nabla_y f(\bar{x}_2, \bar{y}_2))\| \\
&= \|(\nabla_y \nabla_y l(\bar{x}_1, \bar{y}_1) - \nabla_y \nabla_y l(\bar{x}_2, \bar{y}_2))\bar{v}_1 + \nabla_y \nabla_y l(\bar{x}_2, \bar{y}_2)(\bar{v}_1 - \bar{v}_2) - (\nabla_y f(\bar{x}_1, \bar{y}_1) - \nabla_y f(\bar{x}_2, \bar{y}_2))\| \\
&\leq \|\nabla_y \nabla_y l(\bar{x}_1, \bar{y}_1) - \nabla_y \nabla_y l(\bar{x}_2, \bar{y}_2)\|\|\bar{v}_1\| + \|\nabla_y \nabla_y l(\bar{x}_2, \bar{y}_2)\|\|\bar{v}_1 - \bar{v}_2\| \\
&\quad + \|\nabla_y f(\bar{x}_1, \bar{y}_1) - \nabla_y f(\bar{x}_2, \bar{y}_2)\| \\
&\overset{(a)}{\leq} L_{l,2}(\|\bar{x}_1 - \bar{x}_2\| + \|\bar{y}_1 - \bar{y}_2\|)\|\bar{v}_1\| + C_{l_{yy}}\|\bar{v}_1 - \bar{v}_2\| + L_{f,1}(\|\bar{x}_1 - \bar{x}_2\| + \|\bar{y}_1 - \bar{y}_2\|) \\
&\overset{(b)}{\leq} \left(\frac{C_{f_y}L_{l,2}}{\mu} + L_{f,1}\right)(\|\bar{x}_1 - \bar{x}_2\| + \|\bar{y}_1 - \bar{y}_2\|) + C_{l_{yy}}\|\bar{v}_1 - \bar{v}_2\| \\
&\leq \bar{L}_r(\|\bar{x}_1 - \bar{x}_2\| + \|\bar{y}_1 - \bar{y}_2\| + \|\bar{v}_1 - \bar{v}_2\|),
\end{aligned}
\tag{17}
$$

where (a) uses Assumption 3.2, (b) uses Eq. (15) and $\bar{L}_r = \max\{\frac{C_{f_y}L_{l,2}}{\mu} + L_{f,1}, C_{l_{yy}}\}$. Then the proof of 6) is finished.

The proof of the first part of 7) can refer to Lemma 4 in [Yang et al., 2024]; for the second part, we have:

$$
\begin{aligned}
&\|\hat{v}^*(\bar{x}, \bar{y}_1) - \hat{v}^*(\bar{x}, \bar{y}_2)\| \\
&= \|[\nabla_y \nabla_y l(\bar{x}, \bar{y}_1)]^{-1} \nabla_y f(\bar{x}, \bar{y}_1) - [\nabla_y \nabla_y l(\bar{x}, \bar{y}_2)]^{-1} \nabla_y f(\bar{x}, \bar{y}_2)\| \\
&\leq \|[\nabla_y \nabla_y l(\bar{x}, \bar{y}_1)]^{-1} (\nabla_y f(\bar{x}, \bar{y}_1) - \nabla_y f(\bar{x}, \bar{y}_2))\| \\
&\quad + \|([\nabla_y \nabla_y l(\bar{x}, \bar{y}_1)]^{-1} - [\nabla_y \nabla_y l(\bar{x}, \bar{y}_2)]^{-1})\nabla_y f(\bar{x}, \bar{y}_2)\| \\
&\leq \frac{L_{f,1}}{\mu}\|\bar{y}_1 - \bar{y}_2\| + C_{f_y}\|[\nabla_y \nabla_y l(\bar{x}, \bar{y}_1)]^{-1} (\nabla_y \nabla_y l(\bar{x}, \bar{y}_2) - \nabla_y \nabla_y l(\bar{x}, \bar{y}_1)) [\nabla_y \nabla_y l(\bar{x}, \bar{y}_2)]^{-1}\| \\
&\leq \left(\frac{L_{f,1}}{\mu} + \frac{C_{f_y}L_{l,2}}{\mu^2}\right)\|\bar{y}_1 - \bar{y}_2\|.
\end{aligned}
\tag{18}
$$

Thus, the second part of 7) is proved, and the proof of Lemma C.3 is complete. $\qquad\square$

**Lemma C.4 (The Upper Bound of $\sum_{k=0}^{t}\|\mathbf{x}_k - \mathbf{1}\bar{x}_k\|^2$).** *Suppose Assumption 2.1, Assumption 3.1, and Assumption 3.2 hold. Then, the consensus error for the primal variable satisfies:*

$$\sum_{k=0}^{t}\|\mathbf{x}_k - \mathbf{1}\bar{x}_k\|^2 \leq \frac{2\Delta_x^0}{1-\rho_W} + \frac{8\gamma_x^2\rho_W(1+\zeta_q^2)}{(1-\rho_W)^2}\sum_{k=0}^{t}\bar{q}_{k+1}^{-2}\|\bar{\nabla}F(\mathbf{x}_k,\mathbf{y}_k,\mathbf{v}_k)\|^2, \qquad (19)$$

*where $\Delta_0^x = \|\mathbf{x}_0 - \mathbf{1}\bar{x}_0\|^2$ is the initial consensus error for the primary variable $x$, which can be set to 0 with proper initialization.*

*Proof.* By the updating rule of the primal variable, we have:

$$\|\mathbf{x}_{t+1} - \mathbf{1}\bar{x}_{t+1}\|^2$$
$$= \left\|W\left(\mathbf{x}_t - \gamma_x Q_{t+1}^{-1}\bar{\nabla}F(\mathbf{x}_t,\mathbf{y}_t,\mathbf{v}_t)\right) - \mathbf{J}\left(\mathbf{x}_t - \gamma_x Q_{t+1}^{-1}\bar{\nabla}F(\mathbf{x}_t,\mathbf{y}_t,\mathbf{v}_t)\right)\right\|^2$$
$$\overset{(a)}{\leq} (1+\lambda)\rho_W\|\mathbf{x}_t - \mathbf{1}\bar{x}_t\|^2 + 2\left(1+\frac{1}{\lambda}\right)\rho_W\gamma_x^2\bar{q}_{t+1}^{-2}\|\bar{\nabla}F(\mathbf{x}_t,\mathbf{y}_t,\mathbf{v}_t)\|^2$$
$$+ 2\left(1+\frac{1}{\lambda}\right)\rho_W\gamma_x^2\left\|\left(Q_{t+1}^{-1} - \bar{q}_{t+1}^{-1}\mathbf{I}\right)\bar{\nabla}F(\mathbf{x}_t,\mathbf{y}_t,\mathbf{v}_t)\right\|^2$$
$$\overset{(b)}{\leq} \frac{1+\rho_W}{2}\|\mathbf{x}_t - \mathbf{1}\bar{x}_t\|^2 + \frac{2\gamma_x^2(1+\rho_W)\rho_W}{1-\rho_W}\bar{q}_{t+1}^{-2}\|\bar{\nabla}F(\mathbf{x}_t,\mathbf{y}_t,\mathbf{v}_t)\|^2$$
$$+ \frac{2\gamma_x^2(1+\rho_W)\rho_W}{1-\rho_W}\left\|\left(Q_{t+1}^{-1} - \bar{q}_{t+1}^{-1}\mathbf{I}\right)\bar{\nabla}F(\mathbf{x}_t,\mathbf{y}_t,\mathbf{v}_t)\right\|^2, \qquad (20)$$

where (a) uses Young's inequality and we take $\lambda = \frac{1-\rho_W}{2\rho_W}$ in (b).

By the definition of $\zeta_q^2$ in Section 3.2, we have:

$$\left\|\left(Q_{t+1}^{-1} - \bar{q}_{t+1}^{-1}\mathbf{I}\right)\bar{\nabla}F(\mathbf{x}_t,\mathbf{y}_t,\mathbf{v}_t)\right\|^2 \leq \zeta_q^2\bar{q}_{t+1}^{-2}\|\bar{\nabla}F(\mathbf{x}_t,\mathbf{y}_t,\mathbf{v}_t)\|^2. \qquad (21)$$

Thus, summing over $k = 0, \ldots, t-1$, we have:

$$\sum_{k=0}^{t-1}\|\mathbf{x}_{k+1} - \mathbf{1}\bar{x}_{k+1}\|^2$$
$$\leq \sum_{k=0}^{t-1}\left(\frac{1+\rho_W}{2}\right)^k\|\mathbf{x}_0 - \mathbf{1}\bar{x}_0\|^2 + \frac{4\gamma_x^2\rho_W(1+\zeta_q^2)}{1-\rho_W}\sum_{k=0}^{t-1}\left(\frac{1+\rho_W}{2}\right)^k\bar{q}_{k+1}^{-2}\|\bar{\nabla}F(\mathbf{x}_k,\mathbf{y}_k,\mathbf{v}_k)\|^2$$
$$\leq \frac{2}{1-\rho_W}\|\mathbf{x}_0 - \mathbf{1}\bar{x}_0\|^2 + \frac{8\gamma_x^2\rho_W(1+\zeta_q^2)}{(1-\rho_W)^2}\sum_{k=0}^{t-1}\bar{q}_{k+1}^{-2}\|\bar{\nabla}F(\mathbf{x}_k,\mathbf{y}_k,\mathbf{v}_k)\|^2. \qquad (22)$$

Thus, we can get the result in Eq. (19). $\qquad\square$

**Lemma C.5.** *Let variables $x$, $y$, and $v$ be updated over $T_1$, $T_2$, and $T_3$ iterations, respectively. Suppose that the sequences $\{\bar{m}_t^x\}$, $\{\bar{m}_t^y\}$, and $\{\bar{m}_t^v\}$ are generated by Algorithm 1. For any constants $C_{m^x} \geq \bar{m}_0^x$, $C_{m^y} \geq \bar{m}_0^y$, and $C_{m^v} \geq \bar{m}_0^v$, the following statements hold:*

*(1) Either $\bar{m}_t^x \leq C_{m^x}$ for any $t \leq T_1$, or $\exists k_1 \leq T_1$ such that $\bar{m}_{k_1}^x \leq C_{m^x}$, $\bar{m}_{k_1+1}^x > C_{m^x}$.*

*(2) Either $\bar{m}_t^y \leq C_{m^y}$ for any $t \leq T_2$, or $\exists k_2 \leq T_2$ such that $\bar{m}_{k_2}^y \leq C_{m^y}$, $\bar{m}_{k_2+1}^y > C_{m^y}$.*

*(3) Either $\bar{m}_t^v \leq C_{m^v}$ for any $t \leq T_3$, or $\exists k_3 \leq T_3$ such that $\bar{m}_{k_3}^v \leq C_{m^v}$, $\bar{m}_{k_3+1}^v > C_{m^v}$.*

*Proof.* The proof is analogous to that of Lemma 4.1 in [Ward et al., 2020]. We will demonstrate the argument for part (1) concerning $\bar{m}_t^x$; the proofs for parts (2) and (3) follow similarly.

Assume that $\bar{m}_{T_1}^x > C_{m^x}$. Since $C_{m^x} \geq \bar{m}_0^x$ and the sequence $\{\bar{m}_t^x\}$ is monotonically increasing, there must exist an iteration $k_1 \leq T_1$ where $\bar{m}_{k_1}^x \leq C_{m^x}$ and $\bar{m}_{k_1+1}^x > C_{m^x}$. If no such $k_1$ exists, then it must be that $\bar{m}_t^x \leq C_{m^x}$ for all $t \leq T_1$. This completes the proof for part (1). $\qquad\square$

# D   Proofs of Theorem 3.9 and Corollary 3.11

## D.1   Notation

We stack the gradient accumulators as:

$$\mathbf{m}_t^x := [m_{1,t}^x, m_{2,t}^x, \ldots, m_{n,t}^x]^\top \in \mathbb{R}^{n \times p}, \quad \mathbf{m}_t^y := [m_{1,t}^y, m_{2,t}^y, \ldots, m_{n,t}^y]^\top \in \mathbb{R}^{n \times q},$$

$$\mathbf{m}_t^v := [m_{1,t}^v, m_{2,t}^v, \ldots, m_{n,t}^v]^\top \in \mathbb{R}^{n \times q}. \tag{23}$$

Similarly, the gradient vectors are stacked as:

$$\bar{\nabla} F(\mathbf{x}_t, \mathbf{y}_t, \mathbf{v}_t) := \left[ \bar{\nabla} f_1(x_{1,t}, y_{1,t}, v_{1,t}), \bar{\nabla} f_2(x_{2,t}, y_{2,t}, v_{2,t}), \cdots, \bar{\nabla} f_n(x_{n,t}, y_{n,t}, v_{n,t}) \right]^\top,$$

$$\nabla_y L(\mathbf{x}_t, \mathbf{y}_t) := \left[ \nabla_y l_1(x_{1,t}, y_{1,t}), \nabla_y l_2(x_{2,t}, y_{2,t}), \cdots, \nabla_y l_n(x_{n,t}, y_{n,t}) \right]^\top,$$

$$\nabla_v R(\mathbf{x}_t, \mathbf{y}_t, \mathbf{v}_t) := \left[ \nabla_v r_1(x_{1,t}, y_{1,t}, v_{1,t}), \nabla_v r_2(x_{2,t}, y_{2,t}, v_{2,t}), \cdots, \nabla_v r_n(x_{n,t}, y_{n,t}, v_{n,t}) \right]^\top,$$

$$\nabla_y F(\mathbf{x}_t, \mathbf{y}_t) := \left[ \nabla_y f_1(x_{1,t}, y_{1,t}), \nabla_y f_2(x_{2,t}, y_{2,t}), \cdots, \nabla_y f_n(x_{n,t}, y_{n,t}) \right]^\top. \tag{24}$$

For notational simplicity, we also use $\mathbf{g}_t^x$, $\mathbf{g}_t^y$, and $\mathbf{g}_t^v$ to denote $\bar{\nabla} F(\mathbf{x}_t, \mathbf{y}_t, \mathbf{v}_t)$, $\nabla_y L(\mathbf{x}_t, \mathbf{y}_t)$, and $\nabla_v R(\mathbf{x}_t, \mathbf{y}_t, \mathbf{v}_t)$, respectively.

The stepsize discrepancy vectors can be defined as:

$$\tilde{\mathbf{q}}_t^{-1} := \left[ q_{1,t}^{-1} - \bar{q}_t^{-1}, q_{2,t}^{-1} - \bar{q}_t^{-1}, \cdots, q_{n,t}^{-1} - \bar{q}_t^{-1} \right]^\top,$$

$$\tilde{\mathbf{u}}_t^{-1} := \left[ u_{1,t}^{-1} - \bar{u}_t^{-1}, u_{2,t}^{-1} - \bar{u}_t^{-1}, \cdots, u_{n,t}^{-1} - \bar{u}_t^{-1} \right]^\top, \tag{25}$$

$$\tilde{\mathbf{z}}_t^{-1} := \left[ z_{1,t}^{-1} - \bar{z}_t^{-1}, z_{2,t}^{-1} - \bar{z}_t^{-1}, \cdots, z_{n,t}^{-1} - \bar{z}_t^{-1} \right]^\top.$$

Additionally, for ease of presentation, we define the following notations:

$$\begin{cases} \bar{x}_t := \frac{1}{n} \sum_{i=1}^n x_{i,t}, & \bar{y}_t := \frac{1}{n} \sum_{i=1}^n y_{i,t}, & \bar{v}_t := \frac{1}{n} \sum_{i=1}^n v_{i,t}, \\ \bar{q}_t := \frac{1}{n} \sum_{i=1}^n q_{i,t}, & \bar{u}_t := \frac{1}{n} \sum_{i=1}^n u_{i,t}, & \bar{z}_t := \frac{1}{n} \sum_{i=1}^n z_{i,t}. \end{cases} \tag{26}$$

We define the following metrics to represent the level of stepsize inconsistency for the primal, dual, and auxiliary variables:

$$\zeta_q^2 := \sup_{i \in [n], t>0} \frac{\left( q_{i,t}^{-1} - \bar{q}_t^{-1} \right)^2}{\left( \bar{q}_t^{-1} \right)^2}, \quad \zeta_u^2 := \sup_{i \in [n], t>0} \frac{\left( u_{i,t}^{-1} - \bar{u}_t^{-1} \right)^2}{\left( \bar{u}_t^{-1} \right)^2}, \quad \zeta_z^2 := \sup_{i \in [n], t>0} \frac{\left( z_{i,t}^{-1} - \bar{z}_t^{-1} \right)^2}{\left( \bar{z}_t^{-1} \right)^2},$$

$$\sigma_q^2 := \inf_{i \in [n], t>0} \frac{\left( q_{i,t}^{-1} - \bar{q}_t^{-1} \right)^2}{\left( \bar{q}_t^{-1} \right)^2}, \quad \sigma_u^2 := \inf_{i \in [n], t>0} \frac{\left( u_{i,t}^{-1} - \bar{u}_t^{-1} \right)^2}{\left( \bar{u}_t^{-1} \right)^2}, \quad \sigma_z^2 := \inf_{i \in [n], t>0} \frac{\left( z_{i,t}^{-1} - \bar{z}_t^{-1} \right)^2}{\left( \bar{z}_t^{-1} \right)^2}. \tag{27}$$

Below, we define several preset constants for notational convenience at their first use. We first define some Lipschitzness parameters for $\Phi(\bar{x})$ as:

$$L_\Phi := \left( L_{f,1} + \frac{L_{l,2} C_{f_y}}{\mu} \right) \left( 1 + \frac{C_{l_{xy}}}{\mu} \right)^2, \tag{28}$$

$$\bar{L} := \max \left\{ 2 \left( \frac{C_{f_y}^2 L_{l,2}^2}{\mu^2} + L_{f,1}^2 \right)^{\frac{1}{2}}, \sqrt{2} C_{l_{xy}} \right\}. \tag{29}$$

Next, we define the following constants as thresholds for parameters $\bar{m}_t^x$, $\bar{m}_t^y$, and $\bar{m}_t^v$ as:

$$C_{m^x} := \max \left\{ \frac{8 n \gamma_x L_\Phi \left( 1 + \zeta_q^2 \right)}{\bar{z}_0}, \bar{m}_0^x \right\}, \tag{30}$$

$$C_{m^y} := \max \left\{ \frac{2 \gamma_y (1 + \zeta_u^2)(\mu + L_{l,1})}{\sqrt{1 + \sigma_u^2}}, \frac{\gamma_y \mu L_{l,1}}{\mu + L_{l,1}}, \bar{m}_0^y, 64 a_0^2, 1 \right\}, \tag{31}$$

$$C_{m^v} := \max\left\{ \frac{2\gamma_v(1+\zeta_z^2)(\mu+C_{l_{yy}})}{\sqrt{1+\sigma_z^2}}, \frac{2\mu C_{l_{yy}}\gamma_v}{\mu+C_{l_{yy}}}, \bar{m}_0^v, 64a_0^2, 1, C_{l_{yy}} \right\}, \tag{32}$$

$$C_z := C_{m^y} + C_{m^v}. \tag{33}$$

The constant $a_0$ is defined as:

$$
\begin{aligned}
a_0 := & \left( \frac{1}{\mu^2}\left(8 + \frac{4(\mu+C_{l_{yy}})^2}{\mu C_{l_{yy}}}\right)\left(\frac{nL_{l,2}C_{f_y}}{\mu}+L_{f,1}\right)^2 + 1 \right) \\
& \cdot \left( \frac{32\gamma_x^2\rho_W(1+\zeta_q^2)}{n(1-\rho_W)^2}\left(\frac{(\mu+L_{l,1})^3}{\gamma_y\mu^2} + \frac{2L_{l,1}(\mu+L_{l,1})}{\gamma_y\mu}\right) + \frac{8\gamma_x^2L_y^2\left(1+\zeta_q^2\right)(\mu+L_{l,1})^2}{n\gamma_y^2\mu L_{l,1}\sqrt{1+\sigma_u^2}} \right) \\
& + \frac{16(\mu+C_{l_{yy}})}{n\gamma_v\mu^2\sqrt{1+\sigma_u^2}}\left(\frac{nL_{l,2}C_{f_y}}{\mu}+L_{f,1}\right)^2 \\
& \cdot \left( \frac{4\gamma_x^2\rho_W(1+\zeta_q^2)}{\bar{z}_0^2(1-\rho_W)^2}\left(\frac{(\mu+L_{l,1})^2\sqrt{1+\sigma_u^2}}{n\mu^2} + \frac{2L_{l,1}\sqrt{1+\sigma_u^2}}{n\mu}\right) \right. \\
& \left. + \frac{\gamma_x^2L_y^2(1+\zeta_q^2)(\mu+L_{l,1})}{n\bar{z}_0\gamma_y\mu L_{l,1}} \right) + \frac{64\gamma_x^2\rho_W\bar{L}_r^2(\mu+C_{l_{yy}})(1+\zeta_q^2)}{n\mu C_{l_{yy}}\gamma_v(1-\rho_W)^2} + \frac{2\gamma_x^2L_v^2(1+\zeta_q^2)(\mu+C_{l_{yy}})^2}{n\mu C_{l_{yy}}\bar{m}_0^v\gamma_v^2\sqrt{1+\sigma_z^2}}. 
\end{aligned} \tag{34}
$$

### D.2 Descent in Objective Function

**Lemma D.1 (Descent Lemma).** *Suppose Assumption 2.1, Assumption 3.1, and Assumption 3.2 hold. Then, no matter $k_1$ in Lemma C.5 exists or not, we always have:*

$$
\begin{aligned}
& \Phi(\bar{x}_{t+1}) \\
& \leq \Phi(\bar{x}_t) - \frac{\gamma_x\bar{q}_{t+1}^{-1}}{8}\|\nabla\Phi(\bar{x}_t)\|^2 + \frac{\gamma_x\bar{q}_{t+1}^{-1}(2\gamma_x\bar{L}_f^2L_\Phi\bar{q}_{t+1}^{-1}\left(1+\zeta_q^2\right)+\bar{L}_f^2)}{n}\Delta_t \\
& \quad - \left(\frac{\gamma_x}{2} - 2n\gamma_x^2L_\Phi\bar{q}_{t+1}^{-1}\left(1+\zeta_q^2\right)\right)\frac{\|\nabla_x f(\bar{x}_t,\bar{y}_t,\bar{v}_t)\|^2}{\bar{q}_{t+1}} + \frac{\bar{L}^2\gamma_x}{\mu^2}\frac{\|\nabla_v r(\bar{x}_t,\bar{y}_t,\bar{v}_t)\|^2}{\bar{q}_{t+1}} \\
& \quad + \left(\frac{\gamma_x\bar{L}^2}{2\mu^2} + \frac{\gamma_x\bar{L}^2}{\mu^4}\left(\frac{L_{l,2}C_{f_y}}{\mu}+L_{f,1}\right)^2\right)\frac{\|\nabla_y l(\bar{x}_t,\bar{y}_t)\|^2}{\bar{q}_{t+1}} + 2\gamma_x\bar{q}_{t+1}^{-1}\left\|\frac{(\tilde{\mathbf{q}}_{t+1}^{-1})^\top}{n\bar{q}_{t+1}^{-1}}\bar{\nabla}F(\mathbf{x}_t,\mathbf{y}_t,\mathbf{v}_t)\right\|^2.
\end{aligned} \tag{35}
$$

*Additionally, if $k_1$ in Lemma C.5 exists, we have:*

$$
\begin{aligned}
& \Phi(\bar{x}_{t+1}) \\
& \leq \Phi(\bar{x}_t) - \frac{\gamma_x\bar{q}_{t+1}^{-1}}{8}\|\nabla\Phi(\bar{x}_t)\|^2 + \frac{\gamma_x\bar{q}_{t+1}^{-1}(2\gamma_x\bar{L}_f^2L_\Phi\bar{q}_{t+1}^{-1}\left(1+\zeta_q^2\right)+\bar{L}_f^2)}{n}\Delta_t \\
& \quad - \frac{\gamma_x}{4}\frac{\|\nabla_x f(\bar{x}_t,\bar{y}_t,\bar{v}_t)\|^2}{\bar{q}_{t+1}} + \frac{\bar{L}^2\gamma_x}{\mu^2}\frac{\|\nabla_v r(\bar{x}_t,\bar{y}_t,\bar{v}_t)\|^2}{\bar{q}_{t+1}} + 2\gamma_x\bar{q}_{t+1}^{-1}\left\|\frac{(\tilde{\mathbf{q}}_{t+1}^{-1})^\top}{n\bar{q}_{t+1}^{-1}}\bar{\nabla}F(\mathbf{x}_t,\mathbf{y}_t,\mathbf{v}_t)\right\|^2 \\
& \quad + \left(\frac{\gamma_x\bar{L}^2}{2\mu^2} + \frac{\gamma_x\bar{L}^2}{\mu^4}\left(\frac{L_{l,2}C_{f_y}}{\mu}+L_{f,1}\right)^2\right)\frac{\|\nabla_y l(\bar{x}_t,\bar{y}_t)\|^2}{\bar{q}_{t+1}},
\end{aligned} \tag{36}
$$

*where* $\bar{L} := \max\left\{ 2\left(\frac{C_{f_y}^2L_{l,2}^2}{\mu^2} + L_{f,1}^2\right)^{\frac{1}{2}}, \sqrt{2}C_{l_{xy}} \right\}.$

*Proof.* By the smoothness of $\Phi$, we have:

$$\Phi(\bar{x}_{t+1}) - \Phi(\bar{x}_t) \leq \langle\nabla\Phi(\bar{x}_t), \bar{x}_{t+1} - \bar{x}_t\rangle + \frac{L_\Phi}{2}\|\bar{x}_{t+1} - \bar{x}_t\|^2. \tag{37}$$

Noticing that the scalars $\bar{q}_t$, $\bar{u}_t$, and $\bar{z}_t$ are random variables, we then have:

$$\frac{\Phi(\bar{x}_{t+1}) - \Phi(\bar{x}_t)}{\gamma_x \bar{q}_{t+1}^{-1}} \leq -\left\langle \nabla\Phi(\bar{x}_t), \frac{\mathbf{1}^\top}{n}\bar{\nabla}F(\mathbf{x}_t, \mathbf{y}_t, \mathbf{v}_t)\right\rangle - \left\langle \nabla\Phi(\bar{x}_t), \frac{(\tilde{\mathbf{q}}_{t+1}^{-1})^\top}{n\bar{q}_{t+1}^{-1}}\bar{\nabla}F(\mathbf{x}_t, \mathbf{y}_t, \mathbf{v}_t)\right\rangle$$

$$+ \frac{\gamma_x L_\Phi}{2\bar{q}_{t+1}^{-1}}\left\|\left(\frac{\bar{q}_{t+1}^{-1}\mathbf{1}^\top}{n} + \frac{(\tilde{\mathbf{q}}_{t+1}^{-1})^\top}{n}\right)\bar{\nabla}F(\mathbf{x}_t, \mathbf{y}_t, \mathbf{v}_t)\right\|^2. \tag{38}$$

Then, we bound the inner-product terms on the right-hand side (RHS). Firstly,

$$-\left\langle \nabla\Phi(\bar{x}_t), \frac{\mathbf{1}^\top}{n}\bar{\nabla}F(\mathbf{x}_t, \mathbf{y}_t, \mathbf{v}_t)\right\rangle$$

$$= -\left\langle \nabla\Phi(\bar{x}_t), \frac{\mathbf{1}^\top}{n}\bar{\nabla}F(\mathbf{x}_t, \mathbf{y}_t, \mathbf{v}_t) - \frac{\mathbf{1}^\top}{n}\bar{\nabla}F(\mathbf{1}\bar{x}_t, \mathbf{1}\bar{y}_t, \mathbf{1}\bar{v}_t) + \frac{\mathbf{1}^\top}{n}\bar{\nabla}F(\mathbf{1}\bar{x}_t, \mathbf{1}\bar{y}_t, \mathbf{1}\bar{v}_t)\right\rangle$$

$$\leq \frac{1}{4}\|\nabla\Phi(\bar{x}_t)\|^2 + \left\|\frac{\mathbf{1}^\top}{n}\bar{\nabla}F(\mathbf{x}_t, \mathbf{y}_t, \mathbf{v}_t) - \frac{\mathbf{1}^\top}{n}\bar{\nabla}F(\mathbf{1}\bar{x}_t, \mathbf{1}\bar{y}_t, \mathbf{1}\bar{v}_t)\right\|^2$$

$$+ \frac{1}{2}\left(\|\nabla\Phi(\bar{x}_t) - \bar{\nabla}f(\bar{x}_t, \bar{y}_t, \bar{v}_t)\|^2 - \|\nabla\Phi(\bar{x}_t)\|^2 - \|\bar{\nabla}f(\bar{x}_t, \bar{y}_t, \bar{v}_t)\|^2\right)$$

$$\leq -\frac{1}{4}\|\nabla\Phi(\bar{x}_t)\|^2 + \left\|\frac{\mathbf{1}^\top}{n}\bar{\nabla}F(\mathbf{x}_t, \mathbf{y}_t, \mathbf{v}_t) - \frac{\mathbf{1}^\top}{n}\bar{\nabla}F(\mathbf{1}\bar{x}_t, \mathbf{1}\bar{y}_t, \mathbf{1}\bar{v}_t)\right\|^2$$

$$+ \frac{1}{2}\|\nabla\Phi(\bar{x}_t) - \bar{\nabla}f(\bar{x}_t, \bar{y}_t, \bar{v}_t)\|^2 - \frac{1}{2}\|\nabla_x f(\bar{x}_t, \bar{y}_t, \bar{v}_t)\|^2. \tag{39}$$

Additionally, the gradient approximation error satisfies:

$$\|\nabla\Phi(\bar{x}_t) - \bar{\nabla}f(\bar{x}_t, \bar{y}_t, \bar{v}_t)\|^2$$

$$= \|\bar{\nabla}f(\bar{x}_t, y^*(\bar{x}_t), v^*(\bar{x}_t)) - \bar{\nabla}f(\bar{x}_t, \bar{y}_t, \bar{v}_t)\|^2$$

$$\leq 2\|\bar{\nabla}f(\bar{x}_t, y^*(\bar{x}_t), v^*(\bar{x}_t)) - \bar{\nabla}f(\bar{x}_t, \bar{y}_t, v^*(\bar{x}_t))\|^2 + 2\|\bar{\nabla}f(\bar{x}_t, \bar{y}_t, v^*(\bar{x}_t)) - \bar{\nabla}f(\bar{x}_t, \bar{y}_t, \bar{v}_t)\|^2$$

$$\leq 4\|\nabla_x\nabla_y l(\bar{x}_t, y^*(\bar{x}_t))v^*(\bar{x}_t) - \nabla_x\nabla_y l(\bar{x}_t, \bar{y}_t)v^*(\bar{x}_t)\|^2$$

$$+ 4\|\nabla_x f(\bar{x}_t, y^*(\bar{x}_t)) - \nabla_x f(\bar{x}_t, \bar{y}_t)\|^2 + 2\|\nabla_x\nabla_y l(\bar{x}_t, \bar{y}_t)(v^*(\bar{x}_t) - \bar{v}_t)\|^2$$

$$\overset{(a)}{\leq} 4\left(\frac{C_{f_y}^2 L_{l,2}^2}{\mu^2} + L_{f,1}^2\right)\|\bar{y}_t - y^*(\bar{x}_t)\|^2 + 2C_{l_{xy}}^2\|\bar{v}_t - v^*(\bar{x}_t)\|^2$$

$$\leq \bar{L}^2\left(\|\bar{y}_t - y^*(\bar{x}_t)\|^2 + \|\bar{v}_t - v^*(\bar{x}_t)\|^2\right)$$

$$\overset{(b)}{\leq} \frac{\bar{L}^2}{\mu^2}\|\nabla_y l(\bar{x}_t, \bar{y}_t) - \nabla_y l(\bar{x}_t, y^*(\bar{x}_t))\|^2 + \frac{\bar{L}^2}{\mu^2}\|\nabla_v r(\bar{x}_t, \bar{y}_t, \bar{v}_t) - \nabla_v r(\bar{x}_t, \bar{y}_t, v^*(\bar{x}_t))\|^2$$

$$\overset{(c)}{\leq} \frac{\bar{L}^2}{\mu^2}\|\nabla_y l(\bar{x}_t, \bar{y}_t)\|^2 + \frac{2\bar{L}^2}{\mu^2}\|\nabla_v r(\bar{x}_t, \bar{y}_t, \bar{v}_t)\|^2$$

$$+ \frac{2\bar{L}^2}{\mu^2}\|\nabla_v r(\bar{x}_t, \bar{y}_t, v^*(\bar{x}_t)) - \nabla_v r(\bar{x}_t, y^*(\bar{x}_t), v^*(\bar{x}_t))\|^2$$

$$\overset{(d)}{\leq} \frac{\bar{L}^2}{\mu^2}\|\nabla_y l(\bar{x}_t, \bar{y}_t)\|^2 + \frac{2\bar{L}^2}{\mu^2}\|\nabla_v r(\bar{x}_t, \bar{y}_t, \bar{v}_t)\|^2 + \frac{2\bar{L}^2}{\mu^2}\left(\frac{L_{l,2}C_{f_y}}{\mu} + L_{f,1}\right)^2\|\bar{y}_t - y^*(\bar{x}_t)\|^2$$

$$\overset{(e)}{\leq} \left(\frac{\bar{L}^2}{\mu^2} + \frac{2\bar{L}^2}{\mu^4}\left(\frac{L_{l,2}C_{f_y}}{\mu} + L_{f,1}\right)^2\right)\|\nabla_y l(\bar{x}_t, \bar{y}_t)\|^2 + \frac{2\bar{L}^2}{\mu^2}\|\nabla_v r(\bar{x}_t, \bar{y}_t, \bar{v}_t)\|^2, \tag{40}$$

where (a) is using 4) in Lemma C.3 and $\bar{L} := \max\left\{2\left(\frac{C_{f_y}^2 L_{l,2}^2}{\mu^2} + L_{f,1}^2\right)^{\frac{1}{2}}, \sqrt{2}C_{l_{xy}}\right\}$; (b) and (e) use the strong convexity; (c) and (e) result from $\nabla_y l(\bar{x}, y^*(\bar{x})) = 0$ and $\nabla_v r(\bar{x}, y^*(\bar{x}), v^*(\bar{x})) = 0$; (d) uses Lemma C.3. Then substituting Eq. (40) to Eq. (39), we have:

$$-\left\langle \nabla\Phi(\bar{x}_t), \frac{\mathbf{1}^\top}{n}\bar{\nabla}F(\mathbf{x}_t, \mathbf{y}_t, \mathbf{v}_t)\right\rangle$$

$$\overset{(a)}{\leq} -\frac{1}{4}\|\nabla\Phi(\bar{x}_t)\|^2 + \frac{\bar{L}_f^2}{n}\Delta_t + \left(\frac{\bar{L}^2}{2\mu^2} + \frac{\bar{L}^2}{\mu^4}\left(\frac{L_{l,2}C_{f_y}}{\mu} + L_{f,1}\right)^2\right)\|\nabla_y l(\bar{x}_t, \bar{y}_t)\|^2$$

$$+ \frac{\bar{L}^2}{\mu^2}\|\nabla_v r(\bar{x}_t, \bar{y}_t, \bar{v}_t)\|^2 - \frac{1}{2}\|\nabla_x f(\bar{x}_t, \bar{y}_t, \bar{v}_t)\|^2, \tag{41}$$

where (a) uses Lemma C.3 and $\Delta_t := \|\mathbf{x}_t - \mathbf{1}\bar{x}_t\|^2 + \|\mathbf{y}_t - \mathbf{1}\bar{y}_t\|^2 + \|\mathbf{v}_t - \mathbf{1}\bar{v}_t\|^2$ is the consensus error for primal, dual, and auxiliary variables.

For the second inner-product in Eq. (38), using Young's inequality, we get:

$$-\left\langle \nabla\Phi(\bar{x}_t), \frac{(\tilde{\mathbf{q}}_{t+1}^{-1})^\top}{n\bar{q}_{t+1}^{-1}}\bar{\nabla}F(\mathbf{x}_t, \mathbf{y}_t, \mathbf{v}_t)\right\rangle \leq \frac{1}{8}\|\nabla\Phi(\bar{x}_t)\|^2 + 2\left\|\frac{(\tilde{\mathbf{q}}_{t+1}^{-1})^\top}{n\bar{q}_{t+1}^{-1}}\bar{\nabla}F(\mathbf{x}_t, \mathbf{y}_t, \mathbf{v}_t)\right\|^2. \tag{42}$$

For the last term on the RHS of Eq. (38), recalling the definition of stepsize inconsistency in Eq. (27), we have:

$$\frac{\gamma_x L_\Phi}{2\bar{q}_{t+1}^{-1}}\left\|\left(\frac{\bar{q}_{t+1}^{-1}\mathbf{1}^\top}{n} + \frac{(\tilde{\mathbf{q}}_{t+1}^{-1})^\top}{n}\right)\bar{\nabla}F(\mathbf{x}_t, \mathbf{y}_t, \mathbf{v}_t)\right\|^2$$

$$\leq \frac{\gamma_x L_\Phi \bar{q}_{t+1}^{-1}(1+\zeta_q^2)}{n}\|\bar{\nabla}F(\mathbf{x}_t, \mathbf{y}_t, \mathbf{v}_t)\|^2$$

$$\overset{(a)}{\leq} 2n\gamma_x L_\Phi \bar{q}_{t+1}^{-1}(1+\zeta_q^2)\|\bar{\nabla}f(\bar{x}_t, \bar{y}_t, \bar{v}_t)\|^2 + \frac{2\gamma_x \bar{L}_f^2 L_\Phi \bar{q}_{t+1}^{-1}(1+\zeta_q^2)}{n}\Delta_t, \tag{43}$$

where (a) uses $\|\bar{\nabla}F(\mathbf{x}_t, \mathbf{y}_t, \mathbf{v}_t)\|^2 \leq 2\|\bar{\nabla}F(\mathbf{1}\bar{x}_t, \mathbf{1}\bar{y}_t, \mathbf{1}\bar{v}_t)\|^2 + 2\bar{L}_f^2(\|\mathbf{x}_t - \mathbf{1}\bar{x}_t\|^2 + \|\mathbf{y}_t - \mathbf{1}\bar{y}_t\|^2 + \|\mathbf{v}_t - \mathbf{1}\bar{v}_t\|^2)$, $\|\bar{\nabla}F(\mathbf{1}\bar{x}_t, \mathbf{1}\bar{y}_t, \mathbf{1}\bar{v}_t)\|^2 \leq \|\bar{\nabla}F(\mathbf{1}\bar{x}_t, \mathbf{1}\bar{y}_t, \mathbf{1}\bar{v}_t)\|_F^2 = \|n\bar{\nabla}f(\bar{x}_t, \bar{y}_t, \bar{v}_t)\|^2$, and Lemma C.3. By plugging Eq. (41), Eq. (42), and Eq. (43) into Eq. (38), we obtain Eq. (35). Moreover, if $k_1$ in Lemma C.5 exists, then for $t \geq k_1$, we have $\bar{m}^x > C_{m^x} > \frac{8n\gamma_x L_\Phi(1+\zeta_q^2)}{\bar{z}_0}$. Therefore, from Eq. (35) we can get Eq. (36). $\square$

### D.3 The Upper Bound of $\sum_{k=k_2}^t \frac{\|\nabla_y L(\mathbf{x}_k, \mathbf{y}_k)\|^2}{\bar{u}_{k+1}}$

**Lemma D.2.** *Under Assumption 3.1 and Assumption 3.2, for Algorithm 1, suppose the total iteration rounds is $T$. If $k_2$ in Lemma C.5 exists within $T$ iterations, for all integers $t \in [k_2, T]$, we have:*

$$\sum_{k=k_2}^t \frac{\|\nabla_y L(\mathbf{x}_k, \mathbf{y}_k)\|^2}{\bar{u}_{k+1}}$$

$$\leq \frac{2nC_{m^y}^2(\mu + L_{l,1})}{\mu^2\gamma_y\sqrt{1+\sigma_u^2}} + \left(\frac{8\Delta_0^x}{1-\rho_W} + \frac{32\gamma_x^2\rho_W C_{m^x}^2(1+\zeta_q^2)}{\bar{q}_0^2(1-\rho_W)^2}\right)\left(\frac{(\mu+L_{l,1})^3}{\gamma_y\mu^2} + \frac{2L_{l,1}(\mu+L_{l,1})}{\gamma_y\mu}\right)$$

$$+ \frac{32\gamma_x^2\rho_W(1+\zeta_q^2)}{\bar{z}_0(1-\rho_W)^2}\left(\frac{(\mu+L_{l,1})^3}{\gamma_y\mu^2} + \frac{2L_{l,1}(\mu+L_{l,1})}{\gamma_y\mu}\right) + \frac{8\gamma_x^2 L_y^2(1+\zeta_q^2)(\mu+L_{l,1})^2}{\gamma_y^2\mu L_{l,1}\bar{z}_0\sqrt{1+\sigma_u^2}}$$

$$+ \left[\frac{32\gamma_x^2\rho_W(1+\zeta_q^2)}{(1-\rho_W)^2}\left(\frac{(\mu+L_{l,1})^3}{\gamma_y\mu^2} + \frac{2L_{l,1}(\mu+L_{l,1})}{\gamma_y\mu}\right)\right.$$

$$\left.+ \frac{8\gamma_x^2 L_y^2(1+\zeta_q^2)(\mu+L_{l,1})^2}{\gamma_y^2\mu L_{l,1}\sqrt{1+\sigma_u^2}}\right]\sum_{k=\min\{k_1,k_2\}}^t \frac{\|\bar{\nabla}F(\mathbf{x}_k, \mathbf{y}_k, \mathbf{v}_k)\|^2}{[\bar{m}_{k+1}^x]^2\max\{\bar{m}_{k+1}^v, \bar{m}_{k+1}^y\}}. \tag{44}$$

*Proof.* For $k_2 \leq t < T$, we have $\bar{m}_{k_2}^y \leq C_{m^y}$ and $\bar{m}_{t+1}^y > C_{m^y}$. For any positive scalar $\bar{\lambda}_{t+1}$, using Young's inequality, we have:

$$\frac{1}{n}\|\mathbf{y}_{t+1} - \mathbf{1}y^*(\bar{x}_{t+1})\|^2 \leq \frac{(1+\bar{\lambda}_{t+1})}{n}\|\mathbf{y}_{t+1} - \mathbf{1}y^*(\bar{x}_t)\|^2 + \left(1 + \frac{1}{\bar{\lambda}_{t+1}}\right)\|y^*(\bar{x}_t) - y^*(\bar{x}_{t+1})\|^2. \tag{45}$$

For the first term on the RHS of Eq. (45), we have:

$$\frac{1}{n}\|\mathbf{y}_{t+1} - \mathbf{1}y^*(\bar{x}_t)\|^2$$

$$= \frac{1}{n}\left\|\mathbf{y}_t - \gamma_y U_{t+1}^{-1}\nabla_y L(\mathbf{x}_t, \mathbf{y}_t) - \mathbf{1}y^*(\bar{x}_t)\right\|^2$$

$$= \frac{1}{n}\|\mathbf{y}_t - \mathbf{1}y^*(\bar{x}_t))\|^2 + \frac{\gamma_y^2}{n}\|U_{t+1}^{-1}\nabla_y L(\mathbf{x}_t, \mathbf{y}_t)\|^2 - \frac{1}{n}\sum_{i=1}^n 2\bar{u}_{t+1}^{-1}\gamma_y\langle\nabla_y l_i(x_{i,t}, y_{i,t}), y_{i,t} - y^*(\bar{x}_t)\rangle$$

$$- \frac{1}{n}\sum_{i=1}^n 2\bar{u}_{t+1}^{-1}\gamma_y\left\langle\left(\frac{u_{i,t+1}^{-1} - \bar{u}_{t+1}^{-1}}{\bar{u}_{t+1}^{-1}}\right)\nabla_y l_i(x_{i,t}, y_{i,t}), y_{i,t} - y^*(\bar{x}_t)\right\rangle$$

$$\leq \frac{1}{n}\|\mathbf{y}_t - \mathbf{1}y^*(\bar{x}_t))\|^2 + \frac{\gamma_y^2}{n}\|U_{t+1}^{-1}\nabla_y L(\mathbf{x}_t, \mathbf{y}_t)\|^2$$

$$- \frac{1}{n}\sum_{i=1}^n 2\bar{u}_{t+1}^{-1}\gamma_y\langle\nabla_y l_i(x_{i,t}, y_{i,t}) - \nabla_y l_i(x_{i,t}, y^*(\bar{x}_t)), y_{i,t} - y^*(\bar{x}_t)\rangle$$

$$- \frac{1}{n}\sum_{i=1}^n 2\bar{u}_{t+1}^{-1}\gamma_y\langle\nabla_y l_i(x_{i,t}, y^*(\bar{x}_t)) - \nabla_y l_i(\bar{x}_t, y^*(\bar{x}_t)), y_{i,t} - y^*(\bar{x}_t)\rangle$$

$$- \frac{1}{n}\sum_{i=1}^n 2\bar{u}_{t+1}^{-1}\gamma_y\left\langle\left(\frac{u_{i,t+1}^{-1} - \bar{u}_{t+1}^{-1}}{\bar{u}_{t+1}^{-1}}\right)(\nabla_y l_i(x_{i,t}, y_{i,t}) - \nabla_y l_i(x_{i,t}, y^*(\bar{x}_t))), y_{i,t} - y^*(\bar{x}_t)\right\rangle$$

$$- \frac{1}{n}\sum_{i=1}^n 2\bar{u}_{t+1}^{-1}\gamma_y\left\langle\left(\frac{u_{i,t+1}^{-1} - \bar{u}_{t+1}^{-1}}{\bar{u}_{t+1}^{-1}}\right)(\nabla_y l_i(x_{i,t}, y^*(\bar{x}_t)) - \nabla_y l_i(\bar{x}_t, y^*(\bar{x}_t))), y_{i,t} - y^*(\bar{x}_t)\right\rangle$$

$$\overset{(a)}{\leq} \left(\frac{1}{n} - \frac{\gamma_y\bar{u}_{t+1}^{-1}\mu L_{l,1}}{n(\mu + L_{l,1})}\right)\|\mathbf{y}_t - \mathbf{1}y^*(\bar{x}_t))\|^2 + \frac{\gamma_y^2\bar{u}_{t+1}^{-2}(1 + \zeta_u^2)}{n}\|\nabla_y L(\mathbf{x}_t, \mathbf{y}_t)\|^2$$

$$- \frac{2\gamma_y\bar{u}_{t+1}^{-1}\sqrt{1 + \sigma_u^2}}{n(\mu + L_{l,1})}\|\nabla_y L(\mathbf{x}_t, \mathbf{y}_t) \pm \nabla_y L(\mathbf{1}\bar{x}_t, \mathbf{1}y^*(\bar{x}_t)) - \nabla_y L(\mathbf{x}_t, \mathbf{1}y^*(\bar{x}_t))\|^2$$

$$+ \frac{\gamma_y\bar{u}_{t+1}^{-1}(\mu + L_{l,1})\sqrt{1 + \sigma_u^2}}{n\mu L_{l,1}}\|\nabla_y L(\mathbf{x}_t, \mathbf{1}y^*(\bar{x}_t)) - \nabla_y L(\mathbf{1}\bar{x}_t, \mathbf{1}y^*(\bar{x}_t))\|^2$$

$$\overset{(b)}{\leq} \left(\frac{1}{n} - \frac{\gamma_y\bar{u}_{t+1}^{-1}\mu L_{l,1}}{n(\mu + L_{l,1})}\right)\|\mathbf{y}_t - \mathbf{1}y^*(\bar{x}_t))\|^2$$

$$+ \frac{\gamma_y\bar{u}_{t+1}^{-1}}{n}\left(\gamma_y\bar{u}_{t+1}^{-1}(1 + \zeta_u^2) - \frac{\sqrt{1 + \sigma_u^2}}{\mu + L_{l,1}}\right)\|\nabla_y L(\mathbf{x}_t, \mathbf{y}_t)\|^2$$

$$+ \left(\frac{\gamma_y\bar{u}_{t+1}^{-1}(\mu + L_{l,1})\sqrt{1 + \sigma_u^2}}{n\mu L_{l,1}} + \frac{2\gamma_y\bar{u}_{t+1}^{-1}\sqrt{1 + \sigma_u^2}}{n(\mu + L_{l,1})}\right)\|\nabla_y L(\mathbf{x}_t, \mathbf{1}y^*(\bar{x}_t)) - \nabla_y L(\mathbf{1}\bar{x}_t, \mathbf{1}y^*(\bar{x}_t))\|^2$$

$$\overset{(c)}{\leq} \left(\frac{1}{n} - \frac{\gamma_y\bar{u}_{t+1}^{-1}\mu L_{l,1}}{n(\mu + L_{l,1})}\right)\|\mathbf{y}_t - \mathbf{1}y^*(\bar{x}_t))\|^2$$

$$+ \frac{\gamma_y\bar{u}_{t+1}^{-1}}{n}\left(\gamma_y\bar{u}_{t+1}^{-1}(1 + \zeta_u^2) - \frac{\sqrt{1 + \sigma_u^2}}{\mu + L_{l,1}}\right)\|\nabla_y L(\mathbf{x}_t, \mathbf{y}_t)\|^2$$

$$+ \left(\frac{\gamma_y\bar{u}_{t+1}^{-1}L_{l,1}(\mu + L_{l,1})\sqrt{1 + \sigma_u^2}}{n\mu} + \frac{2\gamma_y\bar{u}_{t+1}^{-1}L_{l,1}^2\sqrt{1 + \sigma_u^2}}{n(\mu + L_{l,1})}\right)\|\mathbf{x}_t - \mathbf{1}\bar{x}_t\|^2$$

$$\overset{(d)}{\leq} \left(\frac{1}{n} - \frac{\gamma_y\bar{u}_{t+1}^{-1}\mu L_{l,1}}{n(\mu + L_{l,1})}\right)\|\mathbf{y}_t - \mathbf{1}y^*(\bar{x}_t))\|^2 - \frac{\gamma_y\bar{u}_{t+1}^{-1}\sqrt{1 + \sigma_u^2}}{2n(\mu + L_{l,1})}\|\nabla_y L(\mathbf{x}_t, \mathbf{y}_t)\|^2$$

$$+ \left( \frac{\gamma_y \bar{u}_{t+1}^{-1} L_{l,1}(\mu + L_{l,1})\sqrt{1+\sigma_u^2}}{n\mu} + \frac{2\gamma_y \bar{u}_{t+1}^{-1} L_{l,1}^2 \sqrt{1+\sigma_u^2}}{n(\mu + L_{l,1})} \right) \|\mathbf{x}_t - \mathbf{1}\bar{x}_t\|^2, \tag{46}$$

where (a) employs Lemma 3.11 in [Bubeck et al., 2015], (b) uses Young's inequality, (c) refers to Assumption 3.2, and (d) follows from $\bar{m}_{t+1}^y \geq C_{m^y} \geq \frac{2\gamma_y(1+\zeta_u^2)(\mu+L_{l,1})}{\sqrt{1+\sigma_u^2}}$. By plugging equation Eq. (46) into equation Eq. (45), we have:

$$\frac{1}{n}\|\mathbf{y}_{t+1} - \mathbf{1}y^*(\bar{x}_{t+1})\|^2$$

$$\leq (1 + \bar{\lambda}_{t+1})\frac{1}{n}\left( 1 - \frac{\gamma_y \bar{u}_{t+1}^{-1}\mu L_{l,1}}{\mu + L_{l,1}} \right)\|\mathbf{y}_t - \mathbf{1}y^*(\bar{x}_t))\|^2$$

$$- (1 + \bar{\lambda}_{t+1})\frac{\gamma_y \bar{u}_{t+1}^{-1}\sqrt{1+\sigma_u^2}}{2n(\mu + L_{l,1})}\|\nabla_y L(\mathbf{x}_t, \mathbf{y}_t)\|^2$$

$$+ (1 + \bar{\lambda}_{t+1})\left( \frac{\gamma_y \bar{u}_{t+1}^{-1} L_{l,1}(\mu + L_{l,1})\sqrt{1+\sigma_u^2}}{n\mu} + \frac{2\gamma_y \bar{u}_{t+1}^{-1} L_{l,1}^2 \sqrt{1+\sigma_u^2}}{n(\mu + L_{l,1})} \right)\|\mathbf{x}_t - \mathbf{1}\bar{x}_t\|^2$$

$$+ \left( 1 + \frac{1}{\bar{\lambda}_{t+1}} \right)\|y^*(\bar{x}_t) - y^*(\bar{x}_{t+1})\|^2. \tag{47}$$

By rearranging the terms in Eq. (47), we have:

$$(1 + \bar{\lambda}_{t+1})\frac{\gamma_y \bar{u}_{t+1}^{-1}\sqrt{1+\sigma_u^2}}{2n(\mu + L_{l,1})}\|\nabla_y L(\mathbf{x}_t, \mathbf{y}_t)\|^2$$

$$\leq (1 + \bar{\lambda}_{t+1})\frac{1}{n}\left( 1 - \frac{\gamma_y \bar{u}_{t+1}^{-1}\mu L_{l,1}}{\mu + L_{l,1}} \right)\|\mathbf{y}_t - \mathbf{1}y^*(\bar{x}_t))\|^2 - \frac{1}{n}\|\mathbf{y}_{t+1} - \mathbf{1}y^*(\bar{x}_{t+1})\|^2$$

$$+ (1 + \bar{\lambda}_{t+1})\left( \frac{\gamma_y \bar{u}_{t+1}^{-1} L_{l,1}(\mu + L_{l,1})\sqrt{1+\sigma_u^2}}{n\mu} + \frac{2\gamma_y \bar{u}_{t+1}^{-1} L_{l,1}^2 \sqrt{1+\sigma_u^2}}{n(\mu + L_{l,1})} \right)\|\mathbf{x}_t - \mathbf{1}\bar{x}_t\|^2$$

$$+ \left( 1 + \frac{1}{\bar{\lambda}_{t+1}} \right)\|y^*(\bar{x}_t) - y^*(\bar{x}_{t+1})\|^2. \tag{48}$$

We take $\bar{\lambda}_{t+1} := \frac{\gamma_y \bar{u}_{t+1}^{-1}\mu L_{l,1}}{\mu + L_{l,1}}$. Since $\bar{m}_{t+1}^y > C_{m^y} \geq \frac{\gamma_y \mu L_{l,1}}{\mu + L_{l,1}}$ in Eq. (31), we have $\bar{\lambda}_{t+1} \leq 1$. Then, we have:

$$\frac{\gamma_y \bar{u}_{t+1}^{-1}\sqrt{1+\sigma_u^2}}{2n}\|\nabla_y L(\mathbf{x}_t, \mathbf{y}_t)\|^2$$

$$\leq (1 + \bar{\lambda}_{t+1})\frac{\gamma_y \bar{u}_{t+1}^{-1}\sqrt{1+\sigma_u^2}}{2n}\|\nabla_y L(\mathbf{x}_t, \mathbf{y}_t)\|^2$$

$$\leq \frac{\mu + L_{l,1}}{n}\left( \|\mathbf{y}_t - \mathbf{1}y^*(\bar{x}_t))\|^2 - \|\mathbf{y}_{t+1} - \mathbf{1}y^*(\bar{x}_{t+1})\|^2 \right) + \frac{2(\mu + L_{l,1})}{\bar{\lambda}_{t+1}}\|y^*(\bar{x}_t) - y^*(\bar{x}_{t+1})\|^2$$

$$+ \left( \frac{2\gamma_y \bar{u}_{t+1}^{-1} L_{l,1}(\mu + L_{l,1})^2 \sqrt{1+\sigma_u^2}}{n\mu} + \frac{4\gamma_y \bar{u}_{t+1}^{-1} L_{l,1}^2 \sqrt{1+\sigma_u^2}}{n} \right)\|\mathbf{x}_t - \mathbf{1}\bar{x}_t\|^2$$

$$= \frac{\mu + L_{l,1}}{n}\left( \|\mathbf{y}_t - \mathbf{1}y^*(\bar{x}_t))\|^2 - \|\mathbf{y}_{t+1} - \mathbf{1}y^*(\bar{x}_{t+1})\|^2 \right) + \frac{2(\mu + L_{l,1})^2}{\gamma_y \bar{u}_{t+1}^{-1}\mu L_{l,1}}\|y^*(\bar{x}_t) - y^*(\bar{x}_{t+1})\|^2$$

$$+ \left( \frac{2\gamma_y \bar{u}_{t+1}^{-1} L_{l,1}(\mu + L_{l,1})^2 \sqrt{1+\sigma_u^2}}{n\mu} + \frac{4\gamma_y \bar{u}_{t+1}^{-1} L_{l,1}^2 \sqrt{1+\sigma_u^2}}{n} \right)\|\mathbf{x}_t - \mathbf{1}\bar{x}_t\|^2$$

$$\overset{(a)}{\leq} \frac{\mu + L_{l,1}}{n}\left( \|\mathbf{y}_t - \mathbf{1}y^*(\bar{x}_t))\|^2 - \|\mathbf{y}_{t+1} - \mathbf{1}y^*(\bar{x}_{t+1})\|^2 \right) + \frac{2(\mu + L_{l,1})^2 L_y^2}{\gamma_y \bar{u}_{t+1}^{-1}\mu L_{l,1}}\|\bar{x}_t - \bar{x}_{t+1}\|^2$$

$$+ \left( \frac{2(\mu + L_{l,1})^3 \sqrt{1+\sigma_u^2}}{n\mu^2} + \frac{4L_{l,1}(\mu + L_{l,1})\sqrt{1+\sigma_u^2}}{n\mu} \right)\|\mathbf{x}_t - \mathbf{1}\bar{x}_t\|^2, \tag{49}$$

where (a) uses the Lipschitzness of $y^*(\bar{x})$. Summing the above inequality over $k = k_2, \dots, t$, we have:

$$\sum_{k=k_2}^{t} \frac{\gamma_y \bar{u}_{k+1}^{-1} \sqrt{1+\sigma_u^2}}{2n} \|\nabla_y L(\mathbf{x}_k, \mathbf{y}_k)\|^2$$

$$\leq \sum_{k=k_2-1}^{t} \frac{\gamma_y \bar{u}_{k+1}^{-1} \sqrt{1+\sigma_u^2}}{2n} \|\nabla_y L(\mathbf{x}_k, \mathbf{y}_k)\|^2$$

$$\leq \frac{\mu + L_{l,1}}{n} \|\mathbf{y}_{k_2-1} - \mathbf{1}y^*(\bar{x}_{k_2-1})\|^2 + \frac{2(\mu+L_{l,1})^2 L_y^2}{\gamma_y \mu L_{l,1}} \sum_{k=k_2-1}^{t} \bar{u}_{k+1} \|\bar{x}_k - \bar{x}_{k+1}\|^2$$

$$+ \left( \frac{2(\mu+L_{l,1})^3 \sqrt{1+\sigma_u^2}}{n\mu^2} + \frac{4L_{l,1}(\mu+L_{l,1})\sqrt{1+\sigma_u^2}}{n\mu} \right) \sum_{k=k_2-1}^{t} \|\mathbf{x}_k - \mathbf{1}\bar{x}_k\|^2$$

$$\overset{(a)}{\leq} \frac{\mu + L_{l,1}}{n\mu^2} \sum_{i=1}^{n} \|\nabla_y l_i(x_{i,k_2-1}, y_{i,k_2-1}) - \nabla_y l_i(x_{i,k_2-1}, y^*(\bar{x}_{k_2-1}))\|^2$$

$$+ \left( \frac{2(\mu+L_{l,1})^3 \sqrt{1+\sigma_u^2}}{n\mu^2} + \frac{4L_{l,1}(\mu+L_{l,1})\sqrt{1+\sigma_u^2}}{n\mu} \right) \sum_{k=0}^{t} \|\mathbf{x}_k - \mathbf{1}\bar{x}_k\|^2$$

$$+ \frac{2(\mu+L_{l,1})^2 L_y^2}{\gamma_y \mu L_{l,1}} \sum_{k=k_2-1}^{t} \bar{u}_{k+1} \left\| \gamma_x \bar{q}_{k+1}^{-1} \frac{\mathbf{1}^\top}{n} \bar{\nabla} F(\mathbf{x}_k, \mathbf{y}_k, \mathbf{v}_k) - \gamma_x \frac{(\tilde{\mathbf{q}}_{k+1}^{-1})^\top}{n} \bar{\nabla} F(\mathbf{x}_k, \mathbf{y}_k, \mathbf{v}_k) \right\|^2$$

$$\overset{(b)}{\leq} \frac{\mu + L_{l,1}}{n\mu^2} \sum_{i=1}^{n} \|\nabla_y l_i(x_{i,k_2-1}, y_{i,k_2-1}) - \nabla_y l_i(x_{i,k_2-1}, y^*(\bar{x}_{k_2-1}))\|^2$$

$$+ \frac{4\gamma_x^2 L_y^2 \left(1+\zeta_q^2\right)(\mu+L_{l,1})^2}{n\gamma_y \mu L_{l,1}} \sum_{k=k_2-1}^{t} \bar{u}_{k+1} \bar{q}_{k+1}^{-2} \|\bar{\nabla} F(\mathbf{x}_k, \mathbf{y}_k, \mathbf{v}_k)\|^2 + \frac{16\gamma_x^2 \rho_W \left(1+\zeta_q^2\right)}{(1-\rho_W)^2}$$

$$\cdot \left( \frac{(\mu+L_{l,1})^3 \sqrt{1+\sigma_u^2}}{n\mu^2} + \frac{2L_{l,1}(\mu+L_{l,1})\sqrt{1+\sigma_u^2}}{n\mu} \right) \sum_{k=0}^{t} \bar{q}_{k+1}^{-2} \|\bar{\nabla} F(\mathbf{x}_k, \mathbf{y}_k, \mathbf{v}_k)\|^2$$

$$+ \frac{4\Delta_0^x (\mu+L_{l,1})^3 \sqrt{1+\sigma_u^2}}{n\mu^2(1-\rho_W)} + \frac{8\Delta_0^x L_{l,1}(\mu+L_{l,1})\sqrt{1+\sigma_u^2}}{n\mu(1-\rho_W)}$$

$$\overset{(c)}{\leq} \frac{(\mu+L_{l,1})C_{m^y}^2}{\mu^2} + \left( \frac{4\Delta_0^x}{1-\rho_W} + \frac{16\gamma_x^2 \rho_W C_{m^x}^2 (1+\zeta_q^2)}{\bar{q}_0^2(1-\rho_W)^2} \right)$$

$$\cdot \left( \frac{(\mu+L_{l,1})^3 \sqrt{1+\sigma_u^2}}{n\mu^2} + \frac{2L_{l,1}(\mu+L_{l,1})\sqrt{1+\sigma_u^2}}{n\mu} \right)$$

$$+ \left[ \frac{16\gamma_x^2 \rho_W (1+\zeta_q^2)}{(1-\rho_W)^2} \left( \frac{(\mu+L_{l,1})^3 \sqrt{1+\sigma_u^2}}{n\mu^2} + \frac{2L_{l,1}(\mu+L_{l,1})\sqrt{1+\sigma_u^2}}{n\mu} \right) \right.$$

$$\left. + \frac{4\gamma_x^2 L_y^2 \left(1+\zeta_q^2\right)(\mu+L_{l,1})^2}{n\gamma_y \mu L_{l,1}} \right] \sum_{k=\min\{k_1-1,k_2-1\}}^{t} \frac{\|\bar{\nabla} F(\mathbf{x}_k, \mathbf{y}_k, \mathbf{v}_k)\|^2}{[\bar{m}_{k+1}^x]^2 \max\{\bar{m}_{k+1}^v, \bar{m}_{k+1}^y\}}$$

$$\leq \frac{(\mu+L_{l,1})C_{m^y}^2}{\mu^2} + \left( \frac{4\Delta_0^x}{1-\rho_W} + \frac{16\gamma_x^2 \rho_W C_{m^x}^2 (1+\zeta_q^2)}{\bar{q}_0^2(1-\rho_W)^2} \right)$$

$$\cdot \left( \frac{(\mu+L_{l,1})^3 \sqrt{1+\sigma_u^2}}{n\mu^2} + \frac{2L_{l,1}(\mu+L_{l,1})\sqrt{1+\sigma_u^2}}{n\mu} \right) + \frac{4\gamma_x^2 L_y^2 \left(1+\zeta_q^2\right)(\mu+L_{l,1})^2}{n\gamma_y \mu L_{l,1} \bar{z}_0}$$

$$+ \frac{16\gamma_x^2 \rho_W (1+\zeta_q^2)}{\bar{z}_0(1-\rho_W)^2} \left( \frac{(\mu+L_{l,1})^3 \sqrt{1+\sigma_u^2}}{n\mu^2} + \frac{2L_{l,1}(\mu+L_{l,1})\sqrt{1+\sigma_u^2}}{n\mu} \right)$$

$$+ \left[ \frac{16\gamma_x^2 \rho_W (1+\zeta_q^2)}{(1-\rho_W)^2} \left( \frac{(\mu+L_{l,1})^3 \sqrt{1+\sigma_u^2}}{n\mu^2} + \frac{2L_{l,1}(\mu+L_{l,1})\sqrt{1+\sigma_u^2}}{n\mu} \right) \right.$$

$$\left. + \frac{4\gamma_x^2 L_y^2 (1+\zeta_q^2)(\mu+L_{l,1})^2}{n\gamma_y \mu L_{l,1}} \right] \sum_{k=\min\{k_1,k_2\}}^{t} \frac{\|\bar{\nabla}F(\mathbf{x}_k, \mathbf{y}_k, \mathbf{v}_k)\|^2}{[\bar{m}_{k+1}^x]^2 \max\{\bar{m}_{k+1}^v, \bar{m}_{k+1}^y\}}, \tag{50}$$

where (a) uses Assumption 3.1; (b) refers to Lemma C.4; (c) results from $\|\nabla_y l_i(x_{i,k_2-1}, y_{i,k_2-1})\|^2 \leq [m_{i,k_2}^y]^2 \leq C_{m^y}^2$ and $\|\bar{\nabla}f_i(x_{i,k_1-1}, y_{i,k_1-1}, v_{i,k_1-1})\|^2 \leq [m_{i,k_1}^x]^2 \leq C_{m^x}^2$. Then, the proof is complete. $\qquad \square$

### D.4 The Upper Bound of $\sum_{k=k_3}^{t} \frac{\|\nabla_v R(\mathbf{x}_k, \mathbf{y}_k, \mathbf{v}_k)\|^2}{\bar{z}_{k+1}}$

**Lemma D.3.** *Under Assumption 3.1 and Assumption 3.2, for Algorithm 1, suppose the total iteration rounds is $T$. If $k_3$ in Lemma C.5 exists within $T$ iterations, for all integers $t \in [k_3, T)$, we have:*

$$\sum_{k=k_3}^{t} \frac{\|\nabla_v R(\mathbf{x}_k, \mathbf{y}_k, \mathbf{v}_k)\|^2}{\bar{z}_{k+1}}$$

$$\leq \frac{4nC_{m^y}^2(\mu+C_{l_{yy}})}{\mu^4 \gamma_v \sqrt{1+\sigma_z^2}} \left( \frac{nL_{l,2}C_{f_y}}{\mu} + L_{f,1} \right)^2 + \frac{8nC_{m^v}^2(\mu+C_{l_{yy}})}{\mu^2 \gamma_v \sqrt{1+\sigma_z^2}}$$

$$+ \frac{16(\mu+C_{l_{yy}})}{\gamma_v \mu^2} \left( \frac{nL_{l,2}C_{f_y}}{\mu} + L_{f,1} \right)^2 \left( \frac{\Delta_0^x}{1-\rho_W} + \frac{4\gamma_x^2 \rho_W C_{m^x}^2 (1+\zeta_q^2)}{\bar{q}_0^2(1-\rho_W)^2} \right) \left( \frac{(\mu+L_{l,1})^2}{n\mu^2} + \frac{2L_{l,1}}{n\mu} \right)$$

$$+ \frac{16(\mu+C_{l_{yy}})}{\gamma_v \mu^2 \sqrt{1+\sigma_u^2}} \left( \frac{nL_{l,2}C_{f_y}}{\mu} + L_{f,1} \right)^2 \left[ \frac{4\gamma_x^2 \rho_W (1+\zeta_q^2)}{\bar{z}_0^2(1-\rho_W)^2} \left( \frac{(\mu+L_{l,1})^2 \sqrt{1+\sigma_u^2}}{n\mu^2} \right. \right.$$

$$\left. \left. + \frac{2L_{l,1}\sqrt{1+\sigma_u^2}}{n\mu} \right) + \frac{\gamma_x^2 L_y^2 (1+\zeta_q^2)(\mu+L_{l,1})}{n\bar{z}_0 \gamma_y \mu L_{l,1}} \right] \sum_{k=\min\{k_1-1,k_2-1\}}^{k_3-2} \frac{\|\bar{\nabla}F(\mathbf{x}_k, \mathbf{y}_k, \mathbf{v}_k)\|^2}{[\bar{m}_{k+1}^x]^2}$$

$$+ \left( \frac{64\gamma_x^2 \rho_W \bar{L}_r^2(\mu+C_{l_{yy}})(1+\zeta_q^2)}{\mu C_{l_{yy}} \gamma_v (1-\rho_W)^2} + \frac{2\gamma_x^2 L_v^2(1+\zeta_q^2)(\mu+C_{l_{yy}})^2}{\mu C_{l_{yy}} C_{m^v} \gamma_v^2 \sqrt{1+\sigma_z^2}} \right)$$

$$\cdot \sum_{k=\min\{k_1-1,k_3-1\}}^{t} \frac{\|\bar{\nabla}F(\mathbf{x}_k, \mathbf{y}_k, \mathbf{v}_k)\|^2}{[\bar{m}_{k+1}^x]^2}$$

$$+ \left( \frac{8}{\mu^2} + \frac{4(\mu+C_{l_{yy}})^2}{\mu^3 C_{l_{yy}}} \right) \left( \frac{nL_{l,2}C_{f_y}}{\mu} + L_{f,1} \right)^2 \sum_{k=k_3-1}^{t} \frac{\|\nabla_y L(\mathbf{x}_k, \mathbf{y}_k)\|^2}{\bar{m}_{k+1}^y}$$

$$+ \left( \frac{16\Delta_0^x}{1-\rho_W} + \frac{64\gamma_x^2 \rho_W C_{m^x}^2(1+\zeta_q^2)}{\bar{q}_0^2(1-\rho_W)^2} \right) \left( \frac{\bar{L}_r^2(\mu+C_{l_{yy}})}{\mu C_{l_{yy}} \gamma_v} + \frac{\bar{L}_r^2(\mu+C_{l_{yy}})}{\mu^2 \gamma_v \sqrt{1+\sigma_z^2}} \right). \tag{51}$$

*Proof.* For $k_3 \leq t < T$, we have $\bar{m}_{t+1}^v > C_{m^v}$. For any positive scalar $\hat{\lambda}_{t+1}$, using Young's inequality, we have:

$$\frac{1}{n} \|\mathbf{v}_{t+1} - \mathbf{1}v^*(\bar{x}_{t+1})\|^2 \leq \frac{(1+\bar{\lambda}_{t+1})}{n} \|\mathbf{v}_{t+1} - \mathbf{1}v^*(\bar{x}_t)\|^2 + \left( 1 + \frac{1}{\bar{\lambda}_{t+1}} \right) \|v^*(\bar{x}_t) - v^*(\bar{x}_{t+1})\|^2. \tag{52}$$

For the first term on the RHS of Eq. (52), we have:

$$\frac{1}{n} \|\mathbf{v}_{t+1} - \mathbf{1}v^*(\bar{x}_{t+1})\|^2$$

$$= \frac{1}{n} \left\| \mathcal{P}_\mathcal{V} \left( \mathbf{v}_t - \gamma_v Z_{t+1}^{-1} \nabla_v R(\mathbf{x}_t, \mathbf{y}_t, \mathbf{v}_t) \right) - \mathbf{1}v^*(\bar{x}_t) \right\|^2$$

$$\overset{(a)}{\leq} \frac{1}{n} \left\| \mathbf{v}_t - \gamma_v Z_{t+1}^{-1} \nabla_v R(\mathbf{x}_t, \mathbf{y}_t, \mathbf{v}_t) - \mathbf{1}v^*(\bar{x}_t) \right\|^2$$

$$= \frac{1}{n}\|\mathbf{v}_t - \mathbf{1}v^*(\bar{x}_t)\|^2 + \frac{\gamma_v^2}{n}\|Z_{t+1}^{-1}\nabla_v R(\mathbf{x}_t, \mathbf{y}_t, \mathbf{v}_t)\|^2$$

$$- \frac{1}{n}\sum_{i=1}^n 2\bar{z}_{t+1}^{-1}\gamma_v \langle \nabla_v r_i(x_{i,t}, y_{i,t}, v_{i,t}), v_{i,t} - v^*(\bar{x}_t)\rangle$$

$$- \frac{1}{n}\sum_{i=1}^n 2\bar{z}_{t+1}^{-1}\gamma_y \left\langle \left(\frac{z_{i,t+1}^{-1} - \bar{z}_{t+1}^{-1}}{\bar{z}_{t+1}^{-1}}\right)\nabla_v r_i(x_{i,t}, y_{i,t}, v_{i,t}), v_{i,t} - v^*(\bar{x}_t)\right\rangle, \qquad (53)$$

where (a) uses the non-expansiveness of the projection operator, as established in Lemma 1 of [Nedic et al., 2010]. Then, for the last two terms on the RHS of Eq. (53), we have:

$$- \frac{1}{n}\sum_{i=1}^n 2\bar{z}_{t+1}^{-1}\gamma_v \langle \nabla_v r_i(x_{i,t}, y_{i,t}, v_{i,t}), v_{i,t} - v^*(\bar{x}_t)\rangle$$

$$- \frac{1}{n}\sum_{i=1}^n 2\bar{z}_{t+1}^{-1}\gamma_y \left\langle \left(\frac{z_{i,t+1}^{-1} - \bar{z}_{t+1}^{-1}}{\bar{z}_{t+1}^{-1}}\right)\nabla_v r_i(x_{i,t}, y_{i,t}, v_{i,t}), v_{i,t} - v^*(\bar{x}_t)\right\rangle$$

$$= -\frac{2\gamma_v}{n}\sum_{i=1}^n \bar{z}_{t+1}^{-1}\langle \nabla_v r_i(x_{i,t}, y_{i,t}, v_{i,t}) - \nabla_v r_i(x_{i,t}, y_{i,t}, v_i^*(\bar{x}_t)), v_{i,t} - v_i^*(\bar{x}_t)\rangle$$

$$- \frac{2\gamma_v}{n}\sum_{i=1}^n \bar{z}_{t+1}^{-1}\langle \nabla_v r_i(x_{i,t}, y_{i,t}, v_i^*(\bar{x}_t)) - \nabla_v r_i(x_{i,t}, y_i^*(\bar{x}_t), v^*(\bar{x}_t)), v_{i,t} - v_i^*(\bar{x}_t)\rangle$$

$$- \frac{2\gamma_v}{n}\sum_{i=1}^n \bar{z}_{t+1}^{-1}\left\langle \left(\frac{z_{i,t+1}^{-1} - \bar{z}_{t+1}^{-1}}{\bar{z}_{t+1}^{-1}}\right)(\nabla_v r_i(x_{i,t}, y_{i,t}, v_{i,t}) - \nabla_v r_i(x_{i,t}, y_{i,t}, v_i^*(\bar{x}_t))), v_{i,t} - v_i^*(\bar{x}_t)\right\rangle$$

$$- \frac{2\gamma_v}{n}\sum_{i=1}^n \bar{z}_{t+1}^{-1}\left\langle \left(\frac{z_{i,t+1}^{-1} - \bar{z}_{t+1}^{-1}}{\bar{z}_{t+1}^{-1}}\right)(\nabla_v r_i(x_{i,t}, y_{i,t}, v_i^*(\bar{x}_t)) - \nabla_v r_i(x_{i,t}, y_i^*(\bar{x}_t), v^*(\bar{x}_t))), v_{i,t} - v_i^*(\bar{x}_t)\right\rangle$$

$$\overset{(a)}{\leq} -\frac{2\gamma_v \bar{z}_{t+1}^{-1}\sqrt{1+\sigma_z^2}}{n(\mu + C_{l_{yy}})}\|\nabla_v R(\mathbf{x}_t, \mathbf{y}_t, \mathbf{v}_t) - \nabla_v R(\mathbf{x}_t, \mathbf{y}_t, \mathbf{1}v^*(\bar{x}_t))\|^2$$

$$+ \frac{\gamma_v \bar{z}_{t+1}^{-1}\sqrt{1+\sigma_z^2}(\mu + C_{l_{yy}})}{n\mu C_{l_{yy}}}\|\nabla_v R(\mathbf{x}_t, \mathbf{y}_t, \mathbf{1}v^*(\bar{x}_t)) - \nabla_v R(\mathbf{x}_t, \mathbf{1}y^*(\bar{x}_t), \mathbf{1}v^*(\bar{x}_t))\|^2$$

$$+ \frac{2\mu C_{l_{yy}}\gamma_v \bar{z}_{t+1}^{-1}}{n(\mu + C_{l_{yy}})}\|\mathbf{v}_t - \mathbf{1}v^*(\bar{x}_t)\|^2 - \frac{4\mu C_{l_{yy}}\gamma_v \bar{z}_{t+1}^{-1}}{n(\mu + C_{l_{yy}})}\|\mathbf{v}_t - \mathbf{1}v^*(\bar{x}_t)\|^2$$

$$\overset{(b)}{\leq} -\frac{\gamma_v \bar{z}_{t+1}^{-1}\sqrt{1+\sigma_z^2}}{n(\mu + C_{l_{yy}})}\|\nabla_v R(\mathbf{x}_t, \mathbf{y}_t, \mathbf{v}_t)\|^2 + \frac{2\gamma_v \bar{z}_{t+1}^{-1}\sqrt{1+\sigma_z^2}}{n(\mu + C_{l_{yy}})}\|\nabla_v R(\mathbf{x}_t, \mathbf{y}_t, \mathbf{1}v^*(\bar{x}_t))\|^2$$

$$+ \frac{\gamma_v \bar{z}_{t+1}^{-1}\sqrt{1+\sigma_z^2}(\mu + C_{l_{yy}})}{n\mu C_{l_{yy}}}\|\nabla_v R(\mathbf{x}_t, \mathbf{y}_t, \mathbf{1}v^*(\bar{x}_t)) - \nabla_v R(\mathbf{x}_t, \mathbf{1}y^*(\bar{x}_t), \mathbf{1}v^*(\bar{x}_t))\|^2$$

$$- \frac{2\mu C_{l_{yy}}\gamma_v \bar{z}_{t+1}^{-1}}{n(\mu + C_{l_{yy}})}\|\mathbf{v}_t - \mathbf{1}v^*(\bar{x}_t)\|^2$$

$$\overset{(c)}{=} -\frac{\gamma_v \bar{z}_{t+1}^{-1}\sqrt{1+\sigma_z^2}}{n(\mu + C_{l_{yy}})}\|\nabla_v R(\mathbf{x}_t, \mathbf{y}_t, \mathbf{v}_t)\|^2 - \frac{2\mu C_{l_{yy}}\gamma_v \bar{z}_{t+1}^{-1}}{n(\mu + C_{l_{yy}})}\|\mathbf{v}_t - \mathbf{1}v^*(\bar{x}_t)\|^2$$

$$+ \frac{4\gamma_v \bar{z}_{t+1}^{-1}\bar{L}_r^2\sqrt{1+\sigma_z^2}}{n(\mu + C_{l_{yy}})}\|\mathbf{x}_t - \mathbf{1}\bar{x}_t\|^2 + \left(\frac{4\gamma_v \bar{z}_{t+1}^{-1}\sqrt{1+\sigma_z^2}}{n(\mu + C_{l_{yy}})} + \frac{\gamma_v \bar{z}_{t+1}^{-1}\sqrt{1+\sigma_z^2}(\mu + C_{l_{yy}})}{n\mu C_{l_{yy}}}\right)$$

$$\cdot \|\nabla_v R(\mathbf{x}_t, \mathbf{y}_t, \mathbf{1}v^*(\bar{x}_t)) - \nabla_v R(\mathbf{x}_t, \mathbf{1}y^*(\bar{x}_t), \mathbf{1}v^*(\bar{x}_t))\|^2$$

$$\overset{(d)}{\leq} -\frac{\gamma_v \bar{z}_{t+1}^{-1}\sqrt{1+\sigma_z^2}}{n(\mu + C_{l_{yy}})}\|\nabla_v R(\mathbf{x}_t, \mathbf{y}_t, \mathbf{v}_t)\|^2 - \frac{2\mu C_{l_{yy}}\gamma_v \bar{z}_{t+1}^{-1}}{n(\mu + C_{l_{yy}})}\|\mathbf{v}_t - \mathbf{1}v^*(\bar{x}_t)\|^2$$

$$+ \frac{4\gamma_v \bar{z}_{t+1}^{-1}\bar{L}_r^2\sqrt{1+\sigma_z^2}}{n(\mu + C_{l_{yy}})}\|\mathbf{x}_t - \mathbf{1}\bar{x}_t\|^2 + \left(\frac{2\gamma_v \bar{z}_{t+1}^{-1}\sqrt{1+\sigma_z^2}}{n(\mu + C_{l_{yy}})} + \frac{\gamma_v \bar{z}_{t+1}^{-1}\sqrt{1+\sigma_z^2}(\mu + C_{l_{yy}})}{n\mu C_{l_{yy}}}\right)$$

$$\cdot \left(L_{l,2}\|\mathbf{1}v^*(\bar{x}_t)\| + L_{f,1}\right)^2 \|\mathbf{y}_t - \mathbf{1}y^*(\bar{x}_t)\|^2$$

$$\overset{(e)}{\leq} -\frac{\gamma_v \bar{z}_{t+1}^{-1}\sqrt{1+\sigma_z^2}}{n(\mu+C_{l_{yy}})}\|\nabla_v R(\mathbf{x}_t, \mathbf{y}_t, \mathbf{v}_t)\|^2 - \frac{2\mu C_{l_{yy}}\gamma_v \bar{z}_{t+1}^{-1}}{n(\mu+C_{l_{yy}})}\|\mathbf{v}_t - \mathbf{1}v^*(\bar{x}_t)\|^2$$

$$+ \frac{4\gamma_v \bar{z}_{t+1}^{-1}\bar{L}_r^2\sqrt{1+\sigma_z^2}}{n(\mu+C_{l_{yy}})}\|\mathbf{x}_t - \mathbf{1}\bar{x}_t\|^2 + \left(\frac{2\gamma_v \bar{z}_{t+1}^{-1}\sqrt{1+\sigma_z^2}}{n(\mu+C_{l_{yy}})} + \frac{\gamma_v \bar{z}_{t+1}^{-1}\sqrt{1+\sigma_z^2}(\mu+C_{l_{yy}})}{n\mu C_{l_{yy}}}\right)$$

$$\cdot \left(\frac{nL_{l,2}C_{f_y}}{\mu} + L_{f,1}\right)^2 \|\mathbf{y}_t - \mathbf{1}y^*(\bar{x}_t)\|^2, \tag{54}$$

where (a) follows from Lemma 3.11 in [Bubeck et al., 2015]; (b) uses $-\|a-b\|^2 \leq -\frac{1}{2}\|a\|^2 + \|b\|^2$ since $\|a-b+b\|^2 \leq 2\|a-b\|^2 + 2\|b\|^2$; (c) uses $\nabla_v R(\mathbf{1}\bar{x}_t, \mathbf{1}y^*(\bar{x}_t)), \mathbf{1}v^*(\bar{x}_t)) = 0$; (d) and (e) use Lemma C.3. Plugging Eq. (54) into Eq. (53), we have:

$$\frac{1}{n}\|\mathbf{v}_{t+1} - \mathbf{1}v^*(\bar{x}_t)\|^2$$

$$\leq \frac{1}{n}\left(1 - \frac{2\mu C_{l_{yy}}\gamma_v \bar{z}_{t+1}^{-1}}{\mu+C_{l_{yy}}}\right)\|\mathbf{v}_t - \mathbf{1}v^*(\bar{x}_t)\|^2 + \frac{4\gamma_v \bar{z}_{t+1}^{-1}\bar{L}_r^2\sqrt{1+\sigma_z^2}}{n(\mu+C_{l_{yy}})}\|\mathbf{x}_t - \mathbf{1}\bar{x}_t\|^2$$

$$+ \frac{\gamma_v \bar{z}_{t+1}^{-1}}{n}\left(\gamma_v \bar{z}_{t+1}^{-1}(1+\zeta_z^2) - \frac{\sqrt{1+\sigma_z^2}}{\mu+C_{l_{yy}}}\right)\|\nabla_v R(\mathbf{x}_t, \mathbf{y}_t, \mathbf{v}_t)\|^2$$

$$+ \frac{1}{n}\left(\frac{2\gamma_v\sqrt{1+\sigma_z^2}}{\mu+C_{l_{yy}}} + \frac{\gamma_v\sqrt{1+\sigma_z^2}(\mu+C_{l_{yy}})}{\mu C_{l_{yy}}}\right)\left(\frac{nL_{l,2}C_{f_y}}{\mu} + L_{f,1}\right)^2 \bar{z}_{t+1}^{-1}\|\mathbf{y}_t - \mathbf{1}y^*(\bar{x}_t)\|^2$$

$$\overset{(a)}{\leq} \frac{1}{n}\left(1 - \frac{2\mu C_{l_{yy}}\gamma_v \bar{z}_{t+1}^{-1}}{\mu+C_{l_{yy}}}\right)\|\mathbf{v}_t - \mathbf{1}v^*(\bar{x}_t)\|^2$$

$$- \frac{\gamma_v \bar{z}_{t+1}^{-1}\sqrt{1+\sigma_z^2}}{2n(\mu+C_{l_{yy}})}\|\nabla_v R(\mathbf{x}_t, \mathbf{y}_t, \mathbf{v}_t)\|^2 + \frac{4\gamma_v \bar{z}_{t+1}^{-1}\bar{L}_r^2\sqrt{1+\sigma_z^2}}{n(\mu+C_{l_{yy}})}\|\mathbf{x}_t - \mathbf{1}\bar{x}_t\|^2$$

$$+ \frac{1}{n}\left(\frac{2\gamma_v\sqrt{1+\sigma_z^2}}{\mu+C_{l_{yy}}} + \frac{\gamma_v\sqrt{1+\sigma_z^2}(\mu+C_{l_{yy}})}{\mu C_{l_{yy}}}\right)\left(\frac{nL_{l,2}C_{f_y}}{\mu} + L_{f,1}\right)^2 \bar{z}_{t+1}^{-1}\|\mathbf{y}_t - \mathbf{1}y^*(\bar{x}_t)\|^2, \tag{55}$$

where (a) follows from $\bar{z}_{t+1} \geq \bar{m}_{t+1}^v \geq C_{m^v} \geq \frac{2\gamma_v(1+\zeta_z^2)(\mu+C_{l_{yy}})}{\sqrt{1+\sigma_z^2}}$. Combining Eq. (55) with Eq. (52), we have:

$$\frac{1}{n}\|\mathbf{v}_{t+1} - \mathbf{1}v^*(\bar{x}_{t+1})\|^2$$

$$\leq (1+\hat{\lambda}_{t+1})\frac{1}{n}\left(1 - \frac{2\mu C_{l_{yy}}\gamma_v \bar{z}_{t+1}^{-1}}{\mu+C_{l_{yy}}}\right)\|\mathbf{v}_t - \mathbf{1}v^*(\bar{x}_t)\|^2$$

$$- (1+\hat{\lambda}_{t+1})\frac{\gamma_v \bar{z}_{t+1}^{-1}\sqrt{1+\sigma_z^2}}{2n(\mu+C_{l_{yy}})}\|\nabla_v R(\mathbf{x}_t, \mathbf{y}_t, \mathbf{v}_t)\|^2$$

$$+ (1+\hat{\lambda}_{t+1})\frac{1}{n}\left(\frac{2\gamma_v\sqrt{1+\sigma_z^2}}{\mu+C_{l_{yy}}} + \frac{\gamma_v\sqrt{1+\sigma_z^2}(\mu+C_{l_{yy}})}{\mu C_{l_{yy}}}\right)\left(\frac{nL_{l,2}C_{f_y}}{\mu} + L_{f,1}\right)^2$$

$$\cdot \bar{z}_{t+1}^{-1}\|\mathbf{y}_t - \mathbf{1}y^*(\bar{x}_t)\|^2 + (1+\hat{\lambda}_{t+1})\frac{4\gamma_v \bar{z}_{t+1}^{-1}\bar{L}_r^2\sqrt{1+\sigma_z^2}}{n(\mu+C_{l_{yy}})}\|\mathbf{x}_t - \mathbf{1}\bar{x}_t\|^2$$

$$+ \left(1 + \frac{1}{\hat{\lambda}_{t+1}}\right)\|v^*(\bar{x}_t) - v^*(\bar{x}_{t+1})\|^2. \tag{56}$$

By rearranging the terms in Eq. (56), we have:

$$(1+\hat{\lambda}_{t+1})\frac{\gamma_v \bar{z}_{t+1}^{-1}\sqrt{1+\sigma_z^2}}{2n(\mu+C_{l_{yy}})}\|\nabla_v R(\mathbf{x}_t, \mathbf{y}_t, \mathbf{v}_t)\|^2$$

$$\leq (1+\hat{\lambda}_{t+1})\frac{1}{n}\left(1-\frac{2\mu C_{l_{yy}}\gamma_v \bar{z}_{t+1}^{-1}}{\mu+C_{l_{yy}}}\right)\|\mathbf{v}_t - \mathbf{1}v^*(\bar{x}_t)\|^2 - \frac{1}{n}\|\mathbf{v}_{t+1}-\mathbf{1}v^*(\bar{x}_{t+1})\|^2$$

$$+(1+\hat{\lambda}_{t+1})\frac{1}{n}\left(\frac{2\gamma_v\sqrt{1+\sigma_z^2}}{\mu+C_{l_{yy}}}+\frac{\gamma_v\sqrt{1+\sigma_z^2}(\mu+C_{l_{yy}})}{\mu C_{l_{yy}}}\right)\left(\frac{nL_{l,2}C_{f_y}}{\mu}+L_{f,1}\right)^2$$

$$\cdot \bar{z}_{t+1}^{-1}\|\mathbf{y}_t - \mathbf{1}y^*(\bar{x}_t)\|^2 + (1+\hat{\lambda}_{t+1})\frac{4\gamma_v\bar{z}_{t+1}^{-1}\bar{L}_r^2\sqrt{1+\sigma_z^2}}{n(\mu+C_{l_{yy}})}\|\mathbf{x}_t - \mathbf{1}\bar{x}_t\|^2$$

$$+\left(1+\frac{1}{\hat{\lambda}_{t+1}}\right)\|v^*(\bar{x}_t)-v^*(\bar{x}_{t+1})\|^2. \tag{57}$$

We now take $\hat{\lambda}_{t+1} := \frac{2\mu C_{l_{yy}}\gamma_v\bar{z}_{t+1}^{-1}}{\mu+C_{l_{yy}}}$. Since $\bar{z}_{t+1}\geq \bar{m}_{t+1}^v \geq C_{m^v} \geq \frac{2\mu C_{l_{yy}}\gamma_v}{\mu+C_{l_{yy}}}$ in Eq. (32), we have $\hat{\lambda}_{t+1}\leq 1$. Then we get:

$$\frac{\gamma_v\bar{z}_{t+1}^{-1}\sqrt{1+\sigma_z^2}}{n}\|\nabla_v R(\mathbf{x}_t,\mathbf{y}_t,\mathbf{v}_t)\|^2$$

$$\leq (1+\hat{\lambda}_{t+1})\frac{\gamma_v\bar{z}_{t+1}^{-1}\sqrt{1+\sigma_z^2}}{n}\|\nabla_v R(\mathbf{x}_t,\mathbf{y}_t,\mathbf{v}_t)\|^2$$

$$\overset{(a)}{\leq}\frac{2(\mu+C_{l_{yy}})}{n}\left(\|\mathbf{v}_t-\mathbf{1}v^*(\bar{x}_t)\|^2 - \|\mathbf{v}_{t+1}-\mathbf{1}v^*(\bar{x}_{t+1})\|^2\right)$$

$$+\frac{4(\mu+C_{l_{yy}})}{n}\left(\frac{2\gamma_v\sqrt{1+\sigma_z^2}}{\mu+C_{l_{yy}}}+\frac{\gamma_v\sqrt{1+\sigma_z^2}(\mu+C_{l_{yy}})}{\mu C_{l_{yy}}}\right)\left(\frac{nL_{l,2}C_{f_y}}{\mu}+L_{f,1}\right)^2$$

$$\cdot \bar{z}_{t+1}^{-1}\|\mathbf{y}_t-\mathbf{1}y^*(\bar{x}_t)\|^2 + \frac{16\gamma_v\bar{z}_{t+1}^{-1}\bar{L}_r^2\sqrt{1+\sigma_z^2}}{n}\|\mathbf{x}_t-\mathbf{1}\bar{x}_t\|^2$$

$$+2(\mu+C_{l_{yy}})\left(1+\frac{\mu+C_{l_{yy}}}{2\mu C_{l_{yy}}\gamma_v\bar{z}_{t+1}^{-1}}\right)L_v^2\|\bar{x}_t-\bar{x}_{t+1}\|^2$$

$$\overset{(b)}{\leq}\frac{2(\mu+C_{l_{yy}})}{n}\left(\|\mathbf{v}_t-\mathbf{1}v^*(\bar{x}_t)\|^2-\|\mathbf{v}_{t+1}-\mathbf{1}v^*(\bar{x}_{t+1})\|^2\right)$$

$$+\frac{1}{n}\left(8\gamma_v\sqrt{1+\sigma_z^2}+\frac{4\gamma_v\sqrt{1+\sigma_z^2}(\mu+C_{l_{yy}})^2}{\mu C_{l_{yy}}}\right)\left(\frac{nL_{l,2}C_{f_y}}{\mu}+L_{f,1}\right)^2\bar{z}_{t+1}^{-1}\|\mathbf{y}_t-\mathbf{1}y^*(\bar{x}_t)\|^2$$

$$+\frac{8\bar{L}_r^2(\mu+C_{l_{yy}})\sqrt{1+\sigma_z^2}}{n\mu C_{l_{yy}}}\|\mathbf{x}_t-\mathbf{1}\bar{x}_t\|^2+\frac{2(\mu+C_{l_{yy}})^2L_v^2}{\mu C_{l_{yy}}\gamma_v\bar{z}_{t+1}^{-1}}\|\bar{x}_t-\bar{x}_{t+1}\|^2, \tag{58}$$

where (a) multiplies both sides of Eq. (57) by $2(\mu+C_{l_{yy}})$ and uses $\hat{\lambda}_{t+1}\leq 1$; (b) uses $\bar{z}_{t+1}\geq \bar{m}_{t+1}^v \geq C_{m^v}\geq \frac{2\mu C_{l_{yy}}\gamma_v}{\mu+C_{l_{yy}}}$. Take summation of Eq. (58), then we have:

$$\sum_{k=k_3}^{t}\frac{\gamma_v\bar{z}_{k+1}^{-1}\sqrt{1+\sigma_z^2}}{n}\|\nabla_v R(\mathbf{x}_k,\mathbf{y}_k,\mathbf{v}_k)\|^2$$

$$\leq \sum_{k=k_3-1}^{t}\frac{\gamma_v\bar{z}_{k+1}^{-1}\sqrt{1+\sigma_z^2}}{n}\|\nabla_v R(\mathbf{x}_k,\mathbf{y}_k,\mathbf{v}_k)\|^2$$

$$\leq \frac{2(\mu+C_{l_{yy}})}{n}\|\mathbf{v}_{k_3-1}-\mathbf{1}v^*(\bar{x}_{k_3-1})\|^2+\frac{2(\mu+C_{l_{yy}})^2L_v^2}{\mu C_{l_{yy}}\gamma_v}\sum_{k=k_3-1}^{t}\bar{z}_{k+1}\|\bar{x}_k-\bar{x}_{k+1}\|^2$$

$$+\frac{1}{n}\left(8\gamma_v\sqrt{1+\sigma_z^2}+\frac{4\gamma_v\sqrt{1+\sigma_z^2}(\mu+C_{l_{yy}})^2}{\mu C_{l_{yy}}}\right)\left(\frac{nL_{l,2}C_{f_y}}{\mu}+L_{f,1}\right)^2$$

$$\cdot\sum_{k=k_3-1}^{t}\bar{z}_{k+1}^{-1}\|\mathbf{y}_k-\mathbf{1}y^*(\bar{x}_k)\|^2+\frac{8\bar{L}_r^2(\mu+C_{l_{yy}})\sqrt{1+\sigma_z^2}}{n\mu C_{l_{yy}}}\sum_{k=k_3-1}^{t}\|\mathbf{x}_k-\mathbf{1}\bar{x}_k\|^2$$

$$\leq \frac{2(\mu + C_{l_{yy}})}{n}\|\mathbf{v}_{k_3-1} - \mathbf{1}v^*(\bar{x}_{k_3-1})\|^2 + \frac{8\bar{L}_r^2(\mu + C_{l_{yy}})\sqrt{1+\sigma_z^2}}{n\mu C_{l_{yy}}}\sum_{k=0}^{t}\|\mathbf{x}_k - \mathbf{1}\bar{x}_k\|^2$$

$$+ \frac{2(\mu + C_{l_{yy}})^2 L_v^2}{\mu C_{l_{yy}}\gamma_v}\sum_{k=k_3-1}^{t}\bar{z}_{k+1}\left\|\gamma_x \bar{q}_{k+1}^{-1}\frac{\mathbf{1}^\top}{n}\bar{\nabla}F(\mathbf{x}_k, \mathbf{y}_k, \mathbf{v}_k) - \gamma_x\frac{(\tilde{\mathbf{q}}_{k+1}^{-1})^\top}{n}\bar{\nabla}F(\mathbf{x}_k, \mathbf{y}_k, \mathbf{v}_k)\right\|^2$$

$$+ \frac{1}{n}\left(8\gamma_v\sqrt{1+\sigma_z^2} + \frac{4\gamma_v\sqrt{1+\sigma_z^2}(\mu + C_{l_{yy}})^2}{\mu C_{l_{yy}}}\right)\left(\frac{nL_{l,2}C_{f_y}}{\mu} + L_{f,1}\right)^2$$

$$\cdot \sum_{k=k_3-1}^{t}\bar{z}_{k+1}^{-1}\|\mathbf{y}_k - \mathbf{1}y^*(\bar{x}_k)\|^2$$

$$\overset{(a)}{\leq} \frac{2(\mu + C_{l_{yy}})}{n}\|\mathbf{v}_{k_3-1} - \mathbf{1}v^*(\bar{x}_{k_3-1})\|^2$$

$$+ \frac{2\gamma_x^2 L_v^2(1+\zeta_q^2)(\mu + C_{l_{yy}})^2}{n\mu C_{l_{yy}}\gamma_v}\sum_{k=k_3-1}^{t}\bar{z}_{k+1}\bar{q}_{k+1}^{-2}\left\|\bar{\nabla}F(\mathbf{x}_k, \mathbf{y}_k, \mathbf{v}_k)\right\|^2$$

$$+ \frac{1}{n}\left(8\gamma_v\sqrt{1+\sigma_z^2} + \frac{4\gamma_v\sqrt{1+\sigma_z^2}(\mu + C_{l_{yy}})^2}{\mu C_{l_{yy}}}\right)\left(\frac{nL_{l,2}C_{f_y}}{\mu} + L_{f,1}\right)^2$$

$$\cdot \sum_{k=k_3-1}^{t}[\bar{m}_{k+1}^y]^{-1}\|\mathbf{y}_k - \mathbf{1}y^*(\bar{x}_k)\|^2 + \frac{16\Delta_0^x \bar{L}_r^2(\mu + C_{l_{yy}})\sqrt{1+\sigma_z^2}}{n\mu C_{l_{yy}}(1-\rho_W)}$$

$$+ \frac{64\gamma_x^2\rho_W \bar{L}_r^2(\mu + C_{l_{yy}})(1+\zeta_q^2)\sqrt{1+\sigma_z^2}}{n\mu C_{l_{yy}}(1-\rho_W)^2}\sum_{k=0}^{t}\bar{q}_{k+1}^{-2}\|\bar{\nabla}F(\mathbf{x}_k, \mathbf{y}_k, \mathbf{v}_k)\|^2$$

$$\overset{(b)}{\leq} \frac{2(\mu + C_{l_{yy}})}{n}\|\mathbf{v}_{k_3-1} - \mathbf{1}v^*(\bar{x}_{k_3-1})\|^2$$

$$+ \frac{16\bar{L}_r^2(\mu + C_{l_{yy}})\sqrt{1+\sigma_z^2}}{n\mu C_{l_{yy}}}\left(\frac{\Delta_0^x}{1-\rho_W} + \frac{4\gamma_x^2\rho_W C_{m^x}^2(1+\zeta_q^2)}{\bar{q}_0^2(1-\rho_W)^2}\right)$$

$$+ \left(\frac{64\gamma_x^2\rho_W \bar{L}_r^2(\mu + C_{l_{yy}})(1+\zeta_q^2)\sqrt{1+\sigma_z^2}}{n\mu C_{l_{yy}}(1-\rho_W)^2} + \frac{2\gamma_x^2 L_v^2(1+\zeta_q^2)(\mu + C_{l_{yy}})^2}{n\mu C_{l_{yy}}C_{m^v}\gamma_v}\right)$$

$$\cdot \sum_{k=\min\{k_1-1,k_3-1\}}^{t}\frac{\|\bar{\nabla}F(\mathbf{x}_k, \mathbf{y}_k, \mathbf{v}_k)\|^2}{[\bar{m}_{k+1}^x]^2} + \frac{1}{n}\left(8\gamma_v\sqrt{1+\sigma_z^2} + \frac{4\gamma_v\sqrt{1+\sigma_z^2}(\mu + C_{l_{yy}})^2}{\mu C_{l_{yy}}}\right)$$

$$\cdot \left(\frac{nL_{l,2}C_{f_y}}{\mu} + L_{f,1}\right)^2\frac{1}{\mu^2}\sum_{k=k_3-1}^{t}\frac{\|\nabla_y L(\mathbf{x}_k, \mathbf{y}_k)\|^2}{\bar{m}_{k+1}^y}, \tag{59}$$

where (a) uses Lemma C.4 and (b) refers to Assumption 3.1 and $\|\bar{\nabla}f_i(x_{i,k_1-1}, y_{i,k_1-1}, v_{i,k_1-1})\|^2 \leq [m_{i,k_1}^x]^2 \leq C_{m^x}^2$. Further, the approximation term $\|\mathbf{v}_{k_3-1} - \mathbf{1}v^*(\bar{x}_{k_3-1})\|^2$ satisfies:

$$\|\mathbf{v}_{k_3-1} - \mathbf{1}v^*(\bar{x}_{k_3-1})\|^2$$

$$\overset{(a)}{\leq} \frac{1}{\mu^2}\sum_{i=1}^{n}\|\nabla_v r_i(x_{i,k_3-1}, y_{i,k_3-1}, v_{i,k_3-1}) - \nabla_v r_i(x_{i,k_3-1}, y_{i,k_3-1}, v^*(\bar{x}_{k_3-1}))\|^2$$

$$\leq \frac{2}{\mu^2}\sum_{i=1}^{n}\|\nabla_v r_i(x_{i,k_3-1}, y^*(\bar{x}_{k_3-1}), v^*(\bar{x}_{k_3-1})) - \nabla_v r_i(x_{i,k_3-1}, y_{i,k_3-1}, v^*(\bar{x}_{k_3-1}))\|^2$$

$$+ \frac{2}{\mu^2}\sum_{i=1}^{n}\|\nabla_v r_i(x_{i,k_3-1}, y_{i,k_3-1}, v_{i,k_3-1}) - \nabla_v r_i(x_{i,k_3-1}, y^*(\bar{x}_{k_3-1}), v^*(\bar{x}_{k_3-1}))\|^2$$

$$\overset{(b)}{\leq} \frac{2}{\mu^2}\left(\frac{nL_{l,2}C_{f_y}}{\mu} + L_{f,1}\right)^2\|\mathbf{y}_{k_3-1} - \mathbf{1}y^*(\bar{x}_{k_3-1})\|^2 + \frac{4}{\mu^2}\sum_{i=1}^{n}\|\nabla_v r_i(x_{i,k_3-1}, y_{i,k_3-1}, v_{i,k_3-1})\|^2$$

$$+ \frac{4\bar{L}_r^2}{\mu^2} \|\mathbf{x}_{k_3-1} - \mathbf{1}\bar{x}_{k_3-1}\|^2$$

$$\overset{(c)}{\leq} \frac{2}{\mu^2} \left( \frac{nL_{l,2}C_{f_y}}{\mu} + L_{f,1} \right)^2 \|\mathbf{y}_{k_3-1} - \mathbf{1}y^*(\bar{x}_{k_3-1})\|^2 + \frac{4}{\mu^2} \sum_{i=1}^{n} \|\nabla_v r_i(x_{i,k_3-1}, y_{i,k_3-1}, v_{i,k_3-1})\|^2$$

$$+ \frac{8\bar{L}_r^2}{\mu^2} \left( \frac{\Delta_0^x}{1-\rho_W} + \frac{4\gamma_x^2 \rho_W C_{m^x}^2 (1+\zeta_q^2)}{\bar{q}_0^2 (1-\rho_W)^2} \right), \tag{60}$$

where (a) uses the strong convexity; (b) follows from Lemma C.3 and $\nabla_v r_i(\bar{x}_{k_3-1}, y^*(\bar{x}_{k_3-1}), v^*(\bar{x}_{k_3-1})) = 0$; (c) refers to Lemma C.4. By plugging Eq. (60) into Eq. (59), we have:

$$\sum_{k=k_3}^{t} \frac{\gamma_v \bar{z}_{k+1}^{-1} \sqrt{1+\sigma_z^2}}{n} \|\nabla_v R(\mathbf{x}_k, \mathbf{y}_k, \mathbf{v}_k)\|^2$$

$$\leq \frac{4(\mu + C_{l_{yy}})}{n\mu^2} \left( \frac{nL_{l,2}C_{f_y}}{\mu} + L_{f,1} \right)^2 \|\mathbf{y}_{k_3-1} - \mathbf{1}y^*(\bar{x}_{k_3-1})\|^2$$

$$+ \frac{8(\mu + C_{l_{yy}})}{n\mu^2} \sum_{i=1}^{n} \|\nabla_v r_i(x_{i,k_3-1}, y_{i,k_3-1}, v_{i,k_3-1})\|^2$$

$$+ \left( \frac{64\gamma_x^2 \rho_W \bar{L}_r^2 (\mu + C_{l_{yy}})(1+\zeta_q^2)\sqrt{1+\sigma_z^2}}{n\mu C_{l_{yy}}(1-\rho_W)^2} + \frac{2\gamma_x^2 L_v^2 (1+\zeta_q^2)(\mu+C_{l_{yy}})^2}{n\mu C_{l_{yy}} C_{m^v} \gamma_v} \right)$$

$$\cdot \sum_{k=\min\{k_1-1,k_3-1\}}^{t} \frac{\|\bar{\nabla} F(\mathbf{x}_k, \mathbf{y}_k, \mathbf{v}_k)\|^2}{[\bar{m}_{k+1}^x]^2} + \frac{1}{n} \left( 8\gamma_v \sqrt{1+\sigma_z^2} + \frac{4\gamma_v \sqrt{1+\sigma_z^2}(\mu+C_{l_{yy}})^2}{\mu C_{l_{yy}}} \right)$$

$$\cdot \left( \frac{nL_{l,2}C_{f_y}}{\mu} + L_{f,1} \right)^2 \frac{1}{\mu^2} \sum_{k=k_3-1}^{t} \frac{\|\nabla_y L(\mathbf{x}_k, \mathbf{y}_k)\|^2}{\bar{m}_{k+1}^y}$$

$$+ \left( \frac{16\Delta_0^x}{1-\rho_W} + \frac{64\gamma_x^2 \rho_W C_{m^x}^2 (1+\zeta_q^2)}{\bar{q}_0^2 (1-\rho_W)^2} \right) \left( \frac{\bar{L}_r^2 (\mu + C_{l_{yy}})\sqrt{1+\sigma_z^2}}{n\mu C_{l_{yy}}} + \frac{\bar{L}_r^2 (\mu + C_{l_{yy}})}{n\mu^2} \right). \tag{61}$$

Our next step is bounding $\|\mathbf{y}_{k_3-1} - \mathbf{1}y^*(\bar{x}_{k_3-1})\|^2$ on the RHS of Eq. (61) in two cases. The first case is $\bar{m}_{k_3}^y \leq C_{m^y}$. In this case, by using strong convexity of $l$ and the definition of $\bar{m}_{k_3}^y$, we can easily have:

$$\|\mathbf{y}_{k_3-1} - \mathbf{1}y^*(\bar{x}_{k_3-1})\|^2 \leq \frac{1}{\mu^2} \|\nabla_y l(\mathbf{x}_{k_3-1}, \mathbf{y}_{k_3-1})\|^2 \leq \frac{nC_{m^y}^2}{\mu^2}. \tag{62}$$

For the second case, when $\bar{m}_{k_3}^y > C_{m^y}$, note that $k_2$ exists and $k_3 > k_2$ by Lemma C.5. By plugging $\bar{\lambda}_{k_3-1} := \frac{\gamma_y \bar{u}_{k_3-1}^{-1} \mu L_{l,1}}{\mu + L_{l,1}}$ into Eq. (47) and noting $\bar{\lambda}_{k_3-1} \leq 1$, we have:

$$\|\mathbf{y}_{k_3-1} - \mathbf{1}y^*(\bar{x}_{k_3-1})\|^2$$

$$\leq \|\mathbf{y}_{k_3-2} - \mathbf{1}y^*(\bar{x}_{k_3-2})\|^2 + \frac{2(\mu + L_{l,1})}{\gamma_y \bar{u}_{k_3-1}^{-1} \mu L_{l,1}} \|y^*(\bar{x}_{k_3-2}) - y^*(\bar{x}_{k_3-1})\|^2$$

$$+ \left( \frac{2\gamma_y \bar{u}_{t+1}^{-1} L_{l,1}(\mu + L_{l,1})\sqrt{1+\sigma_u^2}}{n\mu} + \frac{4\gamma_y \bar{u}_{t+1}^{-1} L_{l,1}^2 \sqrt{1+\sigma_u^2}}{n(\mu + L_{l,1})} \right) \|\mathbf{x}_{k_3-2} - \mathbf{1}\bar{x}_{k_3-2}\|^2$$

$$\overset{(a)}{\leq} \|\mathbf{y}_{k_3-2} - \mathbf{1}y^*(\bar{x}_{k_3-2})\|^2 + \frac{2(\mu + L_{l,1})L_y^2}{\gamma_y \bar{u}_{k_3-1}^{-1} \mu L_{l,1}} \|\bar{x}_{k_3-2} - \bar{x}_{k_3-1}\|^2$$

$$+ \left( \frac{2(\mu + L_{l,1})^2 \sqrt{1+\sigma_u^2}}{n\mu^2} + \frac{4L_{l,1}\sqrt{1+\sigma_u^2}}{n\mu} \right) \sum_{k=0}^{k_3-2} \|\mathbf{x}_k - \mathbf{1}\bar{x}_k\|^2$$

$$\overset{(b)}{\leq} \|\mathbf{y}_{k_2-1}-\mathbf{1}y^*(\bar{x}_{k_2-1})\|^2+\frac{4\gamma_x^2L_y^2(1+\zeta_q^2)(\mu+L_{l,1})}{n\gamma_y\mu L_{l,1}}\bar{u}_{k_3-1}\bar{q}_{k_3-1}^{-2}\|\bar{\nabla}F(\mathbf{x}_{k_3-2},\mathbf{y}_{k_3-2},\mathbf{v}_{k_3-2})\|^2$$

$$+\frac{16\gamma_x^2\rho_W(1+\zeta_q^2)}{(1-\rho_W)^2}\left(\frac{(\mu+L_{l,1})^2\sqrt{1+\sigma_u^2}}{n\mu^2}+\frac{2L_{l,1}\sqrt{1+\sigma_u^2}}{n\mu}\right)\sum_{k=0}^{k_3-2}\bar{q}_{k+1}^{-2}\|\bar{\nabla}F(\mathbf{x}_k,\mathbf{y}_k,\mathbf{v}_k)\|^2$$

$$+\frac{4\Delta_0^x(\mu+L_{l,1})^2\sqrt{1+\sigma_u^2}}{n\mu^2(1-\rho_W)}+\frac{8\Delta_0^xL_{l,1}\sqrt{1+\sigma_u^2}}{n\mu(1-\rho_W)}$$

$$\overset{(c)}{\leq}\frac{nC_{m^y}^2}{\mu^2}+\left(\frac{4\Delta_0^x}{1-\rho_W}+\frac{16\gamma_x^2\rho_WC_{m^x}^2(1+\zeta_q^2)}{\bar{q}_0^2(1-\rho_W)^2}\right)\left(\frac{(\mu+L_{l,1})^2\sqrt{1+\sigma_u^2}}{n\mu^2}+\frac{2L_{l,1}\sqrt{1+\sigma_u^2}}{n\mu}\right)$$

$$+\left[\frac{16\gamma_x^2\rho_W(1+\zeta_q^2)}{\bar{z}_0^2(1-\rho_W)^2}\left(\frac{(\mu+L_{l,1})^2\sqrt{1+\sigma_u^2}}{n\mu^2}+\frac{2L_{l,1}\sqrt{1+\sigma_u^2}}{n\mu}\right)\right.$$

$$\left.+\frac{4\gamma_x^2L_y^2(1+\zeta_q^2)(\mu+L_{l,1})}{n\bar{z}_0\gamma_y\mu L_{l,1}}\right]\sum_{k=\min\{k_1-1,k_2-1\}}^{k_3-2}\frac{\|\bar{\nabla}F(\mathbf{x}_k,\mathbf{y}_k,\mathbf{v}_k)\|^2}{[\bar{m}_{k+1}^x]^2}, \tag{63}$$

where (a) uses $L_y$-Lipschitz continuous and $\bar{m}_{k_3+1}^y > C_{m^y} \geq \frac{\gamma_y\mu L_{l,1}}{\mu+L_{l,1}}$; (b) refers to Lemma C.4; (c) follows from Eq. (62) by replacing $k_3$ with $k_2$ since $\bar{m}_{k_2}^y \leq C_{m^y}$ and $\|\bar{\nabla}f_i(x_{i,k_1-1},y_{i,k_1-1},v_{i,k_1-1})\|^2 \leq [m_{i,k_1}^x]^2 \leq C_{m^x}^2$. By combining Eq. (62) and Eq. (63), we obtain a general upper bound for $\|\mathbf{y}_{k_3-1}-\mathbf{1}y^*(\bar{x}_{k_3-1})\|^2$ as:

$$\|\mathbf{y}_{k_3-1}-\mathbf{1}y^*(\bar{x}_{k_3-1})\|^2$$

$$\leq\frac{nC_{m^y}^2}{\mu^2}+\left(\frac{4\Delta_0^x}{1-\rho_W}+\frac{16\gamma_x^2\rho_WC_{m^x}^2(1+\zeta_q^2)}{\bar{q}_0^2(1-\rho_W)^2}\right)\left(\frac{(\mu+L_{l,1})^2\sqrt{1+\sigma_u^2}}{n\mu^2}+\frac{2L_{l,1}\sqrt{1+\sigma_u^2}}{n\mu}\right)$$

$$+\left[\frac{16\gamma_x^2\rho_W(1+\zeta_q^2)}{\bar{z}_0^2(1-\rho_W)^2}\left(\frac{(\mu+L_{l,1})^2\sqrt{1+\sigma_u^2}}{n\mu^2}+\frac{2L_{l,1}\sqrt{1+\sigma_u^2}}{n\mu}\right)\right.$$

$$\left.+\frac{4\gamma_x^2L_y^2(1+\zeta_q^2)(\mu+L_{l,1})}{n\bar{z}_0\gamma_y\mu L_{l,1}}\right]\sum_{k=\min\{k_1-1,k_2-1\}}^{k_3-2}\frac{\|\bar{\nabla}F(\mathbf{x}_k,\mathbf{y}_k,\mathbf{v}_k)\|^2}{[\bar{m}_{k+1}^x]^2}, \tag{64}$$

where we define $\sum_{t=m}^n p_t = 0$ for any $m > n$ and non-negative sequence $\{p_t\}$. By plugging Eq. (64) into Eq. (59) and using $\|\nabla_v r_i(x_{i,k_3-1},y_{i,k_3-1},v_{i,k_3-1})\|^2 \leq [\bar{m}_{k_3}^v]^2 \leq C_{m^v}^2$, we have:

$$\sum_{k=k_3}^t\frac{\gamma_v\bar{z}_{k+1}^{-1}\sqrt{1+\sigma_z^2}}{n}\|\nabla_vR(\mathbf{x}_k,\mathbf{y}_k,\mathbf{v}_k)\|^2$$

$$\leq\frac{4C_{m^y}^2(\mu+C_{l_{yy}})}{\mu^4}\left(\frac{nL_{l,2}C_{f_y}}{\mu}+L_{f,1}\right)^2$$

$$+\frac{8(\mu+C_{l_{yy}})C_{m^v}^2}{\mu^2}+\frac{16(\mu+C_{l_{yy}})}{n\mu^2}\left(\frac{nL_{l,2}C_{f_y}}{\mu}+L_{f,1}\right)^2$$

$$\cdot\left(\frac{\Delta_0^x}{1-\rho_W}+\frac{4\gamma_x^2\rho_WC_{m^x}^2(1+\zeta_q^2)}{\bar{q}_0^2(1-\rho_W)^2}\right)\left(\frac{(\mu+L_{l,1})^2\sqrt{1+\sigma_u^2}}{n\mu^2}+\frac{2L_{l,1}\sqrt{1+\sigma_u^2}}{n\mu}\right)$$

$$+\frac{16(\mu+C_{l_{yy}})}{n\mu^2}\left(\frac{nL_{l,2}C_{f_y}}{\mu}+L_{f,1}\right)^2\left[\frac{4\gamma_x^2\rho_W(1+\zeta_q^2)}{\bar{z}_0^2(1-\rho_W)^2}\left(\frac{(\mu+L_{l,1})^2\sqrt{1+\sigma_u^2}}{n\mu^2}+\frac{2L_{l,1}\sqrt{1+\sigma_u^2}}{n\mu}\right)\right.$$

$$\left.+\frac{\gamma_x^2L_y^2(1+\zeta_q^2)(\mu+L_{l,1})}{n\bar{z}_0\gamma_y\mu L_{l,1}}\right]\sum_{k=\min\{k_1-1,k_2-1\}}^{k_3-2}\frac{\|\bar{\nabla}F(\mathbf{x}_k,\mathbf{y}_k,\mathbf{v}_k)\|^2}{[\bar{m}_{k+1}^x]^2}$$

$$+\left(\frac{64\gamma_x^2\rho_W\bar{L}_r^2(\mu+C_{l_{yy}})(1+\zeta_q^2)\sqrt{1+\sigma_z^2}}{n\mu C_{l_{yy}}(1-\rho_W)^2}+\frac{2\gamma_x^2L_v^2(1+\zeta_q^2)(\mu+C_{l_{yy}})^2}{n\mu C_{l_{yy}}C_{m^v}\gamma_v}\right)$$

$$\cdot \sum_{k=\min\{k_1-1,k_3-1\}}^{t} \frac{\|\bar{\nabla}F(\mathbf{x}_k,\mathbf{y}_k,\mathbf{v}_k)\|^2}{[\bar{m}_{k+1}^x]^2} + \frac{1}{n}\left(8\gamma_v\sqrt{1+\sigma_z^2} + \frac{4\gamma_v\sqrt{1+\sigma_z^2}(\mu+C_{l_{yy}})^2}{\mu C_{l_{yy}}}\right)$$

$$\cdot \left(\frac{nL_{l,2}C_{f_y}}{\mu} + L_{f,1}\right)^2 \frac{1}{\mu^2}\sum_{k=k_3-1}^{t}\frac{\|\nabla_y L(\mathbf{x}_k,\mathbf{y}_k)\|^2}{\bar{m}_{k+1}^y}$$

$$+ \left(\frac{16\Delta_0^x}{1-\rho_W} + \frac{64\gamma_x^2\rho_W C_{m^x}^2(1+\zeta_q^2)}{\bar{q}_0^2(1-\rho_W)^2}\right)\left(\frac{\bar{L}_r^2(\mu+C_{l_{yy}})\sqrt{1+\sigma_z^2}}{n\mu C_{l_{yy}}} + \frac{\bar{L}_r^2(\mu+C_{l_{yy}})}{n\mu^2}\right). \quad (65)$$

Then, the proof is complete. $\qquad\square$

## D.5 The Upper Bounds of $\bar{m}_t^y$ and $\bar{z}_t$

Supported by Lemma D.2 and Lemma D.3, we derive upper bounds of $\bar{m}_t^y$ and $\bar{z}_t$.

**Lemma D.4.** *Suppose the total iteration rounds of Algorithm 1 is $T$. Under Assumption 3.1 and Assumption 3.2, if $k_2$ in Lemma C.5 exists within $T$ iterations, we have:*

$$\bar{m}_{t+1}^y \leq \begin{cases} C_{m^y}, & t < k_2, \\ C_{m^y} + c_0 + d_0\sum_{k=\min\{k_1,k_2\}}^{t}\frac{\|\bar{\nabla}F(\mathbf{x}_k,\mathbf{y}_k,\mathbf{v}_k)\|^2}{[\bar{m}_{k+1}^x]^2\max\{\bar{m}_{k+1}^v,\bar{m}_{k+1}^y\}}, & t \geq k_2. \end{cases} \quad (66)$$

*where $c_0, d_0$ are defined as:*

$$c_0 := \frac{2C_{m^y}^2(\mu+L_{l,1})}{\mu^2\gamma_y\sqrt{1+\sigma_u^2}} + \left(\frac{8\Delta_0^x}{n(1-\rho_W)} + \frac{32\gamma_x^2\rho_W C_{m^x}^2(1+\zeta_q^2)}{n\bar{q}_0^2(1-\rho_W)^2}\right)\left(\frac{(\mu+L_{l,1})^3}{\gamma_y\mu^2} + \frac{2L_{l,1}(\mu+L_{l,1})}{\gamma_y\mu}\right)$$

$$+ \frac{32\gamma_x^2\rho_W(1+\zeta_q^2)}{n\bar{z}_0(1-\rho_W)^2}\left(\frac{(\mu+L_{l,1})^3}{\gamma_y\mu^2} + \frac{2L_{l,1}(\mu+L_{l,1})}{\gamma_y\mu}\right) + \frac{8\gamma_x^2 L_y^2(1+\zeta_q^2)(\mu+L_{l,1})^2}{n\gamma_y^2\mu L_{l,1}\bar{z}_0\sqrt{1+\sigma_u^2}},$$

$$d_0 := \left[\frac{32\gamma_x^2\rho_W(1+\zeta_q^2)}{n(1-\rho_W)^2}\left(\frac{(\mu+L_{l,1})^3}{\gamma_y\mu^2} + \frac{2L_{l,1}(\mu+L_{l,1})}{\gamma_y\mu}\right) + \frac{8\gamma_x^2 L_y^2(1+\zeta_q^2)(\mu+L_{l,1})^2}{n\gamma_y^2\mu L_{l,1}\sqrt{1+\sigma_u^2}}\right],$$
$$(67)$$

*When such $k_2$ does not exist, $\bar{m}_{t+1}^y \leq C_{m^y}$ holds for any $t < T$.*

*Proof.* According to Lemma C.5, the proof can be split into the following three cases:

**Case 1:** $k_2$ does not exist. In this case, based on Lemma C.5, we have $\bar{m}_T^y \leq C_{m^y}$, and hence $\bar{m}_{t+1}^y \leq C_{m^y}$ for any $t < T$ because $\bar{m}_t^y$ is non-decreasing with $t$.

**Case 2:** $k_2$ exists and $t < k_2$: In this case, based on Lemma C.5, we have $\bar{m}_{t+1}^y \leq C_{m^y}$.

**Case 3:** $k_2$ exists and $t \geq k_2$: Using telescoping, we have:

$$\bar{m}_{t+1}^y \overset{(a)}{=} \bar{m}_t^y + \frac{\|\nabla_y L(\mathbf{x}_t,\mathbf{y}_t)\|^2}{n(\bar{m}_{t+1}^y + \bar{m}_t^y)}$$

$$\leq \bar{m}_t^y + \frac{\|\nabla_y L(\mathbf{x}_t,\mathbf{y}_t)\|^2}{n\bar{m}_{t+1}^y}$$

$$= \bar{m}_{k_2}^y + \sum_{k=k_2}^{t-1}\frac{\|\nabla_y L(\mathbf{x}_k,\mathbf{y}_k)\|^2}{n(\bar{m}_{t+1}^y + \bar{m}_t^y)} + \frac{\|\nabla_y L(\mathbf{x}_t,\mathbf{y}_t)\|^2}{n\bar{m}_{t+1}^y}$$

$$\leq \bar{m}_{k_2}^y + \sum_{k=k_2}^{t}\frac{\|\nabla_y L(\mathbf{x}_k,\mathbf{y}_k)\|^2}{n\bar{m}_{k+1}^y}$$

$$\overset{(b)}{\leq} C_{m^y} + \frac{2C_{m^y}^2(\mu+L_{l,1})}{\mu^2\gamma_y\sqrt{1+\sigma_u^2}}$$

$$+ \left( \frac{8\Delta_0^x}{n(1-\rho_W)} + \frac{32\gamma_x^2 \rho_W C_{m^x}^2 (1+\zeta_q^2)}{n\bar{q}_0^2 (1-\rho_W)^2} \right) \left( \frac{(\mu+L_{l,1})^3}{\gamma_y \mu^2} + \frac{2L_{l,1}(\mu+L_{l,1})}{\gamma_y \mu} \right)$$

$$+ \frac{32\gamma_x^2 \rho_W (1+\zeta_q^2)}{n\bar{z}_0 (1-\rho_W)^2} \left( \frac{(\mu+L_{l,1})^3}{\gamma_y \mu^2} + \frac{2L_{l,1}(\mu+L_{l,1})}{\gamma_y \mu} \right) + \frac{8\gamma_x^2 L_y^2 \left(1+\zeta_q^2\right)(\mu+L_{l,1})^2}{n\gamma_y^2 \mu L_{l,1} \bar{z}_0 \sqrt{1+\sigma_u^2}}$$

$$+ \left[ \frac{32\gamma_x^2 \rho_W (1+\zeta_q^2)}{n(1-\rho_W)^2} \left( \frac{(\mu+L_{l,1})^3}{\gamma_y \mu^2} + \frac{2L_{l,1}(\mu+L_{l,1})}{\gamma_y \mu} \right) \right.$$

$$\left. + \frac{8\gamma_x^2 L_y^2 \left(1+\zeta_q^2\right)(\mu+L_{l,1})^2}{n\gamma_y^2 \mu L_{l,1} \sqrt{1+\sigma_u^2}} \right] \sum_{k=\min\{k_1,k_2\}}^{t} \frac{\|\bar{\nabla} F(\mathbf{x}_k, \mathbf{y}_k, \mathbf{v}_k)\|^2}{[\bar{m}_{k+1}^x]^2 \max\left\{\bar{m}_{k+1}^v, \bar{m}_{k+1}^y\right\}}, \tag{68}$$

where (a) employs $(\bar{m}_{t+1}^y + \bar{m}_t^y)(\bar{m}_{t+1}^y - \bar{m}_t^y) = [\bar{m}_{t+1}^y]^2 - [\bar{m}_t^y]^2$ and (b) uses Lemma D.2. Thus, the proof is complete. $\qquad\square$

**Lemma D.5.** *Under Assumption 3.1 and Assumption 3.2, suppose the total iteration rounds of Algorithm 1 is $T$. If at least one of $k_2$ and $k_3$ in Lemma C.5 exists, we denote $k_{min} := \min\{k_2, k_3\}$. Then we have the upper bound of $\bar{z}_t$ as:*

$$\bar{z}_t \leq \begin{cases} C_z, & t \leq k_{min}, \\ a_1 \log(t) + b_1, & t > k_{min}, \end{cases} \tag{69}$$

*where $a_1, b_1$ are defined as:*

$$a_1 := 6a_0, \quad b_1 := 4a_0 \log\left(1 + \frac{nC_{l_{xy}}\bar{b} + nC_{f_x} + \bar{m}_0^x}{nC_{l_{xy}}\bar{a}}\right) + 4a_0 \log\left(nC_{l_{xy}}\bar{a}\right) + 4a_0 + 2b_0, \tag{70}$$

*in which we define constants*

$$\bar{a} := \frac{\sqrt{2n}}{\mu}, \quad \bar{b} := \frac{\sqrt{2n}C_{f_y}}{\mu},$$

$$a_0 := \left( \frac{1}{\mu^2} \left( 8 + \frac{4(\mu+C_{l_{yy}})^2}{\mu C_{l_{yy}}} \right) \left( \frac{nL_{l,2}C_{f_y}}{\mu} + L_{f,1} \right)^2 + 1 \right)$$

$$\cdot \left( \frac{32\gamma_x^2 \rho_W (1+\zeta_q^2)}{n(1-\rho_W)^2} \left( \frac{(\mu+L_{l,1})^3}{\gamma_y \mu^2} + \frac{2L_{l,1}(\mu+L_{l,1})}{\gamma_y \mu} \right) + \frac{8\gamma_x^2 L_y^2 \left(1+\zeta_q^2\right)(\mu+L_{l,1})^2}{n\gamma_y^2 \mu L_{l,1} \sqrt{1+\sigma_u^2}} \right)$$

$$+ \frac{16(\mu+C_{l_{yy}})}{n\gamma_v \mu^2 \sqrt{1+\sigma_u^2}} \left( \frac{nL_{l,2}C_{f_y}}{\mu} + L_{f,1} \right)^2 \left( \frac{4\gamma_x^2 \rho_W (1+\zeta_q^2)}{\bar{z}_0^2 (1-\rho_W)^2} \right.$$

$$\cdot \left( \frac{(\mu+L_{l,1})^2 \sqrt{1+\sigma_u^2}}{n\mu^2} + \frac{2L_{l,1}\sqrt{1+\sigma_u^2}}{n\mu} \right) + \frac{\gamma_x^2 L_y^2 (1+\zeta_q^2)(\mu+L_{l,1})}{n\bar{z}_0 \gamma_y \mu L_{l,1}} \right)$$

$$+ \frac{64\gamma_x^2 \rho_W \bar{L}_r^2 (\mu+C_{l_{yy}})(1+\zeta_q^2)}{n\mu C_{l_{yy}} \gamma_v (1-\rho_W)^2} + \frac{2\gamma_x^2 L_v^2 (1+\zeta_q^2)(\mu+C_{l_{yy}})^2}{n\mu C_{l_{yy}} \bar{m}_0^v \gamma_v^2 \sqrt{1+\sigma_z^2}},$$

$$b_0 := C_{m^y} + C_{m^v} + \frac{4C_{m^y}^2 (\mu+C_{l_{yy}})}{\mu^4 \gamma_v \sqrt{1+\sigma_z^2}} \left( \frac{nL_{l,2}C_{f_y}}{\mu} + L_{f,1} \right)^2$$

$$+ \frac{8(\mu+C_{l_{yy}})C_{m^v}^2}{\mu^2 \gamma_v \sqrt{1+\sigma_z^2}} + \frac{16(\mu+C_{l_{yy}})}{n\gamma_v \mu^2} \left( \frac{nL_{l,2}C_{f_y}}{\mu} + L_{f,1} \right)^2$$

$$\cdot \left( \frac{\Delta_0^x}{1-\rho_W} + \frac{4\gamma_x^2 \rho_W C_{m^x}^2 (1+\zeta_q^2)}{\bar{q}_0^2 (1-\rho_W)^2} \right) \left( \frac{(\mu+L_{l,1})^2}{n\mu^2} + \frac{2L_{l,1}}{n\mu} \right)$$

$$+ \left( \frac{16\Delta_0^x}{n(1-\rho_W)} + \frac{64\gamma_x^2 \rho_W C_{m^x}^2 (1+\zeta_q^2)}{n\bar{q}_0^2 (1-\rho_W)^2} \right) \left( \frac{\bar{L}_r^2 (\mu+C_{l_{yy}})}{\mu C_{l_{yy}} \gamma_v} + \frac{\bar{L}_r^2 (\mu+C_{l_{yy}})}{\mu^2 \gamma_v \sqrt{1+\sigma_z^2}} \right)$$

$$+ \left[\frac{1}{\mu^2}\left(8 + \frac{4(\mu + C_{l_{yy}})^2}{\mu C_{l_{yy}}}\right)\left(\frac{nL_{l,2}C_{f_y}}{\mu} + L_{f,1}\right)^2 + 1\right]\left(\frac{C_{m^y}^2}{\bar{m}_0^y} - \bar{m}_0^y\right)$$

$$+ \left[\frac{1}{\mu^2}\left(8 + \frac{4(\mu + C_{l_{yy}})^2}{\mu C_{l_{yy}}}\right)\left(\frac{nL_{l,2}C_{f_y}}{\mu} + L_{f,1}\right)^2 + 1\right]\left(\frac{2C_{m^y}^2(\mu + L_{l,1})}{\mu^2\gamma_y\sqrt{1 + \sigma_u^2}}\right)$$

$$+ \left(\frac{8\Delta_0^x}{n(1 - \rho_W)} + \frac{32\gamma_x^2\rho_W C_{m^x}^2(1 + \zeta_q^2)}{n\bar{q}_0^2(1 - \rho_W)^2}\right)\left(\frac{(\mu + L_{l,1})^3}{\gamma_y\mu^2} + \frac{2L_{l,1}(\mu + L_{l,1})}{\gamma_y\mu}\right)$$

$$+ \frac{32\gamma_x^2\rho_W(1 + \zeta_q^2)}{n\bar{z}_0(1 - \rho_W)^2}\left(\frac{(\mu + L_{l,1})^3}{\gamma_y\mu^2} + \frac{2L_{l,1}(\mu + L_{l,1})}{\gamma_y\mu}\right) + \frac{8\gamma_x^2 L_y^2(1 + \zeta_q^2)(\mu + L_{l,1})^2}{n\gamma_y^2\mu L_{l,1}\bar{z}_0\sqrt{1 + \sigma_u^2}}\right).$$

$$(71)$$

*When neither $k_2$ nor $k_3$ exists, we have $\bar{z}_t \leq C_z$ for all $t \leq T$.*

*Proof.* To begin with, we first show the following result as the first two lines of Eq. (68): since $\bar{m}_t^y$ and $\bar{m}_t^v$ are positive and increasing monotonically with $t$, we can easily have:

$$0 \leq \min\{[\bar{m}_{t+1}^y]^2, [\bar{m}_{t+1}^v]^2\} - \min\{[\bar{m}_t^y]^2, [\bar{m}_t^v]^2\}$$
$$= \left([\bar{m}_{t+1}^y]^2 + [\bar{m}_{t+1}^v]^2 - \max\{[\bar{m}_{t+1}^y]^2, [\bar{m}_{t+1}^v]^2\}\right) - \left([\bar{m}_t^y]^2 + [\bar{m}_t^v]^2 - \max\{[\bar{m}_t^y]^2, [\bar{m}_t^v]^2\}\right)$$
$$\overset{(a)}{=} \left([\bar{m}_{t+1}^y]^2 + [\bar{m}_{t+1}^v]^2\right) - \left([\bar{m}_t^y]^2 + [\bar{m}_t^v]^2\right) - (\bar{z}_{t+1}^2 - \bar{z}_t^2), \tag{72}$$

where (a) uses the definition $\bar{z}_t := \max\{\bar{m}_t^v, \bar{m}_t^y\}$. Similar to Eq. (68), we have:

$$\bar{z}_{t+1}^2 - \bar{z}_t^2 \leq ([\bar{m}_{t+1}^y]^2 - [\bar{m}_t^y]^2) + ([\bar{m}_{t+1}^v]^2 - [\bar{m}_t^v]^2) = \frac{\|\nabla_y L(\mathbf{x}_t, \mathbf{y}_t)\|^2}{n} + \frac{\|\nabla_v R(\mathbf{x}_t, \mathbf{y}_t, \mathbf{v}_t)\|^2}{n}, \tag{73}$$

which indicates that

$$\bar{z}_{t+1} \leq \bar{z}_t + \frac{\|\nabla_y L(\mathbf{x}_t, \mathbf{y}_t)\|^2}{n(\bar{z}_{t+1} + \bar{z}_t)} + \frac{\|\nabla_v R(\mathbf{x}_t, \mathbf{y}_t, \mathbf{v}_t)\|^2}{n(\bar{z}_{t+1} + \bar{z}_t)}$$
$$\leq \bar{z}_t + \frac{\|\nabla_y L(\mathbf{x}_t, \mathbf{y}_t)\|^2}{n(\bar{m}_{t+1}^y + \bar{m}_t^y)} + \frac{\|\nabla_v R(\mathbf{x}_t, \mathbf{y}_t, \mathbf{v}_t)\|^2}{n\bar{z}_{t+1}}$$
$$\leq \bar{z}_t + \frac{\|\nabla_y L(\mathbf{x}_t, \mathbf{y}_t)\|^2}{n\bar{m}_{t+1}^y} + \frac{\|\nabla_v R(\mathbf{x}_t, \mathbf{y}_t, \mathbf{v}_t)\|^2}{n\bar{z}_{t+1}}. \tag{74}$$

Note that, to simplify the proof, we define $\sum_{t=m}^n p_t = 0$ for any $m > n$ and non-negative sequence $\{p_t\}$. According to the definitions of $k_2$ and $k_3$ in Lemma C.5, the proof can be split into the following four cases.

**Case 1: Neither $k_2$ nor $k_3$ exists.** For any $t \in (0, T)$, we can easily have $\bar{z}_t = \max\{\bar{m}_t^y, \bar{m}_t^v\} \leq \max\{C_{m^y}, C_{m^v}\} \leq C_z$.

**Case 2: $k_2$ exists but $k_3$ does not.** By using the fourth line of Eq. (68), for any $t \in (0, T)$, we have:

$$\bar{z}_{t+1} \leq \bar{m}_{t+1}^y + \bar{m}_{t+1}^v \leq C_{m^y} + \sum_{k=k_2}^t \frac{\|\nabla_y L(\mathbf{x}_k, \mathbf{y}_k)\|^2}{n\bar{m}_{k+1}^y} + C_{m^v}, \tag{75}$$

where we take $\sum_{k=k_2}^t \frac{\|\nabla_y L(\mathbf{x}_k, \mathbf{y}_k)\|^2}{\bar{m}_{k+1}^y} = 0$ for any $t < k_2$.

**Case 3: $k_3$ exists but $k_2$ does not.** From the second line of Eq. (74), for any $t \in (0, T)$, we have:

$$\bar{z}_{t+1} \leq \bar{z}_t + \frac{\|\nabla_y L(\mathbf{x}_t, \mathbf{y}_t)\|^2}{n(\bar{m}_{t+1}^y + \bar{m}_t^y)} + \frac{\|\nabla_v R(\mathbf{x}_t, \mathbf{y}_t, \mathbf{v}_t)\|^2}{n\bar{z}_{t+1}}$$
$$\leq \bar{z}_{k_3} + \sum_{k=k_3}^t \frac{\|\nabla_y L(\mathbf{x}_k, \mathbf{y}_k)\|^2}{n(\bar{m}_{k+1}^y + \bar{m}_k^y)} + \sum_{k=k_3}^t \frac{\|\nabla_v R(\mathbf{x}_k, \mathbf{y}_k, \mathbf{v}_k)\|^2}{n\bar{z}_{k+1}}$$

$$\leq \bar{m}_{k_3}^y + \bar{m}_{k_3}^v + \sum_{k=k_3}^t \frac{\|\nabla_y L(\mathbf{x}_k, \mathbf{y}_k)\|^2}{n(\bar{m}_{k+1}^y + \bar{m}_k^y)} + \sum_{k=k_3}^t \frac{\|\nabla_v R(\mathbf{x}_k, \mathbf{y}_k, \mathbf{v}_k)\|^2}{n\bar{z}_{k+1}}$$

$$\overset{(a)}{=} \bar{m}_{t+1}^y + \bar{m}_{k_3}^v + \sum_{k=k_3}^t \frac{\|\nabla_v R(\mathbf{x}_k, \mathbf{y}_k, \mathbf{v}_k)\|^2}{n\bar{z}_{k+1}}$$

$$\leq C_{m^y} + C_{m^v} + \sum_{k=k_3}^t \frac{\|\nabla_v R(\mathbf{x}_k, \mathbf{y}_k, \mathbf{v}_k)\|^2}{n\bar{z}_{k+1}}, \tag{76}$$

where we take $\sum_{k=k_3}^t \frac{\|\nabla_v R(\mathbf{x}_k, \mathbf{y}_k, \mathbf{v}_k)\|^2}{\bar{z}_{k+1}} = 0$ for any $t < k_3$; (a) uses the first line of Eq. (68).

**Case 4: Both $k_2$ and $k_3$ exist.** From the third line of Eq. (76), for any $t \in (0, T)$, we have:

$$\bar{z}_{t+1} \leq \bar{m}_{k_3}^y + \bar{m}_{k_3}^v + \sum_{k=k_3}^t \frac{\|\nabla_y L(\mathbf{x}_k, \mathbf{y}_k)\|^2}{n\bar{m}_{k+1}^y} + \sum_{k=k_3}^t \frac{\|\nabla_v R(\mathbf{x}_k, \mathbf{y}_k, \mathbf{v}_k)\|^2}{n\bar{z}_{k+1}}$$

$$\overset{(a)}{\leq} \bar{m}_{k_2}^y + \sum_{k=k_2}^{k_3-1} \frac{\|\nabla_y L(\mathbf{x}_k, \mathbf{y}_k)\|^2}{n\bar{m}_{k+1}^y}$$

$$+ C_{m^v} + \sum_{k=k_3}^t \frac{\|\nabla_y L(\mathbf{x}_k, \mathbf{y}_k)\|^2}{n\bar{m}_{k+1}^y} + \sum_{k=k_3}^t \frac{\|\nabla_v R(\mathbf{x}_k, \mathbf{y}_k, \mathbf{v}_k)\|^2}{n\bar{z}_{k+1}}$$

$$= C_{m^y} + C_{m^v} + \sum_{k=k_2}^t \frac{\|\nabla_y L(\mathbf{x}_k, \mathbf{y}_k)\|^2}{n\bar{m}_{k+1}^y} + \sum_{k=k_3}^t \frac{\|\nabla_v R(\mathbf{x}_k, \mathbf{y}_k, \mathbf{v}_k)\|^2}{n\bar{z}_{k+1}}, \tag{77}$$

where (a) uses the fourth line of Eq. (68); we take $\sum_{k=k_2}^{k_3-1} \frac{\|\nabla_y L(\mathbf{x}_k, \mathbf{y}_k)\|^2}{\bar{m}_{k+1}^y} = 0$ when $k_2 \geq k_3$, $\sum_{k=k_2}^t \frac{\|\nabla_y L(\mathbf{x}_k, \mathbf{y}_k)\|^2}{\bar{m}_{k+1}^y} = 0$ for any $t < k_2$, and $\sum_{k=k_3}^t \frac{\|\nabla_v R(\mathbf{x}_k, \mathbf{y}_k, \mathbf{v}_k)\|^2}{\bar{z}_{k+1}} = 0$ for any $t < k_3$. It is easy to see that the upper bound of $\bar{z}_{t+1}$ in Eq. (77) is the largest among all cases. Thus, in the remaining proof, we only explore the upper bound of $\bar{z}_t$ in Case 4.

To further explore the bound of $\bar{z}_t$, we need to use some auxiliary results and bounds. So we split them into three parts as follows:

**Part I: An Auxiliary Bound of $\sum \frac{\|\bar{\nabla} F(\mathbf{x}_t, \mathbf{y}_t, \mathbf{v}_t)\|^2}{[\bar{m}_{t+1}^x]^2}$.** To further explore Case 4, we begin with a common term $\sum_{k=k_0}^t \frac{\|\bar{\nabla} F(\mathbf{x}_k, \mathbf{y}_k, \mathbf{v}_k)\|^2}{[\bar{m}_{k+1}^x]^2}$ for any $k_0 \leq t$. By the strong convexity of $l$ in Assumption 3.1, we have:

$$\sum_{k=1}^t \frac{\mu^2}{n}\|\mathbf{v}_k\|^2 \leq \sum_{k=1}^t \frac{\|\nabla_y \nabla_y L(\mathbf{x}_k, \mathbf{y}_k)\mathbf{v}_k\|^2}{n}$$

$$\leq \sum_{k=1}^t \frac{2\|\nabla_y \nabla_y L(\mathbf{x}_k, \mathbf{y}_k)\mathbf{v}_k - \nabla_y F(\mathbf{x}_k, \mathbf{y}_k)\|^2}{n} + \sum_{k=1}^t \frac{2\|\nabla_y F(\mathbf{x}_k, \mathbf{y}_k)\|^2}{n}$$

$$= \sum_{k=1}^t \frac{2\|\nabla_v R(\mathbf{x}_k, \mathbf{y}_k, \mathbf{v}_k)\|^2}{n} + \sum_{k=1}^t \frac{2\|\nabla_y F(\mathbf{x}_k, \mathbf{y}_k)\|^2}{n}$$

$$\leq 2[\bar{m}_{t+1}^v]^2 + 2tC_{f_y}^2, \tag{78}$$

which indicates that for any $t \geq 0$, $\|\mathbf{v}_t\|$ can be bounded as:

$$\|\mathbf{v}_t\| \leq \frac{\sqrt{2n[\bar{m}_{t+1}^v]^2 + 2ntC_{f_y}^2}}{\mu} \leq \frac{\sqrt{2n[\bar{z}_{t+1}]^2 + 2ntC_{f_y}^2}}{\mu} \leq \frac{\sqrt{2n}\left(\bar{z}_{t+1} + \sqrt{t}C_{f_y}\right)}{\mu}. \tag{79}$$

Then we have:

$$\|\mathbf{v}_t\| \leq \frac{\sqrt{2n}}{\mu}\bar{z}_{t+1} + \frac{\sqrt{2n}C_{f_y}}{\mu}\sqrt{t} =: \bar{a}\bar{z}_{t+1} + \bar{b}\sqrt{t}, \tag{80}$$

where $\bar{a}$ and $\bar{b}$ refer to Eq. (71). According to Lemma C.1, since $\bar{m}_0^x \geq 1$, for any integer $t > 0$, we have:

$$\sum_{k=k_0}^{t} \frac{\|\bar{\nabla}F(\mathbf{x}_k, \mathbf{y}_k, \mathbf{v}_k)\|^2}{[\bar{m}_{k+1}^x]^2}$$

$$\leq \sum_{k=0}^{t} \frac{\|\bar{\nabla}F(\mathbf{x}_k, \mathbf{y}_k, \mathbf{v}_k)\|^2}{[\bar{m}_{k+1}^x]^2}$$

$$\leq \log\left(\sum_{k=0}^{t} \|\bar{\nabla}F(\mathbf{x}_k, \mathbf{y}_k, \mathbf{v}_k)\|^2 + [\bar{m}_0^x]^2\right) + 1$$

$$\leq \log\left(\sum_{k=0}^{t} \left(nC_{l_{xy}}\bar{a}\bar{z}_{k+1} + nC_{l_{xy}}\bar{b}\sqrt{k} + nC_{f_x}\right)^2 + [\bar{m}_0^x]^2\right) + 1$$

$$\leq \log\left(\left(\sum_{k=0}^{t} nC_{l_{xy}}\bar{a}\bar{z}_{k+1} + nC_{l_{xy}}\bar{b}\sqrt{k} + nC_{f_x} + \bar{m}_0^x\right)^2\right) + 1$$

$$= 2\log\left(\sum_{k=0}^{t} nC_{l_{xy}}\bar{a}\bar{z}_{k+1} + nC_{l_{xy}}\bar{b}\sqrt{k} + nC_{f_x} + \bar{m}_0^x\right) + 1$$

$$\leq 2\log\left((t+1)\left(nC_{l_{xy}}\bar{a}\bar{z}_{t+1} + nC_{l_{xy}}\bar{b}\sqrt{t} + nC_{f_x} + \bar{m}_0^x\right)\right) + 1$$

$$\leq 2\log(t+1) + 2\log\left(\left(nC_{l_{xy}}\bar{a}\bar{z}_{t+1} + nC_{l_{xy}}\bar{b} + nC_{f_x} + \bar{m}_0^x\right)\sqrt{t}\right) + 1$$

$$\leq 3\log(t+1) + 2\log\left(nC_{l_{xy}}\bar{a}\bar{z}_{t+1} + nC_{l_{xy}}\bar{b} + nC_{f_x} + \bar{m}_0^x\right) + 1, \tag{81}$$

here we obtain the upper bound of $\sum_{k=k_0}^{t} \frac{\|\bar{\nabla}F(\mathbf{x}_k, \mathbf{y}_k, \mathbf{v}_k)\|^2}{[\bar{m}_{k+1}^x]^2}$ for any $k_0 \leq t$ in Eq. (81). Part I is completed.

**Part II: A More General Bound of $\sum \frac{\|\nabla_y L(\mathbf{x}_t, \mathbf{y}_t)\|^2}{\bar{m}_{t+1}^y}$.**

In Lemma D.2, we show the bound of $\sum_{k=k_2}^{t} \frac{\|\nabla_y L(\mathbf{x}_k, \mathbf{y}_k)\|^2}{\bar{m}_{k+1}^y}$ when $k_2$ exists. In Part II, we further provide a rough bound of $\sum_{k=\tilde{k}}^{t} \frac{\|\nabla_y L(\mathbf{x}_k, \mathbf{y}_k)\|^2}{\bar{m}_{k+1}^y}$ for any potential $\tilde{k} \leq T$. Firstly, if $\tilde{k} \geq k_2$, it is easy to have:

$$\sum_{k=\tilde{k}}^{t} \frac{\|\nabla_y L(\mathbf{x}_k, \mathbf{y}_k)\|^2}{\bar{m}_{k+1}^y} \leq \sum_{k=k_2}^{t} \frac{\|\nabla_y L(\mathbf{x}_k, \mathbf{y}_k)\|^2}{\bar{m}_{k+1}^y}. \tag{82}$$

Secondly, if $\tilde{k} < k_2$, we have:

$$\sum_{k=\tilde{k}}^{t} \frac{\|\nabla_y L(\mathbf{x}_k, \mathbf{y}_k)\|^2}{\bar{m}_{k+1}^y} \leq \sum_{k=\tilde{k}}^{k_2-1} \frac{\|\nabla_y L(\mathbf{x}_k, \mathbf{y}_k)\|^2}{\bar{m}_{k+1}^y} + \sum_{k=k_2}^{t} \frac{\|\nabla_y L(\mathbf{x}_k, \mathbf{y}_k)\|^2}{\bar{m}_{k+1}^y}$$

$$\leq \frac{\sum_{k=\tilde{k}}^{k_2-1} \|\nabla_y L(\mathbf{x}_k, \mathbf{y}_k)\|^2}{\bar{m}_0^y} + \sum_{k=k_2}^{t} \frac{\|\nabla_y L(\mathbf{x}_k, \mathbf{y}_k)\|^2}{\bar{m}_{k+1}^y}$$

$$\leq \frac{n([\bar{m}_{k_2}^y]^2 - [\bar{m}_{\tilde{k}}^y]^2)}{\bar{m}_0^y} + \sum_{k=k_2}^{t} \frac{\|\nabla_y L(\mathbf{x}_k, \mathbf{y}_k)\|^2}{\bar{m}_{k+1}^y}$$

$$\leq \frac{n(C_{m^y}^2 - [\bar{m}_0^y]^2)}{\bar{m}_0^y} + \sum_{k=k_2}^{t} \frac{\|\nabla_y L(\mathbf{x}_k, \mathbf{y}_k)\|^2}{\bar{m}_{k+1}^y}$$

$$= \frac{nC_{m^y}^2}{\bar{m}_0^y} - n\bar{m}_0^y + \sum_{k=k_2}^{t} \frac{\|\nabla_y L(\mathbf{x}_k, \mathbf{y}_k)\|^2}{\bar{m}_{k+1}^y}. \tag{83}$$

Combining these two situations, since $C_{m^y} \geq \bar{m}_0^y$, for any $\tilde{k} \leq t$, we have:

$$\sum_{k=\tilde{k}}^{t} \frac{\|\nabla_y L(\mathbf{x}_k, \mathbf{y}_k)\|^2}{\bar{m}_{k+1}^y}$$

$$\leq \frac{nC_{m^y}^2}{\bar{m}_0^y} - n\bar{m}_0^y + \sum_{k=k_2}^{t} \frac{\|\nabla_y L(\mathbf{x}_k, \mathbf{y}_k)\|^2}{\bar{m}_{k+1}^y}$$

$$\leq \frac{nC_{m^y}^2}{\bar{m}_0^y} - n\bar{m}_0^y + \frac{2nC_{m^y}^2(\mu + L_{l,1})}{\mu^2 \gamma_y \sqrt{1 + \sigma_u^2}}$$

$$+ \left( \frac{8\Delta_0^x}{1 - \rho_W} + \frac{32\gamma_x^2 \rho_W C_{m^x}^2 (1 + \zeta_q^2)}{\bar{q}_0^2 (1 - \rho_W)^2} \right) \left( \frac{(\mu + L_{l,1})^3}{\gamma_y \mu^2} + \frac{2L_{l,1}(\mu + L_{l,1})}{\gamma_y \mu} \right)$$

$$+ \frac{32\gamma_x^2 \rho_W (1 + \zeta_q^2)}{\bar{z}_0 (1 - \rho_W)^2} \left( \frac{(\mu + L_{l,1})^3}{\gamma_y \mu^2} + \frac{2L_{l,1}(\mu + L_{l,1})}{\gamma_y \mu} \right) + \frac{8\gamma_x^2 L_y^2 (1 + \zeta_q^2)(\mu + L_{l,1})^2}{\gamma_y^2 \mu L_{l,1} \bar{z}_0 \sqrt{1 + \sigma_u^2}}$$

$$+ \left[ \frac{32\gamma_x^2 \rho_W (1 + \zeta_q^2)}{(1 - \rho_W)^2} \left( \frac{(\mu + L_{l,1})^3}{\gamma_y \mu^2} + \frac{2L_{l,1}(\mu + L_{l,1})}{\gamma_y \mu} \right) \right.$$

$$\left. + \frac{8\gamma_x^2 L_y^2 (1 + \zeta_q^2)(\mu + L_{l,1})^2}{\gamma_y^2 \mu L_{l,1} \sqrt{1 + \sigma_u^2}} \right] \sum_{k=\min\{k_1, k_2\}}^{t} \frac{\|\bar{\nabla} F(\mathbf{x}_k, \mathbf{y}_k, \mathbf{v}_k)\|^2}{[\bar{m}_{k+1}^x]^2 \max\{\bar{m}_{k+1}^v, \bar{m}_{k+1}^y\}}, \tag{84}$$

where the second inequality uses Lemma D.2. Thus, Part II is completed.

**Part III: The Bound of $\bar{z}_t$ in Case 4.**

Here, we explore the upper bound of $\bar{z}_t$ in Case 4. Recalling Eq. (77), we have:

$$\bar{z}_{t+1} \leq C_{m^y} + C_{m^v} + \sum_{k=k_2}^{t} \frac{\|\nabla_y L(\mathbf{x}_k, \mathbf{y}_k)\|^2}{n\bar{m}_{k+1}^y} + \sum_{k=k_3}^{t} \frac{\|\nabla_v R(\mathbf{x}_k, \mathbf{y}_k, \mathbf{v}_k)\|^2}{n\bar{z}_{k+1}} = C_{m^y} + C_{m^v} = C_z, \tag{85}$$

for $t \leq k_{\min} := \min\{k_2, k_3\}$. For $t > k_{\min}$, we have:

$$\bar{z}_{t+1} \leq C_{m^y} + C_{m^v} + \sum_{k=k_2}^{t} \frac{\|\nabla_y L(\mathbf{x}_k, \mathbf{y}_k)\|^2}{n\bar{m}_{k+1}^y} + \sum_{k=k_3}^{t} \frac{\|\nabla_v R(\mathbf{x}_k, \mathbf{y}_k, \mathbf{v}_k)\|^2}{n\bar{z}_{k+1}}$$

$$\overset{(a)}{\leq} C_{m^y} + C_{m^v} + \frac{4C_{m^v}^2(\mu + C_{l_{yy}})}{\mu^4 \gamma_v \sqrt{1 + \sigma_z^2}} \left( \frac{nL_{l,2} C_{f_y}}{\mu} + L_{f,1} \right)^2$$

$$+ \frac{8(\mu + C_{l_{yy}})C_{m^v}^2}{\mu^2 \gamma_v \sqrt{1 + \sigma_z^2}} + \frac{16(\mu + C_{l_{yy}})}{n\gamma_v \mu^2} \left( \frac{nL_{l,2} C_{f_y}}{\mu} + L_{f,1} \right)^2$$

$$\cdot \left( \frac{\Delta_0^x}{1 - \rho_W} + \frac{4\gamma_x^2 \rho_W C_{m^x}^2 (1 + \zeta_q^2)}{\bar{q}_0^2 (1 - \rho_W)^2} \right) \left( \frac{(\mu + L_{l,1})^2}{n\mu^2} + \frac{2L_{l,1}}{n\mu} \right)$$

$$+ \left( \frac{16\Delta_0^x}{n(1 - \rho_W)} + \frac{64\gamma_x^2 \rho_W C_{m^x}^2 (1 + \zeta_q^2)}{n\bar{q}_0^2 (1 - \rho_W)^2} \right) \left( \frac{\bar{L}_r^2(\mu + C_{l_{yy}})}{\mu C_{l_{yy}} \gamma_v} + \frac{\bar{L}_r^2(\mu + C_{l_{yy}})}{\mu^2 \gamma_v \sqrt{1 + \sigma_z^2}} \right)$$

$$+ \frac{16(\mu + C_{l_{yy}})}{n\gamma_v \mu^2 \sqrt{1 + \sigma_u^2}} \left( \frac{nL_{l,2} C_{f_y}}{\mu} + L_{f,1} \right)^2$$

$$\cdot \left[ \frac{4\gamma_x^2 \rho_W (1 + \zeta_q^2)}{\bar{z}_0^2 (1 - \rho_W)^2} \left( \frac{(\mu + L_{l,1})^2 \sqrt{1 + \sigma_u^2}}{n\mu^2} + \frac{2L_{l,1}\sqrt{1 + \sigma_u^2}}{n\mu} \right) \right.$$

$$\left. + \frac{\gamma_x^2 L_y^2 (1 + \zeta_q^2)(\mu + L_{l,1})}{n\bar{z}_0 \gamma_y \mu L_{l,1}} \right] \sum_{k=\min\{k_1-1, k_2-1\}}^{t} \frac{\|\bar{\nabla} F(\mathbf{x}_k, \mathbf{y}_k, \mathbf{v}_k)\|^2}{[\bar{m}_{k+1}^x]^2}$$

$$+ \left( \frac{64\gamma_x^2 \rho_W \bar{L}_r^2 (\mu + C_{l_{yy}})(1 + \zeta_q^2)}{n\mu C_{l_{yy}} \gamma_v (1 - \rho_W)^2} + \frac{2\gamma_x^2 L_v^2 (1 + \zeta_q^2)(\mu + C_{l_{yy}})^2}{n\mu C_{l_{yy}} C_{m^v} \gamma_v^2 \sqrt{1 + \sigma_z^2}} \right)$$

$$\cdot \sum_{k=\min\{k_1-1,k_3-1\}}^{t} \frac{\|\bar{\nabla} F(\mathbf{x}_k, \mathbf{y}_k, \mathbf{v}_k)\|^2}{[\bar{m}_{k+1}^x]^2}$$

$$+ \left[ \frac{1}{\mu^2} \left( 8 + \frac{4(\mu + C_{l_{yy}})^2}{\mu C_{l_{yy}}} \right) \left( \frac{nL_{l,2} C_{f_y}}{\mu} + L_{f,1} \right)^2 + 1 \right] \left( \frac{C_{m^y}^2}{\bar{m}_0^y} - \bar{m}_0^y \right)$$

$$+ \frac{1}{n} \left[ \frac{1}{\mu^2} \left( 8 + \frac{4(\mu + C_{l_{yy}})^2}{\mu C_{l_{yy}}} \right) \left( \frac{nL_{l,2} C_{f_y}}{\mu} + L_{f,1} \right)^2 + 1 \right] \sum_{k=k_2}^{t} \frac{\|\nabla_y L(\mathbf{x}_k, \mathbf{y}_k)\|^2}{\bar{m}_{k+1}^y}$$

$$\overset{(b)}{\leq} C_{m^y} + C_{m^v} + \frac{4 C_{m^y}^2 (\mu + C_{l_{yy}})}{\mu^4 \gamma_v \sqrt{1 + \sigma_z^2}} \left( \frac{nL_{l,2} C_{f_y}}{\mu} + L_{f,1} \right)^2$$

$$+ \frac{8(\mu + C_{l_{yy}}) C_{m^v}^2}{\mu^2 \gamma_v \sqrt{1 + \sigma_z^2}} + \frac{16(\mu + C_{l_{yy}})}{n\gamma_v \mu^2} \left( \frac{nL_{l,2} C_{f_y}}{\mu} + L_{f,1} \right)^2$$

$$\cdot \left( \frac{\Delta_0^x}{1 - \rho_W} + \frac{4\gamma_x^2 \rho_W C_{m^x}^2 (1 + \zeta_q^2)}{\bar{q}_0^2 (1 - \rho_W)^2} \right) \left( \frac{(\mu + L_{l,1})^2}{n\mu^2} + \frac{2L_{l,1}}{n\mu} \right)$$

$$+ \left( \frac{16\Delta_0^x}{n(1 - \rho_W)} + \frac{64\gamma_x^2 \rho_W C_{m^x}^2 (1 + \zeta_q^2)}{n\bar{q}_0^2 (1 - \rho_W)^2} \right) \left( \frac{\bar{L}_r^2 (\mu + C_{l_{yy}})}{\mu C_{l_{yy}} \gamma_v} + \frac{\bar{L}_r^2 (\mu + C_{l_{yy}})}{\mu^2 \gamma_v \sqrt{1 + \sigma_z^2}} \right)$$

$$+ \left[ \left( \frac{1}{\mu^2} \left( 8 + \frac{4(\mu + C_{l_{yy}})^2}{\mu C_{l_{yy}}} \right) \left( \frac{nL_{l,2} C_{f_y}}{\mu} + L_{f,1} \right)^2 + 1 \right) \right.$$

$$\cdot \left( \frac{32\gamma_x^2 \rho_W (1 + \zeta_q^2)}{n(1 - \rho_W)^2} \left( \frac{(\mu + L_{l,1})^3}{\gamma_y \mu^2} + \frac{2L_{l,1}(\mu + L_{l,1})}{\gamma_y \mu} \right) + \frac{8\gamma_x^2 L_y^2 \left( 1 + \zeta_q^2 \right)(\mu + L_{l,1})^2}{n\gamma_y^2 \mu L_{l,1} \sqrt{1 + \sigma_u^2}} \right)$$

$$+ \frac{16(\mu + C_{l_{yy}})}{n\gamma_v \mu^2 \sqrt{1 + \sigma_u^2}} \left( \frac{nL_{l,2} C_{f_y}}{\mu} + L_{f,1} \right)^2$$

$$\cdot \left( \frac{4\gamma_x^2 \rho_W (1 + \zeta_q^2)}{\bar{z}_0^2 (1 - \rho_W)^2} \left( \frac{(\mu + L_{l,1})^2 \sqrt{1 + \sigma_u^2}}{n\mu^2} + \frac{2L_{l,1} \sqrt{1 + \sigma_u^2}}{n\mu} \right) \right.$$

$$+ \left. \frac{\gamma_x^2 L_y^2 (1 + \zeta_q^2)(\mu + L_{l,1})}{n\bar{z}_0 \gamma_y \mu L_{l,1}} \right) \right] \sum_{k=\{k_1-1,k_2-1\}}^{t} \frac{\|\bar{\nabla} F(\mathbf{x}_k, \mathbf{y}_k, \mathbf{v}_k)\|^2}{[\bar{m}_{k+1}^x]^2}$$

$$+ \left( \frac{64\gamma_x^2 \rho_W \bar{L}_r^2 (\mu + C_{l_{yy}})(1 + \zeta_q^2)}{n\mu C_{l_{yy}} \gamma_v (1 - \rho_W)^2} + \frac{2\gamma_x^2 L_v^2 (1 + \zeta_q^2)(\mu + C_{l_{yy}})^2}{n\mu C_{l_{yy}} \bar{m}_0^v \gamma_v^2 \sqrt{1 + \sigma_z^2}} \right)$$

$$\cdot \sum_{k=\min\{k_1-1,k_3-1\}}^{t} \frac{\|\bar{\nabla} F(\mathbf{x}_k, \mathbf{y}_k, \mathbf{v}_k)\|^2}{[\bar{m}_{k+1}^x]^2}$$

$$+ \left[ \frac{1}{\mu^2} \left( 8 + \frac{4(\mu + C_{l_{yy}})^2}{\mu C_{l_{yy}}} \right) \left( \frac{nL_{l,2} C_{f_y}}{\mu} + L_{f,1} \right)^2 + 1 \right] \left( \frac{C_{m^y}^2}{\bar{m}_0^y} - \bar{m}_0^y \right)$$

$$+ \left[ \frac{1}{\mu^2} \left( 8 + \frac{4(\mu + C_{l_{yy}})^2}{\mu C_{l_{yy}}} \right) \left( \frac{nL_{l,2} C_{f_y}}{\mu} + L_{f,1} \right)^2 + 1 \right] \left( \frac{2C_{m^y}^2 (\mu + L_{l,1})}{\mu^2 \gamma_y \sqrt{1 + \sigma_u^2}} \right)$$

$$+ \left( \frac{8\Delta_0^x}{n(1 - \rho_W)} + \frac{32\gamma_x^2 \rho_W C_{m^x}^2 (1 + \zeta_q^2)}{n\bar{q}_0^2 (1 - \rho_W)^2} \right) \left( \frac{(\mu + L_{l,1})^3}{\gamma_y \mu^2} + \frac{2L_{l,1}(\mu + L_{l,1})}{\gamma_y \mu} \right)$$

$$+ \frac{32\gamma_x^2 \rho_W (1 + \zeta_q^2)}{n\bar{z}_0 (1 - \rho_W)^2} \left( \frac{(\mu + L_{l,1})^3}{\gamma_y \mu^2} + \frac{2L_{l,1}(\mu + L_{l,1})}{\gamma_y \mu} \right) + \frac{8\gamma_x^2 L_y^2 \left( 1 + \zeta_q^2 \right)(\mu + L_{l,1})^2}{n\gamma_y^2 \mu L_{l,1} \bar{z}_0 \sqrt{1 + \sigma_u^2}} \right)$$

$$\overset{(c)}{=}: a_0 \sum_{k=\min\{k_1-1,k_2-1,k_3-1\}}^{t} \frac{\|\bar{\nabla}F(\mathbf{x}_k,\mathbf{y}_k,\mathbf{v}_k)\|^2}{[\bar{m}_{k+1}^x]^2} + b_0,$$

$$\leq a_0 \sum_{k=\min\{k_1,k_2,k_3\}}^{t} \frac{\|\bar{\nabla}F(\mathbf{x}_k,\mathbf{y}_k,\mathbf{v}_k)\|^2}{[\bar{m}_{k+1}^x]^2} + a_0 + b_0$$

$$\overset{(d)}{\leq} a_0 \left( 3\log(t+1) + 2\log\left( \bar{z}_{t+1} + \frac{nC_{l_{xy}}\bar{b} + nC_{f_x} + \bar{m}_0^x}{nC_{l_{xy}}\bar{a}} \right) + 2\log\left(nC_{l_{xy}}\bar{a}\right) + 1 \right)$$
$$+ a_0 + b_0, \tag{86}$$

where (a) uses Lemma D.3 and the first line in Eq. (84) by replacing $\tilde{k}$ with $k_3 - 1$; (b) results from Lemma D.2; (c) refers to Eq. (71); (d) uses Eq. (81). Since $\min\{k_2, k_3\} \leq T$, we have $\bar{z}_{t+1} \geq \min\{C_{m^y}, C_{m^v}\} \geq \max\{64a_0^2, 1\}$, which indicates that

(i) if $8a_0 \leq 1$, we have:

$$4a_0 \log(\bar{z}_{t+1}) \leq \frac{\log(\bar{z}_{t+1})}{2} \leq \frac{\bar{z}_{t+1}}{2} \leq \bar{z}_{t+1}; \tag{87}$$

(ii) if $8a_0 > 1$, we have:

$$\bar{z}_{t+1} - 4a_0 \log(\bar{z}_{t+1}) = \bar{z}_{t+1} - 8a_0 \log(\sqrt{\bar{z}_{t+1}}) \geq 8a_0 \left(\sqrt{\bar{z}_{t+1}} - \log(\sqrt{\bar{z}_{t+1}})\right) \geq 0. \tag{88}$$

Combining (i) and (ii), we have $4a_0 \log(\bar{z}_{t+1}) \leq \bar{z}_{t+1}$. Then we obtain:

$$\bar{z}_{t+1} \leq a_0 \left( 3\log(t+1) + 2\log\left( \bar{z}_{t+1} + \frac{nC_{l_{xy}}\bar{b} + nC_{f_x} + \bar{m}_0^x}{nC_{l_{xy}}\bar{a}} \right) + 2\log\left(nC_{l_{xy}}\bar{a}\right) + 1 \right)$$
$$+ a_0 + b_0$$
$$\leq a_0 \left( 3\log(t+1) + 2\log\left(\bar{z}_{t+1}\right) + 2\log\left( 1 + \frac{nC_{l_{xy}}\bar{b} + nC_{f_x} + \bar{m}_0^x}{nC_{l_{xy}}\bar{a}} \right) + 2\log\left(nC_{l_{xy}}\bar{a}\right) + 1 \right)$$
$$+ a_0 + b_0$$
$$\leq \frac{1}{2}\bar{z}_{t+1} + a_0 \left( 3\log(t+1) + 2\log\left( 1 + \frac{nC_{l_{xy}}\bar{b} + nC_{f_x} + \bar{m}_0^x}{nC_{l_{xy}}\bar{a}} \right) + 2\log\left(nC_{l_{xy}}\bar{a}\right) + 1 \right)$$
$$+ a_0 + b_0, \tag{89}$$

which indicates that

$$\bar{z}_{t+1} \leq 6a_0 \log(t+1) + 4a_0 \log\left( 1 + \frac{nC_{l_{xy}}\bar{b} + nC_{f_x} + \bar{m}_0^x}{nC_{l_{xy}}\bar{a}} \right) + 4a_0 \log\left(nC_{l_{xy}}\bar{a}\right) + 4a_0 + 2b_0$$

$$\overset{(a)}{=} a_1 \log(t+1) + b_1, \tag{90}$$

where (a) refers to Eq. (70). Therefore, we complete the proof of this lemma. $\qquad\square$

## D.6  The Upper Bounds of $\sum \frac{\|\bar{\nabla}F(\mathbf{x}_t,\mathbf{y}_t,\mathbf{v}_t)\|^2}{[\bar{m}_{t+1}^x]^2}$, $\sum \frac{\|\nabla_y L(\mathbf{x}_t,\mathbf{y}_t)\|^2}{\bar{m}_{t+1}^y}$, and $\sum \frac{\|\nabla_v R(\mathbf{x}_t,\mathbf{y}_t,\mathbf{v}_t)\|^2}{\bar{z}_{t+1}}$

**Lemma D.6.** *Under Assumption 3.1 and Assumption 3.2, for any integer $k_0 \in [0, t)$, we have the upper bounds in terms of logarithmic functions as:*

$$\sum_{k=k_0}^{t} \frac{\|\bar{\nabla}F(\mathbf{x}_k,\mathbf{y}_k,\mathbf{v}_k)\|^2}{[\bar{m}_{k+1}^x]^2} \leq 5\log(t+1) + c_2,$$

$$\sum_{k=k_0}^{t} \frac{\|\nabla_y L(\mathbf{x}_k,\mathbf{y}_k)\|^2}{\bar{m}_{k+1}^y} \leq a_2 \log(t+1) + b_2,$$

$$\sum_{k=k_0}^{t} \frac{\|\nabla_v R(\mathbf{x}_k,\mathbf{y}_k,\mathbf{v}_k)\|^2}{\bar{z}_{k+1}} \leq a_3 \log(t+1) + b_3, \tag{91}$$

*where, referring to Eq. (70) and Eq. (71), $c_2$, $a_2$, $b_2$, $a_3$, and $b_3$ are defined as:*

$$c_2 := 2\log\left(nC_{l_{xy}}\bar{a}a_1 + nC_{l_{xy}}\bar{a}b_1 + nC_{l_{xy}}\bar{b} + nC_{f_x} + \bar{m}_0^x\right) + 1,$$

$$a_2 := \frac{160\gamma_x^2\rho_W(1+\zeta_q^2)}{(1-\rho_W)^2}\left(\frac{(\mu+L_{l,1})^3}{\gamma_y\mu^2} + \frac{2L_{l,1}(\mu+L_{l,1})}{\gamma_y\mu}\right) + \frac{40\gamma_x^2L_y^2\left(1+\zeta_q^2\right)(\mu+L_{l,1})^2}{\gamma_y^2\mu L_{l,1}\sqrt{1+\sigma_u^2}},$$

$$b_2 := \frac{nC_{m^y}^2}{\bar{m}_0^y} - n\bar{m}_0^y + \frac{2nC_{m^y}^2(\mu+L_{l,1})}{\mu^2\gamma_y\sqrt{1+\sigma_u^2}}$$

$$+ \left(\frac{8\Delta_0^x}{1-\rho_W} + \frac{32\gamma_x^2\rho_WC_{m^x}^2(1+\zeta_q^2)}{\bar{q}_0^2(1-\rho_W)^2}\right)\left(\frac{(\mu+L_{l,1})^3}{\gamma_y\mu^2} + \frac{2L_{l,1}(\mu+L_{l,1})}{\gamma_y\mu}\right)$$

$$+ \left[\frac{32\gamma_x^2\rho_W(1+\zeta_q^2)}{(1-\rho_W)^2}\left(\frac{(\mu+L_{l,1})^3}{\gamma_y\mu^2} + \frac{2L_{l,1}(\mu+L_{l,1})}{\gamma_y\mu}\right) + \frac{8\gamma_x^2L_y^2\left(1+\zeta_q^2\right)(\mu+L_{l,1})^2}{\gamma_y^2\mu L_{l,1}\sqrt{1+\sigma_u^2}}\right]\left(\frac{1}{\bar{z}_0}+c_2\right),$$

$$a_3 := \frac{80(\mu+C_{l_{yy}})}{\gamma_v\mu^2\sqrt{1+\sigma_u^2}}\left(\frac{nL_{l,2}C_{f_y}}{\mu}+L_{f,1}\right)^2$$

$$\cdot\left(\frac{4\gamma_x^2\rho_W(1+\zeta_q^2)}{\bar{z}_0^2(1-\rho_W)^2}\left(\frac{(\mu+L_{l,1})^2\sqrt{1+\sigma_u^2}}{n\mu^2} + \frac{2L_{l,1}\sqrt{1+\sigma_u^2}}{n\mu}\right)\right.$$

$$\left. + \frac{\gamma_x^2L_y^2(1+\zeta_q^2)(\mu+L_{l,1})}{n\bar{z}_0\gamma_y\mu L_{l,1}}\right) + \frac{320\gamma_x^2\rho_W\bar{L}_r^2(\mu+C_{l_{yy}})(1+\zeta_q^2)}{\mu C_{l_{yy}}\gamma_v(1-\rho_W)^2}$$

$$+ \frac{10\gamma_x^2L_v^2(1+\zeta_q^2)(\mu+C_{l_{yy}})^2}{\mu C_{l_{yy}}C_{m^v}\gamma_v^2\sqrt{1+\sigma_z^2}} + \left(\frac{8a_2}{\mu^2} + \frac{4a_2(\mu+C_{l_{yy}})^2}{\mu^3C_{l_{yy}}}\right)\left(\frac{nL_{l,2}C_{f_y}}{\mu}+L_{f,1}\right)^2,$$

$$b_3 := \frac{nC_{m^v}^2}{\bar{m}_0^v} - n\bar{m}_0^v + \frac{4nC_{m^v}^2(\mu+C_{l_{yy}})}{\mu^4\gamma_v\sqrt{1+\sigma_z^2}}\left(\frac{nL_{l,2}C_{f_y}}{\mu}+L_{f,1}\right)^2 + \frac{8nC_{m^v}^2(\mu+C_{l_{yy}})}{\mu^2\gamma_v\sqrt{1+\sigma_z^2}}$$

$$+ \left(\frac{16\Delta_0^x}{1-\rho_W} + \frac{64\gamma_x^2\rho_WC_{m^x}^2(1+\zeta_q^2)}{\bar{q}_0^2(1-\rho_W)^2}\right)\left(\frac{\bar{L}_r^2(\mu+C_{l_{yy}})}{\mu C_{l_{yy}}\gamma_v} + \frac{\bar{L}_r^2(\mu+C_{l_{yy}})}{\mu^2\gamma_v\sqrt{1+\sigma_z^2}}\right)$$

$$+ \frac{16(\mu+C_{l_{yy}})}{\gamma_v\mu^2}\left(\frac{nL_{l,2}C_{f_y}}{\mu}+L_{f,1}\right)^2\left(\frac{\Delta_0^x}{1-\rho_W} + \frac{4\gamma_x^2\rho_WC_{m^x}^2(1+\zeta_q^2)}{\bar{q}_0^2(1-\rho_W)^2}\right)$$

$$\cdot\left(\frac{(\mu+L_{l,1})^2}{n\mu^2} + \frac{2L_{l,1}}{n\mu}\right) + \frac{16c_2(\mu+C_{l_{yy}})}{\gamma_v\mu^2\sqrt{1+\sigma_u^2}}\left(\frac{nL_{l,2}C_{f_y}}{\mu}+L_{f,1}\right)^2$$

$$\cdot\left(\frac{4\gamma_x^2\rho_W(1+\zeta_q^2)}{\bar{z}_0^2(1-\rho_W)^2}\left(\frac{(\mu+L_{l,1})^2\sqrt{1+\sigma_u^2}}{n\mu^2} + \frac{2L_{l,1}\sqrt{1+\sigma_u^2}}{n\mu}\right) + \frac{\gamma_x^2L_y^2(1+\zeta_q^2)(\mu+L_{l,1})}{n\bar{z}_0\gamma_y\mu L_{l,1}}\right)$$

$$+ \frac{64c_2\gamma_x^2\rho_W\bar{L}_r^2(\mu+C_{l_{yy}})(1+\zeta_q^2)}{\mu C_{l_{yy}}\gamma_v(1-\rho_W)^2} + \frac{2c_2\gamma_x^2L_v^2(1+\zeta_q^2)(\mu+C_{l_{yy}})^2}{\mu C_{l_{yy}}C_{m^v}\gamma_v^2\sqrt{1+\sigma_z^2}}$$

$$+ \left(\frac{8b_2}{\mu^2} + \frac{4b_2(\mu+C_{l_{yy}})^2}{\mu^3C_{l_{yy}}}\right)\left(\frac{nL_{l,2}C_{f_y}}{\mu}+L_{f,1}\right)^2. \tag{92}$$

*Proof.* Based on the results in Lemma D.5, we have the following bounds.

**Part I: Bounding** $\sum \frac{\|\bar{\nabla}F(\mathbf{x}_t,\mathbf{y}_t,\mathbf{v}_t)\|^2}{[\bar{m}_{t+1}^x]^2}$.

Firstly, we bound $\sum_{k=k_0}^t \frac{\|\bar{\nabla}F(\mathbf{x}_k,\mathbf{y}_k,\mathbf{v}_k)\|^2}{[\bar{m}_{k+1}^x]^2}$ for arbitrary $k_0 < t$. Back to Eq. (81), by plugging in Eq. (90), we have:

$$\sum_{k=k_0}^t \frac{\|\bar{\nabla}F(\mathbf{x}_k,\mathbf{y}_k,\mathbf{v}_k)\|^2}{[\bar{m}_{k+1}^x]^2}$$

$$\leq 3\log(t+1) + 2\log\left(nC_{l_{xy}}\bar{a}\bar{z}_{t+1} + nC_{l_{xy}}\bar{b} + nC_{f_x} + \bar{m}_0^x\right) + 1$$

$$\overset{(a)}{\leq} 3\log(t+1) + 2\log\left(nC_{l_{xy}}\bar{a}a_1\log(t+1) + nC_{l_{xy}}\bar{a}b_1 + nC_{l_{xy}}\bar{b} + nC_{f_x} + \bar{m}_0^x\right) + 1$$

$$\leq 3\log(t+1) + 2\log\left(nC_{l_{xy}}\bar{a}a_1(t+1) + nC_{l_{xy}}\bar{a}b_1 + nC_{l_{xy}}\bar{b} + nC_{f_x} + \bar{m}_0^x\right) + 1$$

$$\leq 3\log(t+1) + 2\log\left(\left(nC_{l_{xy}}\bar{a}a_1 + nC_{l_{xy}}\bar{a}b_1 + nC_{l_{xy}}\bar{b} + nC_{f_x} + \bar{m}_0^x\right)(t+1)\right) + 1$$

$$\leq 5\log(t+1) + 2\log\left(nC_{l_{xy}}\bar{a}a_1 + nC_{l_{xy}}\bar{a}b_1 + nC_{l_{xy}}\bar{b} + nC_{f_x} + \bar{m}_0^x\right) + 1$$

$$\overset{(b)}{=:} 5\log(t+1) + c_2, \tag{93}$$

where (a) results from Eq. (90); (b) refers to Eq. (92).

**Part II: Bounding** $\sum \frac{\|\nabla_y L(\mathbf{x}_t, \mathbf{y}_t)\|^2}{\bar{m}_{t+1}^y}$.

Secondly, we bound $\sum_{k=k_0}^{t} \frac{\|\nabla_y L(\mathbf{x}_k, \mathbf{y}_k)\|^2}{\bar{m}_{k+1}^y}$. We split this part into two cases using Lemma C.5.

**Case 1:** If $\bar{m}_{t+1}^y \leq C_{m^y}$, we have:

$$\sum_{k=k_0}^{t} \frac{\|\nabla_y L(\mathbf{x}_k, \mathbf{y}_k)\|^2}{\bar{m}_{k+1}^y}$$

$$\leq \sum_{k=k_0}^{t} \frac{\|\nabla_y L(\mathbf{x}_k, \mathbf{y}_k)\|^2}{\bar{m}_0^y} \leq \frac{n([\bar{m}_{t+1}^y]^2 - [\bar{m}_{k_0}^y]^2)}{\bar{m}_0^y} \leq \frac{n(C_{m^y}^2 - [\bar{m}_0^y]^2)}{\bar{m}_0^y} = \frac{nC_{m^y}^2}{\bar{m}_0^y} - n\bar{m}_0^y \leq b_2. \tag{94}$$

**Case 2:** If $\bar{m}_{t+1}^y > C_{m^y}$, we have $k_2 \leq t$, where $k_2$ refers to Lemma C.5. Then based on Eq. (84), we have:

$$\sum_{k=k_0}^{t} \frac{\|\nabla_y L(\mathbf{x}_k, \mathbf{y}_k)\|^2}{\bar{m}_{k+1}^y}$$

$$\leq \frac{nC_{m^y}^2}{\bar{m}_0^y} - n\bar{m}_0^y + \frac{2nC_{m^y}^2(\mu + L_{l,1})}{\mu^2 \gamma_y \sqrt{1+\sigma_u^2}}$$

$$+ \left(\frac{8\Delta_0^x}{1-\rho_W} + \frac{32\gamma_x^2 \rho_W C_{m^x}^2(1+\zeta_q^2)}{\bar{q}_0^2(1-\rho_W)^2}\right)\left(\frac{(\mu + L_{l,1})^3}{\gamma_y \mu^2} + \frac{2L_{l,1}(\mu + L_{l,1})}{\gamma_y \mu}\right)$$

$$+ \frac{32\gamma_x^2 \rho_W(1+\zeta_q^2)}{\bar{z}_0(1-\rho_W)^2}\left(\frac{(\mu + L_{l,1})^3}{\gamma_y \mu^2} + \frac{2L_{l,1}(\mu + L_{l,1})}{\gamma_y \mu}\right) + \frac{8\gamma_x^2 L_y^2(1+\zeta_q^2)(\mu + L_{l,1})^2}{\gamma_y^2 \mu L_{l,1} \bar{z}_0 \sqrt{1+\sigma_u^2}}$$

$$+ \left[\frac{32\gamma_x^2 \rho_W(1+\zeta_q^2)}{(1-\rho_W)^2}\left(\frac{(\mu + L_{l,1})^3}{\gamma_y \mu^2} + \frac{2L_{l,1}(\mu + L_{l,1})}{\gamma_y \mu}\right) + \frac{8\gamma_x^2 L_y^2(1+\zeta_q^2)(\mu + L_{l,1})^2}{\gamma_y^2 \mu L_{l,1} \sqrt{1+\sigma_u^2}}\right]$$

$$\cdot \sum_{k=\min\{k_1,k_2\}}^{t} \frac{\|\bar{\nabla} F(\mathbf{x}_k, \mathbf{y}_k, \mathbf{v}_k)\|^2}{[\bar{m}_{k+1}^x]^2 \max\{\bar{m}_{k+1}^v, \bar{m}_{k+1}^y\}}$$

$$\overset{(a)}{\leq} \frac{nC_{m^y}^2}{\bar{m}_0^y} - n\bar{m}_0^y + \frac{2nC_{m^y}^2(\mu + L_{l,1})}{\mu^2 \gamma_y \sqrt{1+\sigma_u^2}}$$

$$+ \left(\frac{8\Delta_0^x}{1-\rho_W} + \frac{32\gamma_x^2 \rho_W C_{m^x}^2(1+\zeta_q^2)}{\bar{q}_0^2(1-\rho_W)^2}\right)\left(\frac{(\mu + L_{l,1})^3}{\gamma_y \mu^2} + \frac{2L_{l,1}(\mu + L_{l,1})}{\gamma_y \mu}\right)$$

$$+ \frac{32\gamma_x^2 \rho_W(1+\zeta_q^2)}{\bar{z}_0(1-\rho_W)^2}\left(\frac{(\mu + L_{l,1})^3}{\gamma_y \mu^2} + \frac{2L_{l,1}(\mu + L_{l,1})}{\gamma_y \mu}\right) + \frac{8\gamma_x^2 L_y^2(1+\zeta_q^2)(\mu + L_{l,1})^2}{\gamma_y^2 \mu L_{l,1} \bar{z}_0 \sqrt{1+\sigma_u^2}}$$

$$+ \left[\frac{160\gamma_x^2 \rho_W(1+\zeta_q^2)}{(1-\rho_W)^2}\left(\frac{(\mu + L_{l,1})^3}{\gamma_y \mu^2} + \frac{2L_{l,1}(\mu + L_{l,1})}{\gamma_y \mu}\right)\right.$$

$$\left. + \frac{40\gamma_x^2 L_y^2(1+\zeta_q^2)(\mu + L_{l,1})^2}{\gamma_y^2 \mu L_{l,1} \sqrt{1+\sigma_u^2}}\right]\log(t+1)$$

$$+ \left[ \frac{32\gamma_x^2 \rho_W (1 + \zeta_q^2)}{(1 - \rho_W)^2} \left( \frac{(\mu + L_{l,1})^3}{\gamma_y \mu^2} + \frac{2L_{l,1}(\mu + L_{l,1})}{\gamma_y \mu} \right) + \frac{8\gamma_x^2 L_y^2 \left(1 + \zeta_q^2\right)(\mu + L_{l,1})^2}{\gamma_y^2 \mu L_{l,1} \sqrt{1 + \sigma_u^2}} \right] c_2$$

$$\stackrel{(b)}{=} : a_2 \log(t + 1) + b_2, \tag{95}$$

where (a) uses Eq. (93), and (b) refers to Eq. (95). Since the upper bound of Case 2 is larger, we take Eq. (92) as our final result.

**Part III: Bounding $\sum \frac{\|\nabla_v R(\mathbf{x}_t, \mathbf{y}_t, \mathbf{v}_t)\|^2}{\bar{z}_{t+1}}$.**

Last, we bound $\sum_{k=k_0}^{t} \frac{\|\nabla_v R(\mathbf{x}_k, \mathbf{y}_k, \mathbf{v}_k)\|^2}{\bar{z}_{k+1}}$. We split this part into two cases using Lemma C.5.

**Case 1:** If $\bar{m}_{t+1}^v \leq C_{m^v}$, we have:

$$\sum_{k=k_0}^{t} \frac{\|\nabla_v R(\mathbf{x}_k, \mathbf{y}_k, \mathbf{v}_k)\|^2}{\bar{z}_{k+1}}$$

$$\leq \sum_{k=k_0}^{t} \frac{\|\nabla_v R(\mathbf{x}_k, \mathbf{y}_k, \mathbf{v}_k)\|^2}{\bar{z}_0} \leq \frac{n([\bar{m}_{t+1}^v]^2 - [\bar{m}_0^v]^2)}{\bar{m}_0^v} \leq \frac{nC_{m^v}^2}{\bar{m}_0^v} - n\bar{m}_0^v \leq b_3. \tag{96}$$

**Case 2:** If $\bar{m}_{t+1}^v > C_{m^v}$, we have $k_3 \leq t$, where $k_3$ refers to Lemma C.5.

$$\sum_{k=k_0}^{t} \frac{\|\nabla_v R(\mathbf{x}_k, \mathbf{y}_k, \mathbf{v}_k)\|^2}{\bar{z}_{k+1}}$$

$$\stackrel{(a)}{\leq} \sum_{k=k_0}^{k_3-1} \frac{\|\nabla_v R(\mathbf{x}_k, \mathbf{y}_k, \mathbf{v}_k)\|^2}{\bar{z}_{k+1}} + \sum_{k=k_3}^{t} \frac{\|\nabla_v R(\mathbf{x}_k, \mathbf{y}_k, \mathbf{v}_k)\|^2}{\bar{z}_{k+1}}$$

$$\stackrel{(b)}{\leq} \frac{nC_{m^v}^2}{\bar{m}_0^v} - n\bar{m}_0^v + \frac{4nC_{m^y}^2(\mu + C_{l_{yy}})}{\mu^4 \gamma_v \sqrt{1 + \sigma_z^2}} \left( \frac{nL_{l,2}C_{f_y}}{\mu} + L_{f,1} \right)^2 + \frac{8nC_{m^v}^2(\mu + C_{l_{yy}})}{\mu^2 \gamma_v \sqrt{1 + \sigma_z^2}}$$

$$+ \left( \frac{16\Delta_0^x}{1 - \rho_W} + \frac{64\gamma_x^2 \rho_W C_{m^x}^2 (1 + \zeta_q^2)}{\bar{q}_0^2 (1 - \rho_W)^2} \right) \left( \frac{\bar{L}_r^2(\mu + C_{l_{yy}})}{\mu C_{l_{yy}} \gamma_v} + \frac{\bar{L}_r^2(\mu + C_{l_{yy}})}{\mu^2 \gamma_v \sqrt{1 + \sigma_z^2}} \right)$$

$$+ \frac{16(\mu + C_{l_{yy}})}{\gamma_v \mu^2} \left( \frac{nL_{l,2}C_{f_y}}{\mu} + L_{f,1} \right)^2 \left( \frac{\Delta_0^x}{1 - \rho_W} + \frac{4\gamma_x^2 \rho_W C_{m^x}^2 (1 + \zeta_q^2)}{\bar{q}_0^2 (1 - \rho_W)^2} \right)$$

$$\cdot \left( \frac{(\mu + L_{l,1})^2}{n\mu^2} + \frac{2L_{l,1}}{n\mu} \right) + \frac{16(\mu + C_{l_{yy}})}{\gamma_v \mu^2 \sqrt{1 + \sigma_u^2}} \left( \frac{nL_{l,2}C_{f_y}}{\mu} + L_{f,1} \right)^2$$

$$\cdot \left[ \frac{4\gamma_x^2 \rho_W (1 + \zeta_q^2)}{\bar{z}_0^2 (1 - \rho_W)^2} \left( \frac{(\mu + L_{l,1})^2 \sqrt{1 + \sigma_u^2}}{n\mu^2} + \frac{2L_{l,1}\sqrt{1 + \sigma_u^2}}{n\mu} \right) \right.$$

$$\left. + \frac{\gamma_x^2 L_y^2 (1 + \zeta_q^2)(\mu + L_{l,1})}{n\bar{z}_0 \gamma_y \mu L_{l,1}} \right] \sum_{k=\min\{k_1-1, k_2-1\}}^{k_3-2} \frac{\|\bar{\nabla}F(\mathbf{x}_k, \mathbf{y}_k, \mathbf{v}_k)\|^2}{[\bar{m}_{k+1}^x]^2}$$

$$+ \left( \frac{64\gamma_x^2 \rho_W \bar{L}_r^2(\mu + C_{l_{yy}})(1 + \zeta_q^2)}{\mu C_{l_{yy}} \gamma_v (1 - \rho_W)^2} + \frac{2\gamma_x^2 L_v^2 (1 + \zeta_q^2)(\mu + C_{l_{yy}})^2}{\mu C_{l_{yy}} C_{m^v} \gamma_v^2 \sqrt{1 + \sigma_z^2}} \right)$$

$$\cdot \sum_{k=\min\{k_1-1, k_3-1\}}^{t} \frac{\|\bar{\nabla}F(\mathbf{x}_k, \mathbf{y}_k, \mathbf{v}_k)\|^2}{[\bar{m}_{k+1}^x]^2}$$

$$+ \left( \frac{8}{\mu^2} + \frac{4(\mu + C_{l_{yy}})^2}{\mu^3 C_{l_{yy}}} \right) \left( \frac{nL_{l,2}C_{f_y}}{\mu} + L_{f,1} \right)^2 \sum_{k=k_3-1}^{t} \frac{\|\nabla_y L(\mathbf{x}_k, \mathbf{y}_k)\|^2}{\bar{m}_{k+1}^y}$$

$$\stackrel{(c)}{\leq} \frac{nC_{m^v}^2}{\bar{m}_0^v} - n\bar{m}_0^v + \frac{4nC_{m^y}^2(\mu + C_{l_{yy}})}{\mu^4 \gamma_v \sqrt{1 + \sigma_z^2}} \left( \frac{nL_{l,2}C_{f_y}}{\mu} + L_{f,1} \right)^2 + \frac{8nC_{m^v}^2(\mu + C_{l_{yy}})}{\mu^2 \gamma_v \sqrt{1 + \sigma_z^2}}$$

$$+ \left( \frac{16\Delta_0^x}{1 - \rho_W} + \frac{64\gamma_x^2 \rho_W C_{m^x}^2 (1 + \zeta_q^2)}{\bar{q}_0^2 (1 - \rho_W)^2} \right) \left( \frac{\bar{L}_r^2 (\mu + C_{l_{yy}})}{\mu C_{l_{yy}} \gamma_v} + \frac{\bar{L}_r^2 (\mu + C_{l_{yy}})}{\mu^2 \gamma_v \sqrt{1 + \sigma_z^2}} \right)$$

$$+ \frac{16(\mu + C_{l_{yy}})}{\gamma_v \mu^2} \left( \frac{nL_{l,2}C_{f_y}}{\mu} + L_{f,1} \right)^2 \left( \frac{\Delta_0^x}{1 - \rho_W} + \frac{4\gamma_x^2 \rho_W C_{m^x}^2 (1 + \zeta_q^2)}{\bar{q}_0^2 (1 - \rho_W)^2} \right)$$

$$\cdot \left( \frac{(\mu + L_{l,1})^2}{n\mu^2} + \frac{2L_{l,1}}{n\mu} \right) + \left[ \frac{16(\mu + C_{l_{yy}})}{\gamma_v \mu^2 \sqrt{1 + \sigma_u^2}} \left( \frac{nL_{l,2}C_{f_y}}{\mu} + L_{f,1} \right)^2 \right.$$

$$\cdot \left( \frac{4\gamma_x^2 \rho_W (1 + \zeta_q^2)}{\bar{z}_0^2 (1 - \rho_W)^2} \left( \frac{(\mu + L_{l,1})^2 \sqrt{1 + \sigma_u^2}}{n\mu^2} + \frac{2L_{l,1}\sqrt{1 + \sigma_u^2}}{n\mu} \right) + \frac{\gamma_x^2 L_y^2 (1 + \zeta_q^2)(\mu + L_{l,1})}{n\bar{z}_0 \gamma_y \mu L_{l,1}} \right)$$

$$\left. + \left( \frac{64\gamma_x^2 \rho_W \bar{L}_r^2 (\mu + C_{l_{yy}})(1 + \zeta_q^2)}{\mu C_{l_{yy}} \gamma_v (1 - \rho_W)^2} + \frac{2\gamma_x^2 L_v^2 (1 + \zeta_q^2)(\mu + C_{l_{yy}})^2}{\mu C_{l_{yy}} C_{m^v} \gamma_v^2 \sqrt{1 + \sigma_z^2}} \right) \right] (5\log(t + 1) + c_2)$$

$$+ \left( \frac{8}{\mu^2} + \frac{4(\mu + C_{l_{yy}})^2}{\mu^3 C_{l_{yy}}} \right) \left( \frac{nL_{l,2}C_{f_y}}{\mu} + L_{f,1} \right)^2 (a_2 \log(t + 1) + b_2)$$

$$\overset{(d)}{=} : a_3 \log(t + 1) + b_3, \tag{97}$$

where (a) allows $\sum_{k=k_0}^{k_3 - 1} \frac{\|\nabla_v R(\mathbf{x}_k, \mathbf{y}_k, \mathbf{v}_k)\|^2}{\bar{z}_{k+1}} = 0$ when $k_0 \geq k_3$; (b) uses $C_{m^v} \geq \bar{m}_0^v$ and Lemma D.3; (c) follows from Eq. (93) and Eq. (95); (d) refers to Eq. (92). Since the upper bound of Case 2 is larger, we take Eq. (97) as our final result.

Thus, the proof is complete. $\qquad \square$

### D.7 The Upper Bound of Consensus Errors

**Lemma D.7.** *Suppose Assumption 2.1, Assumption 3.1, and Assumption 3.2 hold. Then, the consensus error $\Delta$ satisfies:*

$$\sum_{k=0}^t \Delta_k \leq \frac{2\Delta_0}{1 - \rho_W} + \frac{8\gamma_x^2 \rho_W (1 + \zeta_q^2)(5\log(t) + c_2)}{(1 - \rho_W)^2}$$

$$+ \frac{8\gamma_y^2 \rho_W (1 + \zeta_u^2)(a_2 \log(t) + b_2)}{(1 - \rho_W)^2} + \frac{8\gamma_v^2 \rho_W (1 + \zeta_z^2)(a_3 \log(t) + b_3)}{(1 - \rho_W)^2}, \tag{98}$$

*where $\Delta_0$ is the initial consensus error, which can be set to 0 with proper initialization.*

*Proof.* According to Lemma C.4, we have:

$$\sum_{k=0}^{t-1} \|\mathbf{x}_{t+1} - \mathbf{1}\bar{x}_{t+1}\|^2 \leq \frac{2}{1 - \rho_W} \|\mathbf{x}_0 - \mathbf{1}\bar{x}_0\|^2 + \frac{8\gamma_x^2 \rho_W (1 + \zeta_q^2)}{(1 - \rho_W)^2} \sum_{k=0}^{t-1} \bar{q}_{k+1}^{-2} \|\bar{\nabla}F(\mathbf{x}_k, \mathbf{y}_k, \mathbf{v}_k)\|^2. \tag{99}$$

With the help of Lemma D.6, we have:

$$\sum_{k=0}^{t-1} \bar{q}_{k+1}^{-2} \|\bar{\nabla}F(\mathbf{x}_k, \mathbf{y}_k, \mathbf{v}_k)\|^2 \leq \sum_{k=0}^{t-1} \frac{\|\bar{\nabla}F(\mathbf{x}_k, \mathbf{y}_k, \mathbf{v}_k)\|^2}{[\bar{m}_{k+1}^x]^2} \leq 5\log(t) + c_2. \tag{100}$$

Similarly, we can get the following inequality for the dual variable:

$$\sum_{k=0}^{t-1} \|\mathbf{y}_{k+1} - \mathbf{1}\bar{y}_{k+1}\|^2 \leq \frac{2}{1 - \rho_W} \|\mathbf{y}_0 - \mathbf{1}\bar{y}_0\|^2 + \frac{8\gamma_y^2 \rho_W (1 + \zeta_u^2)(a_2 \log(t) + b_2)}{(1 - \rho_W)^2}. \tag{101}$$

For the auxiliary variable, we have:

$$v_{t+1} = \mathcal{P}_{\mathcal{V}} \left( W \left( \mathbf{v}_t - \gamma_v Z_{t+1}^{-1} \nabla_v R(\mathbf{x}_t, \mathbf{y}_t, \mathbf{v}_t) \right) \right) = W\mathbf{v}_t - \gamma_v \nabla_v \hat{G}, \tag{102}$$

where

$$\nabla_v \hat{G} = \frac{1}{\gamma_v} \left( \mathcal{P}_\mathcal{V} \left( W \left( \mathbf{v}_t - \gamma_v Z_{t+1}^{-1} \nabla_v R(\mathbf{x}_t, \mathbf{y}_t, \mathbf{v}_t) \right) \right) - W \mathbf{v}_t \right). \tag{103}$$

Using Young's inequality with parameter $\lambda$, we have:

$$\begin{aligned}
&\|\mathbf{v}_{t+1} - \mathbf{1}\bar{v}_{t+1}\|^2 \\
&= \left\| W\mathbf{v}_t - \gamma_v \nabla_v \hat{G} - \mathbf{J} \left( W\mathbf{v}_t - \gamma_v \nabla_v \hat{G} \right) \right\|^2 \\
&\leq (1+\lambda)\rho_W \|\mathbf{v}_t - \mathbf{J}\mathbf{v}_t\|^2 + \left( 1 + \frac{1}{\lambda} \right) \left\| \mathcal{P}_\mathcal{V} \left( W \left( \mathbf{v}_t - \gamma_v Z_{t+1}^{-1} \nabla_v R(\mathbf{x}_t, \mathbf{y}_t, \mathbf{v}_t) \right) \right) - W\mathbf{v}_t \right\|^2 \\
&\leq \frac{1+\rho_W}{2} \|\mathbf{v}_t - \mathbf{J}\mathbf{v}_t\|^2 + \frac{1+\rho_W}{1-\rho_W} \left\| \mathcal{P}_\mathcal{V} \left( W \left( \mathbf{v}_t - \gamma_v Z_{t+1}^{-1} \nabla_v R(\mathbf{x}_t, \mathbf{y}_t, \mathbf{v}_t) \right) \right) - W\mathbf{v}_t \right\|^2. \tag{104}
\end{aligned}$$

Noticing that $W\mathbf{v}_t = \mathcal{P}_\mathcal{V}(W\mathbf{v}_t)$ holds for the convex set $\mathcal{V}$, we get:

$$\begin{aligned}
&\|\mathbf{v}_{t+1} - \mathbf{1}\bar{v}_{t+1}\|^2 \\
&\leq \frac{1+\rho_W}{2} \|\mathbf{v}_t - \mathbf{J}\mathbf{v}_t\|^2 + \frac{1+\rho_W}{1-\rho_W} \left\| \mathcal{P}_\mathcal{V} \left( W \left( \mathbf{v}_t - \gamma_v Z_{t+1}^{-1} \nabla_v R(\mathbf{x}_t, \mathbf{y}_t, \mathbf{v}_t) \right) \right) - \mathcal{P}_\mathcal{V}(W\mathbf{v}_t) \right\|^2 \\
&\overset{(a)}{\leq} \frac{1+\rho_W}{2} \|\mathbf{v}_t - \mathbf{J}\mathbf{v}_t\|^2 + \frac{1+\rho_W}{1-\rho_W} \rho_W \|\gamma_v Z_{t+1}^{-1} \nabla_v R(\mathbf{x}_t, \mathbf{y}_t, \mathbf{v}_t)\|^2, \tag{105}
\end{aligned}$$

where (a) uses the non-expansiveness of the projection operator, as shown in Lemma 1 of [Nedic et al., 2010]. Then, we have:

$$\sum_{k=0}^{t-1} \|\mathbf{v}_k - \mathbf{1}\bar{v}_k\|^2 \leq \frac{2}{1-\rho_W} \|\mathbf{v}_0 - \mathbf{1}v_0\|^2 + \frac{8\gamma_v^2 \rho_W (1+\zeta_z^2)}{(1-\rho_W)^2} \sum_{k=0}^{t-1} \bar{z}_{k+1}^{-2} \|\nabla_v R(\mathbf{x}_k, \mathbf{y}_k, \mathbf{v}_k)\|^2. \tag{106}$$

Similar to the primal and the dual variable, we can bound the last term above, which completes the proof. $\qquad\square$

## D.8 The Upper Bounds of $\sum \frac{\|\bar{\nabla} f(\bar{x}_t, \bar{y}_t, \bar{v}_t)\|^2}{[\bar{m}_{t+1}^x]^2}$, $\sum \frac{\|\nabla_y l(\bar{x}_t, \bar{y}_t)\|^2}{\bar{m}_{t+1}^y}$, and $\sum \frac{\|\nabla_v r(\bar{x}_t, \bar{y}_t, \bar{v}_t)\|^2}{\bar{z}_{t+1}}$

Through Lemma C.3, Lemma D.6, and Lemma D.7, we can derive the upper bound for $\sum_{k=k_0}^{t} \frac{\|\bar{\nabla} f(\bar{x}_k, \bar{y}_k, \bar{v}_k)\|^2}{[\bar{m}_{k+1}^x]^2}$, $\sum_{k=k_0}^{t} \frac{\|\nabla_y l(\bar{x}_k, \bar{y}_k)\|^2}{\bar{m}_{k+1}^y}$, and $\sum_{k=k_0}^{t} \frac{\|\nabla_v r(\bar{x}_k, \bar{y}_k, \bar{v}_k)\|^2}{\bar{z}_{k+1}}$ in the following lemma.

**Lemma D.8.** *Under Assumption 2.1, Assumption 3.1 and Assumption 3.2, for any integer $k_0 \in [0, t)$, we have the upper bounds in terms of logarithmic functions as:*

$$\sum_{k=k_0}^{t} \frac{\|\bar{\nabla} f(\bar{x}_k, \bar{y}_k, \bar{v}_k)\|^2}{[\bar{m}_{k+1}^x]^2} \leq a_4 \log(t+1) + b_4,$$

$$\sum_{k=k_0}^{t} \frac{\|\nabla_y l(\bar{x}_k, \bar{y}_k)\|^2}{\bar{m}_{k+1}^y} \leq a_5 \log(t+1) + b_5,$$

$$\sum_{k=k_0}^{t} \frac{\|\nabla_v R(\bar{x}_k, \bar{y}_k, \bar{v}_k)\|^2}{\bar{z}_{k+1}} \leq a_6 \log(t+1) + b_6, \tag{107}$$

*where*

$$a_4 := \frac{5}{2n^2} + \frac{8\rho_W \bar{L}_f^2 (5\gamma_x^2 (1+\zeta_v^2) + a_2 \gamma_y^2 (1+\zeta_u^2) + a_3 \gamma_v^2 (1+\zeta_z^2))}{[\bar{m}_0^x]^2 (1-\rho_W)^2},$$

$$b_4 := \frac{c_2}{2n^2} + \frac{2\bar{L}_f^2 \Delta_0}{[\bar{m}_0^x]^2 (1-\rho_W)} + \frac{8\rho_W \bar{L}_f^2 (c_2 \gamma_x^2 (1+\zeta_v^2) + b_2 \gamma_y^2 (1+\zeta_u^2) + b_3 \gamma_v^2 (1+\zeta_z^2))}{[\bar{m}_0^x]^2 (1-\rho_W)^2},$$

$$a_5 := \frac{a_2}{2n^2} + \frac{8\rho_W L_{l,1}^2 (5\gamma_x^2 (1+\zeta_v^2) + a_2 \gamma_y^2 (1+\zeta_u^2))}{\bar{m}_0^y (1-\rho_W)^2},$$

$$b_5 := \frac{b_2}{2n^2} + \frac{2L_{l,1}^2(\|\mathbf{x}_0 - \mathbf{1}\bar{x}_0\|^2 + \|\mathbf{y}_0 - \mathbf{1}\bar{y}_0\|^2)}{\bar{m}_0^y(1 - \rho_W)} + \frac{8\rho_W L_{l,1}(c_2\gamma_x^2(1 + \zeta_v^2) + b_2\gamma_y^2(1 + \zeta_u^2))}{\bar{m}_0^y(1 - \rho_W)^2},$$

$$a_6 := \frac{a_3}{2n^2} + \frac{8\rho_W \bar{L}_R^2(5\gamma_x^2(1 + \zeta_v^2) + a_2\gamma_y^2(1 + \zeta_u^2) + a_3\gamma_v^2(1 + \zeta_z^2))}{\bar{m}_0^v(1 - \rho_W)^2},$$

$$b_6 := \frac{b_3}{2n^2} + \frac{2\bar{L}_R^2\Delta_0}{\bar{m}_0^v(1 - \rho_W)} + \frac{8\rho_W \bar{L}_R^2(c_2\gamma_x^2(1 + \zeta_v^2) + b_2\gamma_y^2(1 + \zeta_u^2) + b_3\gamma_v^2(1 + \zeta_z^2))}{\bar{m}_0^v(1 - \rho_W)^2}. \quad (108)$$

*Proof.* According to Lemma C.3, we have:

$$\|\bar{\nabla}F(\mathbf{x}_t, \mathbf{y}_t, \mathbf{v}_t)\|^2 \leq 2\|\bar{\nabla}F(\mathbf{1}\bar{x}_t, \mathbf{1}\bar{y}_t, \mathbf{1}\bar{v}_t)\|^2 + 2\bar{L}_f^2(\|\mathbf{x}_t - \mathbf{1}\bar{x}_t\|^2 + \|\mathbf{y}_t - \mathbf{1}\bar{y}_t\|^2 + \|\mathbf{v}_t - \mathbf{1}\bar{v}_t\|^2),$$
$$(109)$$

$$\|\nabla_y L(\mathbf{x}_t, \mathbf{y}_t)\|^2 \leq 2\|\nabla_y L(\mathbf{1}\bar{x}_t, \mathbf{1}\bar{y}_t)\|^2 + 2L_{l,1}^2(\|\mathbf{x}_t - \mathbf{1}\bar{x}_t\|^2 + \|\mathbf{y}_t - \mathbf{1}\bar{y}_t\|^2), \quad (110)$$

$$\|\nabla_v R(\mathbf{x}_t, \mathbf{y}_t, \mathbf{v}_t)\|^2 \leq 2\|\nabla_v R(\mathbf{1}\bar{x}_t, \mathbf{1}\bar{y}_t, \mathbf{1}\bar{v}_t)\|^2 + 2\bar{L}_r^2(\|\mathbf{x}_t - \mathbf{1}\bar{x}_t\|^2 + \|\mathbf{y}_t - \mathbf{1}\bar{y}_t\|^2 + \|\mathbf{v}_t - \mathbf{1}\bar{v}_t\|^2). \quad (111)$$

Based on Eq. (26), we have $\|\bar{\nabla}F(\mathbf{1}\bar{x}_t, \mathbf{1}\bar{y}_t, \mathbf{1}\bar{v}_t)\|^2 \leq \|\bar{\nabla}F(\mathbf{1}\bar{x}_t, \mathbf{1}\bar{y}_t, \mathbf{1}\bar{v}_t)\|_F^2 = \|n\bar{\nabla}f(\bar{x}_t, \bar{y}_t, \bar{v}_t)\|^2$, $\|\nabla_y L(\mathbf{1}\bar{x}_t, \mathbf{1}\bar{y}_t)\|^2 \leq \|\nabla_y L(\mathbf{1}\bar{x}_t, \mathbf{1}\bar{y}_t)\|_F^2 = \|n\nabla_y l(\bar{x}_t, \bar{y}_t)\|^2$ and $\|\nabla_v R(\mathbf{1}\bar{x}_t, \mathbf{1}\bar{y}_t, \mathbf{1}\bar{v}_t)\|^2 \leq \|\nabla_v R(\mathbf{1}\bar{x}_t, \mathbf{1}\bar{y}_t, \mathbf{1}\bar{v}_t)\|_F^2 = \|n\nabla_v r(\bar{x}_t, \bar{y}_t, \bar{v}_t)\|^2$, then according to Lemma D.6 and Lemma D.7, we have:

$$\sum_{k=k_0}^{t} \frac{\|\bar{\nabla}f(\bar{x}_k, \bar{y}_k, \bar{v}_k)\|^2}{[\bar{m}_{k+1}^x]^2} \leq \frac{5\log(t+1) + c_2}{2n^2} + \frac{\bar{L}_f^2 d_1}{[\bar{m}_0^x]^2} \leq a_4\log(t+1) + b_4,$$

$$\sum_{k=k_0}^{t} \frac{\|\nabla_y l(\bar{x}_k, \bar{y}_k)\|^2}{\bar{m}_{k+1}^y} \leq \frac{a_2\log(t+1) + b_2}{2n^2} + \frac{L_{l,1}^2 d_2}{\bar{m}_0^y} \leq a_5\log(t+1) + b_5,$$

$$\sum_{k=k_0}^{t} \frac{\|\nabla_v r(\bar{x}_k, \bar{y}_k, \bar{v}_k)\|^2}{\bar{z}_{k+1}} \leq \frac{a_3\log(t+1) + b_3}{2n^2} + \frac{\bar{L}_R^2 d_1}{\bar{m}_0^y} \leq a_6\log(t+1) + b_6, \quad (112)$$

where $a_4, b_4, a_5, b_5, a_6, b_6$ can refer to Eq. (108) and

$$d_1 := \frac{2(\|\mathbf{x}_0 - \mathbf{1}\bar{x}_0\|^2 + \|\mathbf{y}_0 - \mathbf{1}\bar{y}_0\|^2 + \|\mathbf{v}_0 - \mathbf{1}\bar{v}_0\|^2)}{1 - \rho_W} + \frac{8\gamma_x^2\rho_W(1 + \zeta_q^2)(5\log(t+1) + c_2)}{(1 - \rho_W)^2},$$

$$+ \frac{8\gamma_y^2\rho_W(1 + \zeta_u^2)(a_2\log(t+1) + b_2)}{(1 - \rho_W)^2} + \frac{8\gamma_v^2\rho_W(1 + \zeta_z^2)(a_3\log(t+1) + b_3)}{(1 - \rho_W)^2}$$

$$d_2 := \frac{2(\|\mathbf{x}_0 - \mathbf{1}\bar{x}_0\|^2 + \|\mathbf{y}_0 - \mathbf{1}\bar{y}_0\|^2)}{1 - \rho_W}$$

$$+ \frac{8\gamma_x^2\rho_W(1 + \zeta_q^2)(5\log(t+1) + c_2)}{(1 - \rho_W)^2} + \frac{8\gamma_y^2\rho_W(1 + \zeta_u^2)(a_2\log(t+1) + b_2)}{(1 - \rho_W)^2}. \quad (113)$$

Thus, the proof is completed. $\qquad \square$

### D.9 The Upper Bound of Stepsize Inconsistencies

**Lemma D.9.** *Suppose Assumption 2.1, Assumption 3.1, and Assumption 3.2 hold. For the proposed Algorithm 1, we have:*

$$\sum_{k=0}^{t-1} \left\| \frac{(\tilde{\mathbf{q}}_{k+1}^{-1})^\top}{n\bar{q}_{k+1}^{-1}} \bar{\nabla}F(\mathbf{x}_k, \mathbf{y}_k, \mathbf{v}_k) \right\|^2$$

$$\leq \frac{(1 + \zeta_q^2)(5\log(t) + c_2)(2(1 + \rho_W)\rho_W)}{n\bar{z}_0^2(1 - \rho_W)^2} \left( \frac{C_{l_{xy}}C_{f_y}}{\mu} + C_{f_x} \right)^2 \left[ C_{l_y}^2 + \left( \frac{C_{l_{yy}}C_{f_y}}{\mu} + C_{f_y} \right)^2 \right],$$

$$\sum_{k=0}^{t-1} \left\| \frac{(\tilde{\mathbf{u}}_{k+1}^{-1})^\top}{n\bar{u}_{k+1}^{-1}} \nabla_y L(\mathbf{x}_k, \mathbf{y}_k) \right\|^2 \leq \frac{(1 + \zeta_u^2)(a_2\log(t) + b_2)(2(1 + \rho_W)\rho_W)C_{l_y}^2}{n\bar{u}_0(1 - \rho_W)^2},$$

$$\sum_{k=0}^{t-1} \left\| \frac{\left(\tilde{\mathbf{z}}_{k+1}^{-1}\right)^\top}{n\bar{z}_{k+1}^{-1}} \nabla_v R(\mathbf{x}_k, \mathbf{y}_k, \mathbf{v}_k) \right\|^2$$

$$\leq \frac{(1+\zeta_z^2)(a_3 \log(t) + b_3)(2(1+\rho_W)\rho_W)}{n\bar{z}_0(1-\rho_W)^2} \left[ C_{l_y}^2 + \left( \frac{C_{l_{yy}}C_{f_y}}{\mu} + C_{f_y} \right)^2 \right]. \tag{114}$$

*Proof.* By the definition of $q_{i,k}$ in Eq. (25), we have:

$$\left\| \frac{\left(\tilde{\mathbf{q}}_{t+1}^{-1}\right)^\top}{n\bar{q}_{t+1}^{-1}} \bar{\nabla} F(\mathbf{x}_t, \mathbf{y}_t, \mathbf{v}_t) \right\|^2$$

$$\leq \frac{1}{n^2} \sum_{i=1}^{n} (\bar{q}_{t+1} - q_{i,t+1})^2 \frac{\|\bar{\nabla} F(\mathbf{x}_t, \mathbf{y}_t, \mathbf{v}_t)\|^2}{q_{i,t+1}^2}$$

$$\leq \sum_{i=1}^{n} \|\bar{q}_{t+1} - q_{i,t+1}\|^2 \frac{1+\zeta_q^2}{n^2} \frac{\|\bar{\nabla} F(\mathbf{x}_t, \mathbf{y}_t, \mathbf{v}_t)\|^2}{\bar{q}_{t+1}^2}$$

$$\leq \sum_{i=1}^{n} \|\bar{q}_{t+1} - q_{i,t+1}\|^2 \frac{1+\zeta_q^2}{n^2 \bar{z}_0^2} \frac{\|\bar{\nabla} F(\mathbf{x}_t, \mathbf{y}_t, \mathbf{v}_t)\|^2}{[\bar{m}_{t+1}^x]^2}. \tag{115}$$

According to Eq. (91), we have:

$$\sum_{k=0}^{t-1} \left\| \frac{\left(\tilde{\mathbf{q}}_{k+1}^{-1}\right)^\top}{n\bar{q}_{k+1}^{-1}} \bar{\nabla} F(\mathbf{x}_k, \mathbf{y}_k, \mathbf{v}_k) \right\|^2$$

$$\leq \frac{1+\zeta_q^2}{n^2 \bar{z}_0^2} \|\mathbf{q}_{k+1} - \mathbf{1}\bar{q}_{k+1}\|^2 \sum_{k=0}^{t-1} \frac{\|\bar{\nabla} F(\mathbf{x}_k, \mathbf{y}_k, \mathbf{v}_k)\|^2}{[\bar{m}_{k+1}^x]^2}$$

$$\overset{(a)}{\leq} \frac{(1+\zeta_q^2)(5\log(t+1) + c_2)}{n^2 \bar{z}_0^2} \|\mathbf{q}_{k+1} - \mathbf{1}\bar{q}_{k+1}\|^2, \tag{116}$$

where (a) uses Lemma D.6. Next, for the term of inconsistency of the stepsize $\|\mathbf{q}_k - \mathbf{1}\bar{q}_k\|^2$, we consider two cases due to the max operator used (i.e., $\mathbf{m}_k^y \geq \mathbf{m}_k^v$ and $\mathbf{m}_k^y < \mathbf{m}_k^v$). First of all, we derive the bound for $\|\mathbf{m}_{k+1}^x - \mathbf{1}\bar{m}_{k+1}^x\|^2$, $\|\mathbf{m}_{k+1}^y - \mathbf{1}\bar{m}_{k+1}^y\|^2$, and $\|\mathbf{m}_{k+1}^v - \mathbf{1}\bar{m}_{k+1}^v\|^2$. For $\|\mathbf{m}_{k+1}^x - \mathbf{1}\bar{m}_{k+1}^x\|^2$, we have:

$$\|\mathbf{m}_{k+1}^x - \mathbf{1}\bar{m}_{k+1}^x\|^2$$

$$\leq \|(W-\mathbf{J})(\mathbf{m}_k^x - \mathbf{1}\bar{m}_k^x)\|^2 + \|(W-\mathbf{J})\mathbf{g}_k^x\|^2$$

$$\leq \frac{1+\rho_W}{2} \|\mathbf{m}_k^x - \mathbf{1}\bar{m}_k^x\|^2 + \frac{(1+\rho_W)\rho_W}{1-\rho_W} \|\mathbf{g}_k^x\|^2$$

$$\overset{(a)}{\leq} \left( \frac{1+\rho_W}{2} \right)^k \|\mathbf{m}_0^x - \mathbf{1}\bar{m}_0^x\|^2 + \frac{n(1+\rho_W)\rho_W}{(1-\rho_W)^2} \left( \frac{C_{l_{xy}}C_{f_y}}{\mu} + C_{f_x} \right)^2 \sum_{t=0}^{k} \left( \frac{1+\rho_W}{2} \right)^{k-t}$$

$$\leq \frac{2n(1+\rho_W)\rho_W}{(1-\rho_W)^2} \left( \frac{C_{l_{xy}}C_{f_y}}{\mu} + C_{f_x} \right)^2, \tag{117}$$

where (a) uses Lemma C.2. For $\|\mathbf{m}_{k+1}^y - \mathbf{1}\bar{m}_{k+1}^y\|^2$, we have:

$$\|\mathbf{m}_{k+1}^y - \mathbf{1}\bar{m}_{k+1}^y\|^2$$

$$\leq \|(W-\mathbf{J})(\mathbf{m}_k^y - \mathbf{1}\bar{m}_k^y)\|^2 + \|(W-\mathbf{J})\mathbf{g}_k^y\|^2$$

$$\leq \frac{1+\rho_W}{2} \|\mathbf{m}_k^y - \mathbf{1}\bar{m}_k^y\|^2 + \frac{(1+\rho_W)\rho_W}{1-\rho_W} \|\mathbf{g}_k^y\|^2$$

$$\overset{(a)}{\leq} \left( \frac{1+\rho_W}{2} \right)^k \|\mathbf{m}_0^y - \mathbf{1}\bar{m}_0^y\|^2 + \frac{nC_{l_y}^2(1+\rho_W)\rho_W}{(1-\rho_W)^2} \sum_{t=0}^{k} \left( \frac{1+\rho_W}{2} \right)^{k-t}$$

$$\leq \frac{2nC_{l_y}^2(1+\rho_W)\rho_W}{(1-\rho_W)^2}, \tag{118}$$

where $C_{l_y} = n^2(C_{m^y} + c_0 + d_0(5\log 2 + c_2))$. The inequality in (a) follows from Lemma D.4 and Lemma D.6. Specifically, since $\|\mathbf{h}_0^y\|^2 \leq \|\mathbf{m}_1^y\|^2 \leq \|n\bar{m}_1^y\|^2 \leq n^2(C_{m^y} + c_0 + d_0(5\log 2 + c_2))$, it follows that the magnitude of the gradient $\|\mathbf{h}_0^y\|$ is upper bounded by $C_{l_y}$. For $\left\|\mathbf{m}_{k+1}^v - \mathbf{1}\bar{m}_{k+1}^v\right\|^2$, we have:

$$
\begin{aligned}
&\left\|\mathbf{m}_{k+1}^v - \mathbf{1}\bar{m}_{k+1}^v\right\|^2 \\
&\leq \|(W-\mathbf{J})(\mathbf{m}_k^v - \mathbf{1}\bar{m}_k^v)\|^2 + \|(W-\mathbf{J})\mathbf{g}_k^v\|^2 \\
&\leq \frac{1+\rho_W}{2}\|\mathbf{m}_k^v - \mathbf{1}\bar{m}_k^v\|^2 + \frac{(1+\rho_W)\rho_W}{1-\rho_W}\|\mathbf{g}_k^v\|^2 \\
&\overset{(a)}{\leq} \left(\frac{1+\rho_W}{2}\right)^k \|\mathbf{m}_0^v - \mathbf{1}\bar{m}_0^v\|^2 + \frac{n(1+\rho_W)\rho_W}{(1-\rho_W)^2}\left(\frac{C_{l_{yy}}C_{f_y}}{\mu} + C_{f_y}\right)^2 \sum_{t=0}^{k}\left(\frac{1+\rho_W}{2}\right)^{k-t} \\
&\leq \frac{2n(1+\rho_W)\rho_W}{(1-\rho_W)^2}\left(\frac{C_{l_{yy}}C_{f_y}}{\mu} + C_{f_y}\right)^2, \tag{119}
\end{aligned}
$$

where (a) uses Lemma C.3.

At iteration $k$, for the case $\mathbf{m}_k^y \geq \mathbf{m}_k^v$ with $\|\mathbf{m}_0^x\mathbf{m}_0^y - \mathbf{1}\bar{m}_0^x\bar{m}_0^y\|^2 = 0$, we have:

$$
\begin{aligned}
\|\mathbf{q}_{k+1} - \mathbf{1}\bar{q}_{k+1}\|^2 &= \left\|\mathbf{m}_{k+1}^x\mathbf{m}_{k+1}^y - \mathbf{1}\bar{m}_{k+1}^x\bar{m}_{k+1}^y\right\|^2 \\
&\leq \|(W-\mathbf{J})(\mathbf{m}_k^x\mathbf{m}_k^y - \mathbf{1}\bar{m}_k^x\bar{m}_k^y)\|^2 + \|(W-\mathbf{J})\mathbf{g}_k^x\mathbf{g}_k^y\|^2 \\
&\leq \frac{1+\rho_W}{2}\|\mathbf{m}_k^x\mathbf{m}_k^y - \mathbf{1}\bar{m}_k^x\bar{m}_k^y\|^2 + \frac{(1+\rho_W)\rho_W}{1-\rho_W}\|\mathbf{g}_k^x\mathbf{g}_k^y\|^2 \\
&\leq \left(\frac{1+\rho_W}{2}\right)^k \|\mathbf{m}_0^x\mathbf{m}_0^y - \mathbf{1}\bar{m}_0^x\bar{m}_0^y\|^2 \\
&\quad + \frac{nC_{l_y}^2(1+\rho_W)\rho_W}{(1-\rho_W)^2}\left(\frac{C_{l_{xy}}C_{f_y}}{\mu} + C_{f_x}\right)^2 \sum_{t=0}^{k}\left(\frac{1+\rho_W}{2}\right)^{k-t} \\
&\leq \frac{2nC_{l_y}^2(1+\rho_W)\rho_W}{(1-\rho_W)^2}\left(\frac{C_{l_{xy}}C_{f_y}}{\mu} + C_{f_x}\right)^2. \tag{120}
\end{aligned}
$$

For the case $\mathbf{m}_k^y < \mathbf{m}_k^v$ with $\|\mathbf{m}_0^x\mathbf{m}_0^v - \mathbf{1}\bar{m}_0^x\bar{m}_0^v\|^2 = 0$, we have:

$$
\begin{aligned}
&\|\mathbf{q}_{k+1} - \mathbf{1}\bar{q}_{k+1}\|^2 \\
&= \left\|\mathbf{m}_{k+1}^x\mathbf{m}_{k+1}^v - \mathbf{1}\bar{m}_{k+1}^x\bar{m}_{k+1}^v\right\|^2 \leq \frac{2n(1+\rho_W)\rho_W}{(1-\rho_W)^2}\left(\frac{C_{l_{xy}}C_{f_y}}{\mu} + C_{f_x}\right)^2\left(\frac{C_{l_{yy}}C_{f_y}}{\mu} + C_{f_y}\right)^2. \tag{121}
\end{aligned}
$$

By summing Eq. (120) and Eq. (121), we obtain the following inequality:

$$\|\mathbf{q}_{k+1} - \mathbf{1}\bar{q}_{k+1}\|^2 \leq \frac{2n(1+\rho_W)\rho_W}{(1-\rho_W)^2}\left(\frac{C_{l_{xy}}C_{f_y}}{\mu} + C_{f_x}\right)^2\left[C_{l_y}^2 + \left(\frac{C_{l_{yy}}C_{f_y}}{\mu} + C_{f_y}\right)^2\right]. \tag{122}$$

Combining Eq. (122) and Eq. (116), we can get the upper bound for $\sum_{k=0}^{t-1}\left\|\left(\tilde{\mathbf{u}}_{k+1}^{-1}\right)^\top \nabla_y L(\mathbf{x}_k, \mathbf{y}_k)/n\bar{u}_{k+1}^{-1}\right\|^2$ in Eq. (114).

Similarly, we have:

$$\|\mathbf{u}_{k+1} - \mathbf{1}\bar{u}_{k+1}\|^2 \leq \frac{2nC_{l_y}^2(1+\rho_W)\rho_W}{(1-\rho_W)^2}, \tag{123}$$

and

$$\|\mathbf{z}_{k+1} - \mathbf{1}\bar{z}_{k+1}\|^2 \leq \frac{2n(1+\rho_W)\rho_W}{(1-\rho_W)^2}\left[C_{l_y}^2 + \left(\frac{C_{l_{yy}}C_{f_y}}{\mu} + C_{f_y}\right)^2\right]. \tag{124}$$

Then combine the results in Lemma D.6, we can get the upper bound for $\sum_{k=0}^{t-1}\left\|(\tilde{\mathbf{z}}_{k+1}^{-1})^\top \nabla_v R(\mathbf{x}_k, \mathbf{y}_k, \mathbf{v}_k)/n\bar{z}_{k+1}^{-1}\right\|^2$ in Eq. (114). Thus, the Lemma D.9 has been proved. $\qquad\square$

## D.10 The Upper Bound of $\bar{m}_t^x$

**Lemma D.10.** *Under Assumption 3.1 and Assumption 3.2, suppose the number of total iteration rounds in Algorithm 1 is $T$. If there exists $k_1 \leq T$ as described in Lemma C.5, then we have:*

$$\bar{m}_t^x \leq \begin{cases} C_{m^x}, & t \leq k_1, \\ C_{m^x} + \left(4\left(\frac{\Phi(\bar{x}_0)-\Phi^*}{\gamma_x}\right) + a_7\log(t+1) + b_7\right)\bar{z}_{t+1}, & t \geq k_1, \end{cases} \tag{125}$$

*where $a_7$ and $b_7$ are defined as:*

$$
\begin{aligned}
a_7 :=\ & \frac{4\bar{L}^2 a_6}{\mu^2 \bar{m}_0^x} + \frac{2\bar{L}^2 a_5}{\mu^2 \bar{m}_0^x}\left(1 + \frac{2}{\mu^2}\left(\frac{L_{l,2}C_{f_y}}{\mu} + L_{f,1}\right)^2\right) \\
& + \frac{80(1+\zeta_q^2)((1+\rho_W)\rho_W)}{n\bar{z}_0^2\bar{q}_0(1-\rho_W)^2}\left(\frac{C_{l_{xy}}C_{f_y}}{\mu} + C_{f_x}\right)^2\left[C_{l_y}^2 + \left(\frac{C_{l_{yy}}C_{f_y}}{\mu} + C_{f_y}\right)^2\right] \\
& + \frac{4\bar{q}_0^{-1}(4\gamma_x\bar{L}_f^2 L_\Phi \bar{q}_0^{-1}\left(1+\zeta_q^2\right) + \bar{L}_f^2)}{n}\left(\frac{2\Delta_0}{1-\rho_W} + \frac{40\gamma_x^2\rho_W(1+\zeta_q^2)}{(1-\rho_W)^2}\right. \\
& \left. + \frac{8a_2\gamma_y^2\rho_W(1+\zeta_u^2)}{(1-\rho_W)^2} + \frac{8a_3\gamma_v^2\rho_W(1+\zeta_z^2)}{(1-\rho_W)^2}\right),
\end{aligned}
$$

$$
\begin{aligned}
b_7 :=\ & \frac{4\bar{L}^2 b_6}{\mu^2 \bar{m}_0^x} + \frac{2\bar{L}^2 b_5}{\mu^2 \bar{m}_0^x}\left(1 + \frac{2}{\mu^2}\left(\frac{L_{l,2}C_{f_y}}{\mu} + L_{f,1}\right)^2\right) \\
& + \frac{16c_2(1+\zeta_q^2)((1+\rho_W)\rho_W)}{n\bar{z}_0^2\bar{q}_0(1-\rho_W)^2}\left(\frac{C_{l_{xy}}C_{f_y}}{\mu} + C_{f_x}\right)^2\left[C_{l_y}^2 + \left(\frac{C_{l_{yy}}C_{f_y}}{\mu} + C_{f_y}\right)^2\right] \\
& + \frac{4\bar{q}_0^{-1}(4\gamma_x\bar{L}_f^2 L_\Phi \bar{q}_0^{-1}\left(1+\zeta_q^2\right) + \bar{L}_f^2)}{n}\left(\frac{2\Delta_0}{1-\rho_W} + \frac{8c_2\gamma_x^2\rho_W(1+\zeta_q^2)}{(1-\rho_W)^2}\right. \\
& \left. + \frac{8b_2\gamma_y^2\rho_W(1+\zeta_u^2)}{(1-\rho_W)^2} + \frac{8b_3\gamma_v^2\rho_W(1+\zeta_z^2)}{(1-\rho_W)^2}\right) + \frac{8n\gamma_x C_{m^x}^2 L_\Phi\left(1+\zeta_q^2\right)}{\bar{q}_0^2}, \tag{126}
\end{aligned}
$$

*and the upper bound of $\bar{z}_t := \max\{\bar{m}_t^y, \bar{m}_t^v\}$ refers to Lemma D.5. When such $k_1$ does not exist, we have $\bar{m}_t^x \leq C_{m^x}$ for any $t \leq T$.*

*Proof.* According to Lemma C.5, the proof can be split into the following three cases:

**Case 1:** If $\bar{m}_T^x \leq C_{m^x}$, for any $t < T$, we have the upper bound of $\bar{m}_{t+1}^x$ as $\bar{m}_{t+1}^x \leq C_{m^x}$.

**Case 2:** If $\bar{m}_T^x > C_{m^x}$, there exists $k_1 \leq T$ described in Lemma C.5. Then we have the upper bound of $\bar{m}_{t+1}^x$ as $\bar{m}_{t+1}^x \leq C_{m^x}$ for any $t < k_1$.

**Case 3:** In the remaining proof, we only consider and explore the case $k_1 \leq t \leq T$ when $\bar{m}_T^x > C_{m^x}$.

From Lemma D.1, for $k \geq k_1$, we have:

$\Phi(\bar{x}_{t+1})$

$$\leq \Phi(\bar{x}_t) - \frac{\gamma_x \bar{q}_{t+1}^{-1}}{8}\|\nabla\Phi(\bar{x}_t)\|^2 + \frac{\gamma_x \bar{q}_{t+1}^{-1}(2\gamma_x \bar{L}_f^2 L_\Phi \bar{q}_{t+1}^{-1}\left(1+\zeta_q^2\right) + \bar{L}_f^2)}{n}\Delta_t$$

$$-\left(\frac{\gamma_x}{2} - 2n\gamma_x^2 L_\Phi \bar{q}_{t+1}^{-1}\left(1+\zeta_q^2\right)\right)\frac{\|\nabla_x f(\bar{x}_t, \bar{y}_t, \bar{v}_t)\|^2}{\bar{q}_{t+1}} + \frac{\bar{L}^2 \gamma_x}{\mu^2}\frac{\|\nabla_v r(\bar{x}_t, \bar{y}_t, \bar{v}_t)\|^2}{\bar{q}_{t+1}}$$

$$+\left(\frac{\gamma_x \bar{L}^2}{2\mu^2} + \frac{\gamma_x \bar{L}^2}{\mu^4}\left(\frac{L_{l,2}C_{f_y}}{\mu} + L_{f,1}\right)^2\right)\frac{\|\nabla_y l(\bar{x}_t, \bar{y}_t)\|^2}{\bar{q}_{t+1}} + 2\gamma_x \bar{q}_{t+1}^{-1}\left\|\frac{(\tilde{\mathbf{q}}_{t+1}^{-1})^\top}{n\bar{q}_{t+1}^{-1}}\bar{\nabla}F(\mathbf{x}_t, \mathbf{y}_t, \mathbf{v}_t)\right\|^2.$$

(127)

In addition, if $k_1$ in Lemma C.5 exists, then for $t \geq k_1$, we have $\bar{m}_{t+1}^x > C_{m^x} \geq \frac{8n\gamma_x L_\Phi\left(1+\zeta_q^2\right)}{\bar{z}_0}$ and

$$\Phi(\bar{x}_{t+1})$$

$$\leq \Phi(\bar{x}_t) - \frac{\gamma_x \bar{q}_{t+1}^{-1}}{8}\|\nabla\Phi(\bar{x}_t)\|^2 + \frac{\gamma_x \bar{q}_{t+1}^{-1}(2\gamma_x \bar{L}_f^2 L_\Phi \bar{q}_{t+1}^{-1}\left(1+\zeta_q^2\right) + \bar{L}_f^2)}{n}\Delta_t$$

$$- \frac{\gamma_x}{4}\frac{\|\nabla_x f(\bar{x}_t, \bar{y}_t, \bar{v}_t)\|^2}{\bar{q}_{t+1}} + \frac{\bar{L}^2 \gamma_x}{\mu^2}\frac{\|\nabla_v r(\bar{x}_t, \bar{y}_t, \bar{v}_t)\|^2}{\bar{q}_{t+1}}$$

$$+\left(\frac{\gamma_x \bar{L}^2}{2\mu^2} + \frac{\gamma_x \bar{L}^2}{\mu^4}\left(\frac{L_{l,2}C_{f_y}}{\mu} + L_{f,1}\right)^2\right)\frac{\|\nabla_y l(\bar{x}_t, \bar{y}_t)\|^2}{\bar{q}_{t+1}} + 2\gamma_x \bar{q}_{t+1}^{-1}\left\|\frac{(\tilde{\mathbf{q}}_{t+1}^{-1})^\top}{n\bar{q}_{t+1}^{-1}}\bar{\nabla}F(\mathbf{x}_t, \mathbf{y}_t, \mathbf{v}_t)\right\|^2,$$

(128)

which indicates that

$$\bar{q}_{t+1}^{-1}\|\nabla_x f(\bar{x}_t, \bar{y}_t, \bar{v}_t)\|^2$$

$$\leq 4\left(\frac{\Phi(\bar{x}_t) - \Phi(\bar{x}_{t+1})}{\gamma_x}\right) + \frac{4\bar{q}_{t+1}^{-1}(2\gamma_x \bar{L}_f^2 L_\Phi \bar{q}_{t+1}^{-1}\left(1+\zeta_q^2\right) + \bar{L}_f^2)}{n}\Delta_t + \frac{4\bar{L}^2}{\mu^2}\cdot\frac{\|\nabla_v r(\bar{x}_t, \bar{y}_t, \bar{v}_t)\|^2}{\bar{q}_{t+1}}$$

$$+ 8\bar{q}_{t+1}^{-1}\left\|\frac{(\tilde{\mathbf{q}}_{t+1}^{-1})^\top}{n\bar{q}_{t+1}^{-1}}\bar{\nabla}F(\mathbf{x}_t, \mathbf{y}_t, \mathbf{v}_t)\right\|^2 + \left(\frac{2\bar{L}^2}{\mu^2} + \frac{4\bar{L}^2}{\mu^4}\left(\frac{L_{l,2}C_{f_y}}{\mu} + L_{f,1}\right)^2\right)\frac{\|\nabla_y l(\bar{x}_t, \bar{y}_t)\|^2}{\bar{q}_{t+1}}.$$

(129)

By taking summation, we have:

$$\sum_{k=k_1}^{t}\bar{q}_{k+1}^{-1}\|\nabla_x f(\bar{x}_k, \bar{y}_k, \bar{v}_k)\|^2$$

$$\leq 4\left(\frac{\Phi(\bar{x}_{k_1}) - \Phi^*}{\gamma_x}\right) + \frac{4\bar{q}_0^{-1}(2\gamma_x \bar{L}_f^2 L_\Phi \bar{q}_0^{-1}\left(1+\zeta_q^2\right) + \bar{L}_f^2)}{n}\sum_{k=k_1}^{t}\Delta_k$$

$$+ \frac{4\bar{L}^2}{\mu^2 \bar{m}_0^x}\sum_{k=k_1}^{t}\frac{\|\nabla_v r(\bar{x}_k, \bar{y}_k, \bar{v}_k)\|^2}{\max\{\bar{m}_{k+1}^y, \bar{m}_{k+1}^v\}} + \frac{8}{\bar{q}_0}\sum_{k=k_1}^{t}\left\|\frac{(\tilde{\mathbf{q}}_{k+1}^{-1})^\top}{n\bar{q}_{k+1}^{-1}}\bar{\nabla}F(\mathbf{x}_k, \mathbf{y}_k, \mathbf{v}_k)\right\|^2$$

$$+ \left(\frac{2\bar{L}^2}{\mu^2 \bar{m}_0^x} + \frac{4\bar{L}^2}{\mu^4 \bar{m}_0^x}\left(\frac{L_{l,2}C_{f_y}}{\mu} + L_{f,1}\right)^2\right)\sum_{k=k_1}^{t}\frac{\|\nabla_y l(\bar{x}_k, \bar{y}_k)\|^2}{\max\{\bar{m}_{k+1}^y, \bar{m}_{k+1}^v\}}.$$

(130)

For $\Phi(\bar{x}_{k_1})$, by telescoping Eq. (127), we get:

$$\Phi(\bar{x}_{k_1}) \leq \Phi(\bar{x}_0) + \frac{\gamma_x \bar{q}_0^{-1}(2\gamma_x \bar{L}_f^2 L_\Phi \bar{q}_0^{-1}\left(1+\zeta_q^2\right) + \bar{L}_f^2)}{n}\sum_{k=0}^{k_1-1}\Delta_k$$

$$+ \frac{\bar{L}^2 \gamma_x}{\mu^2}\sum_{k=0}^{k_1-1}\frac{\|\nabla_v r(\bar{x}_k, \bar{y}_k, \bar{v}_k)\|^2}{\bar{m}_{k+1}^x \max\{\bar{m}_{k+1}^y, \bar{m}_{k+1}^v\}} + \frac{2\gamma_x}{\bar{q}_0}\sum_{k=0}^{k_1-1}\left\|\frac{(\tilde{\mathbf{q}}_{k+1}^{-1})^\top}{n\bar{q}_{k+1}^{-1}}\bar{\nabla}F(\mathbf{x}_k, \mathbf{y}_k, \mathbf{v}_k)\right\|^2$$

$$+ 2n\gamma_x^2 L_\Phi\left(1+\zeta_q^2\right)\sum_{k=0}^{k_1-1}\frac{\|\nabla_x f(\bar{x}_k, \bar{y}_k, \bar{v}_k)\|^2}{[\bar{m}_{k+1}^x]^2 \max\{[\bar{m}_{k+1}^y]^2, [\bar{m}_{k+1}^v]^2\}}$$

$$+ \left(\frac{\gamma_x \bar{L}^2}{2\mu^2} + \frac{\gamma_x \bar{L}^2}{\mu^4}\left(\frac{L_{l,2}C_{f_y}}{\mu} + L_{f,1}\right)^2\right)\sum_{k=0}^{k_1-1}\frac{\|\nabla_y l(\bar{x}_k, \bar{y}_k)\|^2}{\bar{m}_{k+1}^x \max\{\bar{m}_{k+1}^y, \bar{m}_{k+1}^v\}}.$$

(131)

By plugging Eq. (131) into Eq. (130), we have:

$$\sum_{k=k_1}^{t} \bar{q}_{t+1}^{-1} \|\nabla_x f(\bar{x}_k, \bar{y}_k, \bar{v}_k)\|^2$$

$$\leq 4\left(\frac{\Phi(\bar{x}_0) - \Phi^*}{\gamma_x}\right) + \frac{4\bar{q}_0^{-1}(2\gamma_x \bar{L}_f^2 L_\Phi \bar{q}_0^{-1}(1 + \zeta_q^2) + \bar{L}_f^2)}{n}\sum_{k=0}^{t}\Delta_k$$

$$+ \frac{4\bar{L}^2}{\mu^2 \bar{m}_0^x}\sum_{k=0}^{t}\frac{\|\nabla_v r(\bar{x}_k, \bar{y}_k, \bar{v}_k)\|^2}{\max\{\bar{m}_{k+1}^y, \bar{m}_{k+1}^v\}} + \frac{8}{\bar{q}_0}\sum_{k=0}^{t}\left\|\frac{(\tilde{\mathbf{q}}_{k+1}^{-1})^\top}{n\bar{q}_{k+1}^{-1}}\bar{\nabla}F(\mathbf{x}_k, \mathbf{y}_k, \mathbf{v}_k)\right\|^2$$

$$+ \left(\frac{2\bar{L}^2}{\mu^2 \bar{m}_0^x} + \frac{4\bar{L}^2}{\mu^4 \bar{m}_0^x}\left(\frac{L_{l,2}C_{f_y}}{\mu} + L_{f,1}\right)^2\right)\sum_{k=0}^{t}\frac{\|\nabla_y l(\bar{x}_k, \bar{y}_k)\|^2}{\max\{\bar{m}_{k+1}^y, \bar{m}_{k+1}^v\}} + \frac{8n\gamma_x C_{m^x}^2 L_\Phi(1 + \zeta_q^2)}{\bar{q}_0^2}$$

$$\leq 4\left(\frac{\Phi(\bar{x}_0) - \Phi^*}{\gamma_x}\right) + \frac{4\bar{q}_0^{-1}(2\gamma_x \bar{L}_f^2 L_\Phi \bar{q}_0^{-1}(1 + \zeta_q^2) + \bar{L}_f^2)}{n}\sum_{k=0}^{t}\Delta_k$$

$$+ \frac{4\bar{L}^2}{\mu^2 \bar{m}_0^x}\sum_{k=0}^{t}\frac{\|\nabla_v r(\bar{x}_k, \bar{y}_k, \bar{v}_k)\|^2}{\max\{\bar{m}_{k+1}^y, \bar{m}_{k+1}^v\}} + \frac{8}{\bar{q}_0}\sum_{k=0}^{t}\left\|\frac{(\tilde{\mathbf{q}}_{k+1}^{-1})^\top}{n\bar{q}_{k+1}^{-1}}\bar{\nabla}F(\mathbf{x}_k, \mathbf{y}_k, \mathbf{v}_k)\right\|^2$$

$$+ \left(\frac{2\bar{L}^2}{\mu^2 \bar{m}_0^x} + \frac{4\bar{L}^2}{\mu^4 \bar{m}_0^x}\left(\frac{L_{l,2}C_{f_y}}{\mu} + L_{f,1}\right)^2\right)\sum_{k=0}^{t}\frac{\|\nabla_y l(\bar{x}_k, \bar{y}_k)\|^2}{\bar{m}_{k+1}^y} + \frac{8n\gamma_x C_{m^x}^2 L_\Phi(1 + \zeta_q^2)}{\bar{q}_0^2}$$

$$\overset{(a)}{\leq} 4\left(\frac{\Phi(\bar{x}_0) - \Phi^*}{\gamma_x}\right) + \frac{4\bar{L}^2(a_6 \log(t+1) + b_6)}{\mu^2 \bar{m}_0^x}$$

$$+ \frac{2\bar{L}^2(a_5 \log(t+1) + b_5)}{\mu^2 \bar{m}_0^x}\left(1 + \frac{2}{\mu^2}\left(\frac{L_{l,2}C_{f_y}}{\mu} + L_{f,1}\right)^2\right)$$

$$+ \frac{8(1 + \zeta_q^2)(5\log(t) + c_2)(2(1 + \rho_W)\rho_W)}{n\bar{z}_0^2 \bar{q}_0(1 - \rho_W)^2}\left(\frac{C_{l_{xy}}C_{f_y}}{\mu} + C_{f_x}\right)^2\left[C_{l_y}^2 + \left(\frac{C_{l_{yy}}C_{f_y}}{\mu} + C_{f_y}\right)^2\right]$$

$$+ \frac{4\bar{q}_0^{-1}(2\gamma_x \bar{L}_f^2 L_\Phi \bar{q}_0^{-1}(1 + \zeta_q^2) + \bar{L}_f^2)}{n}\left(\frac{2\Delta_0}{1 - \rho_W} + \frac{8\gamma_x^2 \rho_W(1 + \zeta_q^2)(5\log(t+1) + c_2)}{(1 - \rho_W)^2}\right.$$

$$+ \frac{8\gamma_y^2 \rho_W(1 + \zeta_u^2)(a_2 \log(t+1) + b_2)}{(1 - \rho_W)^2} + \frac{8\gamma_v^2 \rho_W(1 + \zeta_z^2)(a_3 \log(t+1) + b_3)}{(1 - \rho_W)^2}\Bigg)$$

$$+ \frac{8n\gamma_x C_{m^x}^2 L_\Phi(1 + \zeta_q^2)}{\bar{q}_0^2}$$

$$=: 4\left(\frac{\Phi(\bar{x}_0) - \Phi^*}{\gamma_x}\right) + a_7 \log(t+1) + b_7, \tag{132}$$

where (a) uses Lemma D.8 and Lemma D.9. This immediately implies that

$$\sum_{k=k_1}^{t}\frac{\|\nabla_x f(\bar{x}_k, \bar{y}_k, \bar{v}_k)\|^2}{\bar{m}_{k+1}^x} \leq \left(4\left(\frac{\Phi(\bar{x}_0) - \Phi^*}{\gamma_x}\right) + a_7 \log(t+1) + b_7\right)\bar{z}_{t+1}. \tag{133}$$

Similarly, we can have the upper bound of $\bar{m}_{t+1}^x$ as:

$$\bar{m}_{t+1}^x \leq \bar{m}_{k_1}^x + \sum_{k=k_1}^{t}\frac{\|\nabla_x f(\bar{x}_k, \bar{y}_k, \bar{v}_k)\|^2}{\bar{m}_{k+1}^x}$$

$$\leq C_{m^x} + \left(4\left(\frac{\Phi(\bar{x}_0) - \Phi^*}{\gamma_x}\right) + a_7 \log(t+1) + b_7\right)\bar{z}_{t+1}. \tag{134}$$

Then the upper bound of $\bar{m}_{t+1}^x$ is proved. $\qquad\square$

## D.11  Proof of Theorem 3.9

Here we still assume the total iteration rounds of Algorithm 1 is $T$. According to Lemma D.1, the proof can be split into the following two cases.

**Case 1:** If $\bar{m}_T^x \le C_{m^x}$, then we have:

$$\bar{q}_{t+1}^{-1}\|\nabla\Phi(\bar{x}_t)\|^2$$

$$\le 8\left(\frac{\Phi(\bar{x}_t) - \Phi(\bar{x}_{t+1})}{\gamma_x}\right) + \frac{8\bar{q}_{t+1}^{-1}(2\gamma_x\bar{L}_f^2 L_\Phi\bar{q}_{t+1}^{-1}\left(1+\zeta_q^2\right)+\bar{L}_f^2)}{n}\Delta_t + \frac{8\bar{L}^2}{\mu^2}\frac{\|\nabla_v r(\bar{x}_t,\bar{y}_t,\bar{v}_t)\|^2}{\bar{q}_{t+1}}$$

$$+ 16n\gamma_x L_\Phi\bar{q}_{t+1}^{-2}\left(1+\zeta_q^2\right)\|\nabla_x f(\bar{x}_t,\bar{y}_t,\bar{v}_t)\|^2 + 16\bar{q}_{t+1}^{-1}\left\|\frac{(\tilde{\mathbf{q}}_{t+1}^{-1})^\top}{n\bar{q}_{t+1}^{-1}}\bar{\nabla}F(\mathbf{x}_t,\mathbf{y}_t,\mathbf{v}_t)\right\|^2$$

$$+\left(\frac{4\bar{L}^2}{\mu^2} + \frac{8\bar{L}^2}{\mu^4}\left(\frac{L_{l,2}C_{f_y}}{\mu}+L_{f,1}\right)^2\right)\frac{\|\nabla_y l(\bar{x}_t,\bar{y}_t)\|^2}{\bar{q}_{t+1}}. \tag{135}$$

By taking the average, we have:

$$\frac{1}{T}\sum_{t=0}^{T-1}\bar{q}_{t+1}^{-1}\|\nabla\Phi(\bar{x}_t)\|^2$$

$$\le \frac{8}{T}\left(\frac{\Phi(\bar{x}_0) - \Phi(\bar{x}_T)}{\gamma_x}\right) + \frac{8\bar{q}_{t+1}^{-1}(2\gamma_x\bar{L}_f^2 L_\Phi\bar{q}_{t+1}^{-1}\left(1+\zeta_q^2\right)+\bar{L}_f^2)}{nT}\sum_{t=0}^{T-1}\Delta_t$$

$$+ \frac{16n\gamma_x L_\Phi\bar{q}_0^{-2}\left(1+\zeta_q^2\right)}{T}\sum_{t=0}^{T-1}\|\nabla_x f(\bar{x}_t,\bar{y}_t,\bar{v}_t)\|^2 + \frac{16\bar{q}_0^{-1}}{T}\sum_{t=0}^{T-1}\left\|\frac{(\tilde{\mathbf{q}}_{t+1}^{-1})^\top}{n\bar{q}_{t+1}^{-1}}\bar{\nabla}F(\mathbf{x}_t,\mathbf{y}_t,\mathbf{v}_t)\right\|^2$$

$$+\left(\frac{4\bar{L}^2}{\mu^2} + \frac{8\bar{L}^2}{\mu^4}\left(\frac{L_{l,2}C_{f_y}}{\mu}+L_{f,1}\right)^2\right)\frac{1}{T}\sum_{t=0}^{T-1}\frac{\|\nabla_y l(\bar{x}_t,\bar{y}_t)\|^2}{\bar{m}_{t+1}^x\max\{\bar{m}_{t+1}^y,\bar{m}_{t+1}^v\}}$$

$$+ \frac{8\bar{L}^2}{\mu^2 T}\sum_{t=0}^{T-1}\frac{\|\nabla_v r(\bar{x}_t,\bar{y}_t,\bar{v}_t)\|^2}{\bar{m}_{t+1}^x\max\{\bar{m}_{t+1}^y,\bar{m}_{t+1}^v\}}$$

$$\le \frac{8}{T}\left(\frac{\Phi(\bar{x}_0) - \Phi(\bar{x}_T)}{\gamma_x}\right) + \frac{8\bar{q}_{t+1}^{-1}(2\gamma_x\bar{L}_f^2 L_\Phi\bar{q}_{t+1}^{-1}\left(1+\zeta_q^2\right)+\bar{L}_f^2)}{nT}\sum_{t=0}^{T-1}\Delta_t$$

$$+ \frac{16n\gamma_x L_\Phi\bar{q}_0^{-2}C_{m^x}^2\left(1+\zeta_q^2\right)}{T} + \frac{16\bar{q}_0^{-1}}{T}\sum_{t=0}^{T-1}\left\|\frac{(\tilde{\mathbf{q}}_{t+1}^{-1})^\top}{n\bar{q}_{t+1}^{-1}}\bar{\nabla}F(\mathbf{x}_t,\mathbf{y}_t,\mathbf{v}_t)\right\|^2$$

$$+\left(\frac{4\bar{L}^2}{\bar{m}_0^x\mu^2} + \frac{8\bar{L}^2}{\bar{m}_0^x\mu^4}\left(\frac{L_{l,2}C_{f_y}}{\mu}+L_{f,1}\right)^2\right)\frac{1}{T}\sum_{t=0}^{T-1}\frac{\|\nabla_y l(\bar{x}_t,\bar{y}_t)\|^2}{\bar{m}_{t+1}^y}$$

$$+ \frac{8\bar{L}^2}{\mu^2 T\bar{m}_0^x}\sum_{t=0}^{T-1}\frac{\|\nabla_v r(\bar{x}_t,\bar{y}_t,\bar{v}_t)\|^2}{\max\{\bar{m}_{t+1}^y,\bar{m}_{t+1}^v\}}$$

$$\overset{(a)}{=} \frac{2}{T}\left(4\left(\frac{\Phi(\bar{x}_0)-\Phi^*}{\gamma_x}\right)+a_7\log(t+1)+b_7\right), \tag{136}$$

where (a) uses Lemma D.8 with $k_0 = 0$.

**Case 2:** If $\bar{m}_T^x > C_{m^x}$, by Lemma C.5, there exists $k_1 \le T_0$ such that $\bar{m}_{k_1}^x \le C_{m^x}$, $\bar{m}_{k_1+1}^x > C_{m^x}$. Then for $t < k_1$ when $\bar{m}_T^x > C_{m^x}$, from Eq. (35), we have:

$$\bar{q}_{t+1}^{-1}\|\nabla\Phi(\bar{x}_t)\|^2$$

$$\le 8\left(\frac{\Phi(\bar{x}_t) - \Phi(\bar{x}_{t+1})}{\gamma_x}\right) + \frac{8\bar{q}_{t+1}^{-1}(2\gamma_x\bar{L}_f^2 L_\Phi\bar{q}_{t+1}^{-1}\left(1+\zeta_q^2\right)+\bar{L}_f^2)}{n}\Delta_t + \frac{8\bar{L}^2}{\mu^2}\frac{\|\nabla_v r(\bar{x}_t,\bar{y}_t,\bar{v}_t)\|^2}{\bar{q}_{t+1}}$$

$$+ 16n\gamma_x L_\Phi \bar{q}_{t+1}^{-2} \left(1 + \zeta_q^2\right) \|\nabla_x f(\bar{x}_t, \bar{y}_t, \bar{v}_t)\|^2 + 16\bar{q}_{t+1}^{-1} \left\| \frac{(\tilde{\mathbf{q}}_{t+1}^{-1})^\top}{n\bar{q}_{t+1}^{-1}} \bar{\nabla} F(\mathbf{x}_t, \mathbf{y}_t, \mathbf{v}_t) \right\|^2$$

$$+ \left( \frac{4\bar{L}^2}{\mu^2} + \frac{8\bar{L}^2}{\mu^4} \left( \frac{L_{l,2} C_{f_y}}{\mu} + L_{f,1} \right)^2 \right) \frac{\|\nabla_y l(\bar{x}_t, \bar{y}_t)\|^2}{\bar{q}_{t+1}}. \tag{137}$$

For $t \geq k_1$ when $\bar{m}_T^x > C_{m^x}$, from Eq. (36), we have:

$$\bar{q}_{t+1}^{-1} \|\nabla \Phi(\bar{x}_t)\|^2$$

$$\leq 8 \left( \frac{\Phi(\bar{x}_t) - \Phi(\bar{x}_{t+1})}{\gamma_x} \right) + \frac{8\bar{q}_{t+1}^{-1}(2\gamma_x \bar{L}_f^2 L_\Phi \bar{q}_{t+1}^{-1} \left(1 + \zeta_q^2\right) + \bar{L}_f^2)}{n} \Delta_t + \frac{8\bar{L}^2}{\mu^2} \frac{\|\nabla_v r(\bar{x}_t, \bar{y}_t, \bar{v}_t)\|^2}{\bar{q}_{t+1}}$$

$$+ 16\bar{q}_{t+1}^{-1} \left\| \frac{(\tilde{\mathbf{q}}_{t+1}^{-1})^\top}{n\bar{q}_{t+1}^{-1}} \bar{\nabla} F(\mathbf{x}_t, \mathbf{y}_t, \mathbf{v}_t) \right\|^2 + \left( \frac{4\bar{L}^2}{\mu^2} + \frac{8\bar{L}^2}{\mu^4} \left( \frac{L_{l,2} C_{f_y}}{\mu} + L_{f,1} \right)^2 \right) \frac{\|\nabla_y l(\bar{x}_t, \bar{y}_t)\|^2}{\bar{q}_{t+1}}. \tag{138}$$

By taking the average, we can merge $t < k_1$ and $t \geq k_1$ as:

$$\frac{1}{T} \sum_{t=0}^{T-1} \bar{q}_{t+1}^{-1} \|\nabla \Phi(\bar{x}_t)\|^2$$

$$= \frac{1}{T} \sum_{t=0}^{k_1-1} \bar{q}_{t+1}^{-1} \|\nabla \Phi(\bar{x}_t)\|^2 + \frac{1}{T} \sum_{t=k_1}^{T-1} \bar{q}_{t+1}^{-1} \|\nabla \Phi(\bar{x}_t)\|^2$$

$$\leq \frac{8}{T} \left( \frac{\Phi(\bar{x}_0) - \Phi(\bar{x}_{k_1})}{\gamma_x} \right) + \frac{8\bar{q}_{t+1}^{-1}(2\gamma_x \bar{L}_f^2 L_\Phi \bar{q}_{t+1}^{-1} \left(1 + \zeta_q^2\right) + \bar{L}_f^2)}{nT} \sum_{t=0}^{k_1-1} \Delta_t$$

$$+ \frac{16n\gamma_x L_\Phi \bar{q}_0^{-2} \left(1 + \zeta_q^2\right)}{T} \sum_{t=0}^{k_1-1} \|\nabla_x f(\bar{x}_t, \bar{y}_t, \bar{v}_t)\|^2 + \frac{16\bar{q}_0^{-1}}{T} \sum_{t=0}^{k_1-1} \left\| \frac{(\tilde{\mathbf{q}}_{t+1}^{-1})^\top}{n\bar{q}_{t+1}^{-1}} \bar{\nabla} F(\mathbf{x}_t, \mathbf{y}_t, \mathbf{v}_t) \right\|^2$$

$$+ \left( \frac{4\bar{L}^2}{\mu^2} + \frac{8\bar{L}^2}{\mu^4} \left( \frac{L_{l,2} C_{f_y}}{\mu} + L_{f,1} \right)^2 \right) \frac{1}{T} \sum_{t=0}^{k_1-1} \frac{\|\nabla_y l(\bar{x}_t, \bar{y}_t)\|^2}{\bar{m}_{t+1}^x \max\{\bar{m}_{t+1}^y, \bar{m}_{t+1}^v\}}$$

$$+ \frac{8\bar{L}^2}{\mu^2 T} \sum_{t=0}^{k_1-1} \frac{\|\nabla_v r(\bar{x}_t, \bar{y}_t, \bar{v}_t)\|^2}{\bar{m}_{t+1}^x \max\{\bar{m}_{t+1}^y, \bar{m}_{t+1}^v\}}$$

$$+ \frac{8}{T} \left( \frac{\Phi(\bar{x}_{k_1}) - \Phi(\bar{x}_T)}{\gamma_x} \right) + \frac{8\bar{q}_{t+1}^{-1}(2\gamma_x \bar{L}_f^2 L_\Phi \bar{q}_{t+1}^{-1} \left(1 + \zeta_q^2\right) + \bar{L}_f^2)}{nT} \sum_{t=k_1}^{T-1} \Delta_t$$

$$+ \frac{16\bar{q}_0^{-1}}{T} \sum_{t=k_1}^{T-1} \left\| \frac{(\tilde{\mathbf{q}}_{t+1}^{-1})^\top}{n\bar{q}_{t+1}^{-1}} \bar{\nabla} F(\mathbf{x}_t, \mathbf{y}_t, \mathbf{v}_t) \right\|^2$$

$$+ \left( \frac{4\bar{L}^2}{\mu^2} + \frac{8\bar{L}^2}{\mu^4} \left( \frac{L_{l,2} C_{f_y}}{\mu} + L_{f,1} \right)^2 \right) \frac{1}{T} \sum_{t=k_1}^{T-1} \frac{\|\nabla_y l(\bar{x}_t, \bar{y}_t)\|^2}{\bar{m}_{t+1}^x \max\{\bar{m}_{t+1}^y, \bar{m}_{t+1}^v\}}$$

$$+ \frac{8\bar{L}^2}{\mu^2 T} \sum_{t=k_1}^{T-1} \frac{\|\nabla_v r(\bar{x}_t, \bar{y}_t, \bar{v}_t)\|^2}{\bar{m}_{t+1}^x \max\{\bar{m}_{t+1}^y, \bar{m}_{t+1}^v\}}$$

$$\leq \frac{8}{T} \left( \frac{\Phi(\bar{x}_0) - \Phi^*}{\gamma_x} \right) + \frac{8\bar{q}_{t+1}^{-1}(2\gamma_x \bar{L}_f^2 L_\Phi \bar{q}_{t+1}^{-1} \left(1 + \zeta_q^2\right) + \bar{L}_f^2)}{nT} \sum_{t=0}^{T-1} \Delta_t$$

$$+ \frac{16n\gamma_x L_\Phi \bar{q}_0^{-2} \left(1 + \zeta_q^2\right)}{T} \sum_{t=0}^{k_1-1} \|\nabla_x f(\bar{x}_t, \bar{y}_t, \bar{v}_t)\|^2 + \frac{16\bar{q}_0^{-1}}{T} \sum_{t=0}^{T-1} \left\| \frac{(\tilde{\mathbf{q}}_{t+1}^{-1})^\top}{n\bar{q}_{t+1}^{-1}} \bar{\nabla} F(\mathbf{x}_t, \mathbf{y}_t, \mathbf{v}_t) \right\|^2$$

$$+ \left( \frac{4\bar{L}^2}{\bar{m}_0^x \mu^2} + \frac{8\bar{L}^2}{\bar{m}_0^x \mu^4} \left( \frac{L_{l,2} C_{f_y}}{\mu} + L_{f,1} \right)^2 \right) \frac{1}{T} \sum_{t=0}^{T-1} \frac{\|\nabla_y l(\bar{x}_t, \bar{y}_t)\|^2}{\bar{m}_{t+1}^y}$$

$$+ \frac{8\bar{L}^2}{\mu^2 T \bar{m}_0^x} \sum_{t=0}^{T-1} \frac{\|\nabla_v r(\bar{x}_t, \bar{y}_t, \bar{v}_t)\|^2}{\max\{\bar{m}_{t+1}^y, \bar{m}_{t+1}^v\}}$$

$$\overset{(a)}{=} \frac{2}{T} \left( 4 \left( \frac{\Phi(\bar{x}_0) - \Phi^*}{\gamma_x} \right) + a_7 \log(t+1) + b_7 \right), \tag{139}$$

where (a) uses Lemma D.8 by plugging in $k_0 = 0$.

Note that Case 1 and Case 2 indicate the same result. Thus, we have:

$$\frac{1}{T} \sum_{t=0}^{T-1} \|\nabla\Phi(\bar{x}_t)\|^2$$

$$\leq \frac{2}{T} \left( 4 \left( \frac{\Phi(\bar{x}_0) - \Phi^*}{\gamma_x} \right) + a_7 \log(T) + b_7 \right) \bar{m}_T^x \bar{z}_T$$

$$\overset{(a)}{\leq} \frac{2}{T} \left[ \left( 4 \left( \frac{\Phi(\bar{x}_0) - \Phi^*}{\gamma_x} \right) + a_7 \log(T) + b_7 \right)^2 \bar{z}_T^2 \right.$$

$$\left. + C_{m^x} \left( 4 \left( \frac{\Phi(\bar{x}_0) - \Phi^*}{\gamma_x} \right) + a_7 \log(T) + b_7 \right) \bar{z}_T \right]$$

$$\overset{(b)}{\leq} \frac{2}{T} \left[ \left( 4 \left( \frac{\Phi(\bar{x}_0) - \Phi^*}{\gamma_x} \right) + a_7 \log(T) + b_7 \right)^2 (a_1 \log(T) + b_1)^2 \right.$$

$$\left. + C_{m^x} \left( 4 \left( \frac{\Phi(\bar{x}_0) - \Phi^*}{\gamma_x} \right) + a_7 \log(T) + b_7 \right) (a_1 \log(T) + b_1) \right]$$

$$= \mathcal{O} \left( \frac{\log^4(T)}{T} \right), \tag{140}$$

where (a) uses Lemma D.10 and (b) uses Lemma D.5. Thus, the proof is finished.

### D.12 Proof of Corollary 3.11

Recall from Theorem 3.9 that there exists a constant $M$ such that:

$$\frac{1}{T} \sum_{t=0}^{T-1} \|\nabla\Phi(x_t)\|^2 \leq \frac{M \log^4(T)}{T}. \tag{141}$$

By setting the total number of iterations $T$ as $T = \frac{ML}{\epsilon} \log^4 \left( \frac{M}{\epsilon} \right)$ and assuming the constant $L = 12^4$, we have:

$$\frac{M \log^4(T)}{T} = \frac{M \log^4 \left( \frac{MN}{\epsilon} \log^4 \left( \frac{M}{\epsilon} \right) \right)}{\frac{MN}{\epsilon} \log^4 \left( \frac{M}{\epsilon} \right)}$$

$$\leq \frac{\left[ \log(N) + \log \left( \frac{M}{\epsilon} \right) + 4 \log \left( \log \left( \frac{M}{\epsilon} \right) \right) \right]^4}{N \log^4 \left( \frac{M}{\epsilon} \right)} \epsilon$$

$$\leq \frac{\left( \log(N) + 2 \log \left( \frac{M}{\epsilon} \right) \right)^4}{N^{1+\frac{1}{4}} \log \left( \frac{M}{\epsilon} \right)} \epsilon \leq \epsilon.$$

Here we have used two key inequalities:

1. $\log \left( \log \left( \frac{M}{\epsilon} \right) \right) \leq \frac{1}{4} \log \left( \frac{M}{\epsilon} \right)$ for sufficiently small $\epsilon$,
2. $\log(L) + 2 \log \left( \frac{M}{\epsilon} \right) \leq L^{1+\frac{1}{4}} \log \left( \frac{M}{\epsilon} \right)$ when $L = 12^4$ and $\epsilon$ is sufficiently small.

Then we can ensure that $\frac{M\log^4(T)}{T} \le \epsilon$. Thus, to achieve an $\epsilon$-accurate stationary point, the required number of iterations is:

$$T = \frac{ML}{\epsilon}\log^4\left(\frac{M}{\epsilon}\right) = \mathcal{O}\left(\frac{1}{\epsilon}\log^4\left(\frac{1}{\epsilon}\right)\right). \tag{142}$$

Finally, the gradient complexity is given by:

$$\mathrm{Gc}(\epsilon) = \Omega(T) = \mathcal{O}\left(\frac{1}{\epsilon}\log^4\left(\frac{1}{\epsilon}\right)\right). \tag{143}$$

Thus, the proof is finished.

## E  Additional Experiments

### E.1  Hyperparameter Optimization Problem

Our experiments are conducted on the following hyperparameter optimization problem:

$$\min_{\lambda \in \mathbb{R}^p} \frac{1}{n}\sum_{i=1}^{n} f_i(\lambda, \omega^*(\lambda)),$$

$$\text{s.t.} \quad \omega^*(\lambda) = \arg\min_{\omega \in \mathbb{R}^q} \frac{1}{n}\sum_{i=1}^{n} l_i(\lambda, \omega),$$

where the goal is to find the optimal hyperparameter $\lambda$, subject to the constraint that $\omega^*(\lambda)$ represents the optimal model given $\lambda$.

### E.2  Synthetic Data Experiments

For the synthetic data experiments, we follow the experimental setups of prior works [Pedregosa, 2016, Grazzi et al., 2020, Chen et al., 2024a]. For any agent $i$, the private objective functions $f_i$ and $l_i$ are defined as:

$$f_i(\lambda, \omega) = \sum_{(x_e, y_e) \in D'_i} \psi(y_e x_e^\top \omega),$$

$$l_i(\lambda, \omega) = \sum_{(x_e, y_e) \in D_i} \psi(y_e x_e^\top \omega) + \frac{1}{2}\sum_{j=1}^{p} e^{\lambda_j}\omega_j^2,$$

where $\psi(x) = \log(1 + e^{-x})$ and $p$ represents the dimensionality of the data. A ground truth vector $\omega^*$ is generated and each $x_e \in \mathbb{R}^p$ is sampled from a normal distribution. The data distribution for $x_e$ at node $i$ follows $\mathcal{N}(0, i^2 \cdot r^2)$, where $r$ quantifies the degree of heterogeneity across agents. The corresponding labels $y_e$ are defined as $y_e = x_e^\top \omega^* + 0.1z$, where $z$ is sampled from a standard normal distribution.

### E.3  Real-World Data Experiments

For the real-world data experiment, we apply our method to hyperparameter optimization on the MNIST dataset [LeCun et al., 1998] and Fashion-MNIST (FMNIST) [Xiao et al., 2017] dataset. Following [Grazzi et al., 2020], the functions $f_i$ and $l_i$ are defined as:

$$f_i(\lambda, \omega) = \frac{1}{|D'_i|} \sum_{(x_e, y_e) \in D'_i} \ell(x_e^\top \omega, y_e),$$

$$l_i(\lambda, \omega) = \frac{1}{|D_i|} \sum_{(x_e, y_e) \in D_i} \ell(x_e^\top \omega, y_e) + \frac{1}{cp}\sum_{j=1}^{c}\sum_{k=1}^{p} e^{\lambda_k}\omega_{jk}^2,$$

where $c = 10$ and $p = 784$ denote the number of classes and features, respectively, $\omega \in \mathbb{R}^{c \times p}$ is the model parameter, and $\ell$ denotes the cross-entropy loss. $D_i$ and $D'_i$ represent the training and validation sets, respectively. The batch size for each computing agent is set to 1,000.

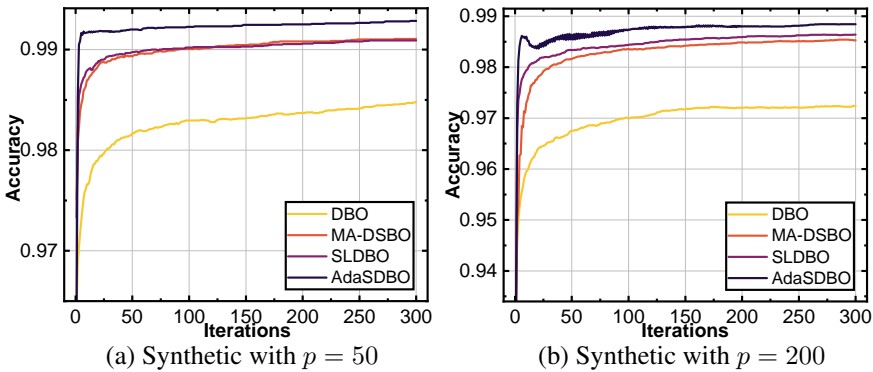

(a) Synthetic with $p = 50$       (b) Synthetic with $p = 200$

Figure 4: Test accuracy on synthetic dataset with $r = 5$.

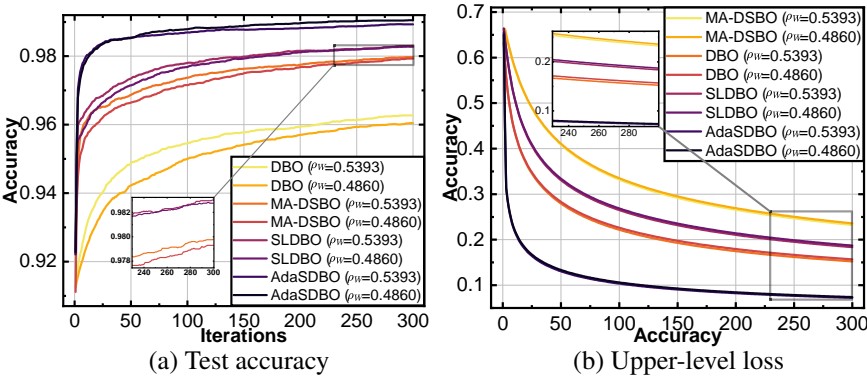

(a) Test accuracy       (b) Upper-level loss

Figure 5: Test accuracy and upper-level loss for synthetic dataset with different $\rho_W$.

### E.4 Decentralized Meta-Learning Experiments

In our decentralized meta-learning experiment, following the MAML framework [Finn et al., 2017], we consider a setting involving $M$ distinct tasks, denoted by $\{T_q\}_{q=1}^{M}$. Each task $T_q$ is associated with a loss function $L(x, y_q)$, where $x$ denotes a shared embedding parameter across tasks, and $y_q$ represents a task-specific parameter. The objective of meta-learning is to identify a universal parameter $x^*$ that facilitates fast adaptation to new tasks by enabling efficient fine-tuning of $y_q$ using a limited number of data points and update steps.

This problem naturally fits within a bilevel optimization framework. At the lower level, given a fixed $x$, each task seeks the corresponding optimal adaptation parameter $y_q^*$ by minimizing the loss over its training data $D_q^{\text{tr}}$. The upper-level optimization then aims to select a shared parameter $x$ such that the adapted models $y_q^*$ perform well on the corresponding validation data $D_q^{\text{val}}$. Let $y^* = \text{col}\{y_1^*, \ldots, y_M^*\}$ denote the collection of all task-specific solutions.

Unlike traditional centralized meta-learning, where all data is accessible at a single location, we consider a decentralized setup in which training and validation data for each task $T_q$ are partitioned across $n$ agents. Specifically, each agent $i \in [n]$ maintains its own local training dataset $D_{i,q}^{\text{tr}}$ and validation dataset $D_{i,q}^{\text{val}}$ for task $T_q$. Given a shared parameter $x$, the local base-learners collaboratively solve for $y_q^*(x)$ using decentralized lower-level optimization. The upper-level meta-update of $x$ is then performed through cooperation among agents based on their local validation losses.

The decentralized bilevel optimization problem is formally expressed as:

$$\min_{x} \quad F(x) := \frac{1}{n} \sum_{i=1}^{n} \frac{1}{M} \sum_{q=1}^{M} f_{i,q}(x, y_q^*(x)),$$

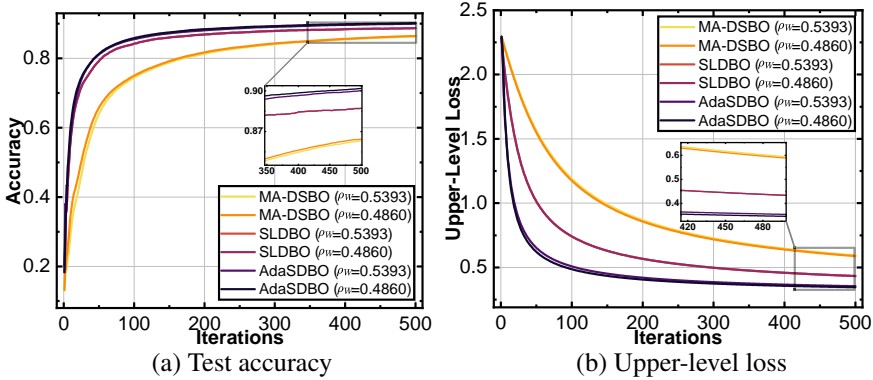

Figure 6: Test accuracy and upper-level loss for MNIST dataset with different $\rho_W$.

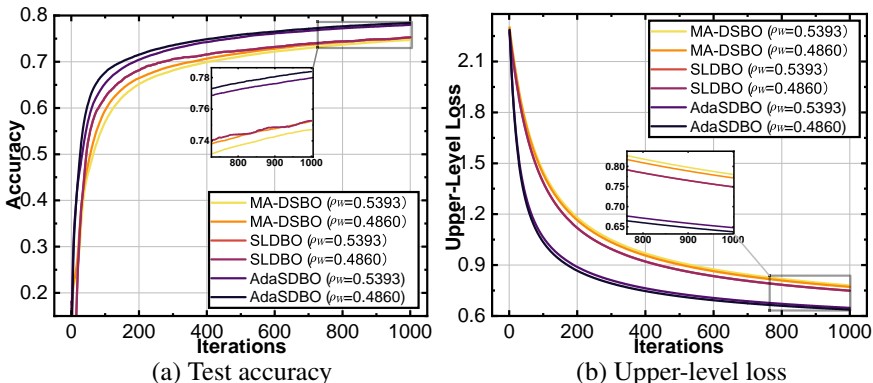

Figure 7: Test accuracy and upper-level loss for FMNIST dataset with different $\rho_W$.

$$\text{s.t.} \quad y_q^*(x) := \arg\min_y \frac{1}{n} \sum_{i=1}^n \frac{1}{M} \sum_{q=1}^M l_{i,q}(x,y),$$

where $f_{i,q}(x, y_q^*) = \frac{1}{|D_{i,q}^{\text{val}}|} \sum_{(x,y_q^*) \in D_{i,q}^{\text{val}}} L(x, y_q^*(x))$ and $l_{i,q}(x,y) = \frac{1}{|D_{i,q}^{\text{tr}}|} \sum_{(x,y_q) \in D_{i,q}^{\text{tr}}} L(x, y_q) + R_{i,x}(y_q)$, with $R_{i,x}(y)$ denoting a strongly-convex regularizer with respect to $y$. The experiment was conducted over 32 batches of tasks across 1,000 iterations. Each task included a training dataset and a validation dataset, both configured for 5-way classification with 50 shots per class. Specifically, the training and validation data were distributed among different agents to enable cooperative learning. For each task, 30% of the data from the $i$-th class was assigned to agent $i$, while the remaining 70% was evenly distributed among the other agents.

### E.5 Configurations

All experiments were performed with $n = 5$ using PyTorch [Paszke et al., 2019]. The network topology was configured as a ring topology, where the weight matrix $W = (w_{ij})$ is defined as:

$$w_{ii} = w, \quad w_{i,i+1} = w_{i,i-1} = \frac{1-w}{2},$$

where $w \in (0,1)$, $w_{1,0} = w_{1,n}$, and $w_{n,n+1} = w_{n,1}$. In this setup, each agent $i$ is only connected to its immediate neighbors $i-1$ and $i+1$ for $i = 1, \cdots, n$, with the indices 0 and $n+1$ representing $n$ and 1, respectively.

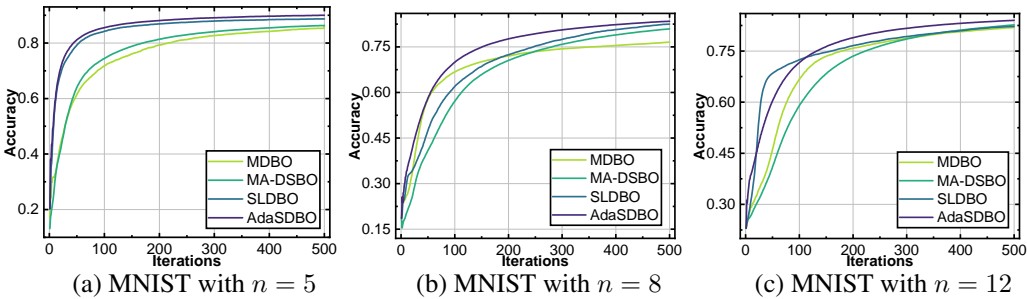

(a) MNIST with $n = 5$      (b) MNIST with $n = 8$      (c) MNIST with $n = 12$

Figure 8: Scalability analysis on the MNIST dataset under varying network sizes ($n = 5, 8, 12$).

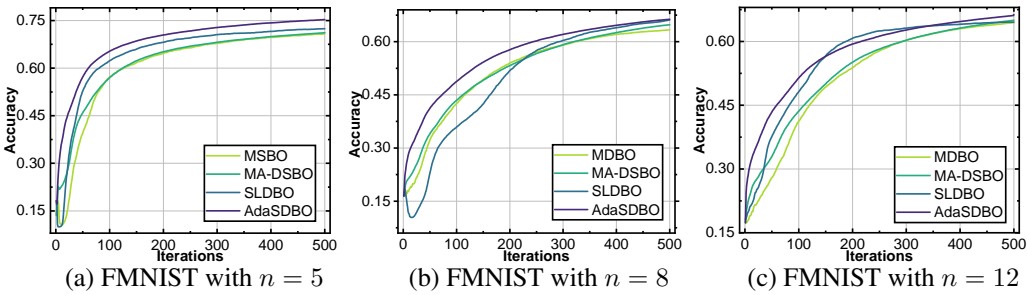

(a) FMNIST with $n = 5$      (b) FMNIST with $n = 8$      (c) FMNIST with $n = 12$

Figure 9: Scalability analysis on the FMNIST dataset under varying network sizes ($n = 5, 8, 12$).

Table 2: Test accuracy on the MNIST and FMNIST datasets under different communication topologies ($n = 8$).

| Algorithms | MNIST | | | FMNIST | | |
|---|---|---|---|---|---|---|
| | Ring | Ladder | Random | Ring | Ladder | Random |
| AdaSDBO | $0.908 \pm 0.001$ | $0.911 \pm 0.001$ | $0.913 \pm 0.001$ | $0.774 \pm 0.003$ | $0.788 \pm 0.003$ | $0.790 \pm 0.003$ |
| SLDBO | $0.871 \pm 0.002$ | $0.871 \pm 0.001$ | $0.871 \pm 0.001$ | $0.758 \pm 0.002$ | $0.758 \pm 0.002$ | $0.758 \pm 0.001$ |
| MA-DSBO | $0.850 \pm 0.001$ | $0.850 \pm 0.001$ | $0.850 \pm 0.002$ | $0.709 \pm 0.002$ | $0.716 \pm 0.001$ | $0.719 \pm 0.002$ |
| MDBO | $0.753 \pm 0.002$ | $0.754 \pm 0.001$ | $0.754 \pm 0.002$ | $0.650 \pm 0.001$ | $0.650 \pm 0.002$ | $0.653 \pm 0.001$ |

For all experiments, except for the test accuracy versus stepsize comparison, we use the following parameter settings. For the baseline methods SLDBO and MA-DSBO, the stepsizes for updating $x$ and $v$ are set to 0.01, while the stepsize for updating $y$ is set to 0.02, following the optimal stepsize order described in [Dong et al., 2023, Chen et al., 2023]. For the baseline methods DBO and MDBO, the stepsizes for updating both $x$ and $y$ are set to 0.01. For AdaSDBO, we set $\gamma_x = \gamma_y = \gamma_v = 1$ and initialize $m_{i,0}^x = m_{i,0}^y = m_{i,0}^v = 10, \forall i \in [n]$. All experiments were conducted on a host machine equipped with an Intel(R) Xeon(R) W9-3475X CPU running at 2.20 GHz (maximum turbo frequency: 4.80 GHz), featuring 36 physical cores and 72 threads. The system was configured with 256 GB of DDR5 ECC RAM and a single NVIDIA(R) RTX(TM) A6000 GPU with 48 GB of memory.

## E.6 Additional Results

For the synthetic dataset, we increased the data heterogeneity parameter $r$ to 5 and analyzed the convergence performance of different methods under two data dimensions ($p = 50$ and $p = 200$). It can be observed in Figure 4 that our proposed algorithm consistently outperforms the baseline methods in both convergence and test accuracy, even as the level of data heterogeneity increases. This superior performance can be attributed to the adaptive stepsizes design, which enables our algorithm to dynamically adjust stepsizes to accommodate varying data distributions. Consequently, our proposed method demonstrates robust performance across different data heterogeneity settings, effectively adapting to changes in the data environment.

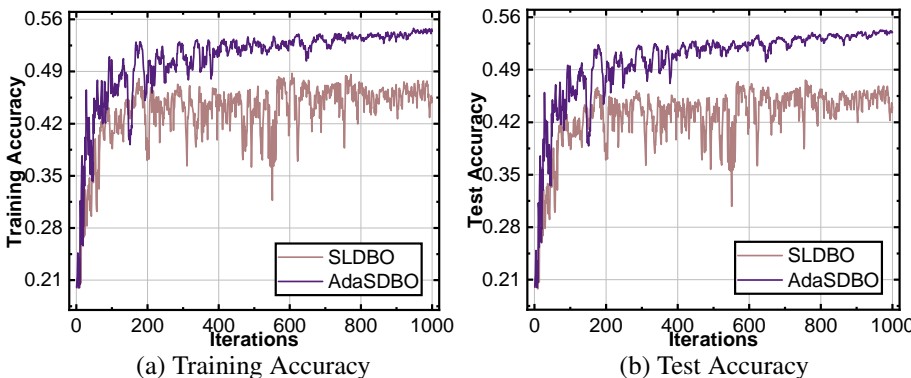

(a) Training Accuracy        (b) Test Accuracy

Figure 10: Performance comparison between AdaSDBO and SLDBO under the MAML framework, where each node employs an identical CNN for 5-way, 50-shot classification on the CIFAR-10 dataset.

Table 3: Performance comparison between AdaSDBO and SLDBO under varying network sizes in the decentralized meta-learning task.

| Algorithms | $n = 10$ | | $n = 20$ | | $n = 30$ | |
|---|---|---|---|---|---|---|
| | Train Acc | Test Acc | Train Acc | Test Acc | Train Acc | Test Acc |
| AdaSDBO | $0.543 \pm 0.002$ | $0.534 \pm 0.002$ | $0.542 \pm 0.002$ | $0.534 \pm 0.001$ | $0.538 \pm 0.001$ | $0.533 \pm 0.001$ |
| SLDBO | $0.503 \pm 0.004$ | $0.488 \pm 0.005$ | $0.472 \pm 0.006$ | $0.461 \pm 0.006$ | $0.486 \pm 0.006$ | $0.475 \pm 0.007$ |

Furthermore, we assessed the performance of different methods across varying network connectivity levels ($\rho_W$) on the synthetic, MNIST, and FMNIST datasets. As depicted in Figure 5, Figure 6, and Figure 7, increasing the network connectivity (i.e., a decrease in $\rho_W$ from 0.5393 to 0.4860), leads to improved accuracy for all methods across the different datasets. Notably, our proposed algorithm maintains superior convergence performance compared to all baseline methods on each dataset, validating its effectiveness and reliability under different levels of network connectivity.

In Figure 8 and Figure 9, we evaluate the broader scalability of our proposed method by varying the number of nodes $n = 5, 8, 12$ on the MNIST and FMNIST datasets. We compare AdaSDBO against several baseline algorithms, including SLDBO [Dong et al., 2023], MA-DSBO [Chen et al., 2023], and MDBO [Gao et al., 2023]. The results show that AdaSDBO achieves convergence performance competitive with these state-of-the-art methods. Furthermore, AdaSDBO maintains stable performance across different system configurations, underscoring its ease of deployment and practical applicability. This stability can be attributed to the problem-parameter-free nature of AdaSDBO, which eliminates the need for manually tuned stepsizes. In contrast, other methods in decentralized settings often suffer from sensitivity to stepsize selection due to their reliance on problem-specific parameters, which are typically unknown or difficult to estimate in practice. As a result, the parameter-free property of AdaSDBO makes it particularly well-suited for real-world decentralized applications.

To assess sensitivity to network structure, we conducted experiments under three commonly used communication topologies—ring, ladder, and random—and compared the performance of AdaSDBO with baseline methods. Structural details for these topologies are provided in the Appendix of [Li et al., 2024]. The results summarized in Table 2 indicate that stronger connectivity leads to faster and more stable convergence: in particular, the random topology, which has the highest connectivity, yields the fastest and most stable convergence for all methods. Across the three topologies, AdaSDBO consistently outperforms the baselines, maintaining strong performance under variations in the communication topology. This robustness is facilitated by the adaptive stepsize mechanism of AdaSDBO, which accommodates topology-induced heterogeneity.

For the decentralized meta-learning experiment, we compared our proposed algorithm with the single-loop decentralized bilevel optimization method SLDBO [Dong et al., 2023]. Figure 10 shows that AdaSDBO consistently outperforms SLDBO in both average training accuracy across all nodes

and test accuracy, demonstrating the effectiveness of our approach in decentralized meta-learning tasks. We further increase the number of agents to 10, 20, and 30; the results in Table 3 indicate that AdaSDBO remains stable and competitive as the network scales, outperforming the baselines even in more complex settings. The superior performance of AdaSDBO stems from its problem-parameter-free design, which enables stepsize selection without requiring knowledge of problem-specific parameters. Furthermore, the adaptive nature of our method allows AdaSDBO to dynamically adjust its learning dynamics and consistently achieve optimal convergence rates—even in decentralized settings where hyperparameter tuning is particularly difficult for other methods. This advantage becomes even more evident in complex decentralized meta-learning tasks, further underscoring the robustness and scalability of our proposed approach.

