# OpenReview forum: "Problem-Parameter-Free Decentralized Bilevel Optimization"
_NeurIPS.cc/2025/Conference — NeurIPS 2025 poster_

### Official Review · Reviewer_Y6eW · 2025-06-27

**Clarity:** 3
**Significance:** 3
**Originality:** 3
**Rating:** 5
**Confidence:** 3

**Summary:**

This paper introduces AdaSDBO, an innovative adaptive single-loop algorithm for decentralized bilevel optimization that eliminates the need for problem-specific hyperparameter tuning. By dynamically adjusting stepsizes based on accumulated gradient norms and incorporating a hierarchical stepsize design, AdaSDBO efficiently balances updates across primal, dual, and auxiliary variables. Theoretically, the algorithm achieves a convergence rate of $\widetilde{\mathcal{O}}(\log^4(T)/T)$, matching state-of-the-art methods while remaining robust to stepsize choices. Extensive experiments on synthetic and real-world datasets, including hyperparameter optimization and meta-learning tasks, demonstrate AdaSDBO’s competitive performance and superior adaptability compared to existing approaches.

**Questions:**

1. The algorithm has 6 hyper-parameters, which is much more than the other decentralized bilevel algorithms. Would this make algorithm hard to tune? How sensitive is performance to their choice, and do you have guidelines for setting them?

2. In algorithm step 6, the communication cost seems to be 6x of the model parameters. Is the proposed algorithm communication-costly than other decentralized bilevel algorithms?

3. Although the hyper-parameters doesn't reply on the problem-parameters, how about the final convergence rate? If the final convergence rate is also affected by these problem-parameters, does it mean the algorithm still need hyper-parameters tuning based on the problem-parameters to achieve similar convergence speed?

4. The experiments use relatively small networks. How does the algorithm scale to hundreds or thousands of nodes?

**Ethical Concerns:**

["NO or VERY MINOR ethics concerns only"]

**Final Justification:**

All my concern are addressed by the authors' responses.

**Limitations:**

The paper didn't discuss the limitation. It may worth to discussing the algorithm communication complexity, memory cost and its applicable in real world tasks.

**Paper Formatting Concerns:**

No formatting concern

**Quality:**

3

**Strengths And Weaknesses:**

Strength: The paper studied the decentralized bilevel problem and proposed a novel algorithm of which the hyper-parameter selection does not rely on problem-specific factors. The rigorous theoretical analysis is provided in the paper with clear format. Numerical experiments are provided to verify the results. Overall, the paper is well-written and with clear novelty.


Weakness:
- How the algorithm become problem-parameter-free is unclear. It may be better to highlight what algorithm improvement achieve this.
- It is better to verify in numerical study that the algorithm is truly problem-parameter-free and outperforms the existing algorithms.

---

> ### Author Rebuttal · Authors · 2025-07-26
>
> **We sincerely appreciate the reviewer for recognizing our contributions and for the constructive comments. Our point-to-point responses to concerns on Weaknesses and Questions are given below.**
>
> **Reply to Weaknesses:**
>
> **1.** Thank you for the helpful comment. In our work, problem-parameter-free refers to the fact that our algorithm achieves a convergence rate of ${\mathcal{O}}(\log^4 T/T)$ without requiring any constraints or tuning on the stepsizes for the optimization variables $x$, $y$, and $v$. This is made possible through two key algorithmic innovations:
>
> - *Hierarchical Adaptive Stepsize Design:*
> We introduce a principled stepsize structure that automatically balances updates across the three optimization levels. Specifically, the stepsize for updating $v$ depends on the gradient magnitudes of both $v$ and $y$, while the update for $x$ is more conservatively adjusted based on the accumulated gradients of $x$, $y$, and $v$.
> This hierarchical design ensures that $x$ and $v$ progress more slowly than $y$, preserving the stability and fidelity of the bilevel dynamics—without needing problem-specific hyperparameter tuning.
>
> - *Stepsize Tracking Mechanism:*
> Decentralized bilevel optimization naturally leads to coupled, agent-specific adaptive stepsizes, which can become inconsistent and degrade convergence in heterogeneous networks. To address this, we incorporate a stepsize tracking mechanism in which agents exchange lightweight scalar gradient accumulators. This ensures that the stepsizes for $x$, $y$, and $v$ remain globally consistent, even in the presence of heterogeneity.
>
> Together, these components enable us to develop the first problem-parameter-free algorithm for decentralized bilevel optimization that achieves optimal convergence without any reliance on problem-specific constants or manual tuning.
>
> **2.** Thank you for the thoughtful comment. We respectfully clarify that we have already conducted a detailed numerical study to verify the problem-parameter-free nature of our algorithm, as presented in Figure 2 of the paper.
>
> In this experiment, we compare the test accuracy of our proposed AdaSDBO algorithm against several baseline methods across a wide range of stepsize values (from $10^{-3}$ to $10^{2}$). The results clearly show that:
>
> - AdaSDBO maintains stable and competitive performance across the entire range of tested stepsizes.
>
> - In contrast, baseline algorithms exhibit much narrower windows of stability and performance, indicating a strong sensitivity to stepsize selection.
>
> These findings highlight the robustness of AdaSDBO with respect to stepsize choices and directly support its problem-parameter-free design. Unlike existing methods that require careful tuning for each optimization variable, our approach consistently performs well without manual adjustment—further demonstrating its practical effectiveness.
>
> **Reply to Questions:**
>
> **1.** Thank you for your question. While our method introduces three adaptive stepsizes—one for each of the variables $x$, $y$, and $v$—we respectfully clarify that no tuning is required for these hyperparameters due to our hierarchical adaptive stepsize design. In simple terms, all three stepsizes can be initialized with the same value, and the algorithm will automatically adjust them throughout training:
>
> - The stepsize for updating $v$ is adaptively adjusted based on the maximum of the gradient accumulators from both $v$ and
> $y$, ensuring that $v$ does not progress faster than $y$.
>
> - The stepsize for updating $x$ is made even more conservative by incorporating the accumulators of $x$, $y$, and $v$, so that $x$ progresses slower than both $v$ and $y$, preserving the hierarchical update structure.
>
> With this adaptive mechanism, our method is highly insensitive to the choice of the initial stepsize. This stands in contrast to existing decentralized bilevel algorithms that often require precise tuning to prevent instability or poor performance.
>
> **2.** Thank you for your insightful question. We agree that Step 6 involves synchronization of adaptive stepsizes across agents. However, this synchronization is highly efficient, as it only requires exchanging a few scalar values that track local gradient information for stepsize adjustment.
> Compared to the communication cost of transmitting full model parameters such as the primal variable $x$, dual variable $y$, and auxiliary variable $v$—which are typically high-dimensional—the added overhead from sharing scalar accumulators is negligible.
>
> Therefore, although our method includes an additional coordination step, it remains communication-efficient in practice. Moreover, this lightweight tracking mechanism ensures global consistency of stepsizes across agents, which is essential for achieving stable and problem-parameter-free convergence in decentralized bilevel optimization.
>
> **3.** Thank you for your valuable question. We respectfully clarify that while the final convergence rate of any optimization algorithm inherently depends on problem-specific properties (e.g., smoothness, convexity, curvature), our key contribution lies in the fact that our stepsize strategy does not rely on these parameters.
>
> In our two-stage convergence analysis, we impose no constraints on the choice of stepsizes. The stepsize scheme employed in our method is inherently adaptive and optimal, allowing us to achieve the ${\mathcal{O}}(\log^4 T/T)$ convergence rate without requiring any knowledge of smoothness and strong convexity constants, the spectral gap of the graph adjacency matrix, or other problem-dependent parameters.
>
> This stands in sharp contrast to problem-parameter-dependent methods, whose optimal convergence guarantees rely on carefully setting stepsizes based on such parameters. If these conditions are violated, those methods can no longer ensure optimal convergence.
>
> In summary, while the convergence rate ultimately reflects the nature of the problem, our algorithm ensures that no hyperparameter tuning is needed based on problem parameters to attain this optimal performance.
>
> **4.** Thank you for your thoughtful question. Due to limited computational resources, we have done our best to evaluate the scalability of our algorithm on larger decentralized networks within a feasible range. Specifically, we conducted experiments on the decentralized meta-learning task with CIFAR-10 dataset using wider network sizes. The results are summarized below:
>
> **Table 1: Convergence Performance under Different Number of Agents with Decentralized Meta-Learning Task**
>
> ---
> | Number of Agents |  | $n=10$  |  | $n=20$ | | $n=30$ |
> |------------------|:-----------------:|:----------------:|:----------------:|:----------------:|:----------------:|:---------------:|
> | Algorithms | Train Acc | Test Acc | Train Acc | Test Acc | Train Acc | Test Acc |
> | AdaSDBO               | $0.543\pm0.002$ | $0.534\pm0.002$|$0.542\pm0.002$|$0.534\pm0.001$ | $0.538\pm0.001$ | $0.533\pm0.001$|
> | SLDBO               | $0.503\pm0.004$ | $0.488\pm0.005$| $0.472\pm0.006$ |$0.461\pm0.006$|$0.486\pm0.006$ |$0.475\pm0.007$|
> ---
>
> These results demonstrate that AdaSDBO maintains consistently superior performance over baselines even as the number of agents increases.
> While this setup may not fully represent large-scale real-world systems with hundreds or thousands of nodes, such experimental scales are commonly adopted in the decentralized optimization literature [1]–[3] due to similar resource constraints. Moreover, AdaSDBO is inherently scalable due to its:
>
> - Problem-parameter-free design, which eliminates the need for manual hyperparameter adjustment across agents.
>
> - Single-loop architecture, avoiding costly nested optimization common in bilevel methods.
>
> These properties suggest that AdaSDBO is well-suited for scaling to larger networks.
>
> **Reply to Limitations:**
>
> Thank you for your constructive comment. We discuss the limitations of our method as follows:
>
> - *Communication Overhead*: Our framework requires the exchange of scalar gradient norms as accumulators, which incurs minimal overhead compared to transmitting high-dimensional variables. While more aggressive communication-saving techniques (e.g., quantized or partially shared accumulators) could further reduce cost, they risk losing the global coordination necessary for stable convergence. We consider this an interesting direction for future exploration.
>
> - *Memory Cost*: Adaptive methods like ours introduce a small additional cost for maintaining gradient accumulators. However, this cost is negligible compared to full gradient evaluations. Nonetheless, reducing memory usage remains a valuable direction for improving overall efficiency.
>
> - *Applicability to Real-World Tasks*: While our method is theoretically grounded in the nonconvex–strongly-convex setting, many real-world tasks involve fully nonconvex landscapes. Extending the framework to such general settings poses additional challenges (e.g., non-uniqueness, non-differentiability [6]) and represents an important open problem in decentralized bilevel optimization.
>
> We have incorporated this discussion into our paper and are enthusiastic about exploring these directions in future work.
>
> **References:**
>
> [1] A Single-Loop Algorithm for Decentralized Bilevel Optimization.
>
> [2] Decentralized Bilevel Optimization over Graphs: Loopless Algorithmic Update and Transient Iteration Complexity.
>
> [3] Decentralized Bilevel Optimization.
>
> [4] Problem-Parameter-Free Decentralized Nonconvex Stochastic Optimization.
>
> [5] Fully First-Order Methods for Decentralized Bilevel Optimization.
>
> [6] On Finding Small Hyper-Gradients in Bilevel Optimization: Hardness Results and Improved Analysis.
>
> **Thank you once again for your thoughtful review and constructive feedback.**

---

> > ### Comment · Reviewer_Y6eW · 2025-08-03
> >
> > Thanks for the detailed responses. All my concerns are addressed.

---

> ### Author Response · Authors · 2025-08-03
>
> Dear Reviewer Y6eW,
>
> Thank you very much for your reply and for recognizing our contributions. We are delighted that your concerns have been resolved. We sincerely appreciate your thoughtful review and constructive feedback.
>
> Sincerely,
>
> Authors of the paper

---

### Official Review · Reviewer_oJg7 · 2025-06-28

**Clarity:** 3
**Significance:** 3
**Originality:** 3
**Rating:** 4
**Confidence:** 3

**Summary:**

The paper introduces AdaSDBO, a novel algorithm designed for decentralized bilevel optimization that does not require prior knowledge of problem parameters. The proposed method addresses the issue of hyperparameter tuning by employing adaptive step sizes based on cumulative gradient norms. This enables the algorithm to update variables simultaneously while adjusting its progress dynamically. The paper provides a thorough theoretical analysis demonstrating that AdaSDBO achieves competitive convergence rates. Furthermore, the experimental results confirm that AdaSDBO is robust and performs well across a range of step size configurations.

**Questions:**

See weakness.

Other questions:
1. In Figure 2, which variable's step size is being depicted? Since the proposed method involves three distinct step sizes, how sensitive is the method to the choice of initial values for these three step sizes?
2. The proposed method is parameter-free, but from Figure 2, it appears that the step size still has a significant impact. Could you explain why this is the case?

**Ethical Concerns:**

["NO or VERY MINOR ethics concerns only"]

**Final Justification:**

My concern has been solved. I will raise my score

**Limitations:**

This paper focus on the algorith and analysis and does not have societal impact .

**Quality:**

3

**Strengths And Weaknesses:**

Strength：
1. The approach of eliminating the need for problem-specific hyperparameter tuning is a significant contribution to decentralized bilevel optimization.
2. The paper includes a detailed theoretical analysis of AdaSDBO, demonstrating its convergence rate and the conditions under which the algorithm performs comparably to state-of-the-art methods.
3. By removing the dependence on problem-specific hyperparameters, AdaSDBO could be particularly valuable in real-world applications, where such parameters are often unknown or costly to determine.
Weakness:
1. Although the experiments are comprehensive, it would be beneficial to see more experiments on large datasets. The experiments on MNIST, FashionMNIST and Cifar10 are not enough.
2. In decentralized algorithms, communication overhead is a critical concern. The paper does not provide a detailed analysis of the communication complexity of AdaSDBO, which could be important for real-world applications involving distributed systems. A comparison of communication cost with other methods could be helpful.
3. The proposed method relies on the strongly convex assumption of the lower level, which is restrictive in the real world.

---

> ### Author Rebuttal · Authors · 2025-07-26
>
> **We sincerely appreciate the reviewer for recognizing our contributions and for the constructive comments. Our point-to-point responses to concerns on Weaknesses and Questions are given below.**
>
> **Reply to Weaknesses:**
>
> **1.** Thank you for your helpful suggestion. To further validate the effectiveness of our proposed method on more complex datasets, we have added new experiments on the CIFAR-100 dataset using ResNet-18 model with varying numbers of agents. The results are summarized below:
>
> **Table 1: Convergence Performance on the CIFAR-100 Dataset with Varying Numbers of Agents**
>
> ---
> | Algorithms |  &nbsp;&nbsp; &nbsp;&nbsp; &nbsp;$n=5$ | &nbsp;&nbsp; &nbsp;&nbsp; &nbsp;$n=8$ |
> |-----------------|:--------------:|:------------:|
> | AdaSDBO   | $0.426\pm0.002$ | $0.417\pm0.003$ |
> | SLDBO     | $0.401\pm0.005$ | $0.388\pm0.005$ |
> ---
>
> Moreover, we have made every effort to further validate the robustness of our method by adding experiments on the MNIST, FashionMNIST, and CIFAR-10 datasets, as shown below:
>
> - Added more experiments on MNIST and FashionMNIST by varying the network topology, with the corresponding structural specifications described in the Appendix of [1], as shown in the following table.
>
> **Table 2: Convergence Performance under Different Topologies ($n=8$)**
>
> ---
> | Dataset |       |  &nbsp; &nbsp;&nbsp; &nbsp;MNIST   |             |      | &nbsp; &nbsp;&nbsp; FMNIST      |      |
> |----------|:------------------:|:--------------------:|:--------------------:|:-----------------------:|:--------------------:|:--------------------:|
> | Algorithms | Ring |Ladder | Complete | Ring |Ladder | Complete |
> | AdaSDBO   | $0.908\pm 0.001$ |$0.911\pm 0.001$| $0.916\pm 0.001$   | $0.774\pm 0.003$ |$0.788\pm 0.003$   | $0.798\pm 0.002$  |
> | SLDBO |  $0.871\pm 0.002$  | $0.871\pm 0.001$  | $0.872\pm 0.001$    | $0.758\pm 0.002$ |$0.758\pm 0.002$   | $0.758\pm 0.001$ |
> | MA-DSBO |  $0.850\pm 0.001$  | $0.850\pm 0.001$ | $0.851\pm 0.002$   | $0.709\pm 0.002$ |$0.716\pm 0.001$   | $0.720\pm 0.002$  |
> | MDBO | $0.753\pm 0.002$ | $0.754\pm 0.001$ | $0.764\pm 0.001$   | $0.650\pm 0.001$ |$0.650\pm 0.002$   | $0.653\pm 0.001$ |
> ---
>
> - Added new experiments on CIFAR-10 by varying the number of agents in the decentralized meta-learning setup, as presented in the table below.
>
> **Table 3: Convergence Performance under Different Number of Agents with Decentralized Meta-Learning Task**
>
> ---
> | Number of Agents |  | $n=10$  |  | $n=20$ | | $n=30$ |
> |------------------|:-----------------:|:----------------:|:----------------:|:----------------:|:----------------:|:---------------:|
> | Algorithms | Train Acc | Test Acc | Train Acc | Test Acc | Train Acc | Test Acc |
> | AdaSDBO               | $0.543\pm0.002$ | $0.534\pm0.002$|$0.542\pm0.002$|$0.534\pm0.001$ | $0.538\pm0.001$ | $0.533\pm0.001$|
> | SLDBO               | $0.503\pm0.004$ | $0.488\pm0.005$| $0.472\pm0.006$ |$0.461\pm0.006$|$0.486\pm0.006$ |$0.475\pm0.007$|
> ---
>
> These additional results further confirm the robustness and superior performance of our proposed AdaSDBO method across diverse settings.
>
> **2.** Thank you for your insightful comment. Through a consolidation of our theoretical analysis, we can get that the communication complexity of AdaSDBO is $\mathcal{O}\left(\frac{1}{\epsilon(1 - \rho_W)^2} \log^4\left(\frac{1}{\epsilon}\right)\right)$, which matches the results of existing methods such as [2]–[4], up to logarithmic factors. Since logarithmic terms grow significantly slower than polynomial ones, they are typically considered negligible in the context of optimization theory (e.g., [1], [5]–[6]).
>
> In sharp contrast to [2]–[4], AdaSDBO is entirely problem-parameter-free, requiring no knowledge of problem-specific constants or manual hyperparameter tuning. This makes it significantly easier to deploy and more robust in heterogeneous, real-world decentralized systems.
> Combined with its competitive communication complexity, this problem-parameter-free design underscores the practical advantages of AdaSDBO over existing decentralized bilevel methods. We have included comparisons of communication cost with existing works in our paper.
>
> **3.** Thank you for your thoughtful comment. We acknowledge that the strong convexity assumption on the lower-level problem may not always hold in real-world applications. However, this assumption remains a standard and widely adopted condition in the bilevel optimization literature [2]-[4].
>
> We agree that extending the theory to handle nonconvex lower-level problems is an important and nontrivial direction. In such cases, the solution set may be non-unique and the overall objective may become non-differentiable [7], requiring fundamentally different algorithmic frameworks. This remains a challenging open problem in decentralized bilevel optimization. We regard this as an important direction for future research and appreciate the reviewer for highlighting this point.
>
> **Reply to Questions:**
>
> **1.** Thank you for your thoughtful question. In Figure 2 of our paper, for baseline methods, we first tune the stepsizes for each optimization variable to their optimal values, and then test sensitivity by scaling these stepsizes up or down by factors of 10. This is necessary because baseline algorithms require precise tuning of individual stepsizes to properly control the progress of different variables.
>
> In contrast, our proposed problem-parameter-free method uses the same initial stepsize for all three variables $x$, $y$, and $v$. This simple choice is supported by our hierarchical adaptive stepsize mechanism:
> - The update for $v$ uses a stepsize inversely proportional to $\max(m^v_{i,t+1}, m^y_{i,t+1})$,
> - The update for $x$ uses a stepsize inversely proportional to $m^x_{i,t+1} \max(m^v_{i,t+1}, m^y_{i,t+1})$, making it more conservative,
>
> where $m^x_{i,t+1}$, $m^y_{i,t+1}$, and $m^v_{i,t+1}$ denote the gradient accumulators for the primal variable $x$, dual variable $y$, and auxiliary variable $v$, respectively. This structure ensures:
> - $v$ progresses no faster than $y$,
> - $x$ progresses no faster than both $y$ and $v$,
>
> thus maintaining a well-coordinated dynamic across levels and preserving the fidelity of the hypergradient approximation. As a result, our method is robust to the initial stepsize setting, and automatically adapts to reach optimal convergence performance without manual tuning.
>
> **2.** Thank you for your insightful question. Through our two-stage theoretical analysis, we have rigorously shown that our problem-parameter-free algorithm guarantees convergence to a stationary point without requiring any specific tuning of stepsize values. However, in practice, the initial stepsize value can still influence the speed of convergence. This behavior is typical of AdaGrad-norm-based adaptive methods. Specifically:
>
> - When the initial stepsize is excessively large, early gradient updates may cause the accumulated gradient norms to grow quickly, which in turn causes the adaptive stepsize to shrink prematurely. This leads to slower progress in subsequent iterations.
>
> - Conversely, when the initial stepsize is too small, the early updates are overly conservative, and the method takes longer to begin meaningful descent—even though the stepsize will eventually adapt.
>
> Despite this, convergence is still ensured in both cases due to the adaptivity of our design. The observed sensitivity only affects the rate of convergence, not the final outcome. With sufficient communication rounds, our method can still achieve optimal performance.
> Importantly, our method is robust across a much wider range of stepsize values compared to existing baselines, many of which fail to converge if the stepsizes are not properly set. This highlights the practical value of our design, which eliminates the need for manual tuning while maintaining convergence guarantees.
>
> **References:**
>
> [1] Problem-Parameter-Free Decentralized Nonconvex Stochastic Optimization.
>
> [2] On the Convergence of Distributed Stochastic Bilevel Optimization Algorithms over a Network.
>
> [3] A Single-Loop Algorithm for Decentralized Bilevel Optimization.
>
> [4] Decentralized Bilevel Optimization.
>
> [5] How Free is Parameter-Free Stochastic Optimization?
>
> [6] Tuning-Free Stochastic Optimization.
>
> [7] On Finding Small Hyper-Gradients in Bilevel Optimization: Hardness Results and Improved Analysis.
>
> **Thank you once again for your thoughtful review and constructive feedback.**

---

> > ### Author Response · Authors · 2025-08-07
> >
> > Dear Reviewer oJg7,
> >
> > We sincerely thank you for your valuable comments and appreciate the time and effort dedicated to providing constructive feedback on our submission. We have carefully considered your suggestions and made significant efforts to address them. Given the limited timeframe of the rebuttal period, we would greatly appreciate it if you could let us know whether any concerns remain. Your insights are invaluable to us as we strive to enhance the quality of our work. Thank you once again for your time and effort in reviewing our paper.
> >
> > Sincerely,
> >
> > Authors of the paper

---

> > > ### Comment · Reviewer_oJg7 · 2025-08-07
> > > **Comments**
> > >
> > > My concern has been solved. I will raise my score to 4.

---

> > > > ### Author Response · Authors · 2025-08-07
> > > >
> > > > Dear Reviewer oJg7,
> > > >
> > > > Thank you very much for your reply and recognition. We are delighted that your concerns have been resolved. We sincerely appreciate your thoughtful review and constructive feedback.
> > > >
> > > > Sincerely,
> > > >
> > > > Authors of the paper

---

### Official Review · Reviewer_Xg2q · 2025-07-03

**Clarity:** 3
**Significance:** 3
**Originality:** 3
**Rating:** 4
**Confidence:** 3

**Summary:**

This paper proposes AdaSDBO, a fully problem-parameter-free algorithm for decentralized bilevel optimization with a single-loop structure. AdaSDBO employs adaptive stepsizes based on cumulative gradient norms to simultaneously update all variables, enabling dynamic adjustment without requiring problem-specific hyperparameter tuning. Theoretical analysis in the nonconvex–strongly-convex bilevel setting shows that AdaSDBO achieves a convergence rate matching that of well-tuned state-of-the-art methods up to polylogarithmic factors. Extensive experiments validate the effectiveness and robustness of AdaSDBO across various tasks and settings.

**Questions:**

1. What is the rationale behind using the accumulators in Lines 178 and 181? Transmitting gradient norms can significantly increase communication overhead. Are there alternative accumulator designs that could reduce this cost?
2. What motivated the design of the primal variable update in Line 196? Could the authors provide more explanation for the choice of the proportional terms used in the update?

**Ethical Concerns:**

["NO or VERY MINOR ethics concerns only"]

**Final Justification:**

As most of my concerns—such as the communication overhead, the scale of the experimental settings, and the practical efficiency of the algorithm—have been addressed, I increase my score by one.

**Limitations:**

Please refer to weaknesses.

**Quality:**

3

**Strengths And Weaknesses:**

AdaSDBO leverages accumulated gradient norms to dynamically adjust stepsizes at each iteration, thereby eliminating the need for hyperparameter tuning. It introduces two key mechanisms: (1) a hierarchical stepsize design that accounts for the interdependence of variables while preserving the autonomy of adaptive updates; and (2) a stepsize tracking scheme that synchronizes gradient-norm accumulators across agents, effectively mitigating stepsize discrepancies in the decentralized setting. The effectiveness of AdaSDBO is supported by both rigorous theoretical analysis and comprehensive experimental results.

**Weaknesses:**
1. The stepsize tracking scheme introduces significant communication overhead due to the additional transmission of gradient norms.
2. AdaSDBO appears to be designed for deterministic bilevel optimization and relies on full gradient information, limiting its applicability in stochastic settings.
3. While often considered negligible, the convergence rate includes an extra $\log^4(T)$ factor, which creates a gap compared to the optimal $O(1/T)$ rate.
4. Each iteration requires Jacobian and Hessian computations, which can be computationally expensive for large-scale problems and may hinder scalability.
5. The experimental evaluation is limited in scale, involving no more than 12 agents and relatively simple datasets and models.
6. The runtime performance of the algorithm is not reported, making it difficult to assess its practical efficiency.
7. The code is not provided, which hinders the reproducibility of the experimental results.

---

> ### Author Rebuttal · Authors · 2025-07-25
>
> **We sincerely appreciate the reviewer for recognizing our contributions and for the constructive comments. Our point-to-point responses to concerns on Weaknesses and Questions are given below.**
>
> **Reply to Weaknesses:**
>
> **1.** Thank you for your thoughtful comment. The stepsize tracking scheme is a key component of our problem-parameter-free framework. Due to the hierarchical nature of decentralized bilevel optimization, multiple coupled adaptive stepsizes naturally arise and may evolve inconsistently across agents. Without proper coordination, this variability may impair convergence. Importantly, our method employs a lightweight tracking mechanism that synchronizes scalar stepsizes across agents. The resulting communication overhead is minimal—only scalar values are exchanged—especially when compared to the cost of transmitting primal variables ($x$), dual variables ($y$), and auxiliary variables ($v$), which are commonly involved in decentralized bilevel methods.
> In summary, our approach does not incur significant communication overhead relative to existing methods, while simultaneously offering a robust and problem-parameter-free solution to decentralized bilevel optimization.
>
> **2.** Thank you for your insightful comment. Extending the proposed methods to stochastic bilevel optimization remains a challenging direction with several open theoretical issues:
>
> - The variance in both first- and second-order gradient estimates can affect the reliability of stepsize bounds and compromise algorithmic stability.
>
> - The two-stage convergence analysis and the coupled stepsize structure adopted in our framework may require additional conditions or variance-reduction techniques to ensure convergence in the stochastic setting.
>
> In this work, our primary aim is to establish **the first problem-parameter-free method for decentralized bilevel optimization**, which lays a solid foundation in the current literature. We are enthusiastic about extending our framework to stochastic settings in future work.
>
> **3.** Thank you for your thoughtful comment. We acknowledge that the $\log^4(T)$ term introduces a slight theoretical gap compared to the $\mathcal{O}(1/T)$ rate. However, in optimization research, such logarithmic factors are generally considered marginal due to their much slower growth compared to polynomial terms [1]–[4]. In our work, the presence of this logarithmic term stems from the accumulated gradient norms used for adaptive stepsize adjustment. Accordingly, it is an inherent consequence of achieving a problem-parameter-free implementation and is a common feature in similar problem-parameter-free algorithms (e.g., [1]–[5]). Given the significant practical advantages of removing the need for problem-specific tuning, this slight theoretical gap is widely regarded as acceptable and well-justified in the current literature.
>
> **4.** Thank you for your thoughtful comment. We agree that computing the full Hessian matrix can be expensive, especially for large-scale problems. However, we respectfully clarify that our method only requires computing an approximate Hessian–vector product, rather than the full Hessian matrix, which is significantly more efficient.
>
> Specifically, we compute the Hessian–vector product  $\nabla^{2} \ell(x, y)  v$ indirectly using the identity $\nabla^{2} \ell(x, y) v = \nabla \left( \nabla \ell(x, y)^{\top} v \right).$
>
> This computation involves the following two steps:
>
> - Compute the inner product  $s := \nabla \ell(x, y)^{\top} v,$ which is a scalar and only requires first-order gradient evaluations.
>
> - Compute the gradient of $s$ with respect to $y$: $\nabla s = \nabla \left( \nabla \ell(x, y)^{\top} v \right) = \nabla^{2} \ell(x, y) v.$
>
> This two-step process avoids forming the full Hessian explicitly and relies only on gradient-like operations. As a result, the computational overhead is much lower than directly computing or storing the full Hessian.
>
> **5.** Thank you for your constructive comment. To address this concern, we have conducted additional experiments on a more challenging decentralized meta-learning task using the CIFAR-10 dataset with a CNN model. In particular, we extended the number of agents to 10, 20, and 30—the widest setting we could support given our limited computational resources.
> We summarize the results in the table below:
>
> **Table 1: Convergence Performance under Different Number of Agents with Decentralized Meta-Learning Task**
>
> ---
> |Number of Agents||$n=10$||$n=20$||$n=30$|
> |---|:---:|:---:|:---:|:---:|:---:|:---:|
> |Algorithms|Train Acc|Test Acc|Train Acc|Test Acc|Train Acc|Test Acc|
> |AdaSDBO|$0.543\pm0.002$|$0.534\pm0.002$|$0.542\pm0.002$|$0.534\pm0.001$|$0.538\pm0.001$|$0.533\pm0.001$|
> |SLDBO|$0.503\pm0.004$|$0.488\pm0.005$|$0.472\pm0.006$|$0.461\pm0.006$|$0.486\pm0.006$|$0.475\pm0.007$|
> ---
>
> These results demonstrate that AdaSDBO remains consistently stable and competitive as the number of agents increases, outperforming the baselines even under more complex settings. We have also made every effort to add new experiments on the CIFAR-100 dataset using a ResNet-18 model with varying numbers of agents. The results, presented in the following table, confirm the sustained superior performance of our proposed method.
>
> **Table 2: Convergence Performance on the CIFAR-100 Dataset with Varying Numbers of Agents**
>
> ---
> |Algorithms|&nbsp;&nbsp;&nbsp;&nbsp;&nbsp;$n=5$|&nbsp;&nbsp;&nbsp;&nbsp;&nbsp;$n=8$|
> |---|:---:|:---:|
> |AdaSDBO|$0.426\pm0.002$|$0.417\pm0.003$|
> |SLDBO|$0.401\pm0.005$|$0.388\pm0.005$|
> ---
>
> **6.** Thank you for your helpful comment. To assess the practical efficiency of our algorithm, we report the total wall-clock runtime required to complete training over 500 communication rounds under different network topologies, using the MNIST and FMNIST datasets. The structural details of these topologies can be found in the Appendix of [1]. The results are summarized below:
>
> **Table 3: Total Runtime Performance under Different Topologies ($n=8$)**
>
> ---
> |Dataset||&nbsp;&nbsp;MNIST|||&nbsp;&nbsp;FMNIST||
> |---|:---:|:---:|:---:|:---:|:---:|:---:|
> |Algorithms|Ring (s)|Ladder (s)|Complete (s)|Ring (s)|Ladder (s)|Complete (s)|
> |AdaSDBO|$1029.6$|$1204.2$|$1502.4$|$1609.2$|$1636.8$|$1812.6$|
> |SLDBO|$999.6$|$1200.2$|$1498.2$|$1428.0$|$1483.8$|$1794.6$|
> |MA-DSBO|$1147.2$|$1215.2$|$1512.0$|$1619.4$|$1702.2$|$1833.6$|
> |MDBO|$1185.0$|$1217.4$|$1654.8$|$1677.6$|$1783.6$|$1906.2$|
> ---
>
> From the results, we observe that AdaSDBO incurs slightly higher total runtime than the single-loop method SLDBO. However, in terms of convergence results, we respectfully refer you to Table 1 in our response to Reviewer 3RRn due to space limitations here, which demonstrates that AdaSDBO consistently achieves superior accuracy and convergence performance compared to SLDBO.
> This implies that AdaSDBO requires less time to reach the same level of accuracy, highlighting its improved overall training efficiency. Moreover, compared to double-loop methods, AdaSDBO remains significantly more efficient in terms of both runtime and convergence performance.
>
> **7.** Thank you for your valuable comment. We will release our complete codebase upon acceptance via GitHub, including thorough documentation, running scripts, and configuration files.
>
> **Reply to Questions:**
>
> **1.** Thank you for your thoughtful question. The accumulators in our framework serve to adaptively adjust stepsizes by accumulating the squared gradients of each variable over time. The accumulator allows the algorithm to assign smaller stepsizes to variables with consistently large gradients, and larger stepsizes to those with smaller or infrequent gradients.
> This is particularly important in decentralized bilevel settings, where gradient magnitudes can vary significantly across agents or levels. This mechanism enhances stability and robustness without requiring manual tuning of stepsize schedules.
>
> We agree that communication efficiency is important in decentralized settings. However, in our framework, only scalar gradient norms are transmitted as accumulators, which incurs minimal overhead compared to exchanging high-dimensional variables.
> While alternative designs (e.g., compressed accumulators) could reduce communication, they risk losing global coordination, which is essential in our problem-parameter-free setting to avoid divergence from inconsistent learning dynamics.
> However, exploring more communication-efficient schemes—such as quantized norms or partial sharing—is an interesting direction for future work.
>
> **2.** Thank you for your question. The update in Line 196 concerns the upper-level variable $x$, which must be carefully coordinated with the lower-level variables $y$ and the auxiliary variable $v$ to maintain a faithful hypergradient approximation in single-loop decentralized bilevel optimization.
>
> In our design, the stepsize for $x$ is conservatively adapted based on the accumulated gradient norms of $x$,
> $y$, and $v$. This ensures the upper-level update does not progress too aggressively relative to the lower level, which is crucial for maintaining stability in a single-loop, problem-parameter-free setting.
> This hierarchical coordination plays a central role in balancing progress across levels and avoiding divergence in decentralized bilevel optimization.
>
> **References:**
>
> [1] Problem-Parameter-Free Decentralized Nonconvex Stochastic Optimization.
>
> [2] Two Sides of One Coin: The Limits of Untuned SGD and the Power of Adaptive Methods.
>
> [3] How Free is Parameter-Free Stochastic Optimization?
>
> [4] Tuning-Free Stochastic Optimization.
>
> [5] Learning-Rate-Free Stochastic Optimization over Riemannian Manifolds.
>
> **Thank you once again for your thoughtful review and constructive feedback.**

---

> > ### Comment · Reviewer_Xg2q · 2025-08-04
> >
> > Thank you to the authors for their responses. As most of my concerns have been addressed, I will increase my score by one.

---

> > > ### Author Response · Authors · 2025-08-05
> > >
> > > Dear Reviewer Xg2q,
> > >
> > > Thank you very much for your reply and recognition. We are delighted that our responses have addressed your concerns. We sincerely appreciate your thoughtful review and constructive feedback.
> > >
> > > Sincerely,
> > >
> > > Authors of the paper

---

### Official Review · Reviewer_3RRn · 2025-07-06

**Clarity:** 3
**Significance:** 3
**Originality:** 3
**Rating:** 5
**Confidence:** 2

**Summary:**

The authors propose AdaSDBO -- a parameter-free single-loop algorithm for decentralized bilevel optimization. Leveraging accumulators and a hierarchical stepsize schedule, the algorithm eliminates the need for knowledge of smoothness and strong-convexity constants. Authors provide theoretical analysis showing that AdaSDBO achieves an $O\left(\frac{\log^4{T}}{T}\right)$ stationarity rate. Experiments on synthetic and real-world problems, and CIFAR-10 meta-learning demonstrate competitive performance compared to baselines.

**Questions:**

1. The second equation on line 174 is not informative — removing it might improve readability.
2. What is the set $\mathcal{V}$?
3. Figure 2 (c & d) — in the legend description, the formula $(n = x)$ is not horizontally centered.

**Ethical Concerns:**

["NO or VERY MINOR ethics concerns only"]

**Final Justification:**

This work represents a relevant contribution to the area of problem-free methods. The authors provide a clear exposition of their results and support their claims with sufficient and meaningful experiments.

**Limitations:**

yes

**Paper Formatting Concerns:**

No major formatting concerns

**Quality:**

3

**Strengths And Weaknesses:**

Strengths:
1. The authors guide the reader through the paper step by step, explaining design choices from a high-level perspective.
2. The algorithm is parameter-free, so practitioners can deploy the algorithm without tuning hyperparameters.
3. The authors establish the SOTA $O(\log^{4}T/T)$ convergence rate, although I have not checked the proofs.

Weaknesses:
1. The experiments cover only a few graph shapes, and they do not disclose how the topology of the network affects the convergence.
2. Many deep-learning tasks are neither smooth nor strongly convex, which limits the direct relevance of the theory.

---

> ### Author Rebuttal · Authors · 2025-07-25
>
> **We sincerely appreciate the reviewer for recognizing our contributions and for the constructive comments. Our point-to-point responses to concerns on Weaknesses and Questions are given below.**
>
> **Reply to Weaknesses:**
>
> **1.** Thank you for your insightful comment. Following your suggestion, we have conducted additional experiments with two additional graph topologies beyond the ring: ladder and complete, using 8 agents on both the MNIST and FashionMNIST datasets. The structural details of these topologies can be found in the Appendix of [1]. A summary of the experimental results across the different topologies is provided below:
>
> **Table 1: Convergence Performance under Different Topologies ($n=8$)**
>
> ---
> | Dataset |       |  &nbsp; &nbsp;&nbsp; &nbsp;MNIST   |             |      | &nbsp; &nbsp;&nbsp; FMNIST      |      |
> |----------|:------------------:|:--------------------:|:--------------------:|:-----------------------:|:--------------------:|:--------------------:|
> | Algorithms | Ring |Ladder | Complete | Ring |Ladder | Complete |
> | AdaSDBO   | $0.908\pm 0.001$ |$0.911\pm 0.001$| $0.916\pm 0.001$   | $0.774\pm 0.003$ |$0.788\pm 0.003$   | $0.798\pm 0.002$  |
> | SLDBO |  $0.871\pm 0.002$  | $0.871\pm 0.001$  | $0.872\pm 0.001$    | $0.758\pm 0.002$ |$0.758\pm 0.002$   | $0.758\pm 0.001$ |
> | MA-DSBO |  $0.850\pm 0.001$  | $0.850\pm 0.001$ | $0.851\pm 0.002$   | $0.709\pm 0.002$ |$0.716\pm 0.001$   | $0.720\pm 0.002$  |
> | MDBO | $0.753\pm 0.002$ | $0.754\pm 0.001$ | $0.764\pm 0.001$   | $0.650\pm 0.001$ |$0.650\pm 0.002$   | $0.653\pm 0.001$ |
> ---
>
> As shown in the table above, better network connectivity leads to faster convergence and improved final performance. The complete topology, having the highest connectivity, achieves the best convergence rate and stability, while the ring topology lags behind due to its minimal connectivity.
>
> **2.** Thank you for your valuable comment. We agree that many deep learning problems do not satisfy smoothness or strong convexity assumptions, particularly in realistic scenarios where both levels of the bilevel problem may be nonconvex. However, due to the inherent complexity of analyzing fully nonconvex bilevel problems, most existing theoretical works on bilevel optimization—such as [2]-[4]—primarily focus on the nonconvex-strongly-convex setting.
>
> We acknowledge that extending the theory to fully nonconvex–nonconvex settings is a highly significant direction, but it remains a challenging open problem in decentralized bilevel optimization. In such cases, the lower-level solution may not be unique, and the upper-level objective can become non-differentiable [5]. Addressing these challenges would require the development of fundamentally different techniques, which are beyond the scope of this work. We plan to investigate this in future work, thank you once again for your insightful comment.
>
> **Reply to Questions:**
>
> **1.** Thank you for your helpful suggestion. Following your advice, we have removed this equation in the revised version to enhance clarity.
>
> **2.** Thank you for your question. The set $\mathcal{V}$ refers to the domain of the auxiliary variable $v$, we have clarified this definition explicitly to improve readability.
>
> **3.** Thank you for pointing this out. We have adjusted the formatting in the legend of Figure 2 (c & d) to ensure that the formula is now properly horizontally centered.
>
> **References:**
>
> [1] Problem-Parameter-Free Decentralized Nonconvex Stochastic Optimization.
>
> [2] Decentralized Bilevel Optimization over Graphs: Loopless Algorithmic Update and Transient Iteration Complexity.
>
> [3] A Single-Loop Algorithm for Decentralized Bilevel Optimization.
>
> [4] SPARKLE: A Unified Single-Loop Primal-Dual Framework for Decentralized Bilevel Optimization.
>
> [5] On Finding Small Hyper-Gradients in Bilevel Optimization: Hardness Results and Improved Analysis.
>
> **Thank you once again for your thoughtful review and constructive feedback.**

---

> > ### Comment · Reviewer_3RRn · 2025-08-04
> >
> > Thanks for your rebuttal.
> >
> > 1. Thank you for running the new experiments. You're right that the ladder and complete topologies differ from the ring used in the original setup. However, the choice of the complete topology seems somewhat unfortunate, as decentralized algorithms are typically replaced by centralized ones in this setting. A fair comparison would likely require benchmarking against centralized alternatives. Similarly, the ladder topology is quite specific. Would it make sense to instead consider random connectivity graphs and report performance over them?
> > 2. I appreciate your perspective on this point. Honestly, I wish the field paid closer attention to this issue—there’s a growing disconnect between optimization theory and the practical settings where we apply these algorithms. That said, you're right: addressing this properly is beyond the scope of this work, even if it makes the experiments with neural networks feel a bit detached from the theory the paper builds on.
> >
> > Thank you as well for addressing the rest of my questions.

---

> > > ### Author Response · Authors · 2025-08-05
> > >
> > > Dear Reviewer 3RRn,
> > >
> > > Thank you very much for your reply. Our point-to-point responses to your remaining questions are given below.
> > >
> > > **1.** Following your suggestion, we conducted additional experiments under a random topology, where each pair of nodes is connected with a probability of 0.8, consistent with the setting used in Appendix of [1]. The results are summarized below:
> > >
> > > **Table 1: Convergence Performance under Different Topologies ($n = 8$)**
> > >
> > > ---
> > > | Dataset || &nbsp;&nbsp;&nbsp;&nbsp;&nbsp;&nbsp;MNIST| || &nbsp;&nbsp;&nbsp;&nbsp;&nbsp;FMNIST | |
> > > |-----|:---:|:---:|:---:|:---:|:---:|:---:|
> > > | Algorithms |Ring  | Ladder   | Random| Ring     | Ladder    | Random |
> > > | AdaSDBO  | $0.908\pm0.001$  | $0.911\pm0.001$  | $0.913\pm0.001$  | $0.774\pm0.003$  | $0.788\pm0.003$  | $0.790\pm0.003$ |
> > > | SLDBO| $0.871\pm0.002$  | $0.871\pm0.001$  | $0.871\pm0.001$  | $0.758\pm0.002$  | $0.758\pm0.002$  | $0.758\pm0.001$  |
> > > | MA-DSBO| $0.850\pm0.001$  | $0.850\pm0.001$  | $0.850\pm0.002$  | $0.709\pm0.002$  | $0.716\pm0.001$  | $0.719\pm0.002$  |
> > > | MDBO| $0.753\pm0.002$  | $0.754\pm0.001$  | $0.754\pm0.002$  | $0.650\pm0.001$  | $0.650\pm0.002$  | $0.653\pm0.001$  |
> > > ---
> > >
> > > Additionally, we conducted a comparison of our proposed method under the complete topology against the centralized single-loop bilevel baseline FSLA [2]. The results on the MNIST and FashionMNIST datasets are presented below:
> > >
> > > **Table 2: Convergence Performance Comparison of AdaSDBO (Complete Topology) and the Centralized Method FSLA**
> > >
> > > ---
> > > |Algorithms|&nbsp;&nbsp;&nbsp;&nbsp;&nbsp;&nbsp;MNIST| &nbsp;&nbsp;&nbsp;&nbsp;FMNIST |
> > > |---|:---:|:---:|
> > > | AdaSDBO | $0.916\pm 0.001$ |$0.798\pm 0.002$ |
> > > | FSLA | $0.889\pm0.002$|$0.791\pm 0.002$ |
> > > ---
> > >
> > > These additional results further demonstrate the robustness and superior performance of our proposed problem-parameter-free algorithm across diverse network topologies, as well as its better overall performance compared to the centralized method.
> > >
> > > **2.** We sincerely appreciate your thoughtful perspective. We would like to clarify that, to align our experiments with the theoretical assumptions, we incorporate a strongly convex regularizer into the lower-level objective—consistent with the experimental setups in prior works such as [3]–[6]. This ensures coherence between the experimental design and the assumptions made in our theoretical analysis. Moreover, we fully agree that extending the current theory to the more general nonconvex–nonconvex regime is both important and meaningful. We are enthusiastic about exploring this direction in future research, including the design of problem-parameter-free algorithms with theoretical guarantees in fully nonconvex landscapes. Thank you for your constructive input.
> > >
> > > Thank you once again for your thoughtful review and helpful dialogue throughout the discussion phase.
> > >
> > > **References:**
> > >
> > > [1] Problem-Parameter-Free Decentralized Nonconvex Stochastic Optimization.
> > >
> > > [2] A Fully Single Loop Algorithm for Bilevel Optimization without Hessian Inverse.
> > >
> > > [3] Locally Differentially Private Decentralized Stochastic Bilevel Optimization with Guaranteed Convergence Accuracy.
> > >
> > > [4] A Single-Loop Algorithm for Decentralized Bilevel Optimization.
> > >
> > > [5] Decentralized Stochastic Bilevel Optimization with Improved per-Iteration Complexity.
> > >
> > > [6] Decentralized Bilevel Optimization over Graphs: Loopless Algorithmic Update and Transient Iteration Complexity.
> > >
> > > Sincerely,
> > >
> > > Authors of the paper

---

> > > > ### Comment · Reviewer_3RRn · 2025-08-05
> > > >
> > > > Thanks for your response. Just to clarify -- did you use only one random graph for the experiments and report the results based on that single random instance? If so, the insights might be limited, as the performance could depend heavily on the specific graph structure. Have you considered running the comparison across multiple random graphs instead? This would provide a more robust and statistically meaningful evaluation. Also, what metric do you think would be most appropriate for comparing the algorithms in such a setup?

---

> > > > > ### Author Response · Authors · 2025-08-07
> > > > >
> > > > > Dear Reviewer 3RRn,
> > > > >
> > > > > Thank you very much for your reply. In our previous experiment, we followed the random graph setting consistent with prior work and used a single instance of a random graph to report experimental results (e.g., [1]). As per your suggestion, to provide a more statistically robust evaluation, we have now conducted a new set of experiments across 5 independently generated random graphs. For each of the four compared methods, we run experiments on the 5 random graphs and report the mean and standard deviation of the final test accuracy across these graph instances. This metric helps better assess the stability and robustness of each method under variations in network topology. We perform these evaluations on both the MNIST and FashionMNIST datasets, and the results are summarized below.
> > > > >
> > > > > **Table 1: Comparison Across 5 Random Graphs ($n = 8$)**
> > > > >
> > > > > ---
> > > > > |Algorithms|&nbsp;&nbsp;&nbsp;&nbsp;AdaSDBO| &nbsp;&nbsp;&nbsp;&nbsp;&nbsp;&nbsp;SLDBO|&nbsp;&nbsp;&nbsp;MA-DSBO|&nbsp;&nbsp;&nbsp;&nbsp;&nbsp;&nbsp;MDBO|
> > > > > |-----|:---:|:---:|:---:|:---:|
> > > > > |MNIST|$0.914\pm0.002$|$0.871\pm0.002$|$0.851\pm0.001$|$0.756\pm0.002$|
> > > > > |FMNIST|$0.790\pm0.002$|$0.757\pm0.004$|$0.719\pm0.002$|$0.655\pm0.003$|
> > > > > ---
> > > > >
> > > > > As shown in the table above, our proposed method consistently achieves the highest final test accuracy, with both competitive average performance and low variance across different random graph realizations. This demonstrates its robustness against variations in network topology.
> > > > >
> > > > > Thank you once again for your thoughtful question.
> > > > >
> > > > > **References:**
> > > > >
> > > > > [1] Problem-Parameter-Free Decentralized Nonconvex Stochastic Optimization.
> > > > >
> > > > > Sincerely,
> > > > >
> > > > > Authors of the paper

---

> > > > > > ### Comment · Reviewer_3RRn · 2025-08-08
> > > > > >
> > > > > > Thank you. I will raise my score.

---

> > > > > > > ### Author Response · Authors · 2025-08-08
> > > > > > >
> > > > > > > Dear Reviewer 3RRn,
> > > > > > >
> > > > > > > Thank you very much for your reply and recognition. We are delighted that our responses have addressed your concerns. We sincerely appreciate your thoughtful review and constructive feedback.
> > > > > > >
> > > > > > > Sincerely,
> > > > > > >
> > > > > > > Authors of the paper

---

### Decision · Program_Chairs · 2025-09-17

**Decision:**

Accept (poster)

**Comment:**

The paper addresses decentralized bilevel optimization and proposes AdaSDBO, a parameter-free single-loop algorithm that removes the need for problem-specific hyperparameter tuning. AdaSDBO leverages accumulated gradient norms together with a hierarchical stepsize scheme to dynamically adjust updates across agents while mitigating stepsize discrepancies in the decentralized setting. The authors provide rigorous theoretical analysis, showing that AdaSDBO achieves an $\mathcal{O}(1/T)$ stationarity rate in the nonconvex–strongly convex setting, matching state-of-the-art methods up to polylogarithmic factors. Extensive experiments on synthetic tasks, real-world problems, and CIFAR-10 meta-learning demonstrate its effectiveness and robustness. Overall, the paper is well written, with clear novelty and strong theoretical and empirical support. I therefore recommend acceptance. Below are some comments that should be clarified in the final version (they have been addressed in the rebuttal).

- AdaSDBO is designed for deterministic bilevel optimization and relies on full gradient information, which limits its applicability in stochastic settings.
- In decentralized algorithms, communication overhead is a critical concern. The paper does not provide a detailed analysis of AdaSDBO’s communication complexity, which could be important for real-world distributed systems. A comparison of communication costs with other methods would strengthen the work.